# Rete ridges form via evolutionarily distinct mechanisms in mammalian skin

Sean M. Thompson[1], Violet S. Yaple[1], Gabriella H. Searle[1], Quan M. Phan[1], Jasson Makkar[1], Xiangzheng Cheng[2], Ruiqi Liu[3], Anna Pulawska-Czub[4], Corin Yanke[1], Natalie M. Williams[1], Isabelle V. Busch[1], Tommy T. Duong[1], Matteo V. Corneto[1], Zachary S. Jordan[5], Debarun Roy[5], Adam B. Salmon[6,7,8], Ov D. Slayden[9], Brian P. Hermann[5], David A. Stoltz[10,11], Michael J. Welsh[10,11,12], UW Birth Defects Research Laboratory*, Ian A. Glass[13], Krzysztof Kobielak[4,14], Qing Nie[3,15], Suoqin Jin[2], Heiko T. Jansen[16], Michela Ciccarelli[1,17], Maksim V. Plikus[3], Iwona M. Driskell[1] & Ryan R. Driskell[1,17✉]

The loss of fur during human evolution has long mystified scientists and the public[1–5]. Reduced hair density coincides with acquisition of epidermal rete ridges, the developmental timing and molecular mechanisms of which are poorly understood despite their prominence in humans[1,6–9]. Examination of human and pig skin development has shown that rete ridges form through a mechanism independent from those of hair follicles[10,11] and sweat glands[3,4,12–15] by establishing interconnected epidermal invaginations. Here we document the occurrence of rete ridges across Mammalia, including in grizzly bears and dolphins, and show that neonatal pig wounds can regenerate them de novo. Multispecies spatiotemporal transcriptomics identifies significant signalling interactions between epidermal and dermal cells during rete ridge morphogenesis, particularly through bone morphogenetic proteins (BMP). We also demonstrate that mouse fingerpad skin forms rete ridges and functionally requires epidermal BMP signalling. We propose that evolution of rete ridges in mammalian skin involved replacement of the molecular program for formation of discrete microscopic appendages, including hair follicles and sweat glands, with a distinct program for the interconnected appendage network. Broad epidermal activation of BMP is required for the development of rete ridge networks organized around underlying dermal pockets. Understanding rete ridge mechanisms may enable development of therapeutic approaches to regenerate epidermal appendages lost during wounding or disease in humans.

During human skin development, the epidermis undergoes a complex series of signalling events that give rise to different types of specialized epidermal appendage, including hair follicles, sweat glands, fingerprint ridges in volar skin, and rete ridges, which support the skin's anatomical complexity and diverse functions[3,4,7,10–14,16,17]. Altered formation of these appendages has been implicated in skin diseases, scarring and ageing[8,10,12,18–22]. Historically, comparative approaches to skin biology have aided identification and classification of the diverse cutaneous structures present in humans and other mammals[16,23–25]. Mice are the dominant model system used to study skin development, wound healing and ageing owing to their ease of handling and an expansive library of transgenic and other technologies that enables genetic manipulation[20].

Technological advances in transgenic and single-cell transcriptomics have further aided identification and validation of new and previously identified molecular and cellular mechanisms underlying the development and regeneration of hair follicles[10,11,20,26–28], sweat glands[3,12–15,17,19] and fingerprint ridges[12]. Mouse trunk skin, unlike that of humans, does not form rete ridges. In addition, previous studies investigating human skin development have failed to precisely record the formation of rete ridges, leaving the cellular and molecular mechanisms required for rete ridge formation unknown[6,7,12,17,29].

Epidermal and dermal signalling programs are critical for the specification of diverse epidermal appendages within the skin of vertebrates, such as scales in reptiles, feathers in birds and hair follicles in

[1]School of Molecular Biosciences, Washington State University, Pullman, WA, USA. [2]School of Mathematics and Statistics, Wuhan University, Wuhan, China. [3]Department of Developmental and Cell Biology, University of California, Irvine, Irvine, CA, USA. [4]Centre of New Technologies, University of Warsaw, Warsaw, Poland. [5]Department of Neuroscience, Developmental and Regenerative Biology, The University of Texas at San Antonio, San Antonio, TX, USA. [6]Barshop Institute for Longevity and Aging Studies, University of Texas Health Science Center, San Antonio, TX, USA. [7]Geriatric Research Education and Clinical Center, South Texas Veterans Healthcare System, San Antonio, TX, USA. [8]Department of Molecular Medicine, University of Texas Health Science Center, San Antonio, TX, USA. [9]Division of Reproductive & Developmental Sciences, Oregon National Primate Research Center, Oregon Health & Science University, Beaverton, OR, USA. [10]Department of Internal Medicine, Roy J. and Lucille A. Carver College of Medicine, University of Iowa, Iowa City, IA, USA. [11]Pappajohn Biomedical Institute, Roy J. and Lucille A. Carver College of Medicine, University of Iowa, Iowa City, IA, USA. [12]Howard Hughes Medical Institute, University of Iowa, Iowa City, IA, USA. [13]Department of Pediatrics, University of Washington School of Medicine, Seattle, WA, USA. [14]Faculty of Medicine, University of Warsaw, Warsaw, Poland. [15]Department of Mathematics, University of California, Irvine, Irvine, CA, USA. [16]Department of Integrative Physiology and Neuroscience, Washington State University, Pullman, WA, USA. [17]Center for Reproductive Biology, Washington State University, Pullman, WA, USA. *A list of authors and their affiliations appears at the end of the paper. ✉e-mail: ryan.driskell@wsu.edu

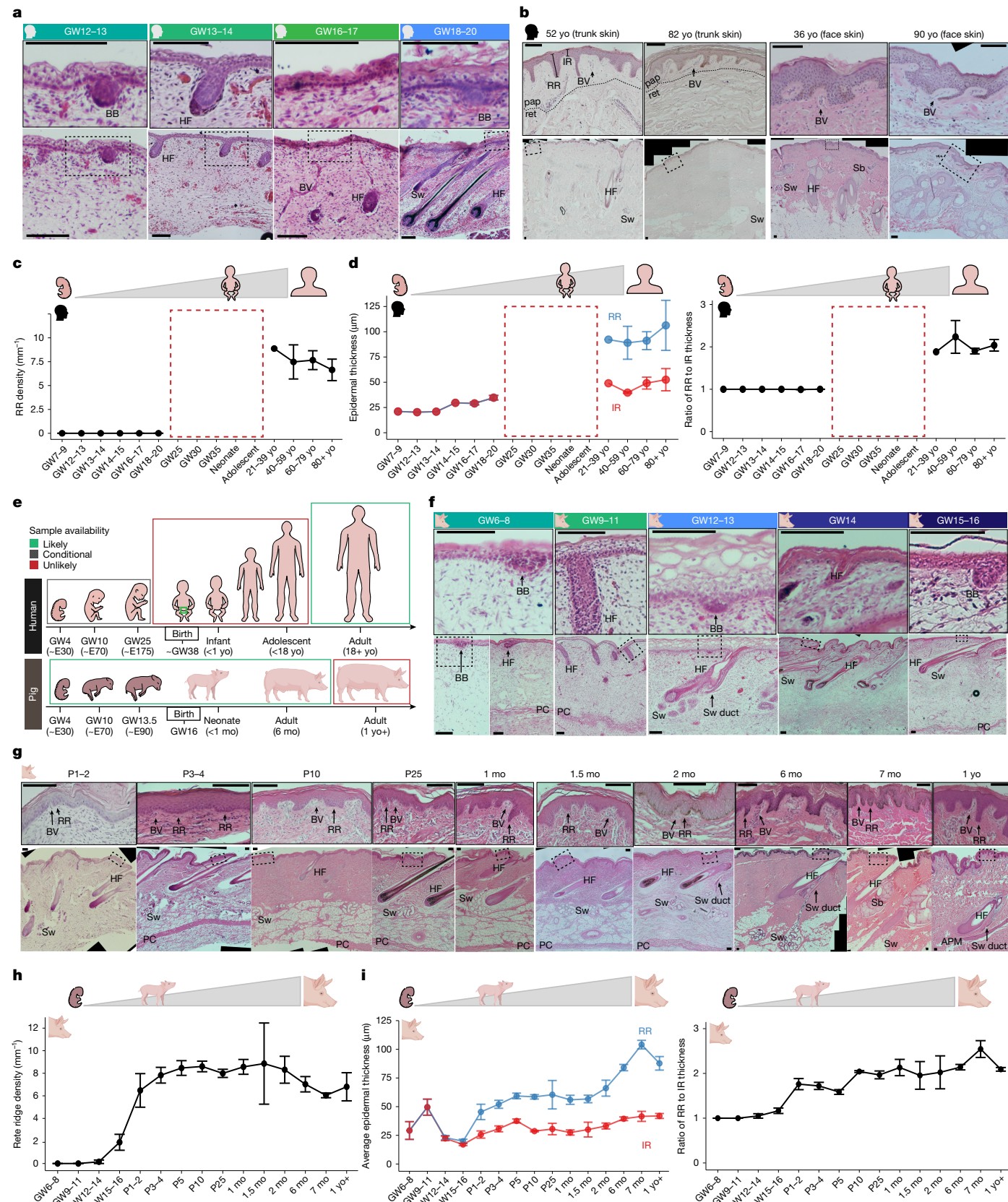

**Fig. 1** | See next page for caption.

mammals[1,3,4,10–12,16,17,19,30,31]. Developing hair follicles, sweat glands and fingerprint ridge placodes share several molecular signals, namely epidermal *EDA/R* and *LEF1/WNT* during their initiation, alongside focal

proliferation supporting appendage elongation[3,10–15,17,31,32]. However, the formation of a dermal condensate beneath the epithelial placode is unique to hair follicles, as sweat glands and fingerprint ridges

**Fig. 1 | Rete ridges form perinatally in human and pig skin. a**, Epidermal placodes form continuously through early-to-mid gestation, but rete ridges do not. Representative H&E stains from human GW12, GW13.5, GW17 and GW19 trunk skin are shown. **b**, Rete ridges have formed and are maintained in adult human skin. Representative H&E stains of samples from trunk and face skin of adult male humans are shown. In **a** and **b**, the dashed box indicates the region of the zoomed-in inlay. **c**,**d**, Quantification of human trunk skin histology at GW7–9 (*n* = 1), GW12–13 (*n* = 1), GW13–14 (*n* = 1), GW14–15 (*n* = 1), GW16–17 (*n* = 7) and GW18–20 (*n* = 5), and trunk or face skin of 21–39-year-old (21–39 yo, *n* = 1), 40–59 yo (*n* = 3), 60–79 yo (*n* = 11) and 80 yo+ (*n* = 8) individuals for rete ridge density (**c**), epidermal thickness (**d**, left), and the ratio of rete ridge (RR) to inter-ridge (IR) thickness (**d**, right). Sample sizes in **c** and **d** are the same as in **a** and **b**. **e**, Graphical representation of human gestation compared with pig gestation. Coloured boxes indicate general availability of tissue samples. **f**,**g**, Rete ridges begin to form perinatally in pig skin (**f**) and rete ridge formation peaks postnatally alongside increased epidermal thickness and dermal vascularization (**g**). Representative H&E stains from skin across fetal (**f**) and postnatal (**g**) pig development from mixed backgrounds are shown (see Methods for full details). **h**,**i**, Quantification of pig histology for rete ridge density, showing rete ridges form continuously across perinatal life in pigs (**h**); epidermal thickness, showing rete ridge formation and maturation drives postnatal epidermal thickening (**i**, left); and the ratio of RR to IR thickness (**i**, right) at GW6–8 (*n* = 7), GW9–11 (*n* = 6), GW12–14 (*n* = 14), GW15–16 (*n* = 4), P1–2 (*n* = 3), P3–4 (*n* = 7), P5 (*n* = 5), P10 (*n* = 2) and P25 (*n* = 2) and in 1 month old (mo; *n* = 9), 1.5 mo (*n* = 2), 2mo (*n* = 2), 6 mo (*n* = 7), 7 mo (*n* = 3) and >1 yo (*n* = 4) individuals. Sample sizes in **h** and **i** are the same as in **f** and **g**. Scale bars, 100 µm. BB, basal bud/epithelial placode; HF, hair follicle; BV, blood vessel; pap, papillary dermis; ret, reticular dermis; Sw, sweat gland; Sb, sebaceous gland; PC, panniculus carnosus; APM, arrector pili muscle. Error bars in line plots in **c**, **d**, **h** and **i** represent s.e.m. Illustrations in **c**–**i** were created using BioRender. Thompson, S. (2026) https://BioRender.com/8rd8cz9.

instead interact with molecularly distinct subpopulations of dermal cells[3,12,15,17,20]. Periodic patterning supports appendage specification and spacing through interactions between numerous signalling pathways, including the WNT, SHH and bone morphogenetic protein (BMP) pathways, which follow Turing principles[3,4,10–12,15,16,31,33]. By contrast, the molecular mechanisms and pattern-forming principles involved in rete ridge formation have remained elusive.

Unlike mouse trunk skin, pig skin closely resembles human skin[24] and scars similarly in adult wound healing contexts[18,34]. However, porcine skin development remains poorly defined, especially in comparison with that of mice. Therefore, we performed a comparative developmental study in humans and pigs to identify when rete ridges form. We generated single-cell transcriptomics (single-cell RNA sequencing; scRNA-seq) and spatial transcriptomics (spatial enhanced resolution omics sequencing; stereo-seq) datasets across pig skin development and reanalysed previously published human skin transcriptomics datasets[12,35,36] to infer shared molecular mechanisms underlying rete ridge development compared with hair follicle, sweat gland and fingerprint ridge formation. We then validated key molecular mechanisms in vivo using transgenic mice, genetic knockout pig models and wound healing approaches. Cellular and signalling interactions underlying rete-ridge-specific development in the skin of humans and pigs support a model for their formation and regeneration that requires epidermal BMP signalling. This model provides a critical foundation for understanding the developmental mechanisms of rete ridge formation in Mammalia and for potential regeneration of rete ridges following loss in disease contexts.

## Rete ridges form perinatally in skin

Epidermal rete ridges are not observed in trunk skin throughout fetal development in humans (Fig. 1a). By approximately gestational week 12 (GW12), hair follicle formation has been initiated via epidermal basal buds and underlying dermal condensates[29] (Fig. 1a). These hair follicles mature by GW19, and a new wave of epithelial placodes lacking dermal condensates becomes visible, probably representing developing sweat glands[3,17] (Fig. 1a). During mid-gestation, dermal and subdermal connective tissues progressively mature, whereas rete ridges have not yet formed (Fig. 1a and Extended Data Fig. 1a). By contrast, young adult and aged human skin prominently features rete ridges; this coincides with notable thickening of the epidermis (Fig. 1b–d). These rete ridges establish a distinctive undulating pattern along the basal side of the epidermis, as observed histologically[8,37] (Fig. 1a,b). In addition, the space beneath the inter-ridge epidermis is occupied by 'dermal pockets', a prominently vascularized region of the papillary dermis (Fig. 1b). Alterations to this dermal microenvironment of rete ridges have been previously associated with skin ageing[8,38] (Fig. 1b).

Critically, owing to ethical and legal limitations with respect to sampling of later fetal, neonatal and adolescent human tissues, the precise timing of rete ridge formation between mid-gestation and young adulthood has remained unresolved (Fig. 1c,e). Thus, how human skin progresses from having a thin epidermis with a smooth basal side during development to forming a thicker and patterned epithelium with interconnecting rete ridges in adulthood[6–8,37] has remained poorly understood (Fig. 1e).

Owing to the greater experimental and ethical availability of pigs, we used porcine skin development as a model for humans to identify the precise timing of rete ridge formation (Fig. 1e). We identified analogous developmental staging between GW6–8 in pigs and GW12 in humans, wherein the first basal buds are visible and the dermis remains immature (Fig. 1a,f and Extended Data Fig. 1a,b). By GW12–13, maturing hair follicles and sweat glands are both present, alongside another wave of basal buds developing subsequent putative sweat glands (Fig. 1f). Dermal connective tissue maturation is mirrored between mid–late gestation in pigs and mid-gestation in humans (Extended Data Fig. 1a,b). In perinatal GW15–16 pigs, small rete ridges first become discernible as regions of thicker undulating epidermis enclosing small dermal pockets (Fig. 1g and Extended Data Fig. 1b). However, rete ridges primarily form during the first week of postnatal life, as the epidermis continues to thicken, and the maturing dermal pockets start to show notable vascularization (Fig. 1g–i and Extended Data Fig. 1c). During the second week of postnatal life, rete ridge density plateaus (Fig. 1h). Maturing rete ridges elongate and are the main contributors to increased epidermal thickness in adulthood, as also seen in humans (Fig. 1b–d,g–i and Extended Data Fig. 1c). Thus, we conclude that pig epidermal and dermal development largely mirrors that of humans. Previous studies have already demonstrated that rete ridges form in human skin within several months of birth[6]. Here we further show that rete ridge formation begins perinatally in both humans and pigs, suggesting that porcine skin development is a close proxy for human skin development. In addition, the temporal overlap in the timing of rete ridge formation and epidermal thickening suggests that these processes may be linked.

## Rete ridges enable thicker epidermis

To understand the evolutionary context of rete ridges, we generated a histological zoo of adult skin of representative terrestrial and aquatic species from diverse orders and families across Mammalia (Fig. 2a and Extended Data Fig. 2a,b). Aquatic cetaceans such as the bottlenose dolphin, short-beaked common dolphin and long-beaked common dolphin have well-documented hairless trunk skin, prominent rete ridges and pronounced epidermal thickness[25,39] (Fig. 2a and Extended Data Fig. 2b,c). In addition to the common domestic pig of mixed backgrounds (Methods), we investigated several other breeds, including the Yucatan 'hairless' miniature pig, which has extremely low hair density; the heritage breed Mangalitsa pig, which has long, curly hair that gives it a woolly appearance; and the Hanford mini-pig, which is also

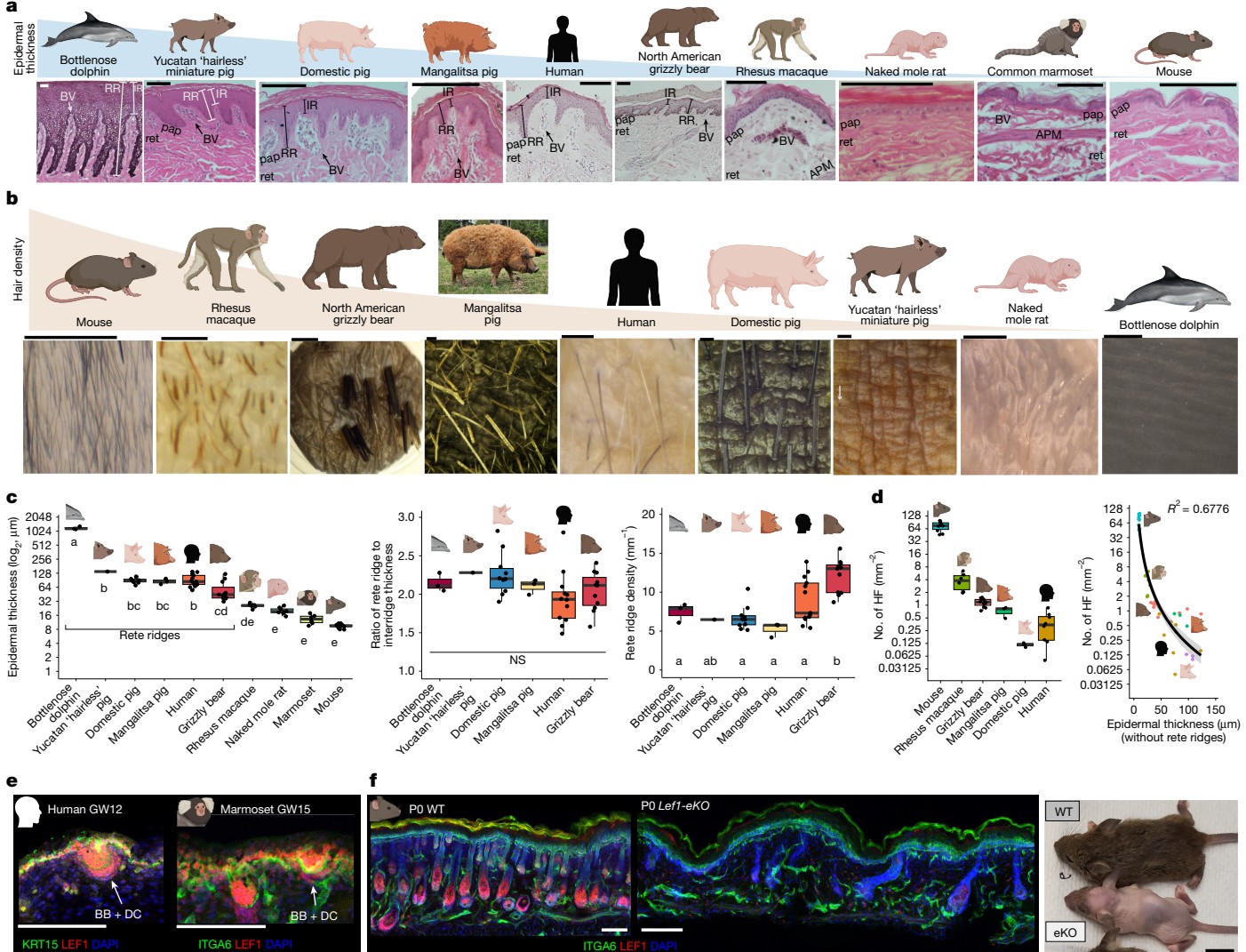

**Fig. 2 | Rete ridges support epidermal thickening. a**, Representative H&E stains of mature skin from across Mammalia: bottlenose dolphin, Yucatan miniaturized 'hairless' pig, domestic pig, Mangalitsa pig, human, North American grizzly bear, rhesus macaque, naked mole rat, common marmoset and 6 mo mouse skin. **b**, Representative images of hair density from across Mammalia. **c**, Rete ridges drive epidermal thickening in mammalian skin. Quantification of trunk skin histology for epidermal thickness (with rete ridges, if applicable) (left), ratio of rete ridge to inter-ridge thickness (middle) and rete ridge density (per mm) (right) in the bottlenose dolphin ($n = 3$), Yucatan miniaturized pig ($n = 1$), 6–7 mo adult domestic pig ($n = 10$), Mangalitsa pig ($n = 3$), human ($n = 13$), North American grizzly bear dorsal rump ($n = 11$), rhesus macaque ($n = 6$), naked mole rat ($n = 8$), common marmoset ($n = 6$) and adult mouse ($n = 15$) are shown. Replicates in **a** are the same as in **c**. Shared letters indicate no significant difference ($P > 0.05$), and different letters indicate significant difference ($P < 0.05$) according to one-way analysis of variance plus Tukey's HSD (exact $P$ values are provided in the source data). For **a**–**c**, see Extended Data Fig. 2a for visual representations of anatomical sites for all species and Methods for more detail. **d**, Rete ridges increase epidermal thickness in non-furry skin. Left, quantification of hair density images from adult mice ($n = 10$), rhesus macaques ($n = 6$), grizzly bears ($n = 11$), Mangalitsa

pigs ($n = 3$), 7 mo domestic pigs ($n = 3$) and adult humans ($n = 8$). Hair density was imaged and quantified as in **b**, except for human samples (see Methods for complete details). Right, correlation between epidermal thickness (with rete ridges, if applicable) and hair density. The correlation statistic shown is the adjusted coefficient of determination, $P = 2.407 \times 10^{-11}$. **e**, LEF1 is broadly expressed in fetal mammalian epidermis. Representative immunostains of GW12 human skin stained for KRT15 and LEF1 (left, $n = 1$) and GW15 marmoset trunk skin stained for ITGA6 and LEF1 (right, $n = 3$) are shown. **f**, Epidermal LEF1 controls hair density by regulating placode formation in trunk skin. Representative immunostains of P0 wild-type (WT) (left, $n = 3$) and P0 *K14-Cre;Lef1^{fl/fl}* (*Lef1-eKO*) (centre, $n = 3$) mouse trunk skin for ITGA6 and LEF1 are shown. Right, representative image of P21 WT (top, $n = 3$) and P21 *Lef1-eKO* (bottom, $n = 3$) female littermates. Histology scale bars, 100 μm (**a**,**e**,**f**); scale bar, 1 mm (**b**); littermate scale bar, 1 cm (**f**). DC, dermal condensate. Photograph in **b** was reproduced with permission from Tania Issa. Cetacean illustrations in were obtained from the National Oceanic and Atmospheric Agency (NOAA) Fisheries Species Directory entries for bottlenose dolphin. Other illustrations in **a**–**f** were created using BioRender. Thompson, S. (2026) https://BioRender.com/8rd8cz9.

miniaturized (Fig. 2a,b and Extended Data Fig. 2b,d,e). All breeds have rete ridges with adjacent vascularized dermal pockets, as well as sweat glands, like adult humans (Fig. 2a and Extended Data Fig. 2b). However, hair density differs among the breeds (Fig. 2b). North American grizzly bear dorsal rump skin contains sweat glands and hair follicles, which are large and organized into dense bundles, leaving expansive regions

of interfollicular (between the hair follicles) epidermis containing rete ridges (Fig. 2a,b and Extended Data Fig. 2b). Non-human primates, which are used in translational research owing to their close genetic similarity to humans, such as the 'Old World' rhesus macaque and 'New World' common marmoset, have hair follicles but lack rete ridges (Fig. 2a,b and Extended Data Fig. 2b). Macaques, but not marmosets,

also have sweat glands[9] (Fig. 2a and Extended Data Fig. 2b). Rodents, such as the naked mole rat (with very low hair density) and furry (with high hair density) mouse, lack rete ridges in their dorsal skin and do not form dermal pockets (Fig. 2a,b and Extended Data Fig. 2b).

Comparison of epidermal thickness across our histological zoo showed that species with rete ridges generally had thicker epidermis than those without them, whereas the inter-ridge thickness (aside from the cetaceans) was not as markedly increased (Fig. 2c and Extended Data Fig. 2f). The ratio of rete ridge to inter-ridge thickness was consistent among rete-ridge-bearing species, with the rete ridge thickness approximately double the inter-ridge thickness (Fig. 2c). As the inter-ridge thickness defines the ceiling of the dermal pocket, this ratio suggests a developmental link between the size of the dermal pocket and the overall epidermal thickness[8,25] (Fig. 2a,c and Extended Data Fig. 2b,f). Rete ridge density was also consistent across species, except in grizzly bears, which exhibited slightly higher density (Fig. 2c). Hair density across species also varied, with rete-ridge-less species generally having higher hair density and thinner epidermis (Fig. 2c,d and Extended Data Fig. 2f). Direct comparison of hair density and epidermal thickness revealed an inverse relationship (Fig. 2d and Extended Data Fig. 2f). As we observed no instance of a species having thick epidermis without rete ridges, the latter seem to be essential for stably increasing epidermal thickness.

## Loss of fur does not induce rete ridges

To test the genetic and developmental determinants of hair density and epidermal thickness, we targeted an evolutionarily conserved step in hair follicle formation, LEF1–WNT- and EDA–EDAR-mediated epidermal placode formation. The WNT signalling transcription factor LEF1 is highly expressed in the fetal epidermis of humans, non-human primates and rodents when hair follicles, sweat glands and fingerprint ridges develop from epithelial placodes[10,12,15] (Fig. 2e). However, epidermal LEF1 expression disappears when placodes stop forming, as in late fetal marmosets, naked mole rats and postnatal mice (Extended Data Fig. 2g,i). Critically, LEF1 can directly regulate *EDA*, whereas WNT and EDA–EDAR signalling exhibit bidirectional cross-talk[15,40]. To investigate the relationship between hair density and epidermal thickness, we generated conditional epidermal *Lef1* knockout (*Lef1-eKO*) mice to disrupt epidermal placode formation (Fig. 2f and Extended Data Fig. 2h,i). *Lef1-eKO* mice showed impaired hair follicle formation, maturation and maintenance (Fig. 2f, Extended Data Fig. 2i and Supplementary Fig. 1a–e). Hair follicle density was markedly reduced in *Lef1-eKO* mice, and the few resulting follicles, potentially representing 'escapers'[10], failed to maintain external hair fibre during the first hair cycle, establishing a cyclical pattern of hair growth and loss during subsequent hair cycles (Supplementary Fig. 1a–c). Changes in the underlying dermis were subtle and probably due to altered paracrine signalling between the much sparser hair follicles and the surrounding dermal compartment[41] (Supplementary Fig. 1d–f). Critically, the markedly reduced hair density in the absence of epidermal *Lef1* did not alter the thickness of the interfollicular epidermis at examined stages of postnatal development and maturation, suggesting that genetic reduction of hair density alone may not directly drive epidermal thickening or rete ridge formation (Supplementary Fig. 1d–f). Studies have also reported that diverse combinations of coding and non-coding sequence variations are associated with fine-tuning of hair density and hair shaft characteristics across Mammalia[5,9,19]. These studies, alongside our mouse model, provide support for combinatorial regulation of skin appendage density and specification; the disparate hair density between naked mole rats and mice does not predict the presence of rete ridges, suggesting that rete ridge formation may be a separate process from modulation of hair density (Fig. 2a–c, Extended Data Fig. 2b–f and Supplementary Fig. 1a–f). Therefore, interfollicular epidermis in mammals that form rete ridges is probably distinct from interfollicular epidermis in mammals that do not.

## Rete ridges are a distinct appendage

As a reduction in hair density does not spontaneously enable rete ridge formation or epidermal thickening, we proposed that rete ridges might form through a molecular mechanism different from the *LEF1–WNT*- and *EDA–EDAR*-mediated processes of hair follicles, sweat glands and fingerprint ridges. *LEF1* was highly expressed in the basal buds of E90 (GW12–13) pig skin long before rete ridge formation, but it was not expressed in the epidermis postnatally, when rete ridges are forming and maturing (Fig. 3a and Extended Data Fig. 3a). In addition, we did not observe any points in rete ridge formation that morphologically resembled basal buds of epithelial placodes (Figs. 1g, 3a and 4a).

Critically, the pig model enables investigation of the neonatal skin environment in greater detail during the critical window from P3 to P10, when skin transitions through rete ridge initiation towards maturation. Therefore, we performed scRNA-seq to capture the single-cell transcriptomes of epidermal and dermal cell lineages in pig skin at E90, P3, P10 and 6 months old (6 mo), and stereo-seq to capture the spatial context at P3, P10 and 6 mo (Supplementary Fig. 2a–d). All these datasets are publicly available and interactive at https://skinregeneration.org/papers/Thompson-et-al-2025/. We classified clusters from the resulting transcriptomics datasets into specific cell types on the basis of their expression of canonical markers (Methods and Supplementary Fig. 2a–d). To determine whether rete ridges formed through molecular mechanisms distinct from those of hair follicles and sweat glands, we integrated fetal and postnatal basal keratinocytes from E90, P3, P10 and 6 mo pig skin and examined whether basal cells involved in rete ridge formation or maturation transcriptionally overlapped with fetal basal bud cells (Fig. 3b and Supplementary Fig. 2e,f). In this analysis, epidermal placode cells, which comprised overwhelmingly E90 fetal keratinocytes, clustered distinctly from both non-dividing fetal basal cells and postnatal basal cells (Fig. 3b and Supplementary Fig. 2b,c,e,f).

We next examined the expression of shared markers for epidermal placode formation in hair follicles, sweat glands and volar fingerprint ridges. *LEF1* and *EDAR*, which are shared markers of placodes involved in the formation of all three appendages[3,10–12,15,17], were highly expressed in E90 basal bud cells forming sweat glands but only sparsely expressed in postnatal basal cells during rete ridge formation and maturation (Fig. 3c and Supplementary Figs. 2e,f and 3b). Postnatal basal cells do not express markers of the other appendages[3,12,13,15] during rete ridge formation and maturation, except for BMP ligands (Fig. 3c and Supplementary Figs. 2e,f and 3a,b). Postnatal basal cells involved in rete ridge formation highly expressed genes encoding BMP ligands, including *BMP7* and *BMP2*, which are not expressed outside the budding sweat glands in fetal skin[3,14] (Fig. 3c). Furthermore, genes encoding NOTCH ligands, including *JAG1* and *DLL1*, were highly expressed in postnatal basal cells but not in fetal cells (Fig. 3c). To investigate conservation of molecular patterns in human skin development, we reanalysed scRNA-seq datasets from fetal human volar skin[12], which forms sweat glands and fingerprint ridges but not hair follicles; neonatal human foreskin[35], which lacks hair follicles; and adult haired trunk skin[36] (Supplementary Fig. 2g–j). Fetal versus postnatal human skin expression patterns of *LEF1*, *EDAR* and *SOX9* resembled the transcriptional dynamics in fetal versus postnatal pig skin (Fig. 2c and Supplementary Fig. 3a–d).

We next aimed to confirm in vivo that rete ridge formation does not require placode-associated signals such as *LEF1*–WNT and *EDA–EDAR*, using both mouse and pig models. In mouse volar skin, sweat glands start to form from LEF1[+] epidermal placodes during fetal development and continue forming briefly after birth[3,14,15] (Supplementary Fig. 4a) Transverse ridges form along the proximal volar surface of mouse digits during fetal development, resembling human fingerprint ridges[12], but they do not form in the fingerpad skin at the digit tip, where we instead observed rete ridge formation postnatally (Supplementary Fig. 4a–c). As in porcine trunk skin, mouse fingerpad rete ridges

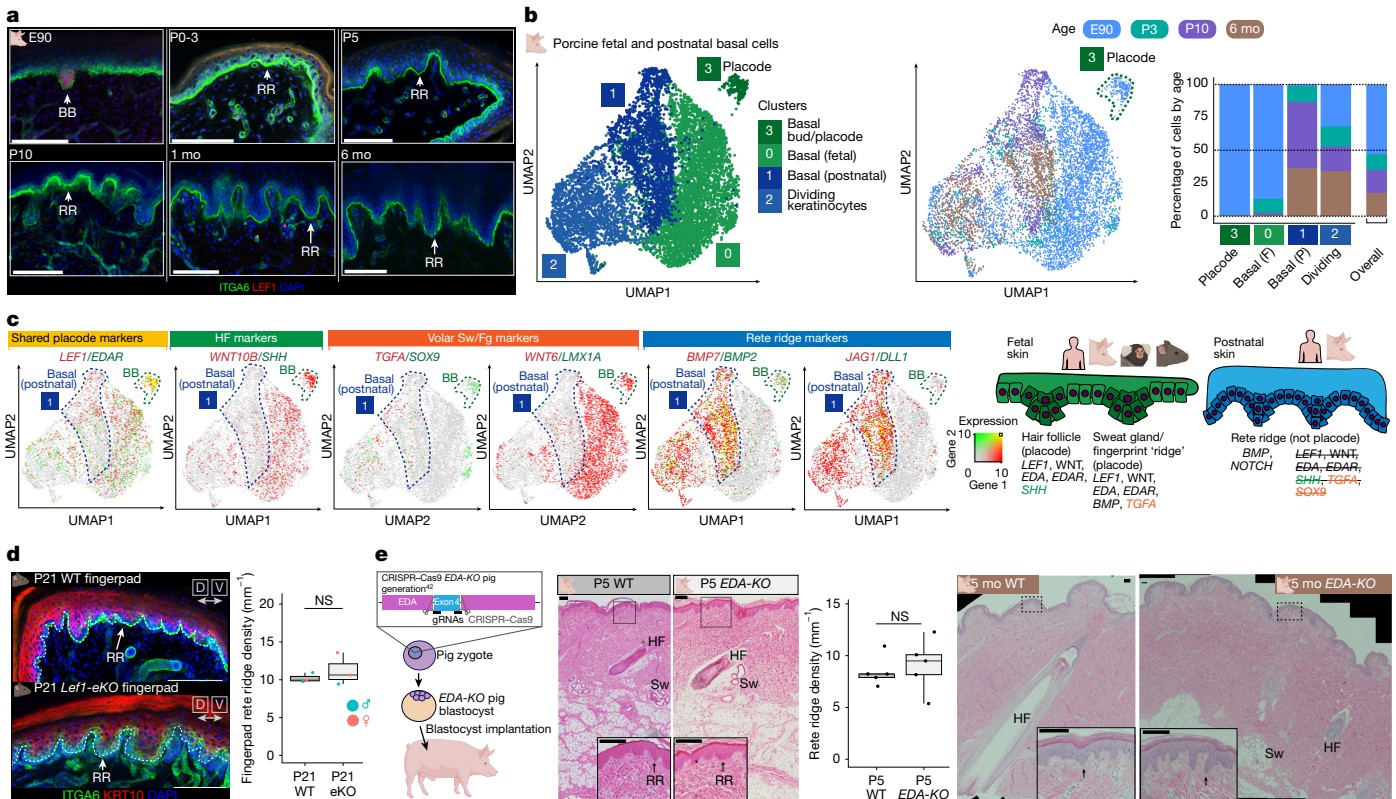

**Fig. 3 | Rete ridge formation does not require *LEF1*–*WNT* or *EDA*–*EDAR*.**
**a**, Rete ridges do not seem to form through LEF1⁺ placodes such as hair follicles and sweat glands. Representative immunostains for ITGA6 and LEF1 in E90 (*n* = 3), P0–3 (*n* = 3), P5 (*n* = 3), P10 (*n* = 2), 1 mo (*n* = 3) and 6 mo (*n* = 3) pig skin are shown. **b**, The epidermal placode transcriptional state is absent from postnatal skin that is forming rete ridges. Integration of E90, P3, P10 and 6 mo pig basal and dividing keratinocytes from scRNA-seq. Left, uniform manifold approximation and projection (UMAP) coloured by cluster. Middle, UMAP coloured by age. Right, bar plots representing the age contribution to each cluster and the overall age contribution to the integrated dataset. **c**, Rete ridge formation seems transcriptionally distinct from hair follicles, sweat glands and fingerprint ridges forming from epidermal placodes. Left, coexpression feature plots of the shared epidermal placode markers *LEF1* and *EDAR*, the hair follicle (HF) markers *WNT10B* and *SHH*, the sweat gland (Sw) or volar fingerprint ridge (Fg) markers *TGFA* and *SOX9*, and *WNT6* and *LMX1A*, and the postnatal rete ridge development and maturation markers *BMP7* and *BMP2*, and *JAG1* and *DLL1*. Expression of the left and right genes is indicated by red and green,

respectively, and coexpression is indicated by yellow. The dashed navy polygon denotes the postnatal basal cell state associated with rete ridge formation and the green polygon the fetal basal bud/placode cell state labelled in **b**. Right, summary graphics of expression in fetal skin versus postnatal skin of marker genes associated with hair follicles, sweat glands, fingerprint ridges and rete ridges. **d**, Epidermal LEF1 is not required for rete ridge formation in mouse fingerpads. Left, representative immunostains of P21 mouse fingerpads from WT and *Lef1-eKO* mice stained for ITGA6 and KRT10 (*n* = 3). Right, quantification of fingerpad rete ridges per millimetre for WT (*n* = 3) and *Lef1-eKO* (eKO, *n* = 3) mouse fingerpads. *P* = 0.5075 from *t*-test. **e**, EDA signalling is not required for rete ridge formation and maturation in porcine skin. Left, schematic of *EDA-KO* pig generation[42]. Middle, representative H&E stains of P5 WT (*n* = 5) and *EDA-KO* (*n* = 5) pig skin and quantification of rete ridge density per millimetre. Each set represents five littermates. Right, representative H&E stains of age-matched 5 mo WT (*n* = 2) and *EDA-KO* (*n* = 2) pigs. *P* = 0.6555 from *t*-test. NS, not significant. Scale bars, 100 μm. Illustrations in **a**–**d** were created using BioRender. Thompson, S. (2026) https://BioRender.com/8rd8cz9.

formed after the completion of sweat gland morphogenesis, when volar epidermis lacked LEF1 expression (Supplementary Fig. 4b). We confirmed the independence of fingerpad rete ridge formation from *LEF1*–WNT signalling using *Lef1-eKO* mice, which exhibited impaired placode development (Fig. 2f, Extended Data Fig. 2i and Supplementary Fig. 1a). Consistent with the hypothesis that rete ridge formation does not require *LEF1*–WNT signalling to form, rete ridges in the fingerpads of juvenile mice were not affected by epidermal ablation of *Lef1* (Fig. 3d and Supplementary Fig. 4c). Next, we used a previously published *EDA-KO* pig model[42] to further validate the independence of rete ridge formation from placode-associated signals during both the critical neonatal window and into adulthood (Fig. 3e). Neonatal *EDA-KO* piglets did not have altered rete ridge formation compared with wild-type (WT) animals, and rete ridges also seemed to mature normally (Fig. 3e). We next tested whether neonatal pig skin could regenerate rete ridges following wounding. Single large square dorsal wounds on neonatal piglets healed to regenerate rete ridges and reform vascularized dermal pockets, suggesting that the neonatal signalling environment has an intrinsic potential to form and reform the epidermal rete

ridge and dermal pocket niches (Supplementary Fig. 5a–g). As neither *LEF1*–WNT nor *EDA*–*EDAR* signalling were required for rete ridge formation in mice and pigs, we concluded that rete ridges must instead form through a distinct mechanism from other placodes (Fig. 4a), and that rete ridges probably represent a distinct type of cutaneous appendage from hair follicles, sweat glands and volar fingerprint ridges.

## Cellular mechanisms of rete ridge formation

Proliferative patterning is a widely conserved element in epidermal appendage initiation and elongation, including in hair follicles, sweat glands and fingerprint ridges[12–14,32]. Basal bud formation for these appendages is prepatterned by clustering of proliferating basal epidermal cells, followed by sustained proliferation along the appendage during downgrowth and elongation[12–14,32] (Supplementary Fig. 6a). However, proliferative patterning of the interfollicular epidermis during rete ridge formation is poorly understood.

We investigated MKI67⁺ cell distribution in postnatal porcine skin and observed divergent spatiotemporal dynamics compared with those of

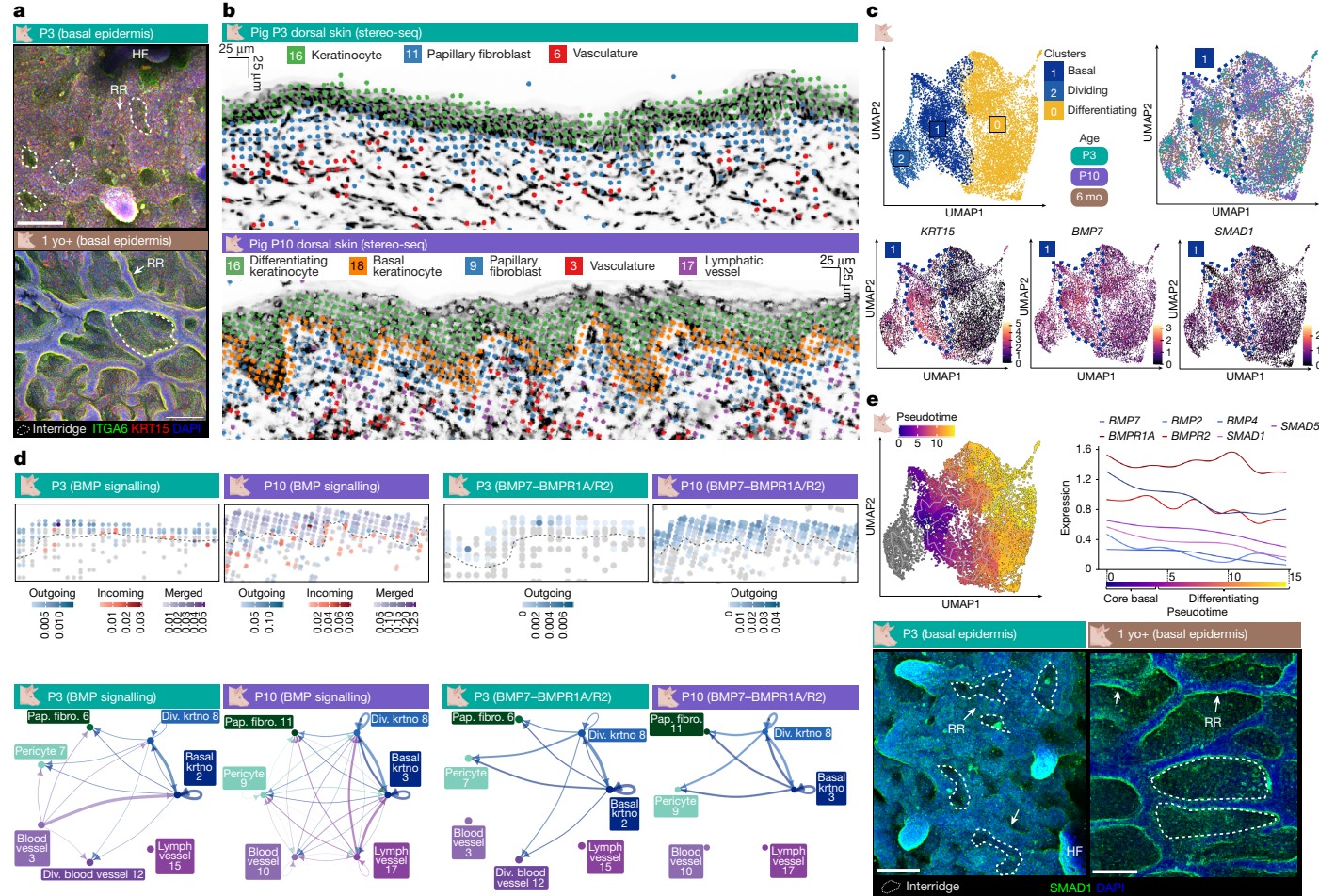

**Fig. 4 | Epidermal BMP signalling activates during rete ridge formation and maturation. a**, Representative immunostains from the basal side of epidermal whole mounts from P3 (*n* = 3) and adult (*n* = 2) pig skin stained for ITGA6 and KRT15, showing the topography of rete ridge formation. **b**, Rete ridge formation establishes an epidermal–dermal signalling niche within the dermal pocket. Representative frame of epidermal–dermal regions from porcine P3 and P10 stereo-seq, visualizing Leiden clusters comprising epidermal and dermal cell lineages, is shown. **c**, BMP signalling is activated in basal cells during rete ridge formation and maturation. UMAPs depicting integrated postnatal interfollicular epidermis keratinocytes from P3, P10 and 6 mo pig scRNA-seq, coloured by cluster (top left) or age (top right) and with visualization of expression of *KRT15*, *BMP7* and *SMAD1* (bottom). The dashed navy polygon outlines the basal cluster 1. **d**, Predicted epidermal–dermal BMP signalling interactions between epidermal keratinocytes, papillary fibroblasts (Pap. fibro.) and vascular/pericyte clusters during rete ridge formation, represented spatially at P3 and

P10 from stereo-seq Spatial CellChat (top row) and as CirclePlots for predicted signalling interactions from P3 and P10 scRNA-seq CellChat (bottom row). The dashed line approximates the epidermal–dermal junction traced from the stereo-seq Leiden-clustered image mask (Supplementary Fig. 8a). **e**, Epidermal BMP signalling activates during rete ridge formation. UMAP depicting pseudotime trajectory from non-dividing basal to differentiated keratinocyte states from integrated P3, P10 and 6 mo pig interfollicular epidermis scRNA-seq, and pseudotime trajectories for expression of BMP signalling ligands, receptors and downstream elements visualized as a LinePlot (middle). Representative immunostains of P3 (*n* = 3) and adult (*n* = 2) pig skin epidermal whole mounts for SMAD1 (bottom). Dashed polygons indicate inter-ridge domains. Div., dividing; IFE, interfollicular epidermis; krtno, keratinocyte; pap. fibro., papillary fibroblasts. Scale bars, 100 μm (**a**, **e**), 25 μm (**b**). Illustrations in **a**–**e** were created using BioRender. Thompson, S. (2026) https://BioRender.com/8rd8cz9.

other cutaneous appendages (Supplementary Fig. 6a,b). At the time of initiation, rete ridges exhibited MKI67⁺ cell distribution across both the rete ridge and inter-ridge domains (Supplementary Fig. 6a–d). At P3, the basal inter-ridge compartment became more proliferative than the basal rete ridge compartment, yet by P5 this pattern flipped, potentially to support early rete ridge thickening (Supplementary Fig. 6a–d). During the transition from development to maturation, proliferation became sporadic in both the rete ridge and inter-ridge domains (Supplementary Fig. 6b–d).

Moreover, by P5, suprabasal proliferation was significantly elevated within the rete ridge domain, suggesting patterned regulation of epidermal proliferation and differentiation within the developing rete ridge (Supplementary Fig. 6b–d). We examined patterns of epidermal differentiation during rete ridge formation by tracking vertical cell movements of BrdU-labelled cells in neonatal pig skin (Supplementary

Fig. 6e,f). BrdU⁺ basal keratinocytes in developing rete ridge and inter-ridge domains showed similar vertical progression, suggestive of both domains functioning similarly in steady-state epidermal maintenance[43] (Supplementary Fig. 6f–i). Therefore, we conclude that spatially patterned proliferation and differentiation support the development and subsequent maintenance of the patterned rete ridge epidermal architecture (Supplementary Fig. 6i).

## Signalling activation in rete ridge formation

Porcine interfollicular epidermis showed greater complexity in stratification over the course of rete ridge formation, with an expansion of KRT10⁺ cell states within the suprabasal compartment of the rete ridges (Supplementary Fig. 7a). By contrast, the interfollicular epidermis of non-rete-ridge-bearing skin more closely resembled that of fetal pig

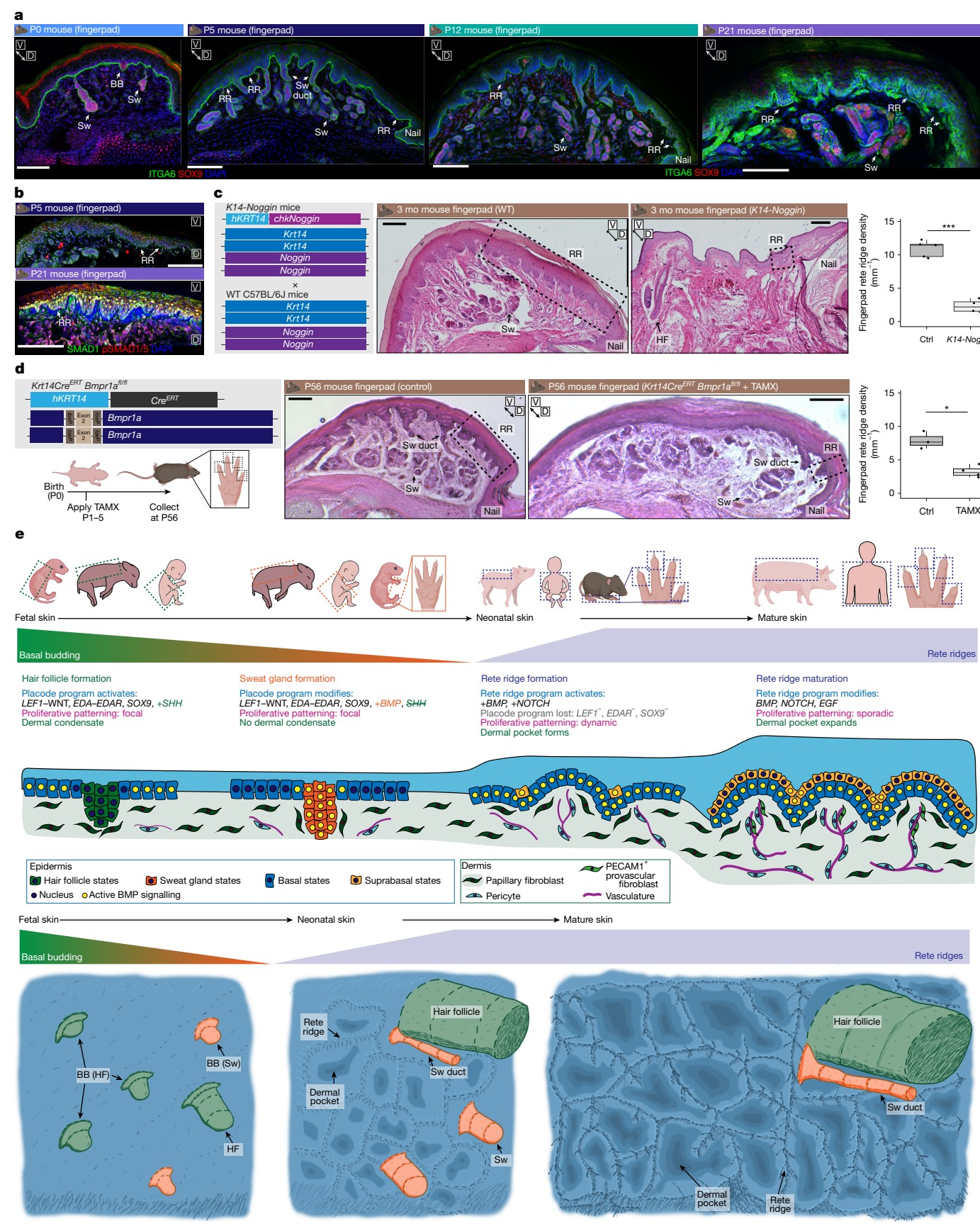

**Fig. 5 | See next page for caption.**

**Fig. 5 | Epidermal BMP signalling is required for rete ridge formation.**
**a**, Fingerpad rete ridges form postnatally after the cessation of sweat gland formation in mice. Representative immunostains of fingerpads from P0, P5, P12 and P21 mice stained for ITGA6 and SOX9 (*n* = 3 for each time point) are shown. **b**, BMP signalling is active in the fingerpad. Representative immunostains of P5 (*n* = 3) and P21 (*n* = 3) fingerpads stained for SMAD1 and phosphorylated SMAD1/5 (pSMAD1/5) are shown. **c**, Epidermal BMP signalling is required for rete ridge formation in mouse fingerpads. Schematic of *K14-Noggin* mouse (left), representative H&E stains of WT and *K14-Noggin* fingerpads (centre) and quantification of fingerpad rete ridges per millimetre (right). *P* = 7.78 × 10⁻⁶ from *t*-test. D and V indicate dorsal and ventral orientation of the digit section.

Zoom-outs of these representative images are in Extended Data Fig. 5h. **d**, Inhibition of epidermal BMP signalling via postnatal *Bmpr1a* KO inhibits rete ridge formation in mouse fingerpads. Schematic of *K14-Cre^ERT*;*Bmpr1a^fl/fl* mouse (left), representative H&E stains of tamoxifen-treated *K14-Cre^ERT* (Ctrl, *n* = 3) and *K14-Cre^ERT*;*Bmpr1a^fl/fl* (TAMX, *n* = 4) fingerpads (centre) and quantification of fingerpad rete ridges per millimetre (right). *P* = 0.01039 from *t*-test. Zoom-outs of these representative images are in Extended Data Fig. 5i. **e**, Proposed models of rete ridge formation in skin represented temporally in two dimensions (top) and morphologically in 2.5 dimensions (bottom). Scale bars, 100 μm. **P* < 0.05, ****P* < 0.001. Illustrations in **a**–**e** were created using BioRender. Thompson, S. (2026) https://BioRender.com/8rd8cz9.

---

skin, with a flatter layering between basal and KRT10⁺ differentiating layers (Supplementary Fig. 7a). Therefore, we examined the spatiotemporal patterning of the basal epidermis during rete ridge formation to identify putative molecular signals that may underly their formation. Spatial transcriptomics showed that rete ridge and inter-ridge basal cells do not form separate clusters (Fig. 4b and Supplementary Fig. 7b). Combining these findings with integrative scRNA-seq analysis of postnatal porcine P3, P10 and 6 mo interfollicular epidermal keratinocytes, we identified a core undifferentiated and non-dividing basal cell state defined by high expression of *KRT15*, NOTCH signalling ligands, BMP signalling ligands and canonical basal markers (Fig. 4a–c and Supplementary Fig. 7b–d). This core basal cell population also expressed signalling ligands associated with recruitment of diverse dermal cells, suggesting that the basal epidermis may engage in signalling to recruit and maintain dermal pocket cells during rete ridge formation and maturation (Supplementary Fig. 7e). The human basal epidermis expressed these markers similarly to that of pigs, suggesting that there is a shared basal cell state between rete ridge and inter-ridge domains in humans and pigs (Fig. 4a–c and Supplementary Fig. 7c–f). These results indicate that morphological patterning of rete ridges may be facilitated by regional signalling nuances across the same basal cell state instead of spatial clustering of two or more distinct basal cell states.

Next, to infer specific signalling interactions potentially involved in rete ridge formation, we examined cell types adjacent to the epidermis between P3 and P10 using stereo-seq. This spatiotemporal approach revealed expansion of dermal cell lineages into the growing dermal pockets between P3 and P10, suggesting distinct epidermal–dermal interactions at the dermal pocket domains (Fig. 4b and Supplementary Fig. 8a). Next, we inferred cell–cell communication with Spatial CellChat on stereo-seq datasets and CellChat[44] on scRNA-seq datasets, focusing on the cross-talk between basal keratinocytes, papillary dermal fibroblasts, pericytes, and blood vessel and lymphatic vessel endothelial cells (Fig. 4b and Supplementary Fig. 8a–c). We observed generally heightened signalling activity in the epidermis and papillary dermis compared with the reticular dermis, suggesting that rete ridges and dermal pockets function as active signalling centres (Supplementary Fig. 8a–c). Further, we identified large-scale signalling changes between fetal and postnatal skin, including postnatal activation of epidermal BMP and NOTCH signalling (Fig. 4d, Extended Data Fig. 4a,b, and Supplementary Figs. 8c,d and 9a,b). BMP signalling was broadly active in postnatal interfollicular epidermis throughout rete ridge formation and maturation, with high expression of genes encoding BMP ligands such as *BMP7*, downstream elements, such as *SMAD1* and *SMAD5*, and BMP receptors, including *BMPR1A* and *BMPR2* (Fig. 4c–e and Extended Data Fig. 4a,b). The rete ridge and inter-ridge regions both expressed SMAD1 in vivo, although the rete ridge regions exhibited higher BMP signalling than the inter-ridge regions (Fig. 4e). Although BMP signalling was most active within the epidermis, further BMP interactions were predicted between epidermal and vascular cell lineages (Fig. 4d, Extended Data Fig. 4a,b and Supplementary Fig. 8c,d). NOTCH signalling is a canonical regulator of epidermal differentiation, and *JAG1* was broadly expressed in adult pig and human basal epidermis, suggesting that NOTCH signalling may contribute to regulation of proliferation and

differentiation within the rete ridge compartment[45] (Supplementary Figs. 8d,e and 9a,b).

Neonatal dermal cells were predicted to interact with the overlying epidermis through pathways including FGF, supporting the notion of bidirectional epidermal–dermal signalling activity during rete ridge formation (Supplementary Figs. 8b–d, 9c,d and 10a–c). From P3 to P10, further epidermal and dermal signalling interactions, such as TGFβ and EGF signalling, activated and were maintained into adulthood (Supplementary Figs. 8c,d and 10a–c), possibly supporting the transition towards rete ridge maturation[46,47]. Rete ridge formation and maturation are also characterized by activity of the PDGF, VEGF and ANGPTL signalling pathways, which are associated with dermal fibroblast and vascular recruitment and maturation[48–50] (Supplementary Figs. 7e, 8c,d and 11a–d). Postnatal recruitment of dermal cell lineages is likely to support epidermal thickening, on the basis of previous in vitro and mouse in vivo experiments[38,48,49,51]. We observed similar expression patterns of rete ridge formation and maturation-associated signalling ligands in human skin scRNA-seq, suggesting that these represent conserved rete ridge signalling activities (Supplementary Figs. 7f and 8e). In addition, we identified the emergence of a distinct PECAM1⁺ fibroblast state in the maturing dermal pocket, which may further encourage vascular recruitment and maintenance during rete ridge maturation (Supplementary Figs. 8c,d and 12a–d). The cellular composition of the vascularized dermal pocket was similar in other species and body regions that contained rete ridges, such as human and dolphin trunk skin, and in the fingerpads and oral mucosa of mice (Supplementary Fig. 13a–d). By contrast, the rete-ridge-less trunk skin of marmosets and mice did not morphologically resemble the dermal pocket (Supplementary Fig. 13e).

Critically, the spatial distribution of epidermal signalling during rete ridge initiation and maturation was broadly basal, rather than compartmentalized into binarized spatial domains (Fig. 4b–e, Extended Data Fig. 4a,b and Supplementary Figs. 7b–e, 9a-b and 11a–d). Thus, the neonatal basal epidermis seems to be broadly supportive of rete ridge formation and dermal recruitment, whereas rete ridge initiation patterning may be cued by nuanced gradients at the protein level or local proximity to differential distribution of underlying dermal fibroblasts and vasculature (Fig. 4b–e, Extended Data Fig. 4a,b, and Supplementary Figs. 9a,b, 10a–c, 11a–d and 12a–d). Overall, these transcriptomic and in vivo observations highlight the existence of distinct postnatal epidermal and dermal signalling activities during rete ridge formation (Fig. 4b–e, Extended Data Fig. 4a,b, and Supplementary Figs. 8c,d, 9a,b, 10a–c, 11a–d and 13a–d). Concurrently, postnatal skin inactivated the fetal signalling programs associated with the formation of placodes for other discrete appendages, such as hair follicles and sweat glands (Figs. 3a–e and 4d,e and Supplementary Fig. 8c–e).

## Epidermal BMP signalling is required

As we had observed that postnatal mouse fingerpads had structures closely resembling rete ridges of humans and pigs, complete with vascularized dermal pockets (Extended Data Fig. 5a and Supplementary Fig. 13d), we investigated mouse fingerpads to define

the developmental timing of their rete ridge formation. As SOX9 is expressed in basal buds, hair follicles[52] and sweat glands[13] but not in rete ridges, we used SOX9 to spatiotemporally resolve when sweat gland formation terminates and rete ridge formation begins (Figs. 3c and 5a and Supplementary Figs. 2b,c and 3a–d). SOX9 effectively labels ductal and secretory components of volar sweat glands and is absent from the intergland epidermis at P5, in parallel with loss of epidermal LEF1 (refs. 13–15; Fig. 5a and Supplementary Fig. 3a,b). We confirmed that mouse fingerpad rete ridges and dermal pockets start to form postnatally after the cessation of sweat gland formation (Fig. 5a, Extended Data Fig. 5a and Supplementary Fig. 4a,b).

Critically, fingerpad rete ridges showed conserved patterning of basal markers such as KRT15/14 and PDGFC, as porcine and human rete ridges do (Fig. 4a, Extended Data Fig. 5b and Supplementary Fig. 7a,c–f). Curiously, mouse trunk skin did not express *Pdgfc*, suggesting that epidermal PDGFC may be a conserved element in rete-ridge-capable skin (Extended Data Fig. 5c). Another signalling difference involved the sweat-gland-regulating transcription factor EN1 (refs. 3,17,19), as human fetal volar epidermis expressed *EN1* but postnatal human foreskin and porcine trunk epidermis did not (Extended Data Fig. 5d). BMP and NOTCH signalling ligands followed similar temporal expression patterns in fetal versus postnatal human and porcine epidermis (Fig. 3c, Extended Data Fig. 5e and Supplementary Fig. 8c–e).

BMP signalling has long been implicated in regulation of cutaneous appendage fate selection and developmental patterning between different species and in different body regions[3,4,12,14,16]. As we had observed BMP activation during porcine rete ridge formation (Fig. 4c–e and Extended Data Fig. 4a,b), we next examined BMP signalling in the postnatal mouse fingerpad. BMP signalling, as indicated by SMAD1/5 phosphorylation, was broadly active in the fingerpad basal epidermis throughout rete ridge formation and, notably, also active within the suprabasal compartment of rete ridges (Fig. 5b). Expression of endogenous BMP signalling antagonists, such as Noggin, is temporally restricted and inactive in skin when sweat glands or rete ridges are forming in both humans and pigs, suggesting that dynamic regulation of BMP signalling may influence the formation of different epithelial appendages[3,4,14,16] (Extended Data Fig. 5f,g).

As epidermal BMP signalling activity during rete ridge formation seemed to be consistent across several species, we functionally tested its role in mouse fingerpads. First, we used *K14-Noggin* mice to overexpress the BMP antagonist Noggin in the epidermis[4] and observed a significant reduction in both rete ridge and sweat gland density in the fingerpad (Fig. 5c and Extended Data Fig. 5h). We also observed conversion of some volar sweat glands to hair follicles (Fig. 5c and Extended Data Fig. 5h), consistent with previous studies that have implicated BMP signalling inhibition through Noggin with supporting hair follicle versus sweat gland fate selection during development[3,4]. Second, we used tamoxifen-inducible *K14-Cre^ERT;Bmpr1a^fl/fl* mice[53] to inhibit postnatal epidermal BMP signalling through deletion of *Bmpr1a*, a key receptor that has been implicated in BMP signalling during rete ridge formation (Figs. 4d,e and 5d and Extended Data Fig. 4a,b). Using this inducible system, we ablated epidermal *Bmpr1a* before the onset of fingerpad rete ridge formation (Fig. 5d) and observed a significant reduction in rete ridge formation (Fig. 5d and Extended Data Fig. 5i). Therefore, we conclude that rete ridge formation requires epidermal BMP signalling activity (Fig. 5e). Collectively, these results implicate BMP activation alongside inactivation of *LEF1*–WNT- and *EDA−EDAR*-mediated processes as a crucial evolutionarily conserved developmental milestone that controls appendage type specification and enables non-furry skin to develop rete ridges (Fig. 5e).

## Discussion

Our results demonstrate that rete ridges form postnatally in several species as a distinct epidermal appendage, in contrast to hair follicles, sweat glands and fingerprint ridges, which form during early or late embryogenesis[3,7,11,12,14,15,29]. We suggest that hair density and rete ridge formation are uncoupled processes, as mutations that ablate or reduce hair density do not cause spontaneous formation of rete ridges, for instance, in Chinese crested dogs or our *Lef1-eKO* mice[9,19,22]. In addition, mutations that increase hair growth, as in human hypertrichosis, do not ablate rete ridge formation[2,5,54]. Mammals with rete ridges generally have thicker epidermis and lower hair density than animals without rete ridges, on the basis of our histological zoo. Thicker epidermis may provide defensive and environmental advantages for the exposed skin surface of species that have reduced hair coverage, including humans, pigs and especially cetaceans[25,39,55]. Critically, rete ridge formation occurs through cellular and molecular mechanisms distinct from those of other epidermal appendages[3,4,10,12,14–16,31]. We found that rete ridge morphogenesis requires broad epidermal BMP signalling, whereas *LEF1*–WNT and *EDA−EDAR* signalling seem to be dispensable. Future studies should aim to understand how rete ridge formation is influenced by the balance of Turing reaction-diffusion patterning and expansion-induction patterning mechanisms[16,31,33]. Rete ridge acquisition seems to be driven not by genetic distinctions between species but rather by the convergence of distinct cellular and molecular characteristics. Consequently, rete ridges appear as an interconnected epidermal appendage acquired de novo over the course of evolution *pari passu* with reductions in hair density.

We also demonstrate that rete ridges possess their own epidermal and dermal niches in human and porcine skin, as well as in mouse volar fingerpads. The 'dermal pocket' is a potential rete ridge niche that may enable dermal cell lineages to assemble beneath the basal epithelium and establish a signalling source that assists with epidermal thickening[8,38,48,49,51]. Furthermore, understanding and implementing the mechanisms of rete ridge formation and maintenance will be important to promote healthy tissue during ageing and wounding, and in diseases in which rete ridges are critically understudied[8,18,21,34,38,56–59]. Future studies will be needed to understand how different dermal cell types contribute to rete ridge formation and patterning compared with other epidermal appendages[3,12,15–17,31,33]. Clinically, the potential contribution of the underlying dermal vasculature to rete ridge patterning can be observed in human infantile haemangiomas, benign skin tumours that show profound dermal hypervascularization yet completely lack rete ridges[57,58]. In conclusion, we have established porcine and mouse fingerpad models that could be used to inform future studies to directly address human-relevant tissue biology and disease. We propose that rete ridges function as a large-scale, interconnected appendage[8,37] that adds "structural and functional complexities to the otherwise flat epithelia"[60].

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

**UW Birth Defects Research Laboratory**

**Ian A. Glass[13]**

## Methods

### Tissue sample collection, preservation and processing

Back skin samples from age-matched adults and one litter of neonatal naked mole rats (*Heterocephalus glaber*) were maintained at the University of Texas Health Science Center at San Antonio for unrelated studies performed under protocols approved by the University of Texas Health Science Center at San Antonio Institutional Animal Care and Use Committee (IACUC; 20210034AR). Rhesus macaques (*Macaca mulatta*) were maintained at the Oregon National Primate Research Center (ONPRC) at Oregon Health and Science University for unrelated studies performed under protocols approved by Oregon Health and Science University IACUC (IP03716, IP03276, IP00367). The ONPRC is accredited by the Association for Assessment and Accreditation of Laboratory Animal Care (Animal Welfare Assurance D16-00195) and registered with the USDA (92-R-001). Rhesus macaque back skin samples were shared through the ONPRC tissue distribution programme. Common marmosets (*Callithrix jacchus*) were maintained at the Southwest National Primate Research Center at Texas Biomedical Research Institute for unrelated studies performed under an approved animal use protocol (assurance number D16-00048). Back skin was recovered from the above species at necropsy after euthanization for unrelated studies. Skin samples were fixed in 4% paraformaldehyde (PFA) overnight and stored in 70% EtOH until tissue processing for paraffin embedding or fixed in 4% PFA, shipped to Washington State University in 1× phosphate-buffered saline (PBS) on ice, and embedded in OCT (Fisher) before cryopreservation at −80 °C. Biological replicates were compiled from samples collected across different litters unless otherwise specified. Histological analyses of bottlenose dolphin (*Tursiops truncatus*), long-beaked common dolphin (*Delphinus capensis*) and short-beaked common dolphin (*Delphinus delphis*) trunk skin were performed using trunk skin samples obtained by the Plikus laboratory at the University of California, Irvine, from the NOAA (Southwest Fisheries Science Center, La Jolla, California) under the destructive loan permit. The analysed specimens included: *D. capensis* (numbers KXD0225, KXD0226, 1741-2023 Dc2301B), *D. delphis* (numbers BLH0012, KXD0357, 585-2022 Dd2202B) and *T. truncatus* (numbers KXD0410, KZP0069, 812-2022 Tt2202B). North American grizzly bears (*Ursus arctos horribilis*), a mix of males and females ranging in age from to 8 to 21 years in the summer of 2023, were housed at the Washington State University Bear Research, Education, and Conservation Center. North American grizzly bears were anaesthetized, and biopsies were collected from the dorsal skin of the rump (lower back) for unrelated studies related to subcutaneous fat performed under Washington State University IACUC-approved protocols (6546). Skin samples used in this study were the whole 6-mm diameters of discarded skin from the biopsies, which were used to sample subcutaneous fat. Representative histology in Fig. 2a and Extended Data Fig. 2b is from a 21-year-old male (haematoxylin and eosin; H&E) and female (Herovici), which were both born in the wild. Mice (*Mus musculus*) used in this study were of a mixed WT C57BL/6 background housed at Washington State University in a 12-h light/dark cycle with food and water ad libitum in a climate-controlled facility set to approximately 68–73 °F and 40% humidity. *K14-Cre;Lef1^{fl/fl}^* (*Lef1-eKO*) mice were generated by crossing Tg(KRT14-cre)1Amc/J (Jackson Laboratory, 004782)[61] with B6.Cg-Lef1tm1Hhx/J (Jackson Laboratory, 030908)[62]. Housing and sample collection was conducted in accordance with Washington State University IACUC-approved protocols (6723, 6724). Paraffin-embedded 3mo *K14-Noggin* and C57BL/6 mouse digits analysed in this study were archived from a previous study[4] and shared with the Driskell laboratory upon request. *K14-Cre^{ERT}^;Bmpr1a^{fl/fl}^* and *K14-Cre^{ERT}^* mice[53,63,64] used in this study were housed at the University of Warsaw. The animal studies were approved by the First Local Ethics Committee: no. 971/2020 as of 28 January 2020, no. 1669/2025 as of 18 March 2025. The studies were conducted in accordance with local legislation and institutional requirements. *K14-Cre^{ERT}^;Bmpr1a^{fl/fl}^* and *K14-Cre^{ERT}^* mice were all treated with tamoxifen (12.5 mg ml$^{-1}$ in 10% EtOH) topically to the paws from P1 to P5 and then aged to P56. *K14-Cre^{ERT}^;Bmpr1a^{fl/fl}^* were labelled TAMX (tamoxifen-induced knockout group), and *K14-Cre^{ERT}^* mice were labelled Ctrl (control, tamoxifen-induced without gene knockout) in Fig. 5d and Extended Data Fig. 5i. Collected digits were fixed for 24 h in 4% PFA at 4 °C, then in 0.5 M EDTA (pH 7) for 6 days at room temperature (to soften the bone and nail), before being moved to 30% sucrose at 4 °C and frozen in OCT. Gestational human tissue samples ranging from GW7 to GW20 were obtained by the Birth Defects Research Laboratory at University of Washington under University of Washington institutional review board (IRB)-approved protocols with maternal written consent (University of Washington STUDY00000380; received under Washington State University 19680). Fetal tissues were obtained surgically, and although there is a possibility of samples being damaged by this process, we did not observe damage in the samples analysed in this study. Adult human tissue samples were obtained by Advanced Dermatology in Spokane, Washington, as surgical discard tissue after informed consent and in accordance with IRB-approved protocols (Washington State University 19796). All human tissue samples were deidentified before receipt at Washington State University and analysed in accordance with Washington State University IRB-approved protocols, as specified above. As they were surgical discards, samples came from a variety of anatomical sites, including the trunk skin of the torso and the skin of the face or head. Human tissue samples were processed for paraffin embedding as described above except for the adult samples, which were fixed overnight in 10% neutral buffered formalin (Fisher) instead of PFA. Pigs (*Sus scrofa*) were housed at Washington State University under approved Washington State University IACUC and USDA protocols in a climate-controlled facility ranging from 70 °F to 80 °F and 30% to 50% humidity. Housing included a 12-h light/dark cycle with regular feeding and ad libitum water. Some fetal pigs were obtained from Biology Products; their age was estimated on the basis of length, and they were exempt from Washington State University IACUC approval. Embryonic day 90 fetal pigs were collected postmortem from a pregnant pig with known date of conception under WSU IACUC-approved protocols (6492). Skin samples from all fetal pigs were collected from the upper back along the dorsal midline. Postnatal pig samples were collected postmortem from pigs either housed at Washington State University, obtained from local farmers, or postmortem from local butchers in accordance with Washington State University IACUC-approved protocols (6492). Skin samples from all postnatal pigs were collected from the upper back along the dorsal midline from multiple litters owing to limited litter sizes and difficulty in obtaining animals. Fetal pigs obtained from Biology Products were generally unpigmented and of unknown background and multiple litters. One litter of known embryonic day 90 pigs (used for histology, immunostaining and scRNA-seq) was collected from an unpigmented pregnant sow raised on a local farm of mixed Yorkshire and Red Duroc background. Postnatal pigs obtained from local farmers or butchers were of mixed backgrounds involving predominantly Yorkshire, Red Duroc and Hampshire breeds, with occasional regional interbreeding with Idaho Pasture pigs and Kunekunes, which have pigmented skin. Notably, Hampshire breeds are known to exhibit banded pigmentation in their skin. As such, most skin we studied was unpigmented, but some piglets or adults had pigmented skin or spots (as visible in Fig. 2b and Supplementary Fig. 5c). Every effort was made to collect skin samples from both males and females across all time points. Adult male skin (6 mo/7 mo) collected from local farmers and butchers was from animals presumably castrated before weaning to avoid boar taint in the meat. Owing to the opportunistic nature of these collections, it is unknown whether postbutchering biological replicates were derived from the same or different litters. Ages were approximated. Postnatal Yucatan miniature hairless pig and Hanford miniature pig skin samples

collected from the upper back along the dorsal midline were received from Sinclair Biosciences, and Mangalitsa pigs were raised on farms, with samples collected from the upper back along the dorsal midline following butchering, and were exempt from Washington State University IACUC approval. *EDA-KO* pig[42] samples were collected from the backs of one litter of pigs at P5 or age-matched at 5 mo by the Welsh and Ostedgaard laboratories at the University of Iowa under University of Iowa IACUC-approved protocols (3071121) and shared upon request. All experiments followed relevant guidelines and regulations of the appropriate ethics committees, as detailed where relevant. A visual summary of tissue sample collection sites can be found in Extended Data Fig. 2a. For all animal tissue sample collections, an individual organism was considered to be one biological replicate. Multiple samples may have been collected from one biological replicate, but these were used only as technical replicates. Unless otherwise specified, tissue samples were fixed in 4% PFA overnight and processed for paraffin embedding as described above, or fixed in 4% PFA, washed in PBS and then embedded in OCT (Fisher) before cryopreservation at −80 °C. Epidermal whole mounts were collected using a process similar to that described in previous studies[65].

### Histological analysis
Paraffin-embedded tissue samples were sectioned at 5 μm (*Lef1-eKO* back skin) or 10 μm (all others) and stained either with H&E according to standard protocols or using Herovici's polychrome in a process adapted from previously published protocols[66]. Coverslips for H&E-stained tissue were mounted using Permount Mounting Medium (Fisher), and those for Herovici-stained tissue were mounted using DPX (Sigma). Slides were bright-field imaged using a Nikon Eclipse E600 fluorescence microscope equipped with a Nikon DS-Fi3 colour camera. Dolphin cryo samples were sectioned at 10 μm, fixed for 30 min at room temperature, washed 3 times with PBS, and rinsed in tap water for several minutes before staining with H&E or Herovici's polychrome, as above. Dolphin H&E and Herovici slide coverslips were mounted in DPX and imaged using a Keyence BZ-X810 wide-field microscope. *K14-Cre^{ERT}* experiment digit samples were sectioned at 13–15 μm using a Leica cryostat and H&E stained using standard protocols.

### Hair density assessment
The surface of skin samples was imaged using an AmScope dissecting microscope with an AmScope colour camera before further processing for paraffin embedding or cryopreservation. Multiple distinct images were captured per biological replicate and treated as technical replicates when skin samples were of sufficient size or quantity. Owing to the surgical discard nature of adult human samples collected in this study, many samples were not large enough to allow accurate assessment of hair density or were collected before the beginning of the hair density assessment, hence the smaller sample size in Fig. 2d (*n* = 8) compared with Fig. 2c. Hair density was also quantified from samples across more anatomical sites, including trunk skin (*n* = 4), face skin (*n* = 2), scalp skin (*n* = 1), and from the base of the nose, which we considered separate from face skin (*n* = 1).

### Porcine wound healing
One litter of neonatal pigs was housed with their mother at Washington State University, were anaesthetized, and received 2.5 × 2.5-cm-square full-thickness wounds under aseptic conditions in accordance with WSU IACUC-approved protocols (6492). Following surgery, piglets were returned to their mother, and the wound site was periodically imaged to assess the size of the wound across the surgery cohort of seven littermates of mixed sexes. Twenty-eight days postwounding (28 dpw), two littermates were euthanized, and the wound site was collected for histological analysis, followed by two more at 43 dpw, and the final three at 58 dpw. Collected wounds were fixed in 4% PFA for several hours, washed in PBS, then embedded and frozen in OCT at −80 °C.

### Porcine BrdU labelling
One litter of neonatal pigs was housed with their mother at Washington State University and injected intraperitoneally twice daily for 3 days from P5 to P7 with 50 mg kg$^{-1}$ of BrdU (AdipoGen Life Sciences, CDX-B0301-G005) dissolved in sterile saline, in strict compliance with WSU IACUC-approved protocols (6492). One piglet was injected with sterile saline only and was considered a negative control. Three BrdU-injected piglets were euthanized, and tissue was collected from the upper back along the dorsal midline for histological analyses at each of three time points: P8 (1 day postinjection, 1 dpi), P12 (5 dpi) and P16 (9 dpi), along with one negative control individual at P16.

### Immunofluorescence analysis of cryo-preserved tissues
For immunostaining of cryo-preserved tissues, frozen tissue samples were sectioned at 60 μm in a Leica cryostat, and staining was performed as described previously[67]. Tissues were stained using the following primary antibodies: Human-ITGA6 rat (1:200, BD Biosciences, catalogue no. 555735, clone GoH3, lot no. 8136528, 3055226, 1214287, 1033227), human LEF1 rabbit (1:200, Cell Signaling, catalogue no. 2230S, clone C12A5, lot no. 8, 9), human-aSMA rabbit (1:1,000, Abcam, catalogue no. ab5694, clone proprietary, lot no. 1038192-2), human-PDGFRA goat (1:250, R&D Systems, catalogue no. AF307NA, clone P16234, lot no. VG0721111) in pig and marmoset samples, mouse-PDGFRA goat (1:250, R&D Systems, catalogue no., clone P26618, lot no. HMQ0222061) in mouse samples, human-KRT10 rabbit (1:250, Dennis Roop), human-KRT15 chicken (1:250, BioLegend, catalogue no. 833904, clone Poly18339, lot no. B353424, B404683), mouse-SOX9 rabbit (1:1,000, EMDMillipore, catalogue no. AB5535, clone P48436 C-term, lot no. 3677685), human-MKI67 rabbit (1:400, Cell Signaling, catalogue no. 9129S, clone D3B5, lot no. 3, 9), BrdU rat (1:200, Abcam, catalogue no. ab6326, clone BU1/75 (ICR1), lot no. 1009715-43, 1009715-48), human-KRT15 rabbit (1:200, Sigma, catalogue no. HPA024554, clone APREST75794, lot no. A119308), human-SMAD1 goat (1:200, R&D, catalogue no. AF2039, clone Q15797, lot no. KOE062503A), human-pSMAD1/5 rabbit (1:200, Cell Signaling, catalogue no. 9516, clone 41D10, lot no. 10), human-KRT14 rabbit (1:250, Dennis Roop), human-KRT14 mouse (1:1,000, R&D, catalogue no. mab3164, clone LL001, lot no. WEY0823012), human-PDGFC goat (1:200, R&D, catalogue no. af1650, clone Q9NRA1, lot no. JDI022406A), mouse-PECAM1 rat (1:200, Thermo Fisher, catalogue no. 12-0311-82, clone 390, lot no. 1989060, 3095164), mouse-PDGFC rat (1:200, R&D, catalogue no. mab1447, clone Q8Cl19, lot no. HYQ022406A). Secondary antibodies used were Alexa Fluor 488 (AF488) anti-rat (1:1,000, Fisher, catalogue no. A21208, clone AB_2535794, lot no. 2482958, 2668657, 2180272), AF488 anti-chicken (1:1,000, Fisher, catalogue no. A11039, clone AB_2534096, lot no. 1899514, 2941307), AF488 anti-rabbit (1:1,000, Fisher, catalogue no. A21206, clone AB_2535792, lot no. 1874771), Alexa Fluor Plus 555 anti-rabbit (1:1,000, Fisher, catalogue no. A32794, clone AB_2762834, lot no. VK307588, VD297829), AF555 anti-rabbit (1:1,000, Fisher, catalogue no. A31572, clone AB_2535849, lot no. 2831376, 2482963), AF555 anti-goat (1:1,000, Fisher, catalogue no. A21432, clone AB_2535853, lot no. 1878842, 2400919), AF Plus 555 anti-rat (1:1,000, Fisher, catalogue no. A48270, clone AB_2896336, lot no. WF333067, ZG398235), AF647 anti-rabbit (1:1,000, Fisher, catalogue no. A31573, clone AB_2536183, lot no. 2544598), AF647 anti-goat (1:1,000, Fisher, catalogue no. A21447, clone AB_2535864, lot no. 1841382, 2297623). DAPI 300 μM stock (1:1,000, BioLegend, catalogue no. 422801, lot no. B222486, B324682) was used alongside secondary antibodies. Slides were imaged using a Leica SP5 or SP8 confocal microscope. Some neonatal pig wound immunostains were also imaged using a Leica DMI8 fluorescence microscope to capture the entire wound section. Dolphin cryo samples were sectioned at 10 μm, fixed in 4% PFA for 30 min, washed in PBS, and treated with 3% hydrogen peroxide and 0.8% potassium hydroxide for 5 min until bleaching occurred. Blocking

was performed using 2.5% bovine serum albumin for 1 h at room temperature. Sections were then incubated overnight at 4 °C with primary antibodies, including antibodies against α-SMA (1:1,000) and keratin 14 (Abcam, ab9220, 1:1,000). After three washes with PBS with Tween-20 (10 min each), sections were incubated with secondary antibodies for 1 hour at room temperature. Following three further washes with PBS with Tween-20, coverslips were mounted using VECTASHIELD (H-1200-10) antifade mounting medium. Dolphin immunostains were imaged using an Olympus FV3000 confocal microscope. Immunofluorescence images were processed in Adobe Photoshop (v.2021, 2023, 2025).

### Immunofluorescence analysis of paraffin-embedded tissues

Paraffin-embedded tissue samples were sectioned at 10 μm, deparaffinized in xylene and rehydrated using a decreasing ethanol gradient, and antigen retrieval was performed by immersing slides in boiling 10 mM sodium citrate buffer for 15–20 min. Antibody staining was performed in a process similar to that described above, and images were acquired on a Leica SP8 confocal microscope as described above.

### Image analysis and quantification

Numbers of biological replicates for representative histology images and for plots presenting quantifications are included in the figure legends. Images were quantified using Fiji ImageJ (v.1.53c) without blinding. For histological quantifications, human and pig 10-μm H&E or Herovici sections were used to determine epidermal thickness, rete ridge density and apical ridge length measurements. Epidermal thickness was measured from the surface of the epidermis to the base of the rete ridge for rete ridge thickness measurements, and from the surface of the epidermis to the base of the inter-ridge epidermis for inter-ridge thickness measurements. In tissues without rete ridges, rete ridge thickness measurements were not possible, so inter-ridge thickness measurements only were performed and used to construct curves (rete ridge thickness = inter-ridge thickness). This was done to enable calculation of the rete ridge thickness to inter-ridge thickness ratio in R, with rete-ridge-less skin possessing a ratio of 1 and the ratio increasing following the emergence of rete ridges and subsequent increase in rete ridge thickness relative to inter-ridge thickness. Rete ridge density was determined by counting the number of rete ridges and dividing it by the basal length of the interfollicular epidermis over the same region. Multiple measurements for each type of quantification were made per slide, multiple slides were used as technical replicates, and the biological replicate values were reported from an average of the multiple technical replicates for that sample. For developmental quantification of human histology in Fig. 1, fetal trunk skin samples and adult trunk and face skin samples were used. For comparison of trunk skin between species in Fig. 2c, only adult human trunk skin samples were used. For quantification of hair density, the total number of hair follicles in a defined area of the sample was divided by the area assessed. Yucatan hairless pigs had a hair density too low to quantify; there were too few hair follicles present in collected samples to enable accurate assessment of hair density. For *Lef1-eKO* mouse quantifications, WT phenotype and *Lef1-eKO* littermates were compared with respect to hair density, and histology quantifications were compared across genotypes; see the legend of Supplementary Fig. 1 for complete details. For quantitative comparison between P5 pig *EDA-KO* and WT rete ridge density, one litter of P5 *EDA-KO* pig skin samples from University of Iowa were compared to one litter of P5 WT pig skin samples collected at Washington State University. Rete ridges were quantified in the fingerpads of randomly selected biological replicates from three different litters of P21 WT versus *Lef1-eKO* littermates, one experiment involving 3 mo WT versus *K14-Noggin* mice, and one experiment with P56 tamoxifen-treated *K14-Cre^ERT^;Bmpr1a^fl/fl^* versus tamoxifen-treated *K14-Cre^ERT^* (control) mice to assess differences in rete ridge density. Rete ridges are generally concentrated in the distal tip of the fingerpad, beneath the nail, where there are fewer sweat glands. For quantification of scar size, the hairless,

pigmentless scar area in the centre of the wounded area was calculated by tracing surface images of the scar using Fiji. For quantification of neonatal pig wounds, immunofluorescence images were quantified in Fiji as described above, using multiple technical replicates (different stained 60 μm sections) per biological replicate. For proliferation and BrdU analyses, MKI67 and BrdU were quantified in Fiji from immunofluorescence images by counting the number of MKI67+ cells in the basal and suprabasal layers per rete ridge or inter-ridge domain. Similarly, BrdU+ cells were counted for each cell layer in the epidermis per rete ridge or inter-ridge domain. It is possible, owing to the dynamic architectural remodelling that occurs during rete ridge formation, that cells within the rete ridge versus inter-ridge boundaries do not represent fixed populations. However, we did not observe signs of lateral migration of BrdU-labelled cells between domains, consistent with previous studies[43]. Biological replicates were pooled from multiple litters across our collective developmental and BrdU studies owing to difficulty in obtaining pig tissue samples.

### Histology and hair density quantification visualization

All quantitative data were graphed using the ggplot package (v.3.4.0) in R (v.4.2.2) with the geom_box() and geom_line() functions for box plots and line plots, respectively. Error bars in line plots represent the standard error of the mean. Hair density and epidermal measurement associations were visualized using geom_line() with addition of a fit line based on a $y \sim \log_2(x)$ function using the generalized additive model smoothing method in the stat_smooth() function. Individual data points were included on all box plots using the geom_point() function.

### Generation of single-cell suspension from pig skin for scRNA-seq

Skin from E90, P3, P10 and 6 mo pigs was collected under approved protocols and processed to generate a single-cell suspension from the epidermis and dermis as described previously[68]. In a modification to the aforementioned protocol, we added elastase (Worthington Biosciences), hyaluronidase (Sigma-Aldrich) and collagenase IV (Worthington Biosciences) to the dermal digestion solution. All porcine epidermal and dermal single-cell suspensions were mixed 1:1 and processed for 10x Genomics scRNA-seq 3′ V3 Kit to generate scRNA-seq libraries.

### Pig scRNA-seq processing and reanalysis of previously published datasets

Pig scRNA-seq libraries were sequenced on an Illumina NovaSeq PE150 by Novogene. Fastq files were aligned to the Sscrofa.11.1 genome assembly[69] (NCBI RefSeq GCF_000003025.6) using 10x Genomics Cell Ranger (v.6.0.0). Cell Ranger outputs were used in downstream analyses. Previously published scRNA-seq datasets were obtained from the GEO or EGA repositories for the respective publications[12,26,35,36] (see our GitHub (https://github.com/DriskellLab/Thompson-et-al.-2025) for more details).

### scRNA-seq analysis

We analysed our pig scRNA-seq datasets and reanalysed previously published human and mouse scRNA-seq datasets[12,26,35,36] using the Seurat package[70] in R. We used standard quality control metrics to filter out low-quality cells, normalized and scaled data using SCTransform, and performed dimensional reduction using UMAP with the SLM algorithm to identify clusters. We annotated clusters on the basis of canonical markers for the cell lineages present in skin alongside differential gene expression analysis, as we and others have described previously[12,26–28,35,36,71–73]. In brief, basal keratinocytes expressed high levels of *KRT15*, *ITGA6*, *ITGB1* and *KRT14*, whereas differentiating keratinocytes expressed low levels of these basal markers and high levels of *KRT10*, *KRT1* and *CALML5* (Supplementary Fig. 2c,f,h,j). Dermal fibroblasts broadly expressed *PDGFRA* and *COL1A1*/*COL3A1*, and papillary fibroblasts were resolved at each time point by their expression of the papillary fibroblast markers *APCDD1*, *AXIN2* and *CRABP1* and

lack of expression of the dermal papillae fate markers *LEF1* and *ALX4* (refs. 3,27,28,36; Supplementary Fig. 2c,h). Pericytes were identifiable by expression of *RGS5* and *ACTA2*, blood vessels by expression of *PECAM1* and *CDH5*, and lymphatic vessels by expression of *LYVE1*, *VEGFR3* (also known as *FLT4*) and *CCL21* (refs. 28,74; Supplementary Fig. 2c,h). Several subpopulations of sweat gland cells, including ductal and secretory components, were identifiable; these were characterized either by *KRT14* and *VIM* coexpression or *SOX9* expression for ductal cells[13], or expression of *KRT18*, *CHIA*, *PHEROC* and *ACTA2* for secretory coil cells[3,14] (Fig. 5a and Supplementary Figs. 2c,h and 3a,b). In addition to its expression in sweat glands, *SOX9* was expressed in outer root sheath keratinocytes, consistent with findings of previous studies[52] (Supplementary Figs. 2c,h and 3a,b). *LHX2* was also used as a hair follicle keratinocyte marker[75] (Extended Data Fig. 5g). We also identified melanocytes and nerve cells by *SOX10* and immune cells by *PTPRC* (Supplementary Fig. 2c,h). The DotPlot() function from Seurat was used to visualize expression of representative marker genes used to define cell types in the UMAP. MKI67 (ENSSSCG00000026302 in porcine datasets owing to genome annotation and renamed MKI67 in figures) was also used to identify dividing cell states.

For integration of porcine epidermal scRNA-seq data for E90, P3, P10 and 6 mo basal cells, basal keratinocyte clusters, defined by high expression of *KRT14/15* and not *SOX9*, *KRT10* or other sweat gland markers, were subsetted from each of the individual datasets and merged in Seurat using merge(). Merged datasets were renormalized using SCTransform. For integration of postnatal porcine interfollicular epidermis clusters and P3, P10 and 6 mo keratinocyte clusters, only keratinocyte clusters that did not express sweat gland or hair follicle markers were included. For integration of porcine dermis, E90, P3, P10 and 6 mo papillary fibroblast, pericyte and vascular clusters were subsetted from the individual datasets and merged in Seurat as described above. The P3–P10–6 mo interfollicular epidermis and E90–P3–P10–6 mo dermal integrations also included Harmony batch correction using RunHarmony[76]. For full details, see our GitHub (https://github.com/DriskellLab/Thompson-et-al.-2025).

The postnatal integrated interfollicular epidermis keratinocytes were also converted from a Seurat object to a CDA object using the SeuratWrappers package to perform pseudotime analysis in Monocle3 (ref. 77). Pseudotime trajectories from non-dividing basal to differentiating states across this dataset were defined with a root in non-dividing basal keratinocyte clusters, which is included on the UMAP-projected pseudotime plots in the figures for reproducibility. To generate line plots of gene expression across pseudotime trajectories, pseudotime gene expression by cell matrices was extracted and converted to a dataframe for visualization of pseudotime gene expression trajectories in ggplot using the geom_line() function accompanied by a trendline generated with the stat_smooth() function using the generalized additive model method and a $y - x$ formula.

To generate cluster-level gene expression heatmaps, the ComplexHeatmap package[78] in R was used. For FeaturePlots for which the gene was not detected in any cell in the dataset, a blank FeaturePlot with the caption 'Gene not detected in this dataset' was created using Adobe Illustrator.

## Stereo-seq analysis
First, 4%-PFA-fixed frozen cryo skin samples from P3, P10 and 6 mo pigs were sectioned at 10 μm for use with a Complete Genomics Stereo-seq T FF v.1.2 kit and sequenced using a DNBSEQ-T7. The pig reference genome for SAW (8.1.1) was prepared using the Sscrofa11.1.112 release from Ensembl and the makeRef function. Next, we used SAW count to generate gene expression matrix files. We manually aligned the gene expression matrix on to the tissue mask using Stereo Map 4 and slightly cropped the tissue mask area of each dataset to enable downstream analysis on our local hardware. Then, we ran SAW realign on the cropped tissue mask image and previous SAW count outputs to obtain the processed stereo-seq tissue mask image and gene expression files used in downstream analysis. SAW realign Visualization output.gef files were loaded into Stereopy (1.5.0) in Python (3.8.20) using read_gef() and bin_size = 20. Data underwent quality control filtration and were normalized using sctransform. Clustering was performed using the Leiden algorithm with resolutions of 1.2 (P3), 1.0 (P10) and 1.0 (6mo). Leiden clusters were assigned cell types on the basis of their spatial expression of canonical markers ('scRNA-seq analysis') using spatial_scatter_by_gene(), find_marker_genes(), and the spatial localization of the cluster on the tissue mask image with the cluster_scatter() function. All source code is publicly available at GitHub (https://github.com/DriskellLab/Thompson-et-al.-2025).

## Cell–cell communication analyses
CellChat[44,79] analysis was used to infer pathway and ligand–receptor interactions among core basal and dividing keratinocyte, papillary fibroblast, pericyte and blood vessel clusters from the E90, P3, P10 and 6 mo porcine skin scRNA-seq datasets in parallel, following the standard CellChat pipeline with the human ligand–receptor database to infer cell–cell communication between these cell groups. For pathways or ligand–receptor pairs not identified as a significant interaction by CellChat, based on the default software threshold of $P < 0.05$ set by the thresh = 0.05 argument in functions like netVisual_aggregate() and extractEnrichedLR(), an empty CirclePlot with the caption 'Interaction Not Predicted' was created using Adobe Illustrator. Other epidermal- or dermal-resident cell types may also contribute to rete ridge formation and maturation, including nerve cell endings and immune cells[25,35,49,56,59,80]; these were beyond the scope of the current study. For the porcine Stereo-seq datasets, Spatial CellChat was used to infer biologically realistic interactions involved in cell–cell communication at cellular resolution on the basis of spatial locations at P3, P10 and 6 mo. Spatial CellChat constrains interactions to biologically realistic distances using the maximal range of molecular diffusion or contact. For secreted signalling, the maximal possible molecular interaction range is the ideal transport distance for small diffusible molecules in a free medium (250 μm by default); for contact-dependent signalling, interactions are restricted to those of molecules in direct contact with each other. The outgoing communication score is defined as the sum of communication probabilities for all outgoing signals, reflecting the role of the cell as a signal sender. Conversely, the incoming communication score is the sum of incoming communication probabilities, representing the role of the cell as a signal receiver. Spatial plots visualizing overall outgoing and incoming secreted signalling strength were constructed for all Leiden clusters from the stereo-seq datasets. Spatial plots visualizing pathway and ligand–receptor-level signalling strength were constructed for a subset of the Stereo-seq data comprising Leiden clusters that were classified as epidermal keratinocyte, papillary fibroblast and vascular clusters: clusters 16, 11 and 6 from P3; clusters 18, 16, 9, 3 and 17 from P10; and clusters 2, 7, 4, 5, 14 and 6 from 6 mo. To provide a visual estimate of the epidermal–dermal junction in Spatial CellChat plots, the bottom boundary was traced beneath the keratinocyte Leiden-clustered spots in Adobe Illustrator using the pen tool and then aligned and superimposed at 50% opacity over the Spatial CellChat plots in Adobe Illustrator (v.2023, 2025). Spatial CellChat plots visualizing pathway and ligand–receptor signalling strength in Fig. 4d, Extended Data Fig. 4a,b and Supplementary Figs. 9–12 were uniformly contrast adjusted by −100% in Adobe Photoshop (v.2023, 2025) to improve readability between the light blue and purple values from the outgoing and merged colour palettes compared with the grey base spot colour.

## Statistics
Statistical analyses were performed in R (v.4.2.2). Sample sizes are listed in the relevant figures and figure legends. No sample size calculations were performed. A minimum of three or more biological replicates, and

a mix of males and females, were obtained when possible. However, at some time points, our human or pig sample sets only consisted of one or two biological replicates owing to model limitations (such as initial pig litter size); these are noted in the figure legends. In mouse experiments, multiple litters and a mix of male and female individuals was used when possible; specifics are included in the figure legends or Methods. In pig experiments, multiple litters were used when possible. Sample sizes in this study were comparable with those used in other studies in the field. $P < 0.05$ was considered to indicate significance; individual $P$ values are given in figure legends. Each experiment was performed once, and no data points were excluded from statistical analysis. In cases in which we could not analyse all littermates from the same litter (for instance, owing to breeding strategy; Fig. 5d), biological replicates were selected randomly for analysis. Hair density and epidermal measurement correlations were computed using the summary() and lm() functions with the $\log_2$ formula. The adjusted coefficient of determination (adjusted $R^2$) and $P$ values are reported in the figures. Zoo histology quantifications, pig wound quantifications between time points, and WT/Het/*Lef1-eKO* histology quantifications were analysed using one-way analysis of variance with the aov() function plus post hoc Tukey's HSD using the TukeyHSD() function. WT/*EDA-KO* pig comparisons, rete ridge versus inter-ridge domain MKI67 and BrdU comparisons, and fingerpad rete ridge or sweat gland density comparisons were performed using Welch's two-sample *t*-test with the t.test() function.

## Reporting summary

Further information on research design is available in the Nature Portfolio Reporting Summary linked to this article.

## Data availability

We have made all the analysed scRNA-seq and stereo-seq datasets in this study available through our interactive webtools at skinregeneration. org (https://skinregeneration.org/papers/Thompson-et-al-2025/). Raw and processed sequencing data files used in our analysis can be found at the NCBI Gene Expression Omnibus (GEO accession GSE305111). Exact *P* values are provided in figure legends or in source data where there is insufficient space in the legend. All other data are available either as Supplementary Information or from the corresponding author upon reasonable request. Source data are provided with this paper.

## Code availability

Code and software package versions used in our analyses are available at GitHub (https://github.com/DriskellLab/Thompson-et-al.-2025).

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

**Acknowledgements** All confocal microscopy was performed on equipment housed and maintained by the Franceschi Microscopy and Imaging Center at WSU. Sequence alignments and some other computations used resources from the Center for Institutional Research Computing at Washington State University. We thank G. Clyde, N. Woodford, J. DenHerder and Washington State University Office of the Campus Veterinarian techs for support with animal experiments; J. Sexton, A. Baylink, T. Issa and Sinclair Biosciences for providing some pig samples; and C. Gwinn, J. Cvancara, J. Sears and C. Sukut from Advanced Dermatology in Spokane, Washington, for collecting adult human tissue samples. We also acknowledge core contributions from L. Madison and the WSU Gene Editing Reagent Core, D. Pouchnik and the WSU Molecular Biology and Genomics Core, the UW Birth Defects Research Laboratory, the San Antonio Nathan Shock Center, the UCLA Translational Pathology Core Lab, and the UCLA Technology Center for Genomics and Bioinformatics; and further assistance from I. Phelps, A. Luo, J. Canfield, J. Lee, S. Spires, E. Gonzalez, S. Penrod and S. Kindl. S.T. thanks N. Law for sharing antibodies. Cetacean graphical representations used in Supplementary Fig. 13c were obtained from the NOAA Fisheries Species Directory entries for bottlenose dolphin, long-beaked common dolphin and short-beaked common dolphin. Photograph of the Mangalitsa pig used in Fig. 2b was provided by T. Issa. R.R.D. was supported by NIH R01 AR078743; S.M.T. and Q.P. by the Poncin Fellowship; and Q.M.P. and J.M. by NIH NIGMS T32GM008336. This research was supported by the Resilient Livestock Initiative, US Department of Agriculture, Agricultural Research Service, award number AWD006105. H.T.J. and the Washington State University Bear Research, Education, and Conservation Center were supported by The Interagency Grizzly Bear Committee, USDA NIFA (McIntire-Stennis project 1018967), Mazuri Exotic Animal Nutrition, the Raili Korkka Brown Bear Endowment, the Nutritional Ecology Endowment, and the Bear Research and Conservation Endowment at Washington State University. G.H.S. and T.D. were supported by EschLEAD funds; M.V.P. by W.M. Keck Foundation grant WMKF-5634988, Chan Zuckerberg Initiative grant AN-0000000062, Horizon Europe grant 101137006, NSF grant IOS-2421118, and NIH grants R01-AR079470, R01-AR079150 and P30-AR075047; and Q.N. by NIH R01GM152494, NIH R01-AR079470 and NSF grant DMS1763272. This work was partly supported by a Major Research Plan of the National Natural Science Foundation of China (no. 92374108 to S.J.), and partly supported by the Foundation for Polish Science (FNP), co-financed by the European Union under the European Regional Development Fund, Team Grant POIR.04.04.00-00-4222/17-00 (to K.K.), National Science Centre, Poland (NCN) Opus Grant 2015/19/B/NZ3/02948 (to K.K.), Opus Grant 2019/33/B/ NZ3/02966 (to K.K.) and Opus Grant 2022/45/B/NZ3/03811 (to K.K.). B.P.H., Z.J. and D.R. were funded by NIH U01 DA054170. SNPRC is funded by P51OD011133. San Antonio Nathan Shock Center was supported by NIH P30 AG013319, and the ONPRC by NIH P51OD011092. I.A.G. was funded by NIH NICD R24HD000836. The Laboratory for Developmental Biology was supported by NIH Award Number 5R24HD000836 (to I.A.G.) from the Eunice Kennedy Shriver National Institute of Child Health & Human Development. M.J.W. was funded by NIH HL163556-01 and HL091842 and the Howard Hughes Medical Institute. D.A.S. was funded by NIH R01HL163556.

**Author contributions** S.M.T., I.M.D. and R.R.D. conceived the study. S.M.T., Q.M.P., J.M., I.M.D. and R.R.D. conducted scRNA-seq experiments, and S.M.T. and R.R.D. conducted stereo-seq experiments. S.M.T. performed scRNA-seq, scRNA-seq CellChat and stereo-seq bioinformatics analyses under supervision of R.R.D, Q.N. and S.J. conceived Spatial CellChat, and X.C. performed Spatial CellChat analysis under supervision of S.J. S.M.T. collected most pig tissue samples with contributions from Q.M.P., J.M., N.M.W., I.M.D. and R.R.D. S.M.T. and I.M.D. collected most mouse tissue samples, and S.M.T., V.S.Y. and G.H.S. performed histology and quantification. For the histological zoo, naked mole rat samples were collected under supervision of A.B.S., rhesus macaque samples were collected under supervision of O.D.S., and common marmoset samples were collected by Z.S.J. and D.R. under supervision of B.P.H. H.T.J. collected bear samples and supervised bear experiments. These samples were then shared with S.M.T. under supervision of R.R.D. for histological processing. R.L. collected and performed histology on cetacean samples under supervision of M.V.P. S.M.T. and V.S.Y. performed most histology and quantification from the histological zoo with contributions from G.H.S., C.Y., N.M.W., I.V.B. and M.V.C. UWBDRL collected gestational human tissue samples under the supervision of I.A.G. S.M.T. and V.S.Y. performed histology and quantification of gestational human tissue histology. S.M.T., V.S.Y. and C.Y. processed, performed histology of, and quantified adult human tissue samples. *EDA-KO* pigs were generated and tissue was collected and some histology performed under supervision of D.A.S. and M.J.W., and S.M.T. and V.S.Y. performed further histology and quantification. S.M.T., M.C., I.M.D. and R.R.D. conducted pilot experiments to validate the methodology for and to conduct pig wound healing and BrdU experiments, and S.M.T. performed histology and quantification. A.P.C. performed tamoxifen-treated paw experiments, collected samples and performed histology from *K14-Cre*ERT;*Bmpr1a* mice under supervision of K.K. M.V.P. provided archived *K14-Noggin* digit samples, and S.M.T.

and G.H.S. performed histology. S.M.T. quantified the *K14-Cre;<sup>ERT</sup>Bmpr1a* and *K14-Noggin* data. S.M.T., T.T.D., R.L., X.C. and A.P.C. curated and stored experimental or bioinformatics data. T.D. and R.R.D. designed the bioinformatics webtool. S.M.T., Q.M.P., J.M., A.B.S., O.D.S., B.P.H., D.A.S., M.J.W., I.A.G., K.K., Q.N., S.J., H.T.J., M.V.P. and R.R.D. acquired funding related to this project. S.M.T. and R.R.D. made the figures with feedback from J.M., M.V.P. and I.M.D. S.M.T. wrote the original draft of the manuscript, and S.M.T., J.M., M.V.P., I.M.D. and R.R.D. reviewed and edited it with input from all authors. S.M.T., V.S.Y., I.M.D. and R.R.D. performed project administration, with overall project supervision from R.R.D.

**Competing interests** S.M.T., R.R.D., and M.V.P. are co-inventors on intellectual property relating to methods that promote epidermal rete ridge formation and regeneration in skin. Washington State University and The Regents of the University of California will file a patent application covering aspects of this work; an application identifier was not available at the time of manuscript approval. The other authors declare no competing interests.

**Additional information**
**Correspondence and requests for materials** should be addressed to Ryan R. Driskell.

## a Dermal Connective Tissue Maturation During Human Skin Development

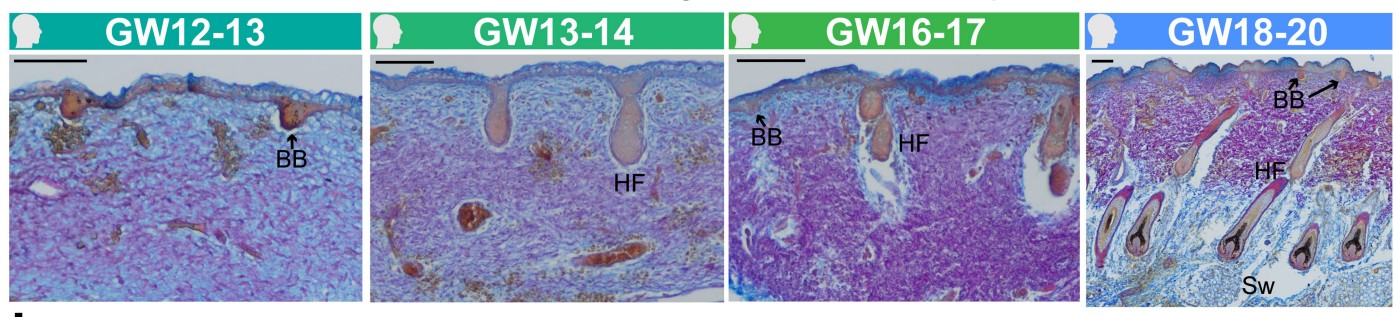

## b Dermal Connective Tissue Maturation During Pig Skin Development

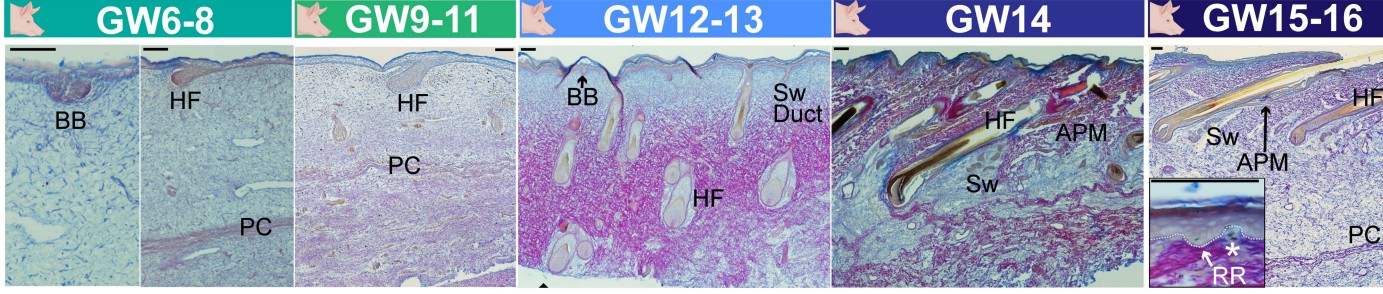

## c Dermal Connective Tissue is Mature in Postnatal Pig Skin

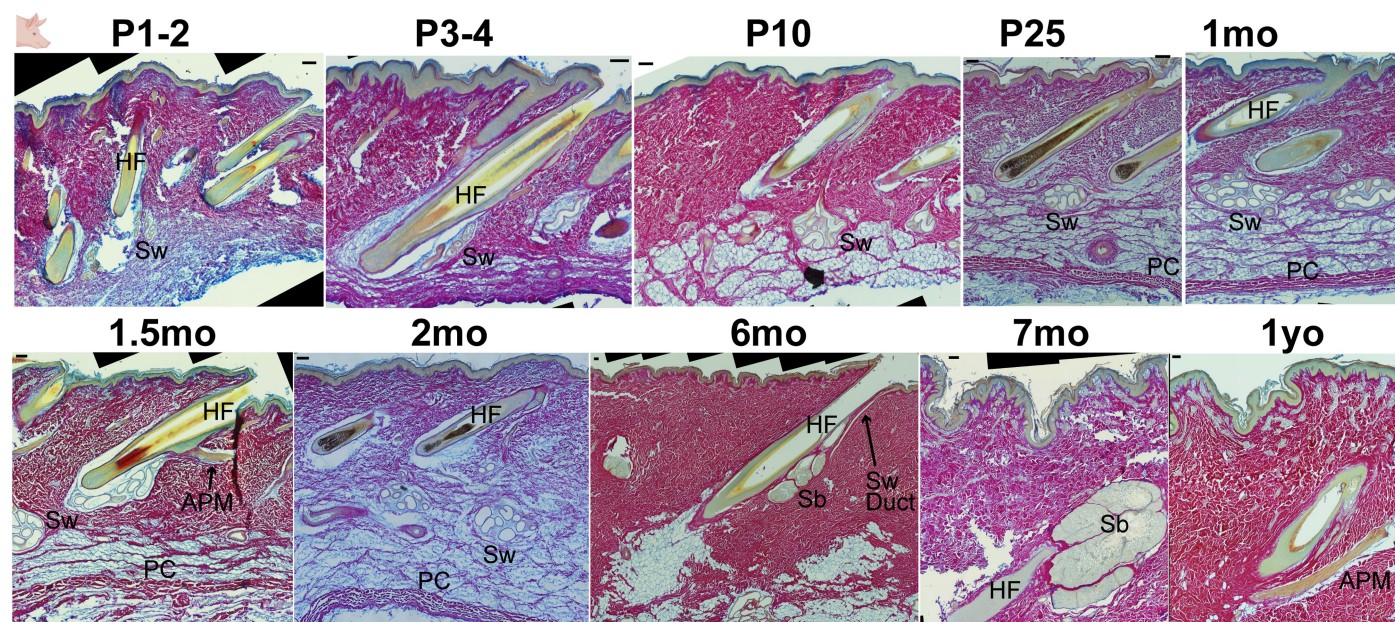

**Extended Data Fig. 1 | Rete ridges form perinatally in human and pig skin.**
(**a**) Representative Herovici stains from human gestational week (GW) 12-13 (n = 1), 13-14 (n = 1), 16-17 (n = 7), 18-20 (n = 5) trunk skin. (**b**) Representative Herovici stains from porcine GW6-8 (n = 7), 9-11 (n = 6), 12-13 (n = 11), 14 (n = 3), 15-16 (n = 4). (**c**) Representative Herovici stains from porcine postnatal day (P) 1-2, 3-4, 10, 25, 1-month-old (1mo), 2mo, 6mo, 7mo, and >1-year-old (1yo). (a-c) Scale bars indicate 100um. BB = basal bud/epithelial placode, HF = hair follicle, BV = blood vessel, Sw = sweat gland, Sb = sebaceous gland, PC = panniculus carnosus, APM = arrector pili muscle. Illustrations in **a–c** were created using BioRender. Thompson, S. (2026) https://BioRender.com/8rd8cz9.

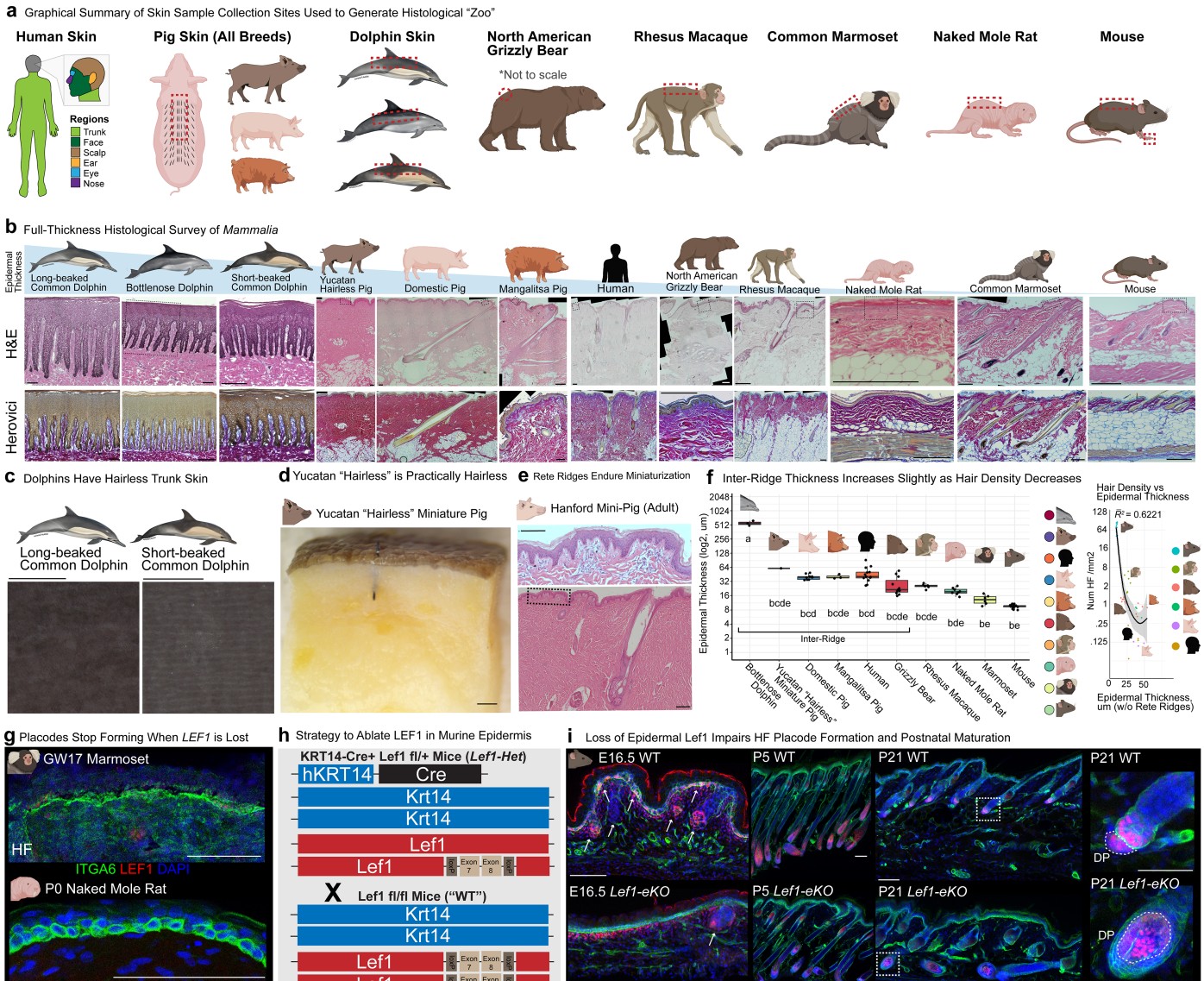

**Extended Data Fig. 2 | Full-thickness Histological "Zoo" From Aquatic and Terrestrial Mammals.** (**a**) Graphical summary representing anatomical sites skin samples were collected from across species to generate the histological "zoo". See Methods for complete details on pig breed backgrounds. (**b**) Representative H&E and Herovici stains from trunk skin of long-beaked common dolphins (n = 3), bottlenose dolphins, short-beaked common dolphins (n = 3), Yucatan miniature "hairless" pig, humans, Mangalitsa pigs, North American grizzly bears, rhesus macaques, naked mole rats, common marmosets, and mice. Same number of biological replicates as Fig. 2a. Dashed rectangles represent regions of zoom-in in Fig. 2a. (**c**) Representative surface images of long-beaked and short-beaked common dolphin trunk skin. (**d**) Representative image from Yucatan miniature "hairless" pig skin sample to illustrate hair density that is too low to reliably quantify. (**e**) Representative H&E stain of skin from an adult Hanford mini-pig (n = 1). Dashed box indicates region of zoomed inlay (top). Zoom-out imaged on 10x objective, zoom-in imaged on 20x objective. (**f**) Quantification of histology for epidermal thickness (left, inter-ridge only, if applicable) and the correlation between epidermal thickness (right, inter-ridge only, if applicable) and hair density. Number of replicates

same as main Fig. 2c-d. Shared letters indicate no difference (p-value > 0.05) and different letters indicate significant difference (p-value < 0.05) from one-way ANOVA plus Tukey's HSD (exact p-values in Source Data). Correlation statistic shown is adjusted coefficient of determination, p-value = 5.532e-10. (**g**) Representative immunostains from fetal gestational week 17 (GW17) marmoset and neonatal naked mole rat for ITGA6/LEF1 and KRT15/LEF1, respectively (n = 3). HF=hair follicle. (**h**) Breeding strategy to generate *K14-Cre;Lef1$^{fl/fl}$* (*Lef1-eKO*) mice from crossing *K14-Cre;Lef1$^{fl/+}$* (*Lef1-Het*) mice with *Lef1$^{fl/fl}$* (WT) mice. (**i**) Representative immunostains (n = 3 each) from embryonic day 16.5 (E16.5), postnatal day 5 (P5), and P21 WT (top) and *Lef1-eKO* (bottom) mice for Itga6/Lef1. Arrows indicate hair follicles. Scale bars in (b) represent 250 um; in (c-d) 1 mm; in (e) zoom-out 250 um and zoom-in 100 um; in (g, i) 100 um. Cetacean illustrations in **a**–**c** and **f** were obtained from the NOAA Fisheries Species Directory entries for bottlenose dolphin, long-beaked common dolphin and short-beaked common dolphin. Illustrations in **a**–**i** were created using BioRender. Thompson, S. (2026) https://BioRender.com/8rd8cz9.

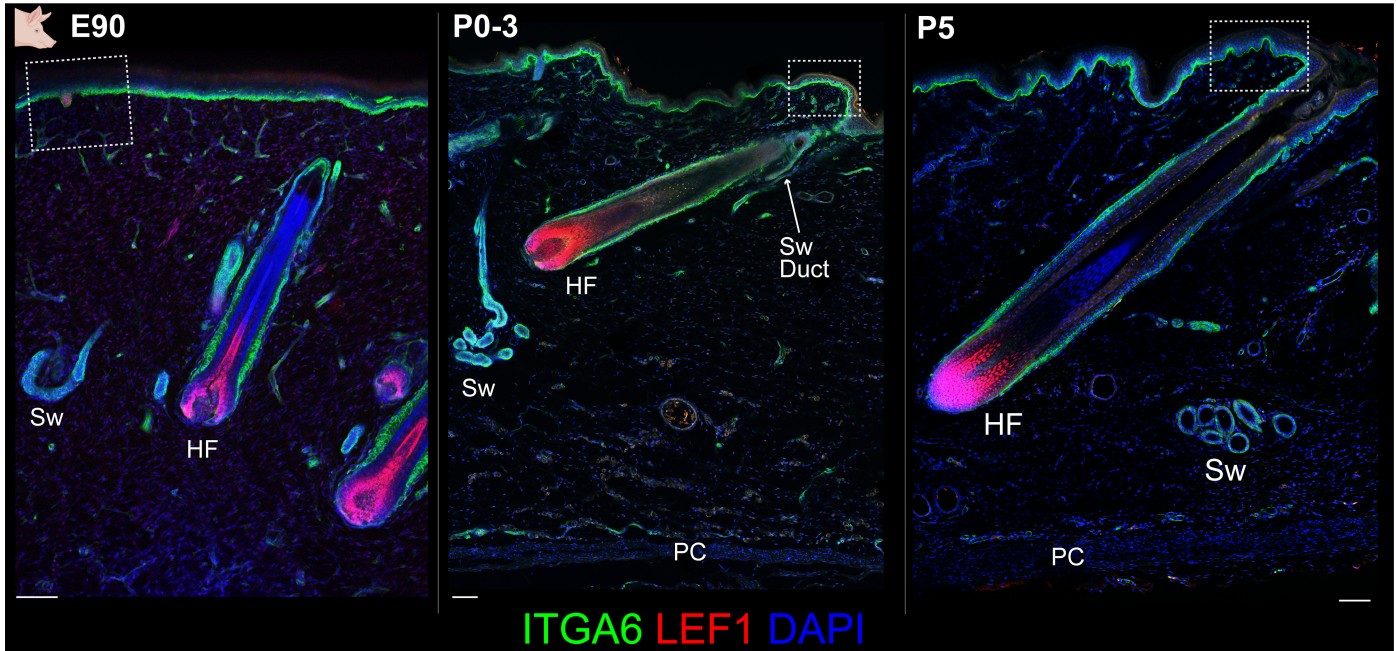

**Extended Data Fig. 3 | Developing Rete Ridges Do Not Express Focal *LEF1*.**
(**a**) Representative immunostains of E90 (n = 3), P0-3 (n = 3), and P5 (n = 3) pig skin stained for ITGA6/LEF1. Sw = sweat gland, HF = hair follicle, PC = panniculus carnosus. Scale bars represent 100 um. Dashed boxes indicate regions of zoom-in in Fig. 3a. Illustrations in **a** were created using BioRender. Thompson, S. (2026) https://BioRender.com/8rd8cz9.

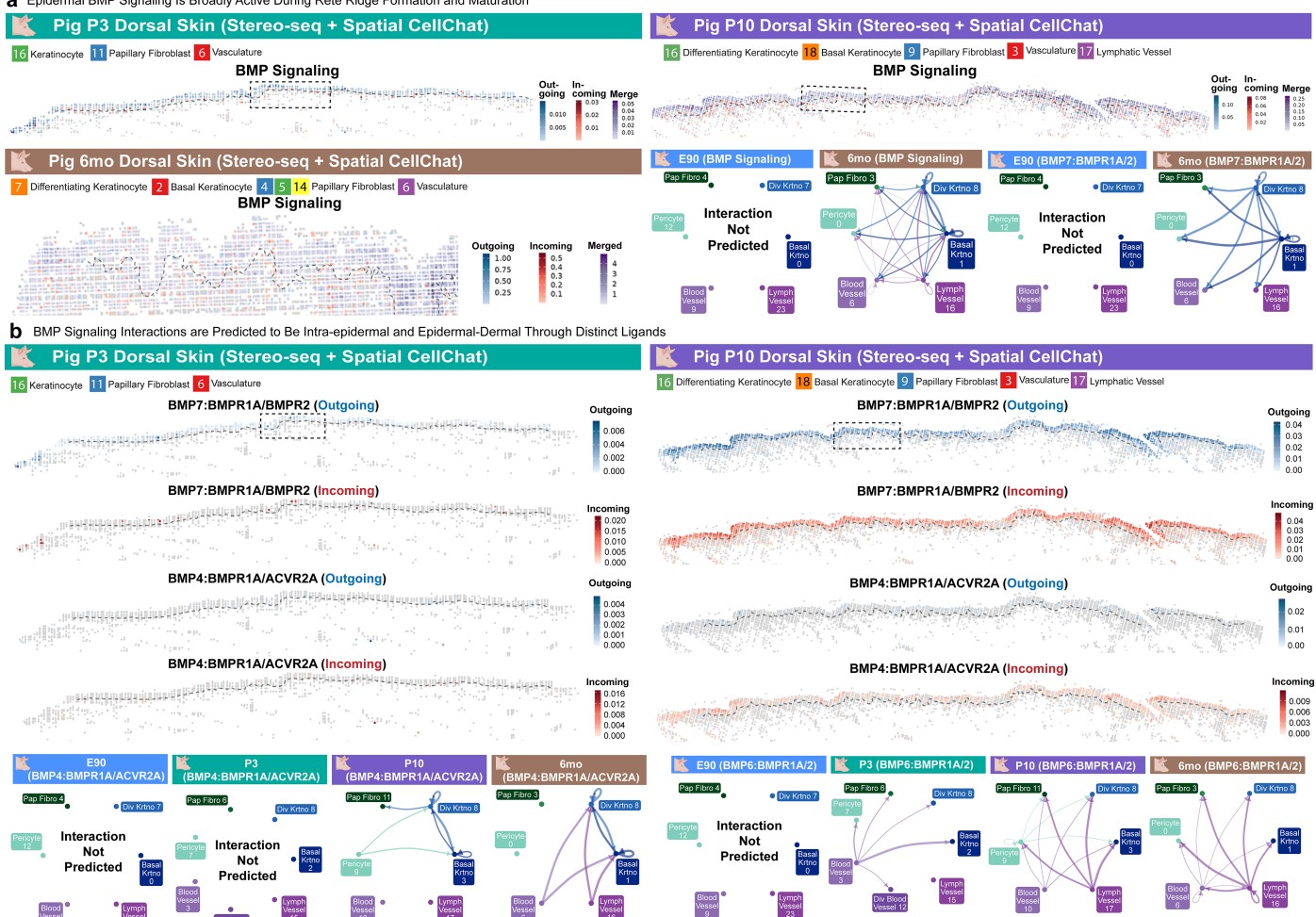

**Extended Data Fig. 4 | BMP Signaling Activates During Rete Ridge Formation.** (**a**) Spatial CellChat outgoing/incoming communication score across the Stereo-seq data at P3, P10, and 6mo (top and left), and (bottom right) scRNA-seq CellChat CirclePlots for predicted signaling interactions at E90 and 6mo for the BMP signaling pathway and BMP7 ligand-receptor interactions. P3 and P10 scRNA-seq CirclePlots for BMP signaling (pathway-level) and BMP7 (ligand-level) are featured in main Fig. 4d. (**b**) Spatial CellChat outgoing/ incoming communication score across the Stereo-seq data at P3 and P10 (top) or (bottom) scRNA-seq CellChat CirclePlots for predicted signaling

interactions from scRNA-seq data at E90, P3, P10, and 6mo for BMP signaling ligand-receptor pairs. (**a-b**) dashed rectangles indicate regions of zoom-in featured in main Fig. 4d. (**a-b**) Dashed line approximates the epidermal-dermal junction traced from the Stereo-seq Leiden-clustered image mask (Supplemental Fig. 8a). Stereo-seq subset utilized in Spatial CellChat analyses consisted of epidermal keratinocytes, papillary fibroblasts, and vascular/ pericyte clusters. Illustrations in **a** and **b** were created using BioRender. Thompson, S. (2026) https://BioRender.com/8rd8cz9.

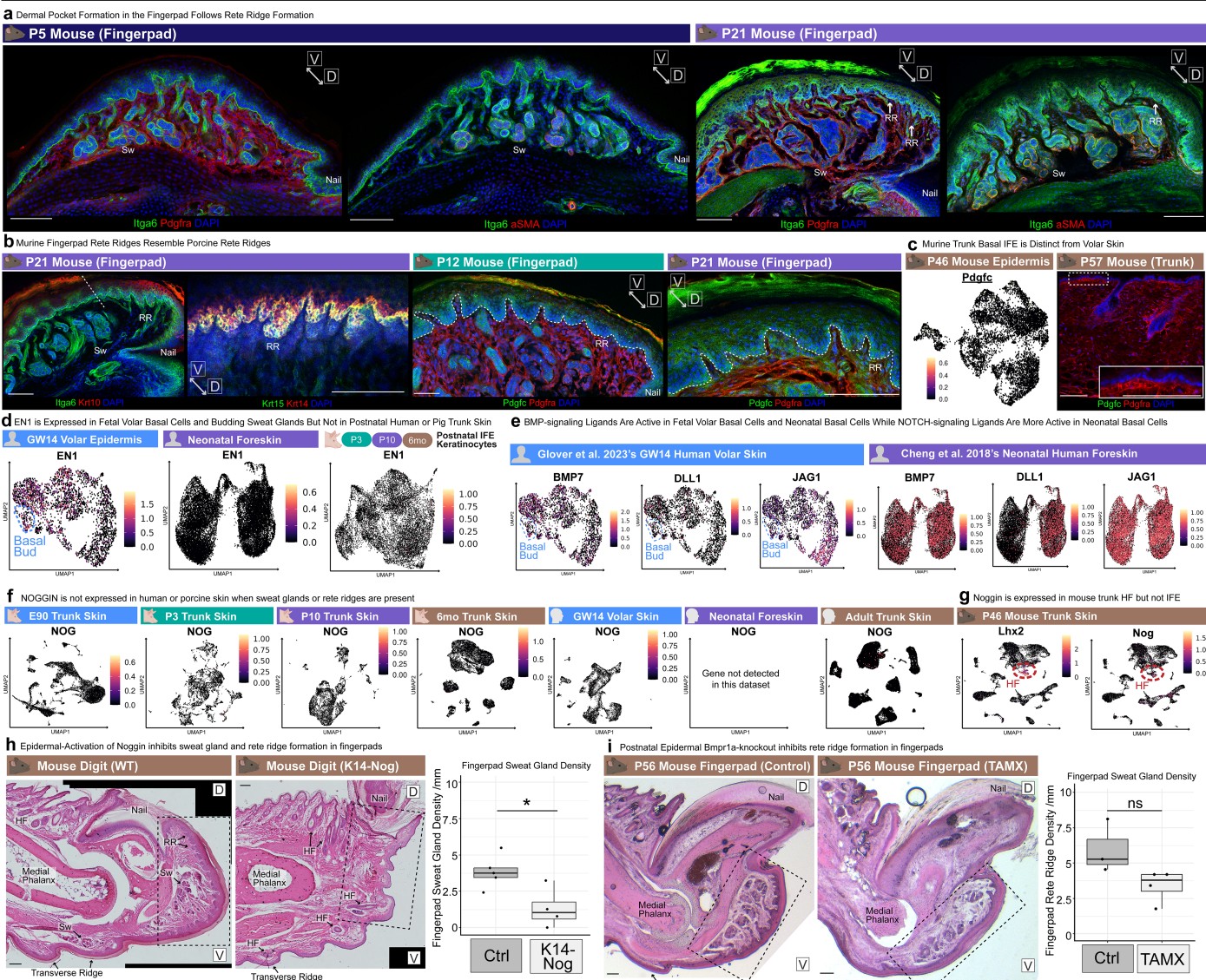

**a** Dermal Pocket Formation in the Fingerpad Follows Rete Ridge Formation

**b** Murine Fingerpad Rete Ridges Resemble Porcine Rete Ridges

**c** Murine Trunk Basal IFE is Distinct from Volar Skin

**d** EN1 is Expressed in Fetal Volar Basal Cells and Budding Sweat Glands But Not in Postnatal Human or Pig Trunk Skin

**e** BMP-signaling Ligands Are Active in Fetal Volar Basal Cells and Neonatal Basal Cells While NOTCH-signaling Ligands Are More Active in Neonatal Basal Cells

**f** NOGGIN is not expressed in human or porcine skin when sweat glands or rete ridges are present

**g** Noggin is expressed in mouse trunk HF but not IFE

**h** Epidermal-Activation of Noggin inhibits sweat gland and rete ridge formation in fingerpads

**i** Postnatal Epidermal Bmpr1a-knockout inhibits rete ridge formation in fingerpads

**Extended Data Fig. 5 | Mouse Volar Fingerpad Rete Ridges Form Postnatally and Require Epidermal BMP Signaling.** (**a**) Representative immunostains of P5 (left) and P21 (right) fingerpads stained for Itga6/Pdgfra (left, n = 3 each) or Itga6/aSMA (right, n = 3 each). (**b**) Representative immunostain of P21 fingerpads stained for Itga6/Krt10 or Krt15/Krt14 (left, n = 3 each), or P12 and P21 fingerpads stained for Pdgfc/Pdgfra (right, n = 3 each). (**c**) FeaturePlot depicting lack of *Pdgfc* expression in Liu et al. 2022's P46 Mouse scRNA-seq epidermal subset (left) and representative immunostain of P57 mouse trunk skin stained for Pdgfc/Pdgfra (right, n = 3). (**d**) FeaturePlots depicting *EN1* expression in Glover et al. 2023's GW14 human volar skin epidermal subset, Cheng et al. 2018's neonatal human foreskin epidermal subset, and integrated P3, P10, and 6mo porcine epidermis. (**e**) FeaturePlots depicting *BMP7*, *DLL1*, and *JAG1* expression in Glover et al. 2023's GW14 human volar skin scRNA-seq epidermal subset, Cheng et al. 2018's neonatal human foreskin scRNA-seq epidermal subset. (**f**) FeaturePlots depicting *NOG* expression in E90, P3, P10, and 6mo porcine trunk skin integrated scRNA-seq (left) and in GW14 human volar skin epidermal subset, neonatal human foreskin epidermal subset, and Solé-Boldo et al. 2020's adult human trunk skin epidermal subset (right). (**g**) FeaturePlots depicting *Lhx2* and *Nog* expression in P46 mouse trunk skin scRNA-seq. The red dashed circle emphasizes this *Lhx2+* follicular keratinocyte cluster. (**h**) Representative H&E stains from 3mo wild-type (WT, n = 5) and *K14-Noggin* (*K14-Nog*, n = 4) digits (left) and quantification of fingerpad sweat gland density (right). * p-value = 0.0265 from t-test. Representative digit transverse ridges are indicated by arrows. (**i**) Representative H&E stains from P56 tamoxifen-applied *K14-Cre^ERT* (Ctrl, n = 3) and *K14-Cre^ERT;Bmpr1a^fl/fl* (TAMX, n = 4) digits (left) and quantification of fingerpad sweat gland density (right). ns=not significant, p-value = 0.1224 from t-test. Dashed boxes in (h) and (i) represent the regions of zoom-in in Figs. 5c and 5d, respectively. (a-b, h-i) Sw=sweat gland, RR=rete ridge, D and V indicate dorsal-ventral orientation. Scale bars represent 100 um. Illustrations in **a**–**i** were created using BioRender. Thompson, S. (2026) https://BioRender.com/8rd8cz9.

# Reporting Summary

## Statistics

For all statistical analyses, confirm that the following items are present in the figure legend, table legend, main text, or Methods section.

| n/a | Confirmed | |
|---|---|---|
| ☐ | ☒ | The exact sample size (*n*) for each experimental group/condition, given as a discrete number and unit of measurement |
| ☐ | ☒ | A statement on whether measurements were taken from distinct samples or whether the same sample was measured repeatedly |
| ☐ | ☒ | The statistical test(s) used AND whether they are one- or two-sided<br>*Only common tests should be described solely by name; describe more complex techniques in the Methods section.* |
| ☒ | ☐ | A description of all covariates tested |
| ☐ | ☒ | A description of any assumptions or corrections, such as tests of normality and adjustment for multiple comparisons |
| ☐ | ☒ | A full description of the statistical parameters including central tendency (e.g. means) or other basic estimates (e.g. regression coefficient) AND variation (e.g. standard deviation) or associated estimates of uncertainty (e.g. confidence intervals) |
| ☒ | ☐ | For null hypothesis testing, the test statistic (e.g. *F*, *t*, *r*) with confidence intervals, effect sizes, degrees of freedom and *P* value noted<br>*Give P values as exact values whenever suitable.* |
| ☒ | ☐ | For Bayesian analysis, information on the choice of priors and Markov chain Monte Carlo settings |
| ☒ | ☐ | For hierarchical and complex designs, identification of the appropriate level for tests and full reporting of outcomes |
| ☒ | ☐ | Estimates of effect sizes (e.g. Cohen's *d*, Pearson's *r*), indicating how they were calculated |

*Our web collection on statistics for biologists contains articles on many of the points above.*

## Software and code

Policy information about availability of computer code

| | |
|---|---|
| Data collection | Image analysis and quantification was performed using Fiji ImageJ (v.1.53c). scRNA-seq data was generated from single-cell suspensions from E90, P3, P10, and 6mo pig skin using 10x Genomics scRNA-seq 3' V3 kit and sequenced on an Ilumina NovaSeq PE150 by Novogene. Stereo-seq data was generated from P3, P10, and 6mo fixed-frozen 10um cryo sections using Complete Genomics Stereo-seq T FF V1.2 kit and sequenced using a DNBSEQ-T7. |
| Data analysis | scRNA-seq Fastq files were aligned to the Sscrofa.11.1 genome assembly using 10x Genomics Cellranger (v6.0.0). Cellranger outputs were used in downstream analysis. Stereo-seq Fastq files were aligned to the Sscrofa.11.1.112 genome release from Ensembl using STOmics SAW (8.1.1) and SAW outputs were used in downstream analysis. Raw and processed data is deposited on GEO as described in our Methods. scRNA-seq data analysis was performed in R (v4.2.1) using the Seurat (v0.3.1), SeuratWrapper (v0.4.3), Monocle3 (v1.3.1), and CellChat (v1.4.0) packages. Statistical analyses of quantifications was analyzed in R (v4.2.2). Source code and more detailed software package and dependency versions is available on our Github page: https://github.com/DriskellLab/Thompson-et-al.-2025 . For Stereo-seq spatial transcriptomics, SAW (8.1.1) was utilized with Sscrofa11.1.112 release with Ensembl and makeRef Function. Tissue mask utilized Stereo Map 4. SAW realign Visualization output.gef files were loaded into and analyzed using Stereopy 1.5.0 in Python 3.8.20.<br>CellChat analysis was used to infer pathway and ligand-receptor interactions among core basal and dividing keratinocyte, papillary fibroblast, pericyte, and blood vessel clusters from porcine skin scRNA-seq datasets. Similarly, Spatial CellChat was used to infer pathway and ligand-receptor interactions from porcine Stereo-seq datasets.<br>Adobe Illustrator (v. 2021, 2023, 2025) and Adobe Photoshop (v. 2021, 2023, 2025) were used as described in the methods section. |

For manuscripts utilizing custom algorithms or software that are central to the research but not yet described in published literature, software must be made available to editors and reviewers. We strongly encourage code deposition in a community repository (e.g. GitHub). See the Nature Portfolio guidelines for submitting code & software for further information.

## Data

Policy information about availability of data

All manuscripts must include a data availability statement. This statement should provide the following information, where applicable:
- Accession codes, unique identifiers, or web links for publicly available datasets
- A description of any restrictions on data availability
- For clinical datasets or third party data, please ensure that the statement adheres to our policy

We have made all the analyzed scRNA-seq and Stereo-seq datasets in this study available via our interactive webtools on skinregeneration.org (https://skinregeneration.org/papers/Thompson-et-al-2025/). Raw and processed sequencing data files used in our analysis can be found at the NCBI's Gene Expression Omnibus (GEO accession number: GSE305111. Data underlying plots in this manuscript are fully available as Source Data, and exact p-values are provided in figure legends or in Source Data where there is insufficient space to include in the legend. All other data is available either as supplementary information or from the corresponding author upon reasonable request.

## Research involving human participants, their data, or biological material

Policy information about studies with human participants or human data. See also policy information about sex, gender (identity/presentation), and sexual orientation and race, ethnicity and racism.

| | |
|---|---|
| Reporting on sex and gender | All human tissue samples were obtained de-identified with basic age and sex data only. Both male and female samples were analyzed whenever possible. |
| Reporting on race, ethnicity, or other socially relevant groupings | Our samples were obtained de-identified without these types of data. |
| Population characteristics | Adult human tissues were obtained as surgical discard tissue from dermatology clinic procedures. Samples were grouped into the following categories 21-39 years old, 40-59 years old, 60-79 years old, and 80+ year old with most samples falling into the 60-79 years old category. Gestational human tissue samples ranging from gestational week 7 up to 20 were obtained by the University of Washington Birth Defects Research Laboratory following maternal consent. A mix of males and females were used in the study whenever possible. |
| Recruitment | Adult subjects were already undergoing a surgical procedure at Advanced Dermatology, Spokane, WA and surgical discard tissue was collected with informed consent in accordance with Washington State University IRB-approved protocols (Washington State University #19796). Gestational human tissue samples ranging from gestational week 7 up to 20 were obtained by the University of Washington Birth Defects Research Laboratory with maternal consent and in accordance with University of Washington IRB-approved protocols (University of Washington STUDY00000380). De-identified samples were received at Washington State University in accordance with Washington State University IRB-approved protocols (Washington State University #19680). Complete details are provided in methods. |
| Ethics oversight | Washington State University IRB (#19796, 19680), University of Washington IRB (STUDY00000380). Complete details are provided in methods. |

Note that full information on the approval of the study protocol must also be provided in the manuscript.

# Field-specific reporting

Please select the one below that is the best fit for your research. If you are not sure, read the appropriate sections before making your selection.

☒ Life sciences  ☐ Behavioural & social sciences  ☐ Ecological, evolutionary & environmental sciences

For a reference copy of the document with all sections, see nature.com/documents/nr-reporting-summary-flat.pdf

# Life sciences study design

All studies must disclose on these points even when the disclosure is negative.

| | |
|---|---|
| Sample size | No sample size calculations were performed. A minimum of 3 or more biological replicates, and a mix of males and females, were obtained when possible, however some timepoints within our human or pig sample sets only consisted of 1-2 biological replicates due to model limitations (e.g. initial pig litter size), which are clearly delineated in the figure legends. In mouse experiments, multiple litters and a mix of male and female individuals were used when possible with specifics included in the figure legends or methods section. In pig experiments, multiple litters were used when possible with specifics included in the figure legends or methods section. Sample sizes in this study are comparable to other studies in our field. |
| Data exclusions | No data was excluded |
| Replication | Sample collection consisted of 3 or more biological replicates when possible to ensure reproducibility within a given condition. Immunostaining was replicated by 3 or more biological replicates when possible to ensure consistency and reproducibility. Some timepoints |

of human or pig samples only had 1-2 replicates due to difficulties in obtaining samples from these organisms, and these are explicitly stated in the Figure Legends or Methods sections. Experiments were repeated more than once when possible. Certain wounding experiments were only performed once but included multiple biological replicates. All replication attempts were successful when experiments were repeated. Complete details for each experiment are included in the figure legends as word limits permit or in the methods section.

**Randomization**

In mouse treatment experiments (tamoxifen induction of epidermal BMPR1A knockout), K14CreERT Bmpr1a fl/fl genotypes were treated with tamoxifen and used as "TAMX" (induced knockout) group while K14CreERT genotypes were treated with tamoxifen and used as control group (induced no knockout). For porcine wound healing and BrdU-labeling experiments, negative control animals and animals selected for collection at discrete timepoints (e.g. 28 days post wounding for wound experiment or 1 day post injection for BrdU-labeling) were chosen randomly.

**Blinding**

Researchers were not blinded to experimental conditions (e.g. genotype) when performing quantifications on images. Many quantifications in this manuscript were exploratory or temporal in nature (e.g. quantifying epidermal thickness or rete ridge density over developmental time) rather than comparisons between experimental groups, which has lower risk of subjective interpretation. For experimental comparisons, many phenotypes in transgenic mouse studies rendered it impossible to "blind" due to dramatic morphological differences: K14-Noggin and tamoxifen-treated K14CreERT Bmpr1a fl/fl mice have major digit tip phenotypes due to the role of BMP signaling in nail or phalanx development and homeostasis which make blinding genotype impossible compared to controls. Similarly Lef1-eKO back skin and digits are also impossible to blind due to their dramatic hair reduction. Similarly, EDA-KO pigs have a hair phenotype which is discernable in histology compared to age-matched wild-type pigs (Ostedgaard et al. 2020. Elife). Finally, statistical analyses were performed after full collection of data from experimental groups and controls. Statistical comparisons between species in Figure 2c were performed multiple times using identical statistical methodology (one-way ANOVA) over the course of the manuscript revisions since additional biological replicates for some species were added between the original submission and the final revision. Critically, all quantifications utilized multiple technical replicates (e.g. different tissue sections or different skin samples collected from the same organism) which were averaged to determine the value of each biological replicate. In all studies, one biological replicate represented one distinct animal. For porcine wound healing and BrdU-labeling experiments, the surgeon or injector, respectively, was blinded to the eventual collection timepoint of the animal.

# Reporting for specific materials, systems and methods

We require information from authors about some types of materials, experimental systems and methods used in many studies. Here, indicate whether each material, system or method listed is relevant to your study. If you are not sure if a list item applies to your research, read the appropriate section before selecting a response.

## Materials & experimental systems

| n/a | Involved in the study |
|---|---|
| ☐ | ☒ Antibodies |
| ☒ | ☐ Eukaryotic cell lines |
| ☒ | ☐ Palaeontology and archaeology |
| ☐ | ☒ Animals and other organisms |
| ☒ | ☐ Clinical data |
| ☒ | ☐ Dual use research of concern |
| ☒ | ☐ Plants |

## Methods

| n/a | Involved in the study |
|---|---|
| ☒ | ☐ ChIP-seq |
| ☒ | ☐ Flow cytometry |
| ☒ | ☐ MRI-based neuroimaging |

## Antibodies

**Antibodies used**

Human-ITGA6 rat (1:200, BD Biosciences, Catalog # 555735, clone GoH3, lot # 8136528, 3055226, 1214287, 1033227), human LEF1 rabbit (1:200, Cell Signaling, Catalog # 2230S, clone C12A5, lot # 8, 9), human-aSMA rabbit (1:1000, Abcam, Catalog # ab5694, clone proprietary, lot # 1038192-2), human-PDGFRA goat (1:250, R&D Systems, Catalog # AF307NA, clone P16234, lot # VG0721111) in pig and marmoset samples, mouse-PDGFRA goat (1:250, R&D Systems, Catalog # AF110, clone P26618, lot # HMQ0222061) in mouse samples, human-KRT10 rabbit (1:250, Dennis Roop), human-KRT15 chicken (1:250, BioLegend, Catalog # 833904, clone Poly18339, lot # B353424, B404683), mouse-SOX9 rabbit (1:1000, EMDMillipore, Catalog # AB5535, clone P48436 C-term, lot # 3677685), human-MKI67 rabbit (1:400, Cell Signaling, Catalog # 9129S, clone D3B5, lot # 3, 9), BrdU rat (1:200, Abcam, Catalog # ab6326, clone BU1/75 (ICR1), lot # 1009715-43, 1009715-48), human-KRT15 rabbit (1:200, Sigma, Catalog # HPA024554, clone APREST75794, lot # A119308), human-SMAD1 goat (1:200, R&D, Catalog # AF2039, clone Q15797, lot # KOE062503A), human-pSMAD1/5 rabbit (1:200, Cell Signaling, Catalog # 9516, clone 41D10, lot # 10), human-KRT14 rabbit (1:250, Dennis Roop), human-KRT14 mouse (1:1000, R&D, Catalog # mab3164, clone LL001, lot # WEY0823012), human-PDGFC goat (1:200, R&D, Catalog # af1650, clone Q9NRA1, lot # JDI022406A), mouse-PECAM1 rat (1:200, Thermofisher, Catalog # 12-0311-82, clone 390, lot # 1989060, 3095164), mouse-PDGFC rat (1:200, R&D, Catalog # mab1447, clone Q8Cl19, lot # HYQ022406A). Secondary antibodies used were AlexaFluor (AF) 488 anti-rat (1:1000, Fisher, Catalog # A21208, clone AB_2535794, lot # 2482958, 2668657, 2180272), AF488 anti-chicken (1:1000, Fisher, Catalog # A11039, clone AB_2534096, lot # 1899514, 2941307), AF488 anti-rabbit (1:1000, Fisher, Catalog # A21206, clone AB_2535792, lot #1874771), AF Plus 555 anti-rabbit (1:1000, Fisher, Catalog # A32794, clone AB_2762834, lot # VK307588, VD297829), AF555 anti-rabbit (1:1000, Fisher, Catalog # A31572, clone AB_2535849, lot # 2831376, 2482963), AF555 anti-goat (1:1000, Fisher, Catalog # A21432, clone AB_2535853, lot # 1878842, 2400919), AF Plus 555 anti-rat (1:1000, Fisher, Catalog # A48270, clone AB_2896336, lot # WF333067, ZG398235), AF647 anti-rabbit (1:1000, Fisher, Catalog # A31573, clone AB_2536183, lot # 2544598), AF647 anti-goat (1:1000, Fisher, Catalog # A21447, clone AB_2535864, lot # 1841382, 2297623). DAPI 300uM stock (1:1000, BioLegend, Catalog # 422801, lot # B222486, B324682) was used alongside secondary antibodies.

**Validation**

We validated antibodies based on known protein localization from ours and previous studies:
Human-ITGA6 rat (BD Biosciences): manufacturer validated in flow cytometry of human, mouse, pig, and dog cells. We and others have previously utilized this antibody for immunofluorescence staining of tissue sections (DOIs: 10.1016/0014-4827(90)90277-h,

10.1177/34.8.2426332, 10.7554/eLife.60066). Human-LEF1 rabbit (Cell Signaling): manufacturer validated in western blotting, immunofluorescence, and flow cytometry of human, mouse, and rat cells. We and others have previously used this antibody for immunofluorescence staining of tissue sections (DOI: 10.7554/eLife.60066). Human-aSMA rabbit (Abcam): manufacturer validated in western blotting, immunofluorescence of human and mouse cells. We and others have previously utilized this antibody for immunofluorescence staining of tissue sections (DOI: 10.7554/eLife.60066). Human-PDGFRA goat (R&D): manufacturer validated in western blotting and immunohistochemistry of human cells (DOI: 10.1016/j.jid.2018.01.016). Mouse-PDGFRA goat (R&D): manufacturer validated in western blotting, immunofluorescence, flow cytometry, and immunohistochemistry. We and others have previously utilized this antibody for immunofluorescence staining of tissue sections (DOI: 10.7554/eLife.60066, 10.1016/j.jid.2018.01.016). human-KRT10 rabbit (Dennis Roop): previously utilized for skin cross section immunofluorescence (DOI: 10.1242/jcs.03298). Human-KRT15 chicken (BioLegend): manufacturer validated for western blotting and immunohistochemistry and published immunofluorescence staining in mouse skin tissue sections (DOI: 10.1038/s41467-022-30800-y). Human-SOX9 rabbit (EMDMillipore): manufacturer validated for ChIP, immunocytochemistry, immunohistochemistry, immunofluorescence, and western blotting of chicken, human, rat, mouse, and predicted for bovine, sheep, feline, equine with some literature validation (DOI: 10.1095/biolreprod.116.139832, 10.1038/s41467-019-10596-0). Human-MKI67 rabbit (Cell Signaling): manufacturer validated immunofluorescence (frozen), immunofluorescence (immunocytochemistry), flow cytometry of human, mouse, and rat cells. We and others have previously utilized this antibody for immunofluorescence staining of tissue sections (DOI: 10.1016/j.devcel.2022.06.005). anti-BrdU rat (Abcam): manufacturer validated for immunohistochemistry, flow cytometry, immunofluorescence (immunocytochemistry) of incorporated BrdU (species agnostic). This antibody has previously been utilized to identify BrdU incorporation in skin cross sections (DOI: 10.1242/dev.064592). Human-KRT15 rabbit (Sigma): manufacturer validated for immunoblotting, immunofluorescence, and immunohistochemistry in humans (DOI: 10.1016/j.devcel.2020.01.033). Human-SMAD1 goat (R&D): manufacturer validated for western blotting and immunohistochemistry (DOI: 10.2337/db17/1043). Human pSMAD1/5 (Cell Signaling): manufacturer validated for western blotting, immunofluorescence (immunocytochemistry), and flow cytometry in humans, mice, and rats. This antibody has previously been used to stain skin sections (DOI: 10.1038/ncb3535; 10.1038/s41467-019-09402-8). Human-KRT14 rabbit (Dennis Roop): previously utilized for skin (DOI: 10.1242/jcs.03298). Human-KRT14 mouse (R&D): manufacturer validated for western blotting, immunocytochemistry, immunohistochemistry in humans (DOI: 10.1016/j.devcel.2022.05.003). Human-PDGFC goat (R&D): manufacturer validated for western blotting and immunohistochemistry in humans. Previously utilized with human cell in vitro (DOI: 10.18632/oncotarget.18706). Mouse-PECAM1 rat (Thermofisher): manufacturer validated for flow cytometry in mouse and published in C. elegans, fish, hamster, human, and mouse (DOI: 10.1038/s41598-018-38366-w). Mouse-PDGFC rat (R&D): manufacturer validated for western blotting in mice and does not cross-react with human-PDGFC (DOI: 10.1074/jbc.M111.222513).

# Animals and other research organisms

Policy information about studies involving animals; ARRIVE guidelines recommended for reporting animal research, and Sex and Gender in Research

| | |
|---|---|
| Laboratory animals | Mice (C57BL/6 mixed background ranging in age from P0 to adults older than 1-year, housed in a climate controlled facility set to ~68-73 degrees Fahrenheit and ~40% humidity. TEMP and humidity, with a 12-hour light/dark cycle with food and water ad libitum). Naked Mole Rat (Heterocephalus glaber), Common Marmoset (Callithrix jacchus), North American Grizzly Bear (Ursus arctos horribilis), Rhesus Macaque (Macaca mulatta), Pig (Sus scrofa) housed in a climate controlled facility ranging ~70-80 degrees Fahrenheit and ~30-50% humidity in a 12-hour light/dark cycle. |
| Wild animals | The study did not involve wild animals. |
| Reporting on sex | Sex is indicated where applicable by colored dots in graph. We made every effort to include equal numbers of each sex when possible. |
| Field-collected samples | No field-collected samples were used in this study. |
| Ethics oversight | Adult naked mole rats (Heterocephalus glaber) were maintained at the University of Texas Health Science Center at San Antonio for unrelated studies performed under protocols approved by the University of Texas Health Science Center at San Antonio IACUC (#20210034AR).<br><br>Rhesus macaques (Macaca mulatta) were maintained at the Oregon National Primate Research Center (ONPRC) at Oregon Health and Science University for unrelated studies performed under protocols approved by Oregon Health and Science University IACUC (IP03716, IP03276, IP00367). Additionally, the ONPRC is accredited by the Association for Assessment and Accreditation of Laboratory Animal Care (AAALAC; Animal Welfare Assurance D16-00195) and registered with the USDA (#92-R-001). Rhesus macaque samples were shared through the ONPRC tissue distribution program.<br><br>Common marmosets (Callithrix jacchus) were maintained at the Southwest National Primate Research Center at Texas Biomedical Research Institute for unrelated studies performed under an approved animal use protocol (Assurance Number D16-00048).<br><br>North American grizzly bears (Ursus arctos horribilis) were housed at the Washington State University Bear Research, Education, and Conservation Center. Bears were anesthetized and biopsies collected for unrelated studies performed under Washington State University IACUC-approved protocols (#6546).<br><br>Mice (Mus musculus) used in this study were of a mixed wild-type C57BL/6 background housed at Washington State University. Mouse housing and skin sample collection were in accordance with Washington State University IACUC-approved protocols (#6723, 6724, 6930).<br><br>Gestational human tissue samples were obtained by the Birth Defects Research Laboratory at University of Washington under University of Washington IRB-approved protocols with maternal written consent (University of Washington STUDY00000380; Washington State University 19680). |

Adult human tissue samples were obtained by Advanced Dermatology in Spokane, Washington as surgical discard tissue after informed consent and in accordance with Washington State University IRB approved protocols (19796). All human tissue samples were deidentified prior to receipt at Washington State University and were analyzed in accordance with Washington State University IRB-approved protocols (19680, 19796).

Pigs (Sus scrofa) were housed at Washington State University under approved IACUC and USDA protocols (6492). Some fetal pigs were obtained from Biology Products (Alexandria, MN) and were exempt from Washington State University IACUC approval. Embryonic day 90 fetal pigs were harvested post-mortem from a pregnant pig with known date of conception under WSU IACUC-approved protocols (6492). Postnatal pig skin samples were collected post-mortem from pigs either housed at Washington State University, obtained from local farmers, or collected post-mortem from local butchers in accordance with Washington State University IACUC-approved protocols (6492). Yucatan Hairless pig and Hanford mini pig tissue samples were received from Sinclair Biosciences (Auxvasse, MO) and Mangalitsa pig tissue samples were obtained following butchering and were exempt from Washington State University IACUC approval. EDA-KO pig samples were collected by the Welsh and Ostedgaard Labs at University of Iowa (Ostedgaard et al., 2020) under University of Iowa IACUC-approved protocols (3071121). All experiments followed relevant guidelines and regulations from the appropriate ethics committees, as detailed above.

Bottlenose dolphin (Tursiops truncatus), long-beaked common dolphin (Delphinus capensis), and short-beaked common dolphin (Delphinus delphis) skin used in this study were obtained by the Plikus Lab at the University of California, Irvine, from NOAA (Southwest Fisheries Science Center, La Jolla, California) under the destructive loan permit. The analyzed specimens include: Delphinus capensis (numbers KXD0225, KXD0226, 1741-2023 Dc2301B), Delphinus delphis (numbers BLH0012, KXD0357, 585-2022 Dd2202B), and Tursiops truncatus (numbers KXD0410, KZP0069, 812-2022 Tt2202B).

K14CreERT Bmpr1a fl/fl tamoxifen-treated and control mice used in this study were housed at the University of Warsaw. The animal studies were approved by the First Local Ethics Committee: No. 971/2020 as of 28 January 2020, No. 1669/2025 as of 18 March 2025.
3mo WT and K14-Noggin mice were analyzed from archived paraffin-embedded digit samples (Plikus et al. 2004) as described in the Methods, and no new animal experiments were performed.

Note that full information on the approval of the study protocol must also be provided in the manuscript.

# Plants

| | |
|---|---|
| Seed stocks | Not applicable. |
| Novel plant genotypes | Not applicable. |
| Authentication | Not applicable. |

