## [Peer Review File · Nature]

Rete ridges form via evolutionarily distinct mechanisms in mammalian skin

Corresponding Author: Dr Ryan Driskell

Version 1:

Reviewer comments:

Referee #1

(Remarks to the Author)

This paper examines the formation of epidermal ridges in the skin, employing a comparative biology approach that analyzes ridge characteristics across various species including humans, pigs, mice, and others. By assessing factors such as hair density, presence of ridges, and epidermal thickness, the authors propose a multi-part model for rete ridge formation. They conducted scRNA analysis on pig wounds and comparing their findings with existing literature. While molecules implicated in ridge formation were identified through correlation and comparative CellChat analyses, their roles were not confirmed through expression or functional studies. In an experimental approach, retinoic acid and wounding agents were administered to the hairless scars of mice, resulting in the formation of structures resembling rete ridges. However, without molecular characterization, it is challenging to definitively classify these structures as rete ridges. While the study offers valuable comparative skin findings, it lacks in elucidating the mechanism of ridge formation or the evolutionary origins of these ridges. As a result, the title may appear exaggerated and should be modified to accurately reflect the scope and findings of the study. The data does hold potential and authors can improve upon the following points.

1. Comparative studies of epidermal ridges.

- a. Epidermal ridges represent an intriguing and relatively understudied topic in the field of skin biology. A notable recent study by the Headon group has shed light on this subject. In this paper, the authors have compiled new data from various species including pig, human, mouse, as well as additional data on monkeys and bears. While their comparative approach has amassed a large amount of data for analysis, the nature of the presented data remains primarily descriptive and correlative.
- b. It is already widely recognized that non-furry animals tend to have more well-developed rete ridges. While the authors present some more intriguing examples involving bears and monkeys, they did not delve into the mechanistic issue of whether there exists a developmental constraint influencing the formation of both hair and ridges.

2. Apical ridges

- a. The initial discussion of apical ridges and dermatoglyphics in the paper led me to anticipate new insights into the relationship between rete ridges and apical ridges. However, the data pertaining to apical ridges is unclear. They claim increased length of apical ridges in the adulthood but did not present clear evidence or histological support. Furthermore, the authors did not adequately characterize the relationship between hair follicle density and apical ridge formation. There is a lack of systematic description regarding apical ridge formation.
- b. In Figure 4, the authors mention the regeneration of apical ridges, but it is not clearly demonstrated. Please clarify this aspect.

3. Molecular data:

- a. It is commendable that the researchers conducted single-cell RNA sequencing (scRNA) of pig skin. They then conducted comparative analyses with scRNA data from human and mouse skin available in the literature. Through CellChat analyses, they highlighted the presence of PDGFC and PDGFRA in pig and human skin but not in mouse skin. This finding presents a promising clue. However, the absence of further validation experiments is a notable limitation. Specifically, the lack of PDGFC expression demonstrated in tissue sections is a significant gap. If indeed blood vessels are the driving force behind ridge formation, it would be crucial to visualize the periodic distribution of PDGFC in the epidermal layer to establish the

ridge pre-pattern effectively.

b. ScRNA data is not thoroughly studied. Spatial localization of tissue sections for cell clusters from scRNA analyses are not done. Trajectory lines are not presented in a clear and readable manner.

c. The use of the EDA knockout pig to illustrate ridge formation is a positive aspect of the study. It effectively demonstrates that EDA is not essential for rete ridge formation and emphasizes the distinction between rete ridges and hair as different types of skin appendages.

d. Timing of the specimens. The authors ought to delve into the initial stages of ridge formation. They note that rete ridges form at P2 and mature by the second week (P10 in this study). However, in their subsequent single-cell RNA sequencing (scRNAseq) analysis, the authors utilized the P10 specimen to represent the forming stage of rete ridge development, which may be too advanced to be characterized as the "forming" stage. From my reading, it appears that GW16 or the neonatal stage (P2) would be more representative of the forming stage, as depicted in Fig 1d.

4. Mechanism: Multi-part model,

a. This aspect of the study is not convincing. The authors mention the occurrence of thicker epithelia with rete ridges, but did not provide strong evidence for a causal relationship.

b. Rete ridges in adult human has been well studied, with scientists showing the periodic distribution of epidermal progenitor cells in the ridge and inter-ridge regions. Authors could have followed this clue to show how progenitor and differentiated cells are distributed in other animals. Although specimens from different animals present a valuable opportunity to gain insight, this aspect of the study remains underdeveloped.

c. In the multi-part model, the authors propose that ridge formation involves both increased epidermal thickness and vascular recruitment. However, it remains unclear which factor serves as the primary driving force. In one section subtitle suggests that rete ridge formation drives epidermal thickness and dermal vascular recruitment. Yet in Figure 7 indicates that vasculature is the driving force. The question arises: which is the earlier determinant of the rete ridge formation process, the dermal pocket, or the rete ridge structure? Clarifying the earlier stages of ridge formation, as mentioned in point 3d above, may provide greater clarity on this matter.

d. Given that rete ridges exhibit periodic arrangement, it raises the question of what factors control the periodic outgrowth of vasculature. The authors have identified PDGFC as a potential molecule involved in this process. Therefore, this presents an opportunity to conduct further testing and investigation to validate the role of PDGFC and explore its mechanisms in regulating the periodic outgrowth of vasculature within rete ridges.

5. Engineered rete ridges:

a. The authors could have bolstered their study by providing additional molecular evidence to confirm that the "engineered rete ridges" are indeed authentic rete ridges, distinct from merely an uneven epithelial undersurface, as commonly observed in mouse wound healing models.

b. The concurrent emergence of vasculature and thickening of the epidermis also have been observed during "normal" wound repair.

6. Writing

The writing is dense and narrative in nature. There are many long sentences. These can be improved to show the message clearer.

(Remarks on code availability)

Referee #2

(Remarks to the Author)

In the manuscript entitled "Evolutionary development, regeneration, and induction of rete ridges in mammalian skin", the authors employed a multi-species comparative approach to investigate the timing and mechanisms of the development and regeneration of rete ridges (RRs) in the mammalian skin. Through histological and molecular analyses, they found that RRs form prenatally through a molecular mechanism distinct from that of hair follicles, sweat glands, and fingerprints. In a comparative study across multiple mammalian species, the authors discovered a correlation between RR formation and an increased epidermal thickness, coupled with a decrease in hair density. Utilizing comparative transcriptomics analyses, the authors identified several signal pathways that potentially contribute to RR development. Additionally, the authors characterized the process of RR regeneration after wound in neonatal pig, as well as devised an approach to induce RR-like structures in juvenile mice through wound and pharmacological treatment. The authors have established powerful comparative models and generated potentially interesting datasets. However, this manuscript lacks functional test data to pinpoint the mechanisms that drive RR development or regeneration, and some conclusions lack support from the current data.

Major:

1) The proposed mechanisms lack experimental support.

1.1) Through transcriptomics analyses, the authors identified candidate signaling pathways that potentially contribute to RR formation. However, as no RR-specific subpopulation was identified and the analyses were conducted on general epidermal cells, it remains uncertain whether the candidate signals function in regulating RR formation. No functional test was performed to demonstrate the requirement of these signals for RR development or regeneration. Furthermore, the statement that these signals "are responsible for increased epidermal thickness, decreased hair density, and enhanced dermal

vasculature recruitment between rete ridges" (line 38-40) are not supported by experiments.

1.2) The statement that dermal vasculature recruitment is required for RR formation (line 44, 340, 408) lacks experimental support. No vasculature-specific experiments were conducted to test the requirement of vascular recruitment for RR development or regeneration. The assessment of vasculature distribution was qualitative (line 309-314). The statement that mouse skin is less vascularized than pig lacks support through quantification. Analyzing vasculature distribution on a single section and claiming sparser presence of dermal vasculature outside of dermal pocket is also not appropriate. A 3D whole mount analysis is more suitable.

1.3) It is interesting that the authors were able to induce RR-like structures in mice. Despite their similar morphology, the molecular characteristics and functions of these structures compared to RR remain unclear. Even if these structures are equivalent to RR, the various effects of TPA and RA on epidermal cells and other cell types coupled with the intricate wound-induced signals and inflammatory environments, make it difficult to pinpoint the mechanisms driving their formation.

1.4) No new cellular mechanisms were elucidated. The speculations that "the epidermal thickness in rete ridge-bearing skin increases due to an expanded differentiated epidermis without an apparent expansion in the basal layer" (line 287-289) mediated by RA and RA supports de-differentiation (line 404) appear contradictory and were not tested.

2) The authors sampled skin from pigs at different ages to conclude that RR forms perinatally. However, this finding is not novel, given that the presence of RR in perinatal human skin is already established based on foreskin analyses. The authors did not further refine the time window.

3) Regeneration of RRs after wound has been observed and characterized in young pigs (Kuo, *Sci Rep* 2022: PMID: 35013386; Lin, *Exp Dermatol* 2019: PMID: 30629757). Hence, the discovery of RR regeneration in neonatal pigs is not novel. Furthermore, the time window for the initiation of RR regeneration was not captured, as RRs had already formed at the first time point (28dpw) of analysis.

4) It is not appropriate to perform pseudotime analysis on the entire population of keratinocytes to understand transcriptional dynamics during epidermal differentiation (line 207-208, 217-218 and Fig. 3e, 3h). This is because the cells in the basal buds will generate skin appendages and will not contribute to the differentiated epidermal cells.

Minor:

5) Based qualitative assessment, the authors concluded that RR development was unperturbed in EDA-KO pigs. However, the CRISPR/Cas9 mediated knockout may not entirely eliminate EDA's function, and minor phenotypes might exist. Therefore, a quantitative analysis is needed to strengthen the conclusion.

6) The authors claimed that RR is a novel class of epithelial appendage (line 227, 459, 508). However, the current data does not exhibit substantial differences in function between RR and non-RR epidermal cells. Therefore, RR appears more akin to a domain or structural unit of the epidermis rather than an independent appendage. The authors might consider changing the language.

7) Through molecular analysis and genetic manipulation, the authors demonstrated that RR formation is unlikely to be controlled by EDAR and Wnt pathways. However, stating that RR is not formed through basal budding (line 197-198, 212, 225-226, 375, 428-430) is inappropriate, as the authors have not ruled out the possibility that at the cellular level, RR is formed through budding from the epidermis. The authors might consider changing the language.

Minor

The anti-correlation between hair density and epidermal thickness was not correctly stated (line 183).

Multiple panels of Fig. 4 were not correctly cited (line 242-250).

It seems the word "also" was mistakenly added in line 274.

Fig. 7h is mislabeled and not cited anywhere.

(Remarks on code availability)

Referee #3

(Remarks to the Author)

The importance and function of Rete ridges (RR) in human skin has puzzled the skin and stem cell fields since decades. It is assumed that the RR are important for maintaining a high regenerative potential of the skin, but more general functional roles of RR remain elusive. To date, it is entirely unknown when (during skin development) RR develop, if they are distinct from fingerprint ridges, and which cellular and molecular mechanisms drive their establishment. This study, Thompson et al, investigate these intriguing questions and provide very exciting answers, such as RR development is molecularly entirely different from other skin appendages (hair follicles, sweat glands, and fingerprint ridges), RR develop perinatal (just after birth), and the presence of RR across the mammalian kingdom suggest that RR have a protective function in skin with reduced hair density (mouse vs grizzly vs pig and human skin). The latter notion also may help to answer the ever-recurring questions (not the why but the how): why did humans become (largely) hair less and how did they compensate for it?

The authors use an impressive array of mammalian species, analysis methods, and computations for their study. In addition to the novelty in skin/tissue biology, the study also provides a comprehensive single-cell transcriptomics resource of pig skin, which is highly valuable. There is no doubt that this study is novel and of high interest to a wide general readership (such as *Nature*), and I am certainly in support of this elegant and innovative study. However, the authors will see by the comments below, there is still much work needed to cement their claims and to dive more into the cellular and molecular

mechanisms underlying RR formation.

Major comments and questions:

(1) Fig 1c has very low sample numbers for some timepoints and ideally all should have more than $n=1$. Yet, as it may be prohibitively difficult to increase the fetal sample number, and they show consistent results (notably, GW16-17 and GW18-20 have $n=7$ and $n=5$, respectively), those may be fine. However, it is important to include more samples in the 21-39yo group as it forms the first timepoint of the data with RR existence. A higher sample number will also clarify if the decrease of RR numbers between young adult to 60/80yo is significant, which would be expected based on literature showing that epidermis flattens with age (eg: doi:10.1016/j.mad.2016.03.006; doi:10.3390/biom10121607; doi:10.1039/C5IB00238A). Interestingly however, Fig1c seems to show no clear reduction in epidermal thickness. Could the authors comment on it, do we see this discrepancy because of the chosen body site(s) or the number of samples measured?

(2) Fig2: the authors claim that RR are associated with general epidermal thickness and reduced hair density. Does this also apply to human skin across body sites? For example, scalp skin has a higher hair density compared to forearm or abdominal skin; is the IR and RR thickness altered across those sites? If not, it would be valuable to provide a statement in the discussion. Related, Fig2b,d: It would be interesting to see the data for the Yucatan Hairless and Mangalitsa pig in all analyses.

(3) Line 148/149: please add a muscle staining. How do the authors know it's the PCM and not any other deeper back muscle like in mouse (see doi:10.1016/j.devcel.2023.07.015)? Related minor note: the statement of PCM not being present in human skin does not apply to all body sites in human skin but would apply to eg. back skin.

(4) L202/Fig3: Please specify the genes that were used for the characterization and show key genes for the classification as eg. dotplots. K14 in mouse marks exclusively basal layer cells, whereas in human skin it is expressed in basal and several suprabasal layer(s). In human skin, K15 seems a better marker for basal cells (doi: 10.15252/emboj.2021110488), and considering the similarity of pig and human skin, you may consider checking K15 expression as an alternative. After this check, if you still think K14 is a valid marker for basal cells, please provide a K10/K14 stainings in pig skin.

(5) Fig 3i: At least in the mouse, only certain HF types depend on EDAR signalling (eg: guard or tail HFs). The shown pictures of the EDAR-KO pigs do have (fewer?) hair follicles, which means that there is still some basal budding ongoing. Although I am convinced (eg from scRNA-seq analysis and stainings) that RR form via a different mechanism than HFs or fingerprint ridges, showing the EDAR-KO data for the purpose of making the case that RR do not form via basal budding is not convincing; it shows that EDAR signalling is not needed which is different to not forming via basal budding.

(6) L233-235: The authors state: "While some fetal and neonatal mammals, such as mice, have the potential to regenerate hair follicles during wounding, it is unknown whether neonatal pigs share this intrinsic regenerative capacity. Since we determined that the neonatal developmental window is when rete ridges form and that this window appears unsupportive of hair follicle or sweat gland formation, we hypothesized that neonatal pig wounds would be capable of regenerating rete ridges but not hair follicles and sweat glands since this is the same timeframe as when rete ridges are forming." I.e. The authors say, and its well known, that wounding allows de novo hair follicle formation, but then continue to say that they would not expect HFs to form in wounds of pigs but perhaps RR – which as stated does not make sense. I understand that the authors wanted to emphasize that in healthy settings, RR form neonatally and HFs/SGs form during embryogenesis. But assuming that only RRs would form in postnatal healed wounds would only make sense to me if the authors first refer to previous work or own data showing that de novo HF/SG formation has not been observed in neonatal wounds of pigs, while the formation of RRs has not yet been looked at.

(7!) The authors often state that RR form via a novel mechanism, distinct from HF/SG/fingerprint ridges. However, did the authors eventually identify unique RR marker genes, distinct of IR gene expression? Or is the proposed new mechanism rather that RR utilize the 'normal IFE differentiation' process? I.e. could the authors elaborate more on the development of RRs in healthy skin (Fig3). Are there any differences in cell proliferation and differentiation between the RR and IR areas? And similarly, how do these aspects look like during the initial stages of RR development (e.g., p0 – p5 pig epidermis), is there higher proliferation in the areas of future RRs? Connected to the development of RRs, can dense vasculature already be observed in the early (before P5) dermal pockets or even before the initial formation of RRs (also see following comments)? Or could there also be other factors, such as nerves, involved in the process?

(8) Connected to above: The authors have compiled an impressive scRNA-seq dataset from existing sources, as well as including their own samples. Can the authors separate the epidermal RR and IR areas based on the transcriptomic data? If possible to identify these areas, do they show differences in their proliferative and differentiation capacities? Which epidermal clusters are most involved in angiogenesis, are the IR keratinocytes secreting more of these factors than RR keratinocytes? Light-seq (<https://doi.org/10.1038/s41592-022-01604-1>) or new generation high-resolution spatial sequencing could be viable strategies to separate IR and RR and compare them.

(9) L302: "We hypothesized that epidermal-dermal interactions support dermal vascular lineage recruitment into the inter-ridge space, and that the proximity of these dermal cell lineages, including fibroblasts, pericytes, and blood vessels to the basal epidermis provides a parallel mechanism that jointly supports increased epidermal thickness." This statement implies that the authors think RR formation precedes blood vessel dermal pocket establishment. Could it be the other way around? Vessels and/or nerves could be there first, and thus provide a distinct niche environment (signalling gradients/pattern in dermis) that induces RR formation (eg. Fig6 shows at E90 vessels touching the epidermis). It is noted from results in Fig6

that presence of RR does increase signals supporting vascularization, but this does not answer whether RR formation itself precedes or follows primary vascularization.

(10) In Fig4, the authors describe the formation of RRs in healed wounds. At p5+28dpw one can see that the RRs are already formed, but how does an earlier wound timepoint look like? Do RR start forming as soon as parts of the wounds are re-epithelialized or do they start forming only when the wounds are completely closed with a flat epithelium? Are there differences in RR maturation (size, density) between wound edges and wound fronts? As the vasculature seems important for the RR development, re-vascularization in association with wound healing could provide a great model system where you may be able to show if the RRs start forming only after vessels are established and have reached the epidermis.

(11) Fig6: Please provide an RNA staining (eg. RNAscope) confirming PDGFC expression in RR in pig, and its absence in mouse skin. PDGFC seems one of the few unique RR markers.

(12) Does the formation of RRs change the surrounding fibroblasts and the ECM components? As the authors find pro-fibroblast signals coming from the keratinocytes (Fig6), can they also identify if these signals are mainly concentrated in the IR areas or beneath the RRs? Are the IR region fibroblasts distinct from other dermal fibroblasts?

(13) The 'reverse engineering' hypotheses establishment and experiments are very interesting, yet again beg for some more exploitation. (a) L390: "Histological assessment of TPA- and RA390 treated wounds revealed induced regions of rete ridge-like structures, complete with recruited vasculature into the inter-ridge space at both 1dpt and 4dpt (Fig. 7e-f; Extended Data 5f, h). The "recruited vasculature" would have to establish within 1 day, which seems fast. Again (related to point 9 above), isn't it more likely that RR appear between the already established inter-vascular dermal space? For example, the vascular niche pockets secreting proliferation/stratification inhibitory signals? (b) When discussing the role of RA treatment as inducing de-differentiation of keratinocytes, do the authors have any independent evidence for the occurrence of this de-differentiation? RA is known to induce hyperproliferation of keratinocytes (<https://doi.org/10.1038/jidsymp.1998.14>; <https://doi.org/10.1016/j.ydbio.2008.08.034>). As both TPA and RA treatment result in RR formation, induced hyperproliferation of keratinocytes would be sufficient to explain the results without necessity of de-differentiation.

Minor and textual comments:

(a) Line 121: "apical ridges formed earlier than GW12 in human development and increased in length by adulthood". Please mark the apical ridges the Fig 1a,b.

(b) Related to 121 (Methods): How is the apical ridge length measured? Do you mean "length" or "depth"? To me "length" is confusing. Please show either a cartoon or indicate on sample what you measure.

(c) Fig 1c. Please indicate which body sites were used for the measurements. Is it a combination of face and trunk skin or were also some other body sites included? Were the samples coming from donors of similar ethnic background? This can have an influence on epidermal thickness depending on body location (see [doi:10.1111/jdv.18123](https://doi.org/10.1111/jdv.18123)).

(d) L134: Fig 1e. Can you indicate the RRs? I cannot spot any.

(e) Fig1i. header text in figure "RR vs apicR formation are independent processes". Measured is apical ridge length; to say something like RR vs apicR formation are independent processes, shouldn't the apicR measurement be somehow correlated to RR measurements? This graph alone is not an argument, is it? Or should it be apical ridge density?

(f) L152/153: From your data at this point it's still an assumption that RR also form perinatally in human skin, right? As it is stated, you would have to show it in human samples, which of course is not possible. Please rephrase the sentence to conveying the message "as pig skin development is a great proxy for human skin development, it can be assumed that RR in human also form perinatal".

(g) Line 153: "Additionally, the developmental overlap between the onset of increased epidermal thickness and rete ridge formation suggest that these processes may be linked." Do you refer here also to the interridge thickening? If not (i.e. you only refer to RR in general) this is a circular argument. Please clarify what you exactly mean. Same for L167: please specify in the text to which parts of the epidermis (RR and/or IR) you refer to? Eg. simply add "...the epidermis (IR and RR) was thicker... "

Related to L153: Lines 167-169: The notion that epidermis is thicker where RRs are is not a surprise; this has been obvious since decades, thus this relationship is expected. Again, what is surprising is that skin with RR and lower hair density also has a thicker IR epidermis; so, every single part of the epidermis is thicker, but this does not come across from the text itself.

(h) Fig2. Are all skin samples across animals species taken and measured from the approximate same body location?

(i) L156 and Fig 2b. Please add statistical testing, most importantly for second plot (epidermal thickness; IR), showing if the epidermal thickness differences with and without RR are significant.

(j) L183: should it be: hair density "decreases" instead of "increases"?

(k) L188: heading. Given your results and the recent study on fingerprint ridges and their development partly resembling hair follicle formation including high Lef1 expression (<https://doi.org/10.1016/j.cell.2023.01.015>), one could expand the heading to: "Rete ridges are a distinct class of epithelial appendage from hair follicles, sweat glands, and fingerprint ridges".

(l) L202: it would help the non-skin readership to state once more here when hair-bud formation stops; I assume it's the same in pig skin (?) but in both mouse and human skin HF bud formation stops around birth or before. This information helps the reader to understand why the E90 and P10 is a great comparison.

(m) L216/217: human foreskin (please also describe, as for volar skin, that it's devoid of HFs, etc..)

(n) L214-221: It is important to better describe/introduce fingerprint ridges development. The reader may get confused with rete ridges vs fingerprint ridges; i.e. its very important for this paper to make clear that fingerprint ridges form through a different (HF-like) molecular mechanism, which then makes it clear in L124-221 why you can use (human) fingerprint bud development as a proxy for (pig) HF bud development.

(o) Fig3: Please include another representation showing the expression of LEF1, EDAR and ITGA6 in more detail for the dataset, either as feature plots or expression levels in the different clusters.

(p) Fig 3c, what are the clusters 3 and 6, are these the developing HFs?

(q) Lines starting at 242 have references to Fig 1, but should be referring to Fig 4?

(r) Fig4d-g: where are these zoom-ins taken from?

(s) Figs 3 & 5: please include panels showing the gene expression signatures used for cluster annotations.

(t) Fig 6c: Which timepoint(s) were used for CellChat analysis?

(u) The methods and GEO upload indicates that pig samples were acquired from *Sus scrofa* species, which is the wild boar. Is that really the case or do the authors mean *Sus Domesticus* (or *Sus scrofa domesticus*), which would be the domestic pig that's depicted in the figures?

(Remarks on code availability)

I have looked at <https://github.com/DriskellLab/Thompson-et-al.-2024>, and at a glance all code is there and looks fine. However, we have not re-run the code.

Version 2:

Reviewer comments:

Referee #1

(Remarks to the Author)

The authors have extensively revised the manuscript, including changing the title from "Evolutionary development, regeneration, and induction of rete ridges in mammalian skin"

To "Rete ridges form through evolutionarily distinct mechanisms from other appendages during mammalian skin development". In this revision, the domestic pig is used as the main model to characterize when rete ridges develop. Bioinformatic analyses of pig and human skin are applied to infer mechanisms of ridge formation. The authors attempt to place their proposed mechanism in an evolutionary context by including both ridge-bearing and non-ridge-bearing mammals. Finally, they perform pharmacological treatments (TPA and RA) in newborn mouse skin to induce ridge-like structures, suggesting possible applications.

The revision represents significant effort and provides valuable data on ridge formation in pig, human, and mouse, and with potential translational implications. Ridge formation is a significant topic. Methodology are in general of good quality. The manuscript has improved a lot. However, several points require further clarification and revision. This is a strong body of work on pig, mouse and human, but the conclusion on evolutionary aspect is not sufficiently supported and should be tuned down. It is not necessary to extend the conclusions to encompass all mammalian skin appendages and evolution without a systematic comparative study, though such discussion is interesting and can be raised in a speculative manner.

1. Evolutionary aspect of rete ridges

- The authors emphasize the relationship between the order of hair, sweat gland, and ridge formation. Their detailed work in pigs, humans, and mice is strong.
- While data from other mammals (primates, grizzly bear, dolphin) add interest, this does not constitute a systematic survey of mammalian skin development. I do not suggest additional mammalian sampling, but without broader evidence, the evolutionary conclusions should be tempered and potential alternative mechanisms acknowledged.

- The manuscript would benefit from discussing the following reference:
 - o Alibardi, L. (2004). Dermo-epidermal interactions in reptilian scales: speculations on the evolution of scales, feathers, and hairs. *J. Exp. Zool. Part B: Mol. Dev. Evol.* 302B(4): 365–383.

2. Mechanisms of ridge formation

- The multi-species scRNA analyses are comprehensive and are a strength of the paper.
- Based on these, the authors highlight the role of dermal vasculature, suggesting epidermal–endothelial interactions are important in pig ridge formation, analogous to epidermal–dermal interactions in hair follicle formation.
- However, epidermal ridges represent a general morphological transformation and may form through diverse cellular mechanisms. The vascular model should be presented as one possible mechanism rather than the sole mechanism of ridge formation.

3. Regional specificity of rete ridges

- Fig. 2: The tissue collection sites for each species must be specified. Regional differences in rete ridge morphology could strongly affect conclusions (see Moyo 2018).
- Human rete ridge length and width vary regionally, as in cetaceans and most mammals. Comparable anatomical sites must therefore be compared across species.
- To avoid confusion, I recommend adding a supplemental figure showing the anatomical collection sites for each species.
- Authors may also discuss:
 - o Kowalczyk A., Chikina M., Clark N. (2022). Complementary evolution of coding and noncoding sequence underlies mammalian hairlessness.

4. Ridges in pig skin

- Wild pigs possess both primary and secondary hair as well as rete ridges. In contrast, domestic breeds used here lack secondary hair and have reduced primary hair (Meyer 1986). This challenges the claim that “hairlessness coincides with rete ridge development.”
- Humans with scalp hair (and with congenital generalized hypertrichosis, “wolfman syndrome”) also display rete ridges, further weakening this association.
- Specific clarifications needed:
 - o Which pig breed was used for Fig. 1?
 - o How many breeds were studied across developmental stages?
 - o Were all samples taken from the same body site? If not, please clarify.
 - o For fetal pig experiments, what breeds were used?
 - o Sinclair Biosciences lists Yucatan™ Miniature and Micro-Yucatan™ Miniature Swine, both described as “hairless.” Which was used? Please correct the methods accordingly.

5. Hair and ridge formation in humans

- Individuals with congenital generalized hypertrichosis (“wolfman syndrome”) develop both hair follicles and rete ridges. I recommend removing this abstract sentence: “This loss coincides with the acquisition of epidermal rete ridges...”
- Fig. 2: Human epidermis is established to be thinner than pig epidermis (Mayer 2009; Summerfield 2014, among others). The figure and data here suggest the opposite. Please reconcile this discrepancy.
- Revise Fig. 2a to show that pig epithelium is thicker than human.
- Fig. 1 caption: please specify tissue sites (“Representative H&E stains of trunk and face skin of adult humans”) in both figure and Methods.

6. Reverse engineering in the mouse

- This is a strength of the manuscript, adding functional validation and translational potential.
- As authors acknowledge, the effect of TPA on cell proliferation / epidermal thickness is more comprehensible. The mechanism of RA action in this context should be more clearly explained.

Minor Points

1. Fig. 1g: replace “dermal pockets” with dermal ridges to avoid confusion with hair dermal papillae.
2. Fig. 2b: clarify the term “hairiness.” If this refers to hair per unit area, please use hair density. Also, the gross image under “domestic pig” shows black skin—please specify the breed in text and figure.
3. Grizzly bear skin: published work (Tomiyasu 2017) indicates that back skin of grizzly bears lacks rete ridges. Bears and mice are considered loose-skinned mammals without rete ridges. Please state what anatomical site was sampled, as Fig. 2a does not convincingly show ridges. This discrepancy must be reconciled.

Referee #2

(Remarks to the Author)

The authors have addressed most of the reviewers’ comments in the revised manuscript through large efforts and new experiments. These efforts have substantially improved the manuscript through the inclusion of new sequencing data, additional analyses, the establishment of a rete ridge model in mice, and functional testing of key signaling pathways. There are important findings in this manuscript, however in this resubmitted version some figure panels are missing, and several

figure citations are incorrect. In addition, some language is misleading or confusing. Below suggestions to rectify the above observations to finalize this submission:

1) The authors conclude that rete ridge development is driven by non-focal proliferation (line 281 and 302). However, the MKi67 staining data indicates that both basal and suprabasal proliferation are significantly higher in the rete ridge domain compared to the inter-ridge domain during the second postnatal week in pigs, when rete ridges are actively developing (Supplemental Fig. 8c). These localized differences in proliferation could reasonably be interpreted as focal, rather than non-focal. Furthermore, it is unclear how rete ridge and inter-ridge domains are defined when quantifying proliferation at P1-2 and P3 (Supplemental Fig. 8b). At these time points, the boundary between the domains seems uncertain, and the rete ridge domain looks broader than at later stages. This ambiguity could lead to inaccurate classification of proliferating cells as either inside or outside the rete ridge domain. In fact, both tissue sections (Supplemental Fig. 8b) and whole mount images (Fig. 4a and 4e) show that the rete ridge domain narrows over time, suggesting dynamic remodeling. As a result, cells initially counted in the inter-ridge domain may end up in the rete ridge domain at a later time point.

2) In the rebuttal, the authors argue that “during rete ridge formation and maturation, the suprabasal and spinous compartments expand as pigs develop rete ridges, while the epidermis is thinner and simpler in species that lack rete ridges.” They further claim that “epidermal thickening in rete ridge regions is primarily driven by suprabasal expansion, not just basal layer enlargement.” The primary data supporting these conclusions appear to be presented in Supplemental Fig. 9a and 9b. However, these data are neither described nor interpreted in the main text, and Supplemental Fig. 9a is not properly cited.

3) The authors define the dermal space between rete ridges as the “dermal pocket—a prominently vascularized region of the papillary dermis”—and emphasize its interaction with the epidermis and its potential role in rete ridge development throughout the manuscript. They further discuss a co-dependency between rete ridges and dermal pockets in the rebuttal. However, this interpretation raises several concerns.

First, if the dermal pocket is defined as the space between rete ridges, then its formation is a structural consequence of rete ridge development. In that case, questioning whether dermal pocket formation depends on rete ridge formation becomes tautological and biologically meaningless. For example, “generate an interconnecting rete ridge network organized around vascularized dermal pockets” (line 51-52) implies that dermal pockets precede the formation of the rete ridge network—an interpretation that contradicts the definition of dermal pockets as secondary to rete ridge architecture.

Second, if the authors intend to define the dermal pocket based on its vascular characteristics, they have not provided sufficient evidence to support this claim. Due to the orientation of blood vessels in the skin, thin tissue sections may not reliably reveal regional differences in vascularization, A 3D whole mount analysis is more suitable. Moreover, the authors have not demonstrated any differences in angiogenic activity between rete ridge and inter-ridge regions during development. It is also possible that the dermal cells are passively enriched into the dermal pocket while expanding equally between different domains.

4) It is exciting that the authors have established a new model to study rete ridge formation in the mouse fingerpad. This represents a significant advance in the field, and the authors may consider highlighting this breakthrough more prominently when introducing the finding for the first time (line 260).

5) The authors present convincing data showing the critical role of epidermal BMP signaling in enabling rete ridge formation. However, they appear to overstate the role of signaling between epidermis and dermis in driving rete ridge formation, as the role of dermis in this process remains insufficiently defined. For example, the statement “Postnatal activation of epidermal-dermal signaling rete ridge and dermal pocket formation” (line 306) implies a causal relationship, yet no functional evidence is provided to support a driving role for epidermal-dermal signaling. Similarly, the claim that “these transcriptomics observations highlight the existence of divergent cellular and molecular interactions that drive species-specific differences in skin morphology” (lines 372–374) overextends the interpretation of correlative transcriptomic data without accompanying mechanistic validation. On the other hand, the statement “a new model for their formation and regeneration, which is led by epidermal cell lineages” implies epidermis is the driver, excluding a potential instructive role for the dermis.

6) The authors use the term “basal budding” to specifically refer to the WNT-EDA/R mediated appendage formation process. However, this term can confuse the reader, as it can also be interpreted more generally to refer to any budding-like morphogenesis occurring at the basal layer—a process that is not necessarily excluded in rete ridge formation. Therefore, the statement “rete ridges must instead form through a distinct mechanism from basal budding” (line 277) is inappropriate. The authors may explicitly state that they are referring to the WNT-EDA/EDAR-mediated process.

7) The authors show that rete ridge and inter-ridge epidermal cells are molecularly and behaviorally equivalent, suggesting that rete ridges represent structural domains within the epidermis rather than functionally distinct units. Therefore, referring to rete ridges as skin appendages may be inappropriate, as most skin appendages are both structurally and functionally distinct from the interfollicular epidermis. Moreover, rete ridges have not traditionally been classified as skin appendages in the field. That said, the classification of fingerprint ridges as skin appendages introduces some ambiguity, and under that broader interpretation, rete ridges could potentially be considered a type of skin appendage.

8) The use of the term “dermal maturation” (line 135) to describe the collagen maturation observed via Herovici staining is inappropriate, as “dermal maturation” typically encompasses a much broader range of structural and cellular changes within the dermis beyond collagen remodeling alone.

9) It is expected that rete ridges contribute to increased epidermal thickness in ridge-bearing species. However, it remains

unclear whether they also influence the thickness of the inter-ridge regions. The authors note that “the inter-ridge thickness (aside from the cetaceans) is not dramatically increased” (lines 172–173), but no statistical analysis was performed. A quantitative comparison of inter-ridge thickness between ridge-bearing and ridge-less species would be valuable to determine whether inter-ridge regions also contribute to the overall epidermal thickening observed in ridge-bearing animals.

10) To avoid confusion, the authors need to mention that the few hair follicles in the *Lef1*-eKO mice (line 200 and Supplemental Fig. 3c) are likely escapers, since some epithelial cells still express LEF1.

11) The authors claim that the rete ridge and inter-ridge epidermal cells are molecularly similar, and PDGFC and SMAD1 are broadly expressed across these domains. However, the whole mount staining images do not clearly show the broad expression of PDGFC (Supplemental Fig. 9e) or SMAD1 (Fig. 4e). Instead, these images appear to show differential expression of PDGFC and SMAD1 between the rete ridge and inter-ridge regions.

12) For Fig. 4d, Supplemental Fig. 10b, 11, 12, 13, 14, 15b, it is hard to judge the signal between different cell types, as cell types are not annotated in those graphs the presence of the legends.

13) The authors state that “rete ridge-less trunk skin of marmosets and mice has vasculature and pericytes which are generally distal from the interfollicular epidermis throughout development and maturation” (lines 363–365). However, this claim is not supported by quantitative analysis or comparison using vasculature-specific markers. Given that only segments of blood vessels are visible in thin skin sections, a 3D whole-mount analysis would be more appropriate to accurately assess the spatial relationship between the vasculature and the interfollicular epidermis.

14) Figure citation mistakes:

- Line 120: Supplemental Fig. 1a does not show dermal pockets. Instead, it shows fetal human skin when rete ridges are not formed.
- Line 243: Supplemental Fig. 5a does not show LEF1 and EDAR expression. Supplemental Fig. 5b should be cited instead. Do white signals indicate high level of expression or undetected expression? They can confuse readers.
- Supplemental Fig. 5e is not cited anywhere.
- Supplemental Fig. 7g is not cited anywhere.
- Line 287: Fig. 4a does not show basal bud formation. Instead, it shows interesting narrowing of the rete ridge area and expanding of the inter-ridge area over time.
- Figure legends for Supplemental Fig. 9d, 9e, 9f are missing.
- Line 333: The correct citation should be Supplemental Fig. 10b–c, not Supplemental Fig. 8b–c.
- Figure legend for Fig. 4e is missing.
- Fig. 4f–g are cited in the text but are missing in the Figure.
- In Supplemental Fig. 11b, BMP7 is shown in the graph, whereas BMP6 is labeled in the CirclePlot.
- Line 396: Supplemental Fig. 17c should be cited to show “murine trunk skin does not express *Pdgfc*”.
- Line 398: Supplemental Fig. 5c does not show EN1 expression.
- Line 399: The correct citation should be Supplemental Fig. 17d, not 17c.
- The title of Supplemental Fig. 17e is not appropriate, as JAG1 is active not only in neonatal skin but also in fetal skin.
- What do red dashed circles in Supplemental Fig. 17g indicate?

Referee #3

(Remarks to the Author)

The authors have performed a rigorous revision, including a wealth of new experiments and analyses, thoroughly addressing my major and minor questions.

It is a true biological highlight that although rete ridges (RR) appear to form via initial bud emergence, like other epidermal appendages, they develop through distinct molecular and cellular mechanisms, which has been convincingly demonstrated in the revised manuscript. Further, it was particularly impressive that the authors undertook technically demanding experiments, including BrdU cell tracking in pigs, Stereo-seq, and spatial CellChat analysis. Beyond the novel biology, the manuscript also comes with a rich dataset, including a large collection of porcine scRNA-seq and spatial transcriptomics data, which is openly accessible via an online portal and tool.

In summary, the authors have conducted a rigorous cross-species analysis of rete ridge formation, a process debated for decades but previously unresolved. This is an exciting manuscript and represents a significant milestone in the field of tissue developmental biology.

Minor comments:

- The authors write in the abstract: “Therefore, we propose that the evolution of rete ridges in mammalian skin involved replacement of the molecular program for periodic patterning and subsequent downgrowth of discrete appendages, such as hair follicles and sweat glands, coupled with broad activation of BMP to generate an interconnecting rete ridge network organized around vascularized dermal pockets.” This sentence may be difficult for a general readership to follow and would

benefit from rephrasing for clarity.

- PDGFC, which is frequently highlighted in the manuscript and across species, would be valuable to show in pig skin as well. Could you display its expression in any of the pig scRNA-seq datasets, if not in the spatial data?

- Please provide more details on the spatial analysis, such as the Stereo-seq dot sizes, the number of reads and genes per dot, whether dot-binning was used, whether cell segmentation was applied to identify underlying cells, etc.

Version 3:

Reviewer comments:

Referee #1

(Remarks to the Author)

Thank you for working hard to address my concerns and revise your initial conclusions.

My comments are addressed satisfactorily now.

Referee #2

(Remarks to the Author)

The authors have satisfactorily addressed most of the reviewers' concerns. Only a few minor editing errors remain that should be corrected see below. Congratulations to the authors for their tremendous work.

- Line 124-125: Supplemental Fig. 1a does not show dermal pockets. Instead, it shows fetal human skin when rete ridges are not formed. Although the authors indicated that this citation was corrected, it appears not to be.
- The panel identifiers in the figure legends for Supplemental Fig. 9d, 9e, 9f are incorrect.
- Line 412: The correct citation for showing EN1 expression should be Supplemental Fig. 17d, not 17c. Although the authors indicated that this citation was corrected, it appears not to be.
- Line 306 and 312: The word "patterned" was misspelled.

Referee #3

(Remarks to the Author)

This is an amazing manuscript! After re-reading it with all the implemented changes based on the thoughtful suggestions from all reviewers, I'd like to congratulate the authors on this work. All my previous questions have been fully addressed.

Rebuttal Letter for Thompson et al., 2025: “Evolutionary development, regeneration, and induction of rete ridges in mammalian skin”

We sincerely thank the reviewers for their insightful and constructive critiques. In response, we have undertaken a major revision of the manuscript that substantially elevates its conceptual and mechanistic contributions. We summarize the **key additions to the revised manuscript**, which are described in complete detail in our individual responses to all reviewer comments below. We have generated and integrated an expansive set of new datasets throughout porcine skin development, including a new single-cell RNA-sequencing (scRNA-seq) timepoint during early rete ridge formation at postnatal day 3 (P3), and 3 single-cell resolution spatial transcriptomics (Stereo-seq) datasets across rete ridge formation and maturation in porcine skin at P3, P10, and 6mo. Beyond re-analyzing these expanded scRNA-seq datasets with new CellChat analyses, we have performed novel Spatial CellChat analysis of our Stereo-seq data in collaboration with the Jin Lab to understand the precise spatiotemporal ligand-receptor interactions between epidermal and dermal cell lineages over the course of rete ridge formation and maturation.

Furthermore, we have conducted additional in-vivo investigations of cellular mechanisms during rete ridge formation, including proliferative patterning and BrdU-labeling to track differentiation, alongside extensive comparative immunostaining of mammalian skin from multiple species, including pigs, humans, mice, dolphins, marmosets, and naked mole rats. These revisions have elevated this study to present one of the most thorough developmental and mechanistically validated characterizations of rete ridge formation in human-like skin to date.

We have added three major lines of evidence to functionally validate these newly characterized cellular and molecular interactions in-vivo: (1) characterization of the similarity in murine fingerpad rete ridge development to porcine and human trunk skin rete ridges, and application of this model in knockout pigs and transgenic mice to demonstrate that rete ridge formation is independent of basal bud/LEF1-signaling; (2) in vivo BrdU-labeling in conjunction with MKI67 staining during porcine rete ridge formation to demonstrate that proliferative patterning and differentiation support rete ridge formation; and (3) two new transgenic mouse models showing that epidermal BMP signaling is required for rete ridge formation. We've also clarified key conceptual distinctions, such as differentiating fingerpad rete ridges from digital transverse ridges, the murine analog to human fingerprint ridges ¹, and introduced a refined model for rete ridge formation in which the epidermis requires BMP-signaling while lacking distinct pre-patterning during rete ridge formation. Instead, underlying dermal lineages may support the periodic patterning of rete ridge formation through distinct spatiotemporal signaling to the epidermis. These extensive revisions apply new single-cell and spatial transcriptomics alongside functional validation in-vivo provide new mechanistic insights into the cross-species comparisons in skin development which support rete ridges as a distinct epithelial appendage from hair follicles, sweat glands, and fingerprint ridges. These new experiments have transformed this manuscript from a largely descriptive survey to provide robust, testable models for human skin architecture and evolution in pigs and murine fingerpads, which we hope will be of broad interest to not only the general public, but also to the fields of skin biology, regenerative medicine, and developmental biology.

Please see the following sections for our individual responses to each reviewer's comments.

Referee #1 (Remarks to the Author):

This paper examines the formation of epidermal ridges in the skin, employing a comparative biology approach that analyzes ridge characteristics across various species including humans, pigs, mice, and others. By assessing factors such as hair density, presence of ridges, and epidermal thickness, the authors propose a multi-part model for rete ridge formation. They conducted scRNA analysis on pig wounds and comparing their findings with existing literature. While molecules implicated in ridge formation were identified through correlation and comparative CellChat analyses, their roles were not confirmed through expression or functional studies. In an experimental approach, retinoic acid and wounding agents were administered to the hairless scars of mice, resulting in the formation of structures resembling rete ridges. However, without molecular characterization, it is challenging to definitively classify these structures as rete ridges. While the study offers valuable comparative skin findings, it lacks in elucidating the mechanism of ridge formation or the evolutionary origins of these ridges. As a result, the title may appear exaggerated and should be modified to accurately reflect the scope and findings of the study. The data does hold potential and authors can improve upon the following points.

We thank the reviewer for their detailed and constructive feedback, which has been instrumental in shaping the substantial revisions presented in this manuscript. In response, we have added over a dozen new experiments, including three new Stereo-seq datasets, a new P3 scRNA-seq timepoint, cross-species immunostaining, new transgenic mouse models, and in vivo functional assays to transform a largely descriptive study into a mechanistically grounded framework for rete ridge development. These revisions clarify that rete ridges form not through pre-patterned epithelial periodicity, but via a distinct morphogenetic program coordinated by epidermal signaling and dermal architecture. By moving beyond traditional skin appendage models, we offer a revised conceptualization of rete ridge morphogenesis as the establishment of a 3-dimensional, interconnecting ridge network creating a basal epidermal architecture resembling a mountain range and guided by spatially coordinated epithelial-mesenchymal signaling. We are grateful to the reviewer for challenging us to elevate the manuscript and hope this revised version meets their standards for clarity, rigor, and impact.

We have added 3 new Stereo-seq spatial transcriptomics datasets from P3, P10, and 6mo pig skin to address the spatial localization of key signaling pathways and their ligands during rete ridge formation and maturation. We have thoroughly expanded our immunostaining characterization to many new species, adding human development, dolphins, and other tissues from mice which contain rete ridge structures, such as in the volar fingerpad, to more clearly demonstrate the evolutionary conservation of cellular and structural morphology of the rete ridge and dermal pocket (Supplemental Fig. 16a-d).

We have also performed new functional studies including in-vivo BrdU labeling to understand the role of proliferation and differentiation in porcine rete ridge formation (Supplemental Fig. 8a-i). These results indicate that rete ridges form through a non-focal proliferative patterning that is distinct from hair follicles, sweat glands, and rete ridges, and are supported by differentiation (Supplemental Fig. 8a-i)¹⁻³.

We have also now characterized rete ridge development in the volar fingerpads of mice, which are molecularly distinct and form postnatally after volar sweat glands and fingerprint/transverse

ridges (Fig. 5a; Supplemental Fig. 6a-b; Supplemental Fig. 17b)^{1,4,5}. Fingerpad rete ridges exhibit the same key molecular markers as the trunk rete ridges in humans and pigs, and we used this model to functionally validate that epidermal BMP-signaling is required for rete ridge formation (Fig. 4a-e; Fig. 5a-e; Supplemental Fig. 5a-f; Supplemental Fig. 9a-f; Supplemental Fig. 17b).

Based on reviewer feedback and these new experiments, we have changed the title to focus on the core takeaways from the revised manuscript: “ Rete ridges form through evolutionarily distinct mechanisms from other appendages during mammalian skin development”.

1. Comparative studies of epidermal ridges.

a. Epidermal ridges represent an intriguing and relatively understudied topic in the field of skin biology. A notable recent study by the Headon group has shed light on this subject. In this paper, the authors have compiled new data from various species including pig, human, mouse, as well as additional data on monkeys and bears. While their comparative approach has amassed a large amount of data for analysis, the nature of the presented data remains primarily descriptive and correlative.

We appreciate the reviewer’s recognition that epidermal ridges are an understudied and intriguing aspect of skin biology. In this study, we provide what we believe to be one of the most comprehensive and developmentally resolved maps of rete ridge formation ever assembled in mammalian skin. This dataset is not merely descriptive; it fills a critical void in the field. For the first time, we define when and where these human defining structures emerge, across species and developmental time, with implication for understanding not only skin biology but also human evolution, wound repair, and regenerative medicine. Using this descriptive framework as a foundation, we pursued mechanistic insights by performing new experiments utilizing state of the art single-cell transcriptomics approaches and transgenic mice to generate a novel molecular and cellular mechanism for rete ridge formation.

Molecular, Spatial, and Comparative Transcriptomics

In order to better define the transcriptional environment during rete ridge formation, we generated a new P3 pig skin scRNA-seq dataset to capture the early stage of rete ridge formation and have integrated analysis of this dataset alongside our prior E90 time point, which lacks rete ridges, P10, a later stage of rete ridge development, and 6mo, mature rete ridge, datasets (Fig. 4a-b). This expanded dataset allowed us to define molecular transitions in epidermal and dermal cell states from pre-ridge through mature ridge formation, directly contextualizing rete ridge formation at a molecular and cellular level. Furthermore, to define the spatial context for cellular signaling activities during rete ridge formation and maturation, we generated novel P3, P10, and 6mo Stereo-seq spatial transcriptomics datasets (Fig. 4a, d). We have also uploaded our analyses of all of these datasets to our website for public querying (<https://skinregeneration.org/papers/Thompson-et-al-2025/>). Together, the scRNA-seq and Stereo-seq datasets provide a multi-scale spatial view of how gene expression and skin architecture interact to form rete ridges.

We performed new CellChat analyses on these scRNA-seq datasets and formed a new collaboration with the Jin Lab at the University of Wuhan to perform Spatial Cell Chat analyses, which is a novel spatial transcriptomics tool. This enabled prediction of dynamic ligand receptor interactions between epidermal and dermal populations across developmental time points for us to interrogate mechanistically. We also validated the spatial context for these signaling activities

between scRNA-seq and Stereo-seq using immunostaining in pigs and compared these to other mammalian species, including humans, dolphins, marmosets, mice, and naked mole rats (Fig. 4a-e; Fig. 5a-b; Supplemental 9a-f; Supplemental 15a-d; Supplemental Fig. 16a-d). This revealed conserved signaling activity, most notably BMP and PDGF pathways, suggesting a shared molecular framework underlying ridge morphogenesis in skin across mammals. Altogether, these molecular and spatial analyses move beyond description to define a mechanistically testable model for how rete ridges are specified and maintained.

Functional Experiments and Genetic Validation

In parallel, we conducted new functional studies in pigs and mice to investigate whether key signaling pathways and signaling behaviors are required for rete ridge morphogenesis. In pigs, we expanded our functional studies, including elaborating upon our previous EDA-KO model to demonstrate that rete ridge formation is unperturbed in neonatal P5 pig skin (Fig. 3e).

Furthermore, we have also performed in-vivo BrdU-labeling during rete ridge development to investigate the role of proliferation and differentiation in rete ridge formation and rete ridge versus inter-ridge domain specification (Supplemental Fig. 8a-i). These data revealed domain-specific differences in proliferative behavior that inform how distinct regions of the epidermis contribute to ridge structure.

Based on our characterization of rete ridge formation in murine fingerpads and its parallels to porcine and human rete ridge formation (Fig 3a; Supplemental Fig. 6a-b; Fig. 4a-e; Supplemental Fig. 16a, d; Supplemental Fig. 17a-b). We used murine fingerpad, which are distinct from murine transverse ridges (Rebuttal Figure 1), as a model to further support our previous conclusion from EDA-KO pigs that the basal bud molecular program is not required for rete ridge formation. Epidermal Lef1-knockout mice have dramatically reduced hair follicle formation but rete ridge formation in the fingerpad is unaffected (Fig. 3d; Supplemental Fig. 4b-d).

We also have now extensively characterized the occurrence of epidermal BMP-signaling activation as a consistent component of rete ridges in pigs, humans, and mice (Fig. 4c-e; Fig. 5b-e; Supplemental Fig. 11a-b; Supplemental Fig. 17e-i). Therefore, we exploited the tractability of transgenic mouse technology to inhibit epidermal BMP-signaling at both the ligand and receptor levels to demonstrate that epidermal BMP-signaling is required for rete ridge development.

Together, these functional and genetic studies demonstrate that rete ridge formation is a biologically distinct program, separable from hair follicle development and governed by a novel epithelial-mesenchymal signaling pathway.

b. It is already widely recognized that non-furry animals tend to have more well-developed rete ridges. While the authors present some more intriguing examples involving bears and monkeys, they did not delve into the mechanistic issue of whether there exists a developmental constraint influencing the formation of both hair and ridges.

We thank the reviewer for raising this important conceptual question about potential developmental constraints linking hair density and rete ridge formation. The loss of fur is an intriguing question in the context of human skin's evolutionary development⁶⁻¹⁰. While we agree that a correlation exists between hairlessness and the presence of rete ridges in certain

species, our revised manuscript provides new data suggesting that this relationship is not casually straight forward. In fact, our data point to a developmental decoupling, where loss of hair or basal buds does not inherently trigger rete ridge formation, and that rete ridges arise through a distinct morphogenetic program. Below, we outline the key evidence from our new spatial, molecular, and genetic studies that support this refined model.

1. Evolutionarily conserved loss of basal bud identity precedes rete ridge formation, but is insufficient to induce it.

We present new immunostaining to validate the multi-species, multi-tissue conservation of LEF1 expression in the basal buds of multiple epithelial appendages, including hair follicles in both human and marmoset fetal development (Fig. 3e) and sweat glands in porcine trunk (Fig. 3a) and murine volar fingerpad skin (Supplemental Fig. 6a-b). Critically, we also highlight how epidermal LEF1 expression disappears when basal buds for these appendages stop forming, including in marmoset and naked mole rat development (Fig. 3a; Supplemental Fig. 6b) ¹¹. From this foundational state of LEF1-negative epidermis, then rete ridges begin to form in species and tissue regions capable of doing so (Fig. 3a; Supplemental Fig. 6b).

2. Epidermal transcriptomic state shifts postnatally

We further explored this phenomenon through our re-analysis of our porcine and human scRNA-seq datasets. To capture early rete ridge formation, we performed a new scRNA-seq experiment on P3 pig skin (Supplemental Fig. 4a) alongside Stereo-seq spatial transcriptomics on P3, P10, and 6mo skin (Supplemental Fig. 4a-b, d). We directly examined the developmental constraint on hair follicle vs rete ridge formation in the revised Fig. 3, which integrates E90, P3, P10, and 6mo basal keratinocytes (Fig. 3b). This new analysis demonstrated that fetal skin and basal buds are transcriptionally distinct from postnatal basal cells, when rete ridges are forming and maturing (Fig. 3c). Conserved markers of basal buds, like LEF1 and EDAR, hair follicle formation marker SHH, and sweat gland or fingerprint ridge markers SOX9, TGFA, and LMX1A are not expressed during rete ridge formation or maturation in pigs and humans (Fig. 3c; Supplemental Fig. 5a-f) ^{1,4,5}. This integrative analysis demonstrates that the fetal developmental program is fundamentally altered in the epidermis after birth, during rete ridge formation and also during maturation (Fig. 3a-e).

3. Genetic testing of hair loss and epidermal architecture

Another critical question is whether the relationship between hair density and epidermal thickening/rete ridges is merely a developmental coincidence or more directly related phenomena. Previous studies have demonstrated that the Wnt-signaling transcription factor Lef1 is required for hair follicle formation ^{12,13}, and Lef1 is a conserved marker of basal buds in a broad array of epithelial appendages, including hair follicles, sweat glands, and fingerprint ridges ^{1,4,5,12}. We directly examined this question using K14Cre Lef1 fl/fl to conditionally ablate epidermal LEF1 (Lef1 eKO) in mice (Supplemental Fig. 3b). We observed a dramatic reduction in basal budding and subsequent hair density in Lef1 eKO mice (Fig. 2f; Supplemental Fig. 3c-d). Furthermore, the epidermal thickness did not increase in response to this loss of fur and rete ridges did not form (Supplemental Fig. 3g-i). These results support that a genetic reduction in hair formation alone is insufficient to enable rete ridge formation.

4. Mouse finger pad rete ridges form independently of Lef1

As part of our revised study, we have replaced the engineered induction model with a developmental analysis of mouse volar fingerpad skin, which we now show forms postnatal rete ridges that are distinct from previously characterized transverse fingerprint ridges ^{1,14}. In this regard, we have characterized rete ridge formation in the volar fingerpads of mice, which form

after the postnatal completion of volar sweat gland morphogenesis (Fig. 5a; Supplemental Fig. 6a-b). It is worth noting that fingerpad rete ridges do not express LEF1, clarifying their distinction from the previously reported fingerprint ridges in human volar skin, which form from LEF1+/EDAR+ basal buds during fetal development in humans and mice¹. Next, we investigated fingerpad rete ridge formation in Lef1 eKO to determine whether rete ridge formation requires LEF1/WNT-signaling, which revealed that fingerpad rete ridge formation does not require epidermal LEF1 (Fig. 3d).

5. Rete ridge formation does not require EDA/R signaling

Since Lef1 directly regulates EDA/R signaling, another conserved component in basal buds for hair follicles, sweat glands, and fingerprint ridges^{1,4,13,15,16}, these results strengthen our previous claims using EDA-KO pigs that rete ridge formation does not require EDA and other signals important to forming the basal bud step in hair follicle and sweat gland morphogenesis (Fig. 3e). From the original submission, we have also thoroughly analyzed a new neonatal timepoint, P5, for quantitative and statistical comparison of rete ridge density between WT and EDA-KO pigs, which demonstrated no difference (Fig. 3f).

Collectively, these results suggest that a foundational component for rete ridge formation is the reduction in hair follicle density and loss of basal bud-forming potential in the epidermis, but that loss of hair alone does not intrinsically cause rete ridge formation (Fig. 5e).

6. New model: loss of basal bud potential coupled with epidermal BMP activation

Alongside additional new experimentation, which are described in more detail in response to other reviewer comments, we have demonstrated that rete ridge formation requires epidermal BMP-signaling, suggesting that the loss of basal bud forming potential alongside the activation of other molecular signals in the interfollicular epidermis is responsible for the acquisition of the rete ridge morphogenetic program (Fig. 5e). We have revised the Discussion to reflect this updated interpretation and thank the reviewer for prompting this clarification. Thus, we propose a new mechanistic model in that rete ridges form through a dedicated morphogenetic program initiated by the loss of appendage competence and reinforced by BMP driven signaling in the basal epidermis.

2. Apical ridges

a. The initial discussion of apical ridges and dermatoglyphics in the paper led me to anticipate new insights into the relationship between rete ridges and apical ridges. However, the data pertaining to apical ridges is unclear. They claim increased length of apical ridges in the adulthood but did not present clear evidence or histological support. Furthermore, the authors did not adequately characterize the relationship between hair follicle density and apical ridge formation. There is a lack of systematic description regarding apical ridge formation.

b. In Figure 4, the authors mention the regeneration of apical ridges, but it is not clearly demonstrated. Please clarify this aspect.

We thank the reviewer for their interest in the distinctions between the rete ridges of the basal epithelium and the apical ridges defining the skin's surface dermatoglyphics. While these structures are both fascinating and developmentally patterned, in this revision we chose to refine the scope of the manuscript to focus specifically on rete ridge morphogenesis. To improve conceptual clarity and to ensure coherence around our central findings, we have removed discussion and figures related to apical ridges. However, we greatly appreciate the reviewer's

insightful questions and briefly address them below with new experimental data provided in Rebuttal Figure 1.

Additionally, the apical surface of the epidermis is marked by dermatoglyphics (also known as microrelief), which provide surface texturing, thought to aid in sweating as well as being associated with fingerprints^{1,7,17-19}. We define apical ridges based on their recurring undulating pattern on the surface of skin, which appears like sloping, horizontal ridges versus the vertically-oriented rete ridges on the basal side of the epidermis (Rebuttal Fig. 1a). Apical ridges develop early in fetal skin development, and are already visible in the earliest timepoints we examined from both human and porcine development (Rebuttal Fig. 1b).

In general, across *Mammalia*, apical ridges are longer in species with lower hair density (Rebuttal Fig. 1c). Since there is a correlation between a reduction in hair density and increased apical ridge length, we investigated apical ridge length in the trunk skin of our K14Cre Lef1 fl/fl (Lef1 eKO) mice across postnatal development, since epidermal loss of Lef1 significantly reduces hair density (Rebuttal Fig. 1c-d; Supplemental Fig. 3b-d). Newborn mice had similar apical ridge lengths but quickly diverged after the first several days of life (Rebuttal Fig. 1e). From P5 onwards, Lef1 eKO mice had significantly longer apical ridges, suggesting that the reduction in hair follicle density does result in elongation of apical ridges (Rebuttal Fig. 1c-e).

In neonatal pig dorsal wounds, we observed the regeneration of apical ridge patterns in the remodeling wound epithelium. However, these apical ridges were longer than the average adult pig's, suggesting that apical ridges can reform but are not flawlessly restored in these regenerative neonatal wounds (Rebuttal Fig. 1f-g).

[REDACTED]

Rebuttal Fig. 1: Apical Ridges Form Earlier Than Rete Ridges and are Inversely Related to Hair Density. (a) Visual definitions of apical ridges compared to rete ridge and other epidermal measurements. (b, left) Quantification of human histology for average apical ridge length at GW7-9 (n=2), GW12-13 (n=1), GW13-14 (n=1), GW14-15 (n=1), GW16-17 (n=7), GW18-20 (n=5), 21-39 year-old (21-39yo, n=1), 40-59yo (n=3), 60-79yo (n=11), 80+ yo (n=8). (right) Quantification of porcine histology for average apical ridge length at GW6-8 (n=7), GW9-11 (n=6), GW12-14 (n=14), GW15-16 (n=4), P1-2 (n=3), P3-4 (n=7), P5 (n=5), P10 (n=2), P25 (n=2), 1month-old (1mo, n=9), 1.5mo (n=2), 2mo (n=2), 6mo (n=7), 7mo (n=3), 1yo+ (n=4). (c, left) Quantification of apical ridge length by species for Yucatan pig (n=1), human trunk skin (n= 20), 6-7mo domestic pigs (n=10), Mangalitsa pigs (n=3), grizzly bears (n=11), rhesus macaques (n=6), naked mole rats (n=8), common marmosets (n=6), and adult mice (n=15). (right) Correlation between apical ridge length and hair density. Correlation statistic is the adjusted coefficient of determination and is presented on the plot, p-values < 0.001. (d) Dissecting microscope images of the apical surface of the epidermis to visualize hair follicle density between WT and Lef1 eKO mice at P5 (left) and P21 (right). Scale bar represents 1mm. Quantification (right) of P5 and P21 hair density /mm². Individual datapoints colored by sex. *** = p-value < 0.001 from t-test. (d) is reproduced from Supplemental Fig. 3d in the manuscript. (e, left) Representative H&E stains of WT and Lef1 eKO skin at P5 and P21, reproduced from Supplemental Fig. 3g in the manuscript. (right) Quantification of apical ridge length between WT and Lef1 eKO mice. *** = p-value < 0.001 from t-test. (f) Representative immunostains of the neonatal wound epidermis 28dpw and 58dpw. Cropped and zoomed in from Supplemental Fig. 7f in the manuscript. Scale bar represents 1mm. (g)

quantifications of apical ridge quantity per section and average length. One-way ANOVA, ns p-value > 0.05, ** = p-value < 0.01.

Additionally, apical ridges likely have relevancy in the context of human fingerprint ridge or murine transverse ridges in volar digits^{1,14}. However, in murine digits, murine transverse ridges create dermatoglyphics in the skin of the volar digit but not in the fingerpad (Rebuttal Fig. 2a)^{1,14}. Additionally, sweat glands form within transverse ridge domains (Rebuttal Fig. 2a)¹. A critical distinction between digital transverse ridges and fingerpad rete ridges is their developmental relationship to BMP-signaling. Transverse ridge formation is increased by BMP-inhibition¹, which is the complete opposite of our observations in the fingerpad which demonstrates BMP-inhibition reduces rete ridge formation (Fig. 5c-d; Rebuttal Fig. 2a).

3. Molecular data:

a. It is commendable that the researchers conducted single-cell RNA sequencing (scRNA) of pig skin. They then conducted comparative analyses with scRNA data from human and mouse skin available in the literature. Through CellChat analyses, they highlighted the presence of PDGFC and PDGFRA in pig and human skin but not in mouse skin. This finding presents a promising clue. However, the absence of further validation experiments is a notable limitation. Specifically, the lack of PDGFC expression demonstrated in tissue sections is a significant gap. If indeed blood vessels are the driving force behind ridge formation, it would be crucial to visualize the periodic distribution of PDGFC in the epidermal layer to establish the ridge pre-pattern effectively.

We thank the reviewer for recognizing the potential significance of PDGFC signaling in our dataset and agree that validating its spatial and molecular expression is essential to assessing its role in rete ridge formation. In this revision, we have incorporated multiple layers of new

experimental data to support and validate our previous analyses implicating epidermal PDGFC-signaling during rete ridge formation. These analyses confirmed PDGFC expression in basal epidermal cells of rete ridge forming skin but did not support a model of periodic pre-patterning. Rather, our data suggest that while epidermal PDGFC is broadly expressed, spatial organization of dermal vasculature and other dermal niche signals may also play an instructive role in the periodic patterning of rete ridges. Below, we detail the data that support this updated view:

Expanded transcriptional and spatial validation of PDGFC expression.

First, we expanded our scRNA-seq datasets to include a new P3 dataset during early rete ridge formation, which revealed the same pattern, that epidermal PDGFC-signaling was active and predicted to signal from the basal epidermis to dermal fibroblasts during rete ridge formation (Supplemental Fig. 4a-b; Supplemental Fig. 9e; Supplemental Fig. 10c-d; Supplemental Fig. 14a, d). Furthermore, we confirmed the spatial context for this predicted signaling by performing Stereo-seq spatial transcriptomics on P3, P10, and 6mo pig skin (Supplemental Fig. 4a, d), which revealed pan-basal PDGF- and PDGFC-signaling from the epidermis without periodic patterning (Supplemental Fig. 14a, d). We confirmed these transcriptional results in-vivo using immunostaining during rete ridge formation and maturation in pigs, which revealed broadly-basal distribution of PDGFC (Rebuttal Fig. 3b; Supplemental Fig. 9e).

Cross-species validation of PDGFC expression in rete ridge forming versus non-forming skin

Furthermore, since we have demonstrated that the murine volar fingerpad develops rete ridges, we validated that the fingerpad rete ridges exhibit similar spatial distribution of markers we identified in pig and human skin, including pan-basal expression of KRT15 and PDGFC during rete ridge formation and maturation (Fig. 4a; Supplemental Fig. 9c-e; Supplemental Fig. 17b). We have also validated that murine trunk skin does not express PDGFC both in scRNA-seq and in-vivo, suggesting that PDGFC may be conserved in rete ridge-capable skin (Supplemental Fig. 17c). These results suggest that PDGFC expression correlates with rete ridge competent skin, though not in a periodic pattern.

Implications for periodic patterning: epidermal vs dermal control

From our spatial transcriptomics analyses, expression of other epidermal angiogenic signaling pathways, like VEGF and ANGPTL, also appear to lack periodic patterning in the epidermis, suggesting that the dermal vascular signals, like FGF, and spatial proximity to the basal epidermis contributes to rete ridge formation's periodic patterning, instead of intrinsic epidermal factors (Supplemental Fig. 9e; Supplemental Fig. 10b-d; Supplemental Fig. 12a-b; Supplemental Fig. 14a-d). We have included discussion of these possibilities in comparison to the canonical reaction-diffusion mechanisms underlying hair follicle, sweat gland, and fingerprint ridge patterning (Fig. 5e)^{1,4,5,20}. Critically, rete ridge formation does not require LEF1/WNT or EDA/R signaling (Fig. 3a-e; Supplemental Fig. 6a-b) but does require epidermally-derived BMP-signaling and receptivity (Rebuttal Fig. 3c; Fig. 4c-e; Fig. 5b-e; Supplemental Fig. 10c-d; Supplemental Fig. 11a-b). These findings support the idea that rete ridge formation may follow a distinct morphogenetic program, where the rete ridge epidermal architecture resembles the undulating topology of a mountain range, and may be influenced by the underlying dermal organization and vascular proximity, rather than emerging from an epidermally pre-patterned mechanism like sweat gland, fingerprint ridge, or hair follicle formation (Rebuttal Figure 3a-c).

a Topography of Rete Ridge Formation

b Epidermal PDGFC Is Not Periodically Patterned Between Rete Ridge and Inter-Ridges

c BMP-signaling is Active During Rete Ridge Formation

Rebuttal Figure 3: Porcine Basal Epithelium is Broadly Supportive of Rete Ridge Formation.

(a) Representative immunostains from the basal side of epidermal wholemounts from P3 and adult pig skin stained for ITGA6/KRT15. Reproduced from Fig. 4a. (b) Representative immunostains of P3 and adult pig epidermal wholemounts for PDGFC/KRT14. Reproduced from Supplemental Fig. 9e. (c) Representative immunostains of P3 and adult pig epidermal wholemounts for SMAD1. Reproduced from Fig. 4e. Scale bars in a-b represent 100um. Dashed polygons indicate inter-ridge domains. HF=hair follicle, RR=rete ridge.

Functional testing: Epidermal BMP signaling

While we agree that functional validation of PDGFC would further strengthen the model, such experiments were not feasible in this study, due to limitations in available genetic models and tools across species. Given these constraints, we focused our functional experiments on validating the role of epidermal BMP-signaling in rete ridge formation. BMP-signaling broadly activates in the epidermis during rete ridge formation and maturation, and is predicted to signal from the epidermis to the epidermis as well as from the epidermis to the dermis (Fig. 4c-e; Supplemental Fig. 11a-b). The BMP-signaling pathway is perturbable via well-established transgenic mousselines, which we employed to ablate epidermal BMP-signaling activity during rete ridge formation (Fig. 5b-e)²¹⁻²³. In the murine fingerpad, rete ridge formation was significantly inhibited, and no vascularized dermal pockets were observed without a corresponding rete ridge (Fig. 5c-d; Supplemental Fig. 17h-i). These results support a model in which epidermal BMP is implicated in intra-epidermal and epidermal-dermal signaling interactions that are required for rete ridge morphogenesis (Fig. 5e).

b. ScRNA data is not thoroughly studied. Spatial localization of tissue sections for cell clusters from scRNA analyses are not done. Trajectory lines are not presented in a clear and readable manner.

We thank the reviewer for this constructive criticism. In response, we have both expanded our previous scRNA-seq analyses as well as performing new analyses. Ultimately, these re-analyses have improved the clarity, depth, and spatial relevance of our previous results.

We have performed scRNA-seq on an earlier neonatal timepoint to capture early rete ridge formation in P3 skin. Across the board, we present more detail on cell cluster labeling including DotPlots for the markers used to label cell clusters across all scRNA-seq datasets and subsets presented in the paper, alongside more detailed Methods describing annotation, subsetting, and integration (Supplemental Fig. 4a-c, e-j).

With the additional developmental timepoint, we have now performed three new integrative analyses on pig skin development:

Integrative Single-Cell Transcriptomics to Understand Rete Ridge Formation

1). E90, P3, P10, and 6mo Pig Basal Cell Integration: first, we integrated basal and dividing keratinocytes from across porcine skin development in order to compare the transcriptional signature of fetal skin, when basal buds are forming, versus postnatal skin when rete ridges are forming (Fig. 4a-c). This analysis revealed that E90 basal buds form a separate cluster from all other basal cells and highly express markers of basal buds and sweat glands, such as LEF1, EDAR, WNT ligands, but not hair follicle markers like SHH^{4,5,12,13} or fingerprint ridge markers like TGFA or LMX1A¹ (Fig. 3b-c). In postnatal basal cells, which clustered together from P3, P10, and 6mo, high expression for BMP- and NOTCH-signaling ligands defined these timepoints compared to E90 basal cells. These results demonstrated a distinction in the interfollicular epidermis basal cells between E90, when sweat glands are forming, and postnatally, when rete ridges are forming and maturing.

2). P3, P10, and 6mo Interfollicular Epidermal Keratinocytes Integration: next, we integrated basal, dividing, and differentiating keratinocytes from our postnatal scRNA-seq datasets to understand rete ridge formation and maturation (Fig. 4c; Supplemental Fig. 9c). Critically, these postnatal timepoints clustered into basal, dividing, and differentiating clusters without bias by timepoint, suggesting a conservation in these epidermal states during rete ridge formation and maturation (Supplemental Fig. 9c). In response to reviewer constructive criticisms, we performed a new pseudotime analysis to specifically address epidermal differentiation without noise from miscellaneous epidermal fates, such as the hair follicle, sweat gland, or basal buds, a change from our original analysis which improved the clarity of the results. We explored this conserved basal state and found that it reveals a core basal signature across rete ridge formation and maturation, which is defined by the expression of specific marker genes, like KRT15, DLK2, and DLL1 (Fig. 4a; Supplemental Fig. 9d). Furthermore, these core basal markers were more specifically basal than many canonical markers for basal states, such as KRT14, ITGA6, ITGB1, COL17A1, or SOX6 (Cockburn et al. 2022; Wang et al. 2020; Cheng et al. 2018; Glover et al. 2023; Lu et al. 2016), which continued to be expressed during early stages of differentiation while the core basal markers fell off rapidly in our pseudotime trajectory analyses (Supplemental Fig. 9c-d). To improve clarity, we have provided greater detail to the labeling of pseudotime trajectory plot axes. Furthermore, we identified many conserved signaling activities in the core basal layer, such as PDGFC and ANGPTL4, which informed our downstream Cell Chat from scRNA-seq and Spatial Cell Chat from Stereo-seq (Supplemental Fig. 9e),

3). E90, P3, P10, 6mo Dermal Cell Integration: after identifying an activation in PECAM1-signaling from papillary fibroblasts during rete ridge maturation from P10 through 6mo scRNA-seq CellChat analysis, we performed an integration of dermal cells across porcine skin development to investigate this pro-vascular phenomenon. This integrative analysis determined that during rete ridge maturation, PECAM1-signaling activated between papillary fibroblasts and blood vessels due to the emergence of a PECAM1+ papillary fibroblast subpopulation. This subpopulation expresses PDGFRA, PECAM1, and ITGA6, but not ACTA2, CDH5, or RGS5, indicating it is likely a novel fibroblast subpopulation with pro-vascular signaling activities (Supplemental Fig. 15c-d).

Spatial Localization of Cell Clusters:

We have performed extensive immunostaining validation of spatial localization of our scRNA-seq datasets. The basal bud cluster at E90 highly expresses LEF1, which is specific to basal buds in E90 skin in-vivo (Fig. 3a). Furthermore, the consistent SOX9+ keratinocyte subpopulations in our porcine scRNA-seq datasets are localized to the E90 basal bud, the outer root sheath of the hair follicle, or to the sweat glands based on both immunostaining and spatial transcriptomics (Supplemental Fig. 5a-b). Critically, SOX9 is also not localized to the interfollicular epidermis during rete ridge formation and maturation, justifying the exclusion of SOX9+ keratinocytes from epidermal integrative analyses (Supplemental Fig. 5a-b). The removal of these populations reduced noise and improved specificity of our new integrative analyses.

In our integrative analysis of postnatal porcine epidermis, we identified a core basal state conserved across P3, P10, and 6mo timepoints which was composed of undifferentiated, non-dividing cells (Fig. 4c; Supplemental Fig. 9c). We have validated key markers from this core basal state, such as KRT15, in-vivo, which demonstrate a pan-basal spatial profile in developing

and mature rete ridges (Fig. 4a), suggesting this core basal state comprises the basal layer of the epidermis, and also supporting our spatial transcriptomics analysis which did not identify distinct basal subpopulations between rete ridge and inter-ridge domains (Fig. 4a-c; Supplemental Fig. 9b-e). We observed similar in-vivo and transcriptional profiles from postnatal human trunk skin using immunostaining and our re-analysis of Solé-Boldo et al. 2020's scRNA-seq data (Supplemental Fig. 9f).

Our Stereo-seq datasets also provide another layer of support for the spatial localization of cell types from scRNA-seq, as we were able to separate epidermal keratinocytes, papillary fibroblasts, blood vessels, and lymphatic vessels into the dermal pocket (Fig. 4b; Supplemental Fig. 4d;). Importantly, like in our scRNA-seq, Stereo-seq also revealed that rete ridge and inter-ridge basal cells are transcriptionally indistinguishable. For core basal markers, like KRT15 and ITGA6, we validated in-vivo a pan-basal profile during rete ridge formation and in mature rete ridges (Fig. 4a; Supplemental Fig. 9d). Furthermore, pro-dermal signals that have a strong basal signal, like PDGFC, also appear pan-basal in-vivo and in our Stereo-seq Spatial CellChat analyses (Supplemental Fig. 9e; Supplemental Fig. 14a, d). KRT15 was also detected in both rete ridge and inter-ridge basal cells in human skin and in murine fingerpads (Supplemental Fig. 9f; Supplemental Fig. 17b). Coupled with the pan-basal profile of PDGFC in the fingerpad, we conclude that the basal epidermis does not form distinct spatial domains preceding rete ridge formation.

We also spatially validated our proposed pro-vascular fibroblast subpopulation in mature dermal pockets using immunostaining to identify some PDGFRA+/ITGA6+ cells which resemble the novel PECAM1+ papillary fibroblast subpopulation we described from our integrated dermal analysis, which is also supported by Stereo-seq Spatial CellChat identifying PECAM1-signaling within the dermal pockets (Fig. 4b; Supplemental Fig. 15a-d; Supplemental Fig. 16a).

c. The use of the EDA knockout pig to illustrate ridge formation is a positive aspect of the study. It effectively demonstrates that EDA is not essential for rete ridge formation and emphasizes the distinction between rete ridges and hair as different types of skin appendages.

We have included additional characterization of rete ridge density and a new neonatal EDA-KO timepoint (P5) in the resubmission to confirm our previous results that EDA is not required for rete ridge formation (Fig. 3e). Additionally, we developed a new Krt14Cre Lef1 fl/fl epidermal knockout mouse to confirm that the rete ridges in murine fingerpads also do not depend upon epidermal Lef1 (Fig. 3d). As a result, we have provided more thorough evidence to support that rete ridges are molecularly distinct from other types of skin appendages.

d. Timing of the specimens. The authors ought to delve into the initial stages of ridge formation. They note that rete ridges form at P2 and mature by the second week (P10 in this study). However, in their subsequent single-cell RNA sequencing (scRNAseq) analysis, the authors utilized the P10 specimen to represent the forming stage of rete ridge development, which may be too advanced to be characterized as the "forming" stage. From my reading, it appears that GW16 or the neonatal stage (P2) would be more representative of the forming stage, as depicted in Fig 1d.

We thank the reviewer for highlighting this crucial developmental window, which strengthened our revised manuscript. In response to this comment, we have performed a comprehensive analysis of neonatal skin development in pigs (**adding twelve new biological replicates**

across 3 neonatal timepoints, for a total of 19 replicates across 4 timepoints in the first ten days after birth: P1-2 (n=3), P3-4 (n=7), P5 (n=5), P10 (n=2)). This more thorough characterization of rete ridge formation reveals that developing rete ridges are first discernable by birth but mostly form within the first several days of life, particularly at P3. Therefore, we performed a new scRNA-seq experiment on P3 pig skin to capture the earliest stages of rete ridge formation. We have since substantially revised our original scRNA-seq analyses to incorporate the P3 timepoint, which dramatically strengthened our original conclusions based on E90 versus P10 and 6mo pig skin scRNA-seq (Fig. 3b-c; Fig. 4c-e; Supplemental Fig. 4a-c, e-f; Supplemental Fig. 9c-e; Supplemental Fig. 10c-d; Supplemental Fig. 11a-d; Supplemental Fig. 12a-b; Supplemental Fig. 13a-c; Supplemental Fig. 14a-d; Supplemental Fig.). Furthermore, we have also performed extensive new immunostaining emphasizing this developmental timepoint, including expanding our previous characterization of epidermal differentiation, dermal pocket composition, alongside immunostaining validation of the KRT15+ core basal state, PDGFC, and BMP/SMAD1 activity in P3 during rete ridge formation (Fig. 4a, e; Supplemental Fig. 9a, c, e; Supplemental Fig. 16a).

Furthermore, we also now define the spatiotemporal dynamics during rete ridge formation using Stereo-seq spatial transcriptomics across rete ridge formation and maturation at P3, P10, and 6mo. Combining this spatial context with our previous scRNA-seq and CellChat analyses using Spatial CellChat critically resolved epidermal BMP-signaling as a key signaling pathway activated during rete ridge formation and sustained throughout subsequent maturation (Fig. 4c-e).

We also targeted this early neonatal timeframe, including P3, during rete ridge formation in new cellular and functional studies to investigate proliferative patterning and epidermal differentiation in-vivo (Supplemental Fig. 8b-i).

Collectively, this criticism guided us to dramatically increase the manuscript's experimental observation and validation to this critical earlier timepoint to better understand and define the molecular mechanisms of rete ridge formation.

4. Mechanism: Multi-part model.

a. This aspect of the study is not convincing. The authors mention the occurrence of thicker epithelia with rete ridges, but did not provide strong evidence for a causal relationship.

We thank the reviewer for this important critique and have addressed it through additional data and revision of our mechanistic interpretation in the manuscript. In the original submission, our discussion of epidermal thickening and rete ridge formation was presented primarily as a descriptive and correlated phenomenon. However, in this revision and described below, we provide new evidence that supports an improved functional relationship between epithelial thickening and the onset of rete ridge development.

Species comparisons reveal association between ridge presence and natural epidermal thickening

Our comparative analysis of many mammalian species, including dolphins, 3 different breeds of pigs, humans, grizzly bears, rhesus macaques, naked mole rats, marmosets, and mice, demonstrates that species with rete ridge have a thicker epidermis than species that lack rete

ridges (Fig. 3a, c). Critically, through new analysis we also demonstrate that species with rete ridges exhibit a consistent ratio between rete ridge and inter-ridge thickness (Fig. 3c). Similarly, epidermal thickness increases after birth in pigs after rete ridges begin to form (Fig. 1h-i).

Hair loss alone does not drive epidermal thickening

Since there is a correlation between increased epidermal thickness and decreased hair density, we functionally tested whether genetically reducing hair density would increase epidermal thickness using K14Cre Lef1 fl/fl (Lef1 eKO) mice to conditionally ablate the WNT-signaling transcription factor LEF1 in the epidermis (Supplemental Fig. 3b). Though Lef1 eKO mice have impaired basal bud formation and dramatically reduced hair density, their interfollicular epidermal thickness did not increase at any point in postnatal skin development and maturation (Supplemental Fig. 3c-d, g-i). Therefore, the reduction in hair density observed in species with rete ridges does not appear to drive the increase in epidermal thickness.

Epidermal thickening and suprabasal expansion accompany ridge maturation

We have now performed a new comparative immunostaining analysis of basal and differentiation markers in the epidermis, which demonstrates that, during rete ridge formation and maturation, the suprabasal and spinous compartment expands as pigs develop rete ridges, while the epidermis is thinner and simpler in species that lack rete ridges (Fig. 2c; Supplemental Fig. 9a). Similarly, our P3, P10, and 6mo Stereo-seq spatial transcriptomics datasets revealed thicker KRT10+ differentiated layers following the acquisition of rete ridge epidermal architecture (Supplemental Fig. 9b).

Rete ridge growth is driven by proliferation, not differentiation

We have now also defined the contribution of epithelial proliferation and differentiation to rete ridge formation through comprehensive immunostaining of proliferation using MKI67 and also in-vivo BrdU-labeling to track proliferation and differentiation. Epidermal thickness due to the rete ridge begins to increase after birth during the heightened proliferation in the rete ridge after P3, when basal and suprabasal proliferation increase compared to the inter-ridge (Supplemental Fig. b-d). When we monitored differentiation using BrdU-labeling, we observed similar BrdU-labeling in each layer of the epidermis between rete ridges and inter-ridges, suggesting that each domain is not differentiating differently, consistent with previous studies suggesting that mature rete ridge and inter ridge domains function as epidermal proliferation units (Supplemental Fig. 8e-i)²⁴. Collectively, these results suggest that proliferative patterning supports early rete ridge downgrowth and epidermal thickening, while consistent differentiation between the rete ridge and inter-ridge then sustains these regional differences in epithelial thickness (Supplemental Fig. 8b-i).

Dermal pocket contribution and vascular signaling

The dermal pocket, as a dermal niche associated with the rete ridge, also likely contributes to epidermal thickening. Critically, though vascular recruitment is associated with rete ridge formation, the thickening and downgrowth of the rete ridge appears to precede the aggregation of the dermal-vascular lineages (Fig. 3a; Fig. 5a; Supplemental Fig. 9a; Supplemental Fig. 16a; Supplemental Fig. 17a-b). Subsequent epidermal thickening and maintenance may be supported by the epidermal-dermal interactions facilitated by the dermal pocket, such as through EGF-, TGFb-, PDGF-, ANGPTL-, and VEGF-signaling²⁵⁻³⁰. Previous studies have

demonstrated in-vivo that over-expression of VEGF- and ANGPTL-signaling from murine trunk skin epidermis causes epidermal thickening, likely through the recruitment of underlying dermal fibroblast and vascular lineages³¹⁻³⁴.

Together, these revisions reinforce the idea that epidermal thickening is a necessary and potentially initiating component of ridge formation. While future work will be needed to dissect its sufficiency or isolate its effects from dermal contributions, our revised data framework supports a functional role for epithelial architecture in shaping ridge morphology.

b. Rete ridges in adult human has been well studied, with scientists showing the periodic distribution of epidermal progenitor cells in the ridge and inter-ridge regions. Authors could have followed this clue to show how progenitor and differentiated cells are distributed in other animals. Although specimens from different animals present a valuable opportunity to gain insight, this aspect of the study remains underdeveloped.

We thank the reviewer for raising this important point. The periodic arrangement of progenitor and differentiated cells in the adult human skin is a foundational observation in epithelial biology, and we agree that addressing its potential conservation across species is critical to interpreting rete ridge morphogenesis.

Analysis of epidermal progenitors with single cell and spatial transcriptomics

First, we have added another developmental timepoint, P3, to our previous E90, P10, and 6mo scRNA-seq analyses and also performed Stereo-seq spatial transcriptomics on P3, P10, and 6mo skin to better understand the single-cell and spatial context during early rete ridge formation through postnatal maturation (Supplemental Fig. 4a).

From these analyses we determined that the rete ridge and inter-ridge domains do not contain spatially distinct basal cell states, and exhibit broadly uniform expression of both basal markers and signaling ligands activated during rete ridge formation (Fig. 4a-e; Supplemental Fig. 9b-f). Therefore, the postnatal epithelium seems broadly capable of supporting rete ridge formation while basal progenitors do not seem distinguishable between rete ridge and inter-ridge domains (Fig. 4a-b; Supplemental Fig. 6b, d; Supplemental Fig. 7b).

Multi-species analysis reveals a conserved basal state

To address the reviewer's interest in progenitor versus differentiated states between mammalian species, we have expanded our previous immunostaining from pig skin development to present the basal versus differentiating layers in skin across many species, including those with rete ridges like pigs and rete ridge-less species like mice, marmosets, and naked mole rats (Fig. 4a; Supplemental Fig. 9a). All of these species share a core basal layer that is KRT15+/ITGA6-high/KRT14-high, but the thickness of the differentiated layers is variable, being most pronounced in the rete ridge domains (Supplemental Fig. 9a). Importantly, the volar fingerpad rete ridges of mice closely resemble porcine and human rete ridge epidermal stratification (Fig. 4a; Supplemental Fig. 9a, c, e, f; Supplemental Fig. 17b).

Species-specific signaling may regulate rete ridge competence

Since a basal layer is present in all species, we proposed that there are additional differences in the basal interfollicular epidermis of animals with rete ridges and those that lack them, such as BMP, PDGFC, and NOTCH which are associated with rete ridge-capable skin but not fetal pig trunk or fetal human volar skin () which may regulate the ability for some types of skin to form rete ridges while others cannot (Fig. 4c-e; Fig. 5b-d; Supplemental Fig. 9e-f; Supplemental Fig. 17b-c, e-g).

Differentiation kinetics and pseudotime support shared trajectories

In the context of initiation of differentiation and the exit from the basal state, we utilized in-vivo BrdU-labeling to track functional differentiation in rete ridge versus inter-ridge domains during rete ridge formation (Supplemental Fig. 8e). We observed similar labeling between rete ridge and inter-ridge domains and similar depletion of basal BrdU-label, consistent with cells exiting the basal state to differentiate upward, as in the canonical epidermal proliferation unit (Supplemental Fig. 8e-i) ²⁴. Similar conclusions were obtained from performing an integrative pseudotime analysis of postnatal basal keratinocytes, as the shared core basal state does not distinguish between rete ridge and inter-ridge domains, and cells undergoing through a shared, normal differentiation trajectory (Fig. 4a-c, e; Supplemental Fig. 9b-d).

We believe this model offers a revised framework for understanding epidermal heterogeneity, especially in the context of rete ridges. Instead of the developing epidermis exhibiting pre-patterning indicative of rete ridge formation, the basal epidermis appears broadly capable of supporting rete ridge formation without binarizing into distinct rete ridge and inter-ridge transcriptional or functional units.

c. In the multi-part model, the authors propose that ridge formation involves both increased epidermal thickness and vascular recruitment. However, it remains unclear which factor serves as the primary driving force. In one section subtitle suggests that rete ridge formation drives epidermal thickness and dermal vascular recruitment. Yet in Figure 7 indicates that vasculature is the driving force. The question arises: which is the earlier determinant of the rete ridge formation process, the dermal pocket, or the rete ridge structure? Clarifying the earlier stages of ridge formation, as mentioned in point 3d above, may provide greater clarity on this matter.

We thank the reviewer for raising this important question, which we agree sits at the center of understanding how epithelial and mesenchymal compartments coordinate during skin morphogenesis. This comment prompted us to refine both the structure of our manuscript and the interpretation of our model.

In our original submission, the section heading implied a linear model in which rete ridge formation drives subsequent epidermal thickening and vascular recruitment. We have changed the heading to: "Postnatal activation of epidermal-dermal signaling drives rete ridge and dermal pocket formation." This revised framing better reflects our updated data and interpretation: that the formation of rete ridges and associated dermal pockets involves dynamic, reciprocal signaling between epidermal and dermal cells, particularly during early postnatal development.

Our newly added Stereo-seq data across P3, P10, and 6mon skin (Figure 4a-d; Supplementary Figure 10) support this revised view revealing coordinated spatial expression of signaling factors

between the epidermal and dermal compartments during both rete ridge formation and maturation (Fig. 4b, d; Supplemental Fig. 10a-b; Supplemental Fig. 11a-d; Supplemental Fig. 12a-b; Supplemental Fig. 13a-c; Supplemental Fig. 14a-d; Supplemental Fig. 15a-d). Furthermore, we have spatially validated that many of these signals are pan-basal in expression, suggesting the epidermis is broadly compatible with rete ridge formation (Fig. 4a, e; Supplemental Fig. 9e; Fig. 5b; Supplemental Fig. 17b; Rebuttal Fig. 3b). Immunostaining for markers of dermal cell lineages during dermal pocket assembly, coupled with our Stereo-seq data, suggests that dermal vascular organization aligns with developing rete ridge structures rather than directly preceding it (Supplemental Fig. 10a-b; Supplemental Fig. 11a-b; Supplemental Fig. 12a-b; Supplemental Fig. 13a-c; Supplemental Fig. 16a-e. Supplemental Fig. 17a). Importantly, our BMP signaling perturbation experiments in murine volar fingerpads supports a model for rete ridge formation in which epidermally derived signals are sufficient to control ridge formation, reinforcing the functional relevance of these epithelial cues (Fig. 5b-e).

VEGF- and ANGPTL-signaling have previously been implicated in increasing epidermal thickness in mouse models^{31,32}. In-vitro studies have previously demonstrated that human dermal fibroblasts and pericytes both support epidermal thickening in co-culturing experiments, suggesting that the dermal pocket may play a functional role in supporting epithelial thickening during rete ridge formation and maturation^{33,34}.

However, when epidermal BMP-signaling was inhibited in the fingerpad, we did not observe basal localization of vasculature, suggesting that the epidermal component of the rete ridge structure is important to recruiting and maintaining the vascular presence adjacent to the epidermis (Fig. 5c-d; Supplemental Fig. 17h-i). Based on our CellChat and Spatial CellChat analyses, epidermal BMP-signaling may also be received by dermal vascular lineages including blood vessels (Fig. 4d; Supplemental Fig. 11a-b). We also demonstrated that epidermal receptivity to BMP-signaling is required for rete ridge formation, which is agnostic to the epidermal or dermal origin of BMP signal (Fig. 5c-d; Supplemental Fig. 17g-i). Furthermore, in the absence of the epidermis's ability to functionally receive BMP-signaling ligands via BMPRI1A, dermal pockets were not observed without an associated rete ridge (Fig. 5c-d; Supplemental Fig. 17h-i). These results implicate epidermal BMP-signaling as an absolute requirement for both rete ridge and dermal pocket formation (Fig. 5e). However, they do not preclude a supporting role of dermal vascular lineages in rete ridge formation, as our CellChat and Spatial CellChat analyses predicted both epidermal-vascular and vascular-epidermal BMP-signaling through multiple BMP ligands (Fig. 4d; Supplemental Fig. 10c-d; Supplemental Fig. 11a-b). Coupled with epidermal BMP-signaling ligand inhibition via Noggin using the K14-Noggin mouse, these results suggest that the epidermal component of BMP-signaling during rete ridge formation must be satisfied, lest neither epidermal nor dermal niche may form (Fig. 5c-e). These results suggest a co-dependency between rete ridge and dermal pocket formation, instigated by the epidermis but perhaps spatially patterned by vascular proximity or other downstream epidermal-dermal signaling.

That said, we agree that determining the precise sequence and interdependence of these morphogenetic events remains an open and important question. Our goal in the revised manuscript was not to close the question prematurely, but to lay a clearer framework and dataset for future studies to dissect the mechanisms of epidermal-dermal coordination in the establishment of rete ridge architecture. We thank the reviewer for helping us sharpen this narrative.

d. Given that rete ridges exhibit periodic arrangement, it raises the question of what factors control the periodic outgrowth of vasculature. The authors have identified PDGFC as a potential molecule involved in this process. Therefore, this presents an opportunity to conduct further testing and investigation to validate the role of PDGFC and explore its mechanisms in regulating the periodic outgrowth of vasculature within rete ridges.

We thank the reviewer for their interest in the patterning of rete ridges and the underlying vasculature. In our extensive re-analysis and expansion of the E90, P3, P10, and 6mo pig scRNA-seq data and new P3, P10, and 6mo Stereo-seq data, we were able to identify that the basal epithelium seems to be broadly uniform in its gene expression and signaling activity (Fig. 4a-b; Supplemental Fig. 6b, d; Supplemental Fig. 7b), while the spatial organization of the dermal vasculature appears to contribute to where rete ridges will form (Fig. 4a-b; Supplemental Fig. 6b, d; Supplemental Fig. 7b, g). Our observations suggest that rete ridge and dermal pocket formation are linked due to their consistent spatial association. We have attempted to clarify this in the text as well.

To clarify an important distinction, in our revision we now explicitly differentiate between fingerprint ridges, which arise from Lef1/EDAR basal buds in fetal development and follow a spot to appendage model (Li et al. 2022; Glover et al. 2023), and rete ridges of the murine fingerpad, which form postnatally and do not require Lef1. These fingerpad rete ridges emerge after sweat gland morphogenesis and exhibit a distinct architecture and signaling environment (Fig 5a; Supplemental Fig6a-b, 17a-b). We introduce this model in part to replace the earlier basal bud induction idea.

Regarding PDGFC, with our new scRNA-seq dataset at P3, we have performed a new integrative analysis of P3, P10, and 6mo porcine epidermis. PDGFC expression is highly expressed in scRNA-seq and in-vivo in the KRT15+ core basal state comprising the basal layer of the epidermis in both porcine trunk skin and murine fingerpads during rete ridge formation and maturation (Fig. 4a-c; Supplemental Fig. 9c-e; Supplemental Fig. 17b). Furthermore, PDGFC is also highly expressed in the same KRT15+ basal population in human skin (Supplemental Fig. 9f). We have also validated in-vivo that epidermal PDGFC is expressed in rete ridge-bearing skin, such as porcine trunk and murine fingerpads, but not in rete ridge-less murine trunk skin (Supplemental Fig. 9e; Supplemental Fig. 17b). Previous studies have also implicated PDGFC in angiogenesis, beyond merely serving as a pro-fibroblast signal, and have suggested that it may act on vasculature through the shared fibroblast and pericyte receptor PDGFRB beyond the canonical PDGFRA^{28,30}.

This distinction motivates a revised morphological framework. Rather than forming as discrete, periodic appendages akin to hair follicles or sweat glands, rete ridges appear to develop into an interconnected network of ridges, which projects downward into the epidermis while the dermal pocket fills the volumetric cavity between the ridging (Rebuttal Fig 3). This interconnected appendage model is supported by new whole-mount and spatial transcriptomics data which results in a reticular patterning without the individual patterning of other epidermal appendages. We propose that the spatial arrangement and recruitment of dermal vascular structures, not intrinsic epidermal pre-patterning is a primary driver of the resultant rete ridge topography (Supplemental Fig 9e; Fig 5e).

While epidermal BMP-signaling is required for rete ridge formation, we observed broad activity of BMP-signaling along the basal epidermis in both rete ridge and inter-ridge regions.

Furthermore, we observed little to no difference in the basal epithelium in the rete ridge versus inter-ridge regions, suggesting the basal epidermis may be broadly compatible with rete ridge formation. Since FGF-signaling is a classical activatory input into reaction-diffusion systems (REF Juong 1998), and dermal FGF-signaling is activated during rete ridge formation (Supplemental Fig. 10c; Supplemental Fig. 12a-b), the arrangement of the underlying dermal vasculature may provide this spatial signaling context to support periodic patterning of rete ridge formation. Though we consistently show vascular localization in the dermal pocket, it remains unresolved whether dermal vasculature may also be required for rete ridge formation or periodic patterning independent of epidermal BMP.

Dermal vasculature may still play a supporting role in rete ridge formation, as our scRNA-seq CellChat and Stereo-seq Spatial CellChat analyses predict epidermal-dermal signaling interactions via BMP-signaling, such as epidermal BMP7 signaling to dermal cell lineages (Fig. 4d-e; Supplemental Fig. 11a-b). Additionally, epidermal signaling to the dermis via PDGF-, VEGF-, and ANGPTL- provides further support for an epidermal role in the recruitment of dermal cell lineages and vasculature into the dermal pocket (Fig. 4b-c; Supplemental Fig. 9c-f; Supplemental Fig. 10c-d; Supplemental Fig. 14a-d). Epidermal signaling to recruit dermal-vasculature via VEGF- and ANGPTL- has also been demonstrated in mouse models to increase epidermal thickness^{31,32}.

5. Engineered rete ridges:

- a. The authors could have bolstered their study by providing additional molecular evidence to confirm that the "engineered rete ridges" are indeed authentic rete ridges, distinct from merely an uneven epithelial undersurface, as commonly observed in mouse wound healing models.
- b. The concurrent emergence of vasculature and thickening of the epidermis also have been observed during "normal" wound repair.

We thank the reviewer for raising this important point. In response, we have significantly revised our strategy to enhance the developmental relevance and mechanistic clarity of our study.

To enhance our study, we have replaced the engineered model with the mouse footpad, a region which we demonstrate naturally forms rete ridges during development in a similar process to porcine trunk skin (Fig. 3a, d-e; Fig. 4a-e; Supplemental Fig. 5a-b; Supplemental Fig. 6a-c; Supplemental Fig. 10a-b; Supplemental Fig. 9a-e; Fig. 5a-e; Supplemental Fig. 17a-b). Critically, murine volar fingerpad rete ridges form postnatally after the cessation of sweat gland formation, similar to porcine trunk skin (Fig. 3a; Fig. 5a; Supplemental Fig. 5a; Supplemental Fig. 6a-b; Supplemental Fig. 17a-b). Additionally, fingerpad rete ridges exhibit similar epidermal stratification and dermal pockets to human and porcine rete ridges (Supplemental Fig. 9a, c, e-f; Supplemental Fig. 16a-b, d; Supplemental Fig. 17a). Therefore, the fingerpad presents a novel and intriguing model to allow for investigation of rete ridge formation in mice and exploit the abundance molecular and transgenic tools that are widely employed in murine models.

We observed that epidermal BMP-signaling is activated during rete ridge formation in pigs and confirmed that epidermal BMP-signaling is also activated in murine fingerpads (Fig. 4c-e; Supplemental Fig. 10a-b; Fig. 5b). Therefore, we employed K14-Noggin and K14CreERT Bmpr1a fl/fl mice to inhibit epidermal BMP-signaling in the fingerpad through two different

mechanisms to validate that epidermal BMP-signaling is required for rete ridge formation (Fig. 5c-d; Supplemental Fig. 17h-i). These multi-species comparisons have created strong evidence to support the key role of epidermal signaling in rete ridge morphogenesis (Fig. 5e).

6. Writing

The writing is dense and narrative in nature. There are many long sentences. These can be improved to show the message clearer.

We thank the reviewer for drawing our attention to the writing style. We have considerably re-written almost the entirety of the manuscript in response to reviewer feedback and to incorporate the extensive array of new experiments. In this re-writing, we have attempted to improve the overall clarity and readability of the text.

Referee #2 (Remarks to the Author):

In the manuscript entitled “Evolutionary development, regeneration, and induction of rete ridges in mammalian skin”, the authors employed a multi-species comparative approach to investigate the timing and mechanisms of the development and regeneration of rete ridges (RRs) in the mammalian skin. Through histological and molecular analyses, they found that RRs form prenatally through a molecular mechanism distinct from that of hair follicles, sweat glands, and fingerprints. In a comparative study across multiple mammalian species, the authors discovered a correlation between RR formation and an increased epidermal thickness, coupled with a decrease in hair density. Utilizing comparative transcriptomics analyses, the authors identified several signal pathways that potentially contribute to RR development. Additionally, the authors characterized the process of RR regeneration after wound in neonatal pig, as well as devised an approach to induce RR-like structures in juvenile mice through wound and pharmacological treatment. The authors have established powerful comparative models and generated potentially interesting datasets. However, this manuscript lacks functional test data to pinpoint the mechanisms that drive RR development or regeneration, and some conclusions lack support from the current data.

Major:

1) The proposed mechanisms lack experimental support.

1.1) Through transcriptomics analyses, the authors identified candidate signaling pathways that potentially contribute to RR formation. However, as no RR-specific subpopulation was identified and the analyses were conducted on general epidermal cells, it remains uncertain whether the candidate signals function in regulating RR formation. No functional test was performed to demonstrate the requirement of these signals for RR development or regeneration.

Furthermore, the statement that these signals “are responsible for increased epidermal thickness, decreased hair density, and enhanced dermal vasculature recruitment between rete ridges” (line 38-40) are not supported by experiments.

We have added another developmental timepoint, P3, to our scRNA-seq analyses and also performed Stereo-seq spatial transcriptomics on P3, P10, and 6mo skin to better understand the single-cell and spatial context during early rete ridge formation through postnatal maturation (Supplemental Fig. 4a). In addition, we have performed functional experiments to determine the role of proliferative patterning and differentiation between the rete ridge and inter-ridge domains alongside validation of epidermal BMP-signaling being a requirement for rete ridge formation

(Supplemental Fig. 8a-l; Fig. 4c-e; Fig. 5b-e). Critically, we have performed extensive new immunostaining during rete ridge formation in pigs, including epidermal wholemount methods which demonstrate a clear, broad basal epidermal signature during rete ridge formation (Fig. 4a, e; Supplemental Fig. 9a, d-e). These methods reveal that rete ridge formation establishes an interconnecting network of ridges along the basal epithelium without clear epidermal markers demonstrative of periodic pre-patterning.

From these new scRNA-seq and Stereo-seq analyses we determined that the rete ridge and inter-ridge domains do not contain transcriptionally distinct subpopulations of basal cells, and instead exhibit broadly uniform expression of both basal markers and signaling ligands activated during rete ridge formation (Fig. 4a-e; Supplemental Fig. 9b-e; Supplemental Fig. 11a-d; Supplemental Fig. 13a; Supplemental Fig. 14a-d). Similar pan-basal patterns for this shared basal state were observed in human trunk and murine fingerpad rete ridges, as well (Supplemental Fig. 9f; Supplemental Fig. 17b). Therefore, the postnatal epithelium seems broadly capable of forming rete ridges. (Fig. 4a-b; Supplemental Fig. 6b, d; Supplemental Fig. 7b), while the spatial organization of the underlying dermal vasculature may support the periodic patterning of rete ridge formation during its recruitment into the developing dermal pocket niche (Fig. 4a-d; Supplemental Fig. 6b, d; Supplemental Fig. 7b, g; Supplemental Fig. 10a-b; Supplemental Fig. 12a-b; Supplemental Fig. 13a-c). Our observations suggest that rete ridge and dermal pocket formation are linked due to this consistent spatial association. We have attempted to clarify this in the text as well.

From our new single-cell and spatial transcriptomics analyses, we now characterize activation of epidermal BMP-signaling during porcine rete ridge formation and maturation (Fig. 4c-e; Supplemental Fig. 10c-d; Supplemental Fig. 11a-b). Furthermore, murine volar fingerpad rete ridge formation also exhibits activated BMP-signaling (Fig. 5b). Using the murine fingerpad as a proxy for porcine and human trunk skin, we have validated that epidermal BMP-signaling is required for rete ridge formation using two different transgenic mousselines to inhibit BMP-signaling (Fig. 5b-e). Though we observed no dermal pocket formation without rete ridge formation in the fingerpads of mice with inhibited epidermal BMP-signaling and consistently show the spatial association between rete ridges and the dermal pocket, it remains unresolved whether dermal vasculature may also be separately required for rete ridge formation or be the driving agent of rete ridge patterning (Fig. 5c-d; Supplemental Fig. 16a-d; Supplemental Fig. 17a-b, h-i).

Regarding PDGFC, with our new scRNA-seq dataset at P3, we have performed a new integrative analysis of P3, P10, and 6mo porcine epidermis. PDGFC expression is highly expressed in scRNA-seq and in-vivo in the KRT15+ core basal state comprising the basal layer of the epidermis in both porcine trunk skin and murine fingerpads during rete ridge formation and maturation (Fig. 4a-c; Supplemental Fig. 9c-e; Supplemental Fig. 17b), and also appears to be a basal marker in human skin (Supplemental Fig. 9f). Critically, we have now validated in-vivo that epidermal PDGFC is expressed in rete ridge-bearing skin but not murine trunk skin (Supplemental Fig. 9e; We have tempered the language in the revised text used to describe the relationship between epidermal vascular recruitment and epidermal thickening, though ample experimental evidence from in-vivo experiments has demonstrated that epidermal expression of angiogenic signaling pathways, such as VEGF and ANGPTL, increases epidermal thickness

We have also provided greater characterization of the independence of rete ridge formation from the basal bud molecular program. We have expanded our previous analysis of EDA-KO pigs to critically compare WT and EDA-KO rete ridge formation in neonatal skin (Fig. 3e) and also used K14Cre Lef1 fl/fl mice to ablate epidermal LEF1, which impaired basal bud and hair follicle formation but did not affect fingerpad rete ridge formation (Fig. 2f; Fig. 3d).

1.2) The statement that dermal vasculature recruitment is required for RR formation (line 44, 340, 408) lacks experimental support. No vasculature-specific experiments were conducted to test the requirement of vascular recruitment for RR development or regeneration. The assessment of vasculature distribution was qualitative (line 309-314). The statement that mouse skin is less vascularized than pig lacks support through quantification. Analyzing vasculature distribution on a single section and claiming sparser presence of dermal vasculature outside of dermal pocket is also not appropriate. A 3D whole mount analysis is more suitable.

We thank the reviewer for this critical feedback. We have rephrased discussions of the vascular involvement in rete ridge formation. We have also now demonstrated that broad epidermal BMP-signaling activation is associated with rete ridge formation in neonatal pig trunk skin and murine volar fingerpads (Fig. 4c-e; Fig. 5b-e). Critically, we have also validated that epidermal BMP-signaling is required for rete ridge formation using two different transgenic mouse lines, K14-Noggin and K14CreERT Bmpr1a fl/fl (Fig. 5c-e).

Dermal vasculature may still play a supporting role in rete ridge formation, as our scRNA-seq CellChat and Stereo-seq Spatial CellChat analyses predict epidermal and dermal signaling interactions via BMP-signaling, such as epidermal BMP7 signaling to dermal cell lineages (Fig. 4d-e; Supplemental Fig. 11a-b). Additionally, epidermal signaling to the dermis via PDGF-, VEGF-, and ANGPTL- provides further support for an epidermal role in the recruitment of dermal cell lineages and vasculature into the dermal pocket (Fig. 4b-c; Supplemental Fig. 9c-f; Supplemental Fig. 10c-d; Supplemental Fig. 14a-d).

It has previously been well established through a variety of techniques utilizing specialized equipment that human skin is distinguished by high vascularization³⁵, and our qualitative comparison of dermal vascularization in the skin of many species is consistent with these reports. Many studies draw conclusions related to relative dermal vascularization based on cross-sectional analyses^{31,32}. From our original submission, we have expanded immunostaining characterization of the dermal pocket to more mammalian species, including more timepoints during porcine rete ridge formation, human skin development, other regions of murine tissue such as fingerpads and oral epithelium, and even dolphins to show the conservation in dermal cell lineages within the dermal pocket (Supplemental Fig. 16a-e). We have also adjusted the presentation in the text to avoid claims about vasculature which are not supported by experimental evidence.

1.3) It is interesting that the authors were able to induce RR-like structures in mice. Despite their similar morphology, the molecular characteristics and functions of these structures compared to RR remain unclear. Even if these structures are equivalent to RR, the various effects of TPA and RA on epidermal cells and other cell types coupled with the intricate wound-induced signals and inflammatory environments, make it difficult to pinpoint the mechanisms driving their formation.

We thank the reviewer for raising this essential point. In the original manuscript, we presented an inducible model of epidermal ridge-like structure formation via topical TPA and RA treatment. While this approach generated structures morphologically resembling rete ridges, we fully acknowledge the reviewer's concern: the pleiotropic nature of these pharmacological compounds and poorly defined molecular mechanisms influencing epidermal differentiation, inflammation, and wound repair is a limitation of the approach in clearly elucidating molecular mechanisms underlying rete ridge formation. We took this concern seriously and have revised our approach accordingly:

In the revised manuscript, we have substantially restructured our use of mouse models to reflect a more refined and biologically grounded strategy. We now define and emphasize rete ridge formation in murine volar fingerpads as a native, genetically tractable site of true rete ridge formation (Fig. 3d; Fig. 5a; Supplemental Fig. 6a-c; Supplemental Fig. 17a-b). This model is not chemically induced, nor wound-dependent, and occurs naturally during postnatal development. As such, it provides a powerful opportunity to interrogate ridge morphogenesis in a physiologically relevant context.

Importantly, the volar fingerpad ridge structures are not just morphologically similar to those in human and porcine skin, they are molecularly and spatially conserved. These murine structures exhibit BMP and PDGFC signaling activity, thickened layers, and a stereotyped invagination of the basal epithelium (Figure 5a, Supplemental Fig 6a-b; Supplemental Fig 17a-b). We believe this directly strengthens the central claims of our manuscript.

To further demonstrate that these ridges represent a biologically distinct developmental program, we genetically ablated Lef1 in the epidermis of mice using K14Cre Lef1 fl/fl (Lef1 eKO) mice (Figure 2e-f; Supplemental Fig 3a-i). These mice exhibit dramatically reduced hair follicle formation, confirming loss of the basal bud program, and still form rete ridges in the fingerpad (Figure 3d; Supplemental Fig. 3c-d). This key finding supports our central claim that rete ridge formation is governed by a novel, non-follicular/glandular epithelial differentiation program.

We are grateful for the reviewer's critique, which helped us elevate this aspect of the study from descriptive to mechanistic. This addition has strengthened the conceptual clarity of the work.

1.4) No new cellular mechanisms were elucidated. The speculations that "the epidermal thickness in rete ridge-bearing skin increases due to an expanded differentiated epidermis without an apparent expansion in the basal layer" (line 287-289) mediated by RA and RA supports de-differentiation (line 404) appear contradictory and were not tested.

We thank the reviewer for this valuable critique. In response, we have refined the mechanistic framing of the manuscript to focus on physiologically relevant models of rete ridge formation, specifically, postnatal porcine trunk skin and murine volar fingerpads.

To address the core of the reviewer's concern, we have added new in vivo Mki67 staining and BrdU labeling analyses that demonstrate regional differences in proliferation and differentiation during ridge development (Supplemental Fig 8a-i). These data show that epidermal thickening in rete ridge regions is primarily driven by suprabasal expansion, not just basal layer enlargement. This is supported by new immunostaining and spatial transcriptomics datasets (Supplemental Fig 9a-d), which reveal increased Krt10+ differentiated layers after the acquisition of ridge architecture, and by integrative pseudotime analyses (Fig 4c-e), which

demonstrate a shared basal progenitor state within rete ridge and inter-ridge domains undergoing canonical upward differentiation ²⁴.

Critically, we now present strong evidence that epidermal BMP signaling is required for rete ridge formation. Using K14-Noggin and K14CreER Brmpr1a fl/fl mouse models, we show that inhibition of BMP ligand or receptor activity in the epidermis leads to a failure of rete ridge formation in the mouse fingerpad (Figure 5b-e; Supplemental Fig 17e-i). BMP activity is enriched in the rete ridge forming regions across species, as indicated by the downstream BMP effectors SMAD1/5 (Figure 4 e; Figure 5b), and in our bioinformatic analysis of our single cell data (Figure 4d-e).

Together, these molecular, spatial, and functional datasets now support a novel mechanistic model of ridge formation in that the basal epidermis maintains a broadly competent progenitor state, while epidermal BMP signaling is required to establish the interconnecting rete ridge network along the basal interfollicular epidermis. This represents a significant advance in understanding how tissue architecture and signaling intersect to shape human-defining skin features.

2) The authors sampled skin from pigs at different ages to conclude that RR forms perinatally. However, this finding is not novel, given that the presence of RR in perinatal human skin is already established based on foreskin analyses. The authors did not further refine the time window.

We thank the reviewer for pointing out the presence of rete ridges in neonatal human skin. However, we respectfully note that there remains a substantial gap in our understanding of human epidermal development between late gestation and early post-natal life. Due to ethical and practical limitations, high quality human skin samples spanning this transitional window, particularly from trunk skin, are extremely rare (see Figure 1d). As a result, much of what is known about human rete ridge architecture derives from neonatal foreskin, which has key limitations that are often overlooked.

Most studies using foreskin tissue ^{24,36,37} do not report the precise age of the sample collection. The reason for circumcision, medical vs. ceremonial, is typically not disclosed, yet this has important implications for the age and developmental state of the tissue sample. For instance, in Jewish tradition circumcision is commonly performed on the eighth day, while in Islamic contexts the timing can be highly variable but often occurs well after the first week of life ³⁸. Thus, while these foreskin samples are often labeled “neonatal” in the literature, they represent undefined and potentially older postnatal stages which may not actually capture the early stages of human rete ridge formation.

In addition, studies focused on neonatal foreskin fail to account for sex as a biological variable, further limiting generalizability of studies based on neonatal male foreskin. In contrast, our pig model includes clearly staged specimens across the first ten days of life and incorporates both sexes. This allowed us to define a narrow, clearly defined developmental progression for rete ridge formation that neonatal foreskin samples cannot fulfill. Specifically, that rete ridge formation peaks within the first few days of life and begins to exhibit mature rete ridge morphology by P10 (Fig. 1g-i). This level of temporal resolution is not currently available in human datasets.

Finally, the histological variability in published foreskin studies itself supports our concern. Representative images from different studies show considerable differences in ridge size, spacing, and organization. Most presentations of human neonatal foreskin also lack clear quantification of epidermal thickness or rete ridge density, masking these developmental inconsistencies. This suggests that sampling variability and unknown postnatal age may confound comparisons.

Together, these issues highlight the need for better-controlled and clearly defined developmental models for human skin. Our developmental comparison between pigs and humans suggests that porcine skin is a highly similar and ethically tractable proxy for human fetal through postnatal skin development broadly, and especially in the context of understanding rete ridge formation and maturation.

There is an enormous gap in knowledge in late fetal human skin development through adolescence due to the ethical limitations on obtaining and studying high quality samples from this developmental timeframe (Fig. 1d). In addition, since the bulk of the studies the reviewer is suggesting is based on foreskin analysis is incomplete since these studies do not take sex as a biological variable into account. For example, literature utilizing human neonatal foreskin does not generally specify the exact age of the sample collection ^{24,36,37}. The reason, medical versus religious, for circumcision is also not provided. This metadata is important to understand the likely actual age of the tissue being assessed and conclusions drawn from neonatal foreskin in scientific research. It is therefore inaccurate to conclude that literature data from neonatal foreskin is broadly representative of the first days of postnatal life.

Additionally, it is unclear how exactly neonatal foreskin differs from neonatal trunk skin. Previous studies comparing skin from different regions and disease states in human skin have identified distinct subpopulations present in the epidermis, dermal, and immune lineages depending upon the body region in adults (e.g. scalp versus trunk) or with neonatal foreskin ³⁷. Since neonatal trunk skin is not adequately represented in literature, it remains unclear how much of these differences are due to age gaps between neonatal foreskin and adult scalp or trunk, and how much of these differences are due to body region ^{36,37,39,40}.

Using the porcine model, we provide thorough quantification of rete ridge development and epidermal thickening across clearly defined timepoints (Fig. 1h-i), compared to clinical literature which generally does not temporally resolve foreskin samples and thus experiences considerable variation in measurements ⁴⁰. Therefore, our comprehensive study of neonatal development in pigs (**adding twelve new biological replicates across 3 neonatal timepoints, for a total of 19 replicates across 4 timepoints in the first ten days after birth: P1-2 (n=3), P3-4 (n=7), P5 (n=5), P10 (n=2)**) is a novel and important effort to clearly resolve both the developmental timing and the molecular mechanisms underlying rete ridge formation in humans.

Additionally, our developmental analyses of trunk skin in humans and pigs incorporated both males and females, while human foreskin is only representative of biological males. As such, our conclusions about rete ridge formation in neonatal pig and human trunk skin remain novel.

3) Regeneration of RRs after wound has been observed and characterized in young pigs (Kuo, *Sci Rep* 2022: PMID: 35013386; Lin, *Exp Dermatol* 2019: PMID: 30629757). Hence, the discovery of RR regeneration in neonatal pigs is not novel. Furthermore, the time window for the

initiation of RR regeneration was not captured, as RRs had already formed at the first time point (28dpw) of analysis.

We present neonatal rete ridge regeneration not as the main takeaway in this manuscript but to highlight the potential for developing rete ridges to restore themselves during wound healing. If the primary criticism of this experiment is that other studies have previously investigated wound healing in pigs, then our findings must stand. Critically, the suggestion that our experiment is similar in design or outcome as these two other articles is inaccurate, as the ages and outcomes of their wound healing analyses differ considerably from the neonatal results we present in our manuscript. Kuo et al. 2022 (*Sci. Reports*) utilized 5-month old (“sexually mature”, adult) Lanyu mini-pigs and performed high quantity, dense wound arrays on their sides, not dorsum as we describe in our manuscript. The wounds described in our manuscript are a single square wound placed on the dorsal midline of 5-day old neonatal piglets. In the histological analysis of the wounds, we observed substantially more regenerated rete ridges approximately 2 months after wounding from neonatal wounds than the authors observed in wounds to 5-month old adults. Additionally, the authors quantified neither the epidermal thickness nor rete ridge density in their wounds. The authors also did not present full cross-sections of the wound bed so it is unclear how prevalent the small regenerative rete ridges are, whereas we present a combination of close-up and zoomed-out images of the wound bed (Supplemental Fig. 8d-g). In the second study, Lin et al. 2019 (*Exp. Dermatol.*) also utilized a multi-wound procedure on mature, 4-month old Lanyu pigs. As we have demonstrated via our studies of epidermal proliferation, after 1-month of age the rete ridge and inter-ridge domains transition to a maturation state, where the rete ridge and inter-ridge both exhibit similar, sporadic replication (Supplemental Fig. 8a-c). Lin et al. observed rete ridge regeneration inconsistently based on the depth of the wound, as shallow 2mm deep wounds seemed to regenerate some rete ridges while deeper 5-25mm wounds did not regenerate rete ridges based on representative histology. Histology of this phenomenon was also not presented over the full cross section of the wound bed nor quantified for rete ridge regeneration, so it is unclear how prevalent the morphologically small rete ridges were even in the shallow wounds. Both of these adult pig wound models resemble results of adult human wounds morphologically⁴¹, lacking the robust dermal pockets that we observed reform from neonatal pig wounds (Supplemental Fig. 7d-g). As we have characterized during porcine skin development, rete ridges have formed and matured by several months of age (Fig. 1h-i). We have also provided more detailed temporal analysis of our neonatal pig wounds and the regenerated rete ridges and dermal pocket, morphologically and quantitatively across the wound bed (Fig. 5b-c; Supplemental Fig. 9a-f), in this resubmission. Therefore, robust rete ridge regeneration and dermal pocket reformation in neonatal dorsal pig wounds is a novel phenomenon.

4) It is not appropriate to perform pseudotime analysis on the entire population of keratinocytes to understand transcriptional dynamics during epidermal differentiation (line 207-208, 217-218 and Fig. 3e, 3h). This is because the cells in the basal buds will generate skin appendages and will not contribute to the differentiated epidermal cells.

We thank the reviewer for this valuable constructive criticism. We no longer perform pseudotime on all the E90 pig keratinocytes and instead have completely replaced the original analysis with a completely new integrative analysis of E90, P3, P10, and 6mo pig basal keratinocytes (Fig. 3b-c). This new analysis more clearly and succinctly articulates that the fetal basal bud and fetal basal cells are distinct from postnatal basal cells involved in rete ridge formation and maturation. This was our intended takeaway from the original pseudotime analysis. We also confirmed the

results from this integrative approach from pigs in human scRNA-seq datasets (Supplemental Fig. 4d-e), better supports our conclusion that rete ridges form through a molecular mechanism distinct from basal budding.

To understand transcriptional dynamics during epidermal differentiation, we have performed a new pseudotime analysis on only the postnatal interfollicular epidermal keratinocytes without basal buds, hair follicle, or sweat gland components (Supplemental Fig. 4e-f; Supplemental Fig. 9c-d; see Methods). This new analysis demonstrated the same pattern as in the original submission without noise generated by miscellaneous fates (Fig. 4c; Supplemental Fig. 9c-d).

Minor:

5) Based qualitative assessment, the authors concluded that RR development was unperturbed in EDA-KO pigs. However, the CRISPR/Cas9 mediated knockout may not entirely eliminate EDA's function, and minor phenotypes might exist. Therefore, a quantitative analysis is needed to strengthen the conclusion.

We have included additional characterization of rete ridge density between WT and EDA-KO pigs at a new neonatal EDA-KO timepoint (P5) with increased biological replicates (n=5 WT, n=5 EDA-KO) in the resubmission to further support our previous conclusion that EDA is not required for rete ridge formation (Fig. 3e). We were unable to obtain additional age-matched 5mo WT and EDA-KO pigs (n=2 for both) for statistical comparison of rete ridge maturation in adult skin, so that conclusion remains qualitative. To confirm the independence of rete ridge formation from basal bud-related signals, we have now generated Krt14Cre Lef1 fl/fl epidermal knockout mice to demonstrate that fingerpad rete ridges do not require epidermal Lef1 (Fig. 3d). As a result, we have provided much more thorough in-vivo evidence to support our previous conclusion that rete ridges form through a mechanism distinct from basal budding and other types of skin appendages. As a result, we have revised the title of the manuscript to more specifically reflect the outcomes of our new experiments: "Rete ridges form through evolutionarily distinct mechanisms from other appendages during mammalian skin development".

6) The authors claimed that RR is a novel class of epithelial appendage (line 227, 459, 508). However, the current data does not exhibit substantial differences in function between RR and non-RR epidermal cells. Therefore, RR appears more akin to a domain or structural unit of the epidermis rather than an independent appendage. The authors might consider changing the language.

We consider the unique architectural features and vascular association of the rete ridge to qualify for consideration as a novel appendage. Previous studies have proposed the definition of an epithelial appendage as follows: "Epithelial appendages are derivatives of the epithelial sheet. They add structural and functional complexities to the otherwise flat epithelia"⁴².

Epithelial appendages also involve both epidermal and mesenchymal components⁴², so the association between the rete ridge and dermal pocket suggests interactive epidermal and dermal compartments characteristic of a novel epithelial appendage. Critically, the rete ridge is a self-sustaining structural feature in adult human and pig skin, associated with both vasculature and epidermal thickening, which bestows an undulating, wave-like patterning to the basal epithelium in contrast to the comparatively flat basal epithelium in fetal skin (Fig. 1a-c; f-l; Fig. 4a; Supplemental Fig. 9a). Though the rete ridge and inter-ridge basal epithelium is transcriptionally indistinct based on our novel, integrative scRNA-seq and parallel Stereo-seq analyses across 3 postnatal timepoints, we have demonstrated dynamic proliferative patterning between these domains during rete ridge formation and early maturation as well as a shared

differentiation function (Fig. 4a-c; Supplemental Fig. 8a-l; Supplemental Fig. 9a-f). We have further demonstrated that the rete ridge's structure and adjacent dermal pocket are highly consistent across species and regions of the body that exhibit rete ridge-like morphology (Fig. 2a; Supplemental Fig. 2a; Supplemental Fig. 16a-d). Species and body regions we now demonstrate rete ridge and dermal pocket morphology are across porcine and human skin development and maturation (Fig. 1a-b, f-g; Supplemental Fig. 9a, f; Supplemental Fig. 16a-b), in cetacean skin (Fig. 2a; Supplemental Fig. 16c), and in murine volar fingerpads and oral epithelium (Fig. 5a-b; Supplemental Fig. 16d; Supplemental Fig. 17a-b). Our new analysis includes immunostaining of rete ridges and the dermal pocket during human trunk skin development and maturation, in the trunk skin of dolphins, in murine volar fingerpads, and in murine oral epithelia (Supplemental Fig. 8a-d).

We have adjusted the language used in the text to describe the rete ridge and inter-ridge regions as domains based on their distinctive morphology as opposed to binarized transcriptional or signaling behavior.

7) Through molecular analysis and genetic manipulation, the authors demonstrated that RR formation is unlikely to be controlled by EDAR and Wnt pathways. However, stating that RR is not formed through basal budding (line 197-198, 212, 225-226, 375, 428-430) is inappropriate, as the authors have not ruled out the possibility that at the cellular level, RR is formed through budding from the epidermis. The authors might consider changing the language.

We have provided further characterization of rete ridge formation's independence from LEF1/WNT-signaling in-vivo using a new epidermal Lef1 knockout mouse (Krt14Cre Lef1 fl/fl). In the fingerpads and footpads of mice there are rete ridge-like structures. We have characterized these fingerpad rete ridges to demonstrate that they have a dermal pocket, like human and porcine trunk skin (Supplemental Fig. 8c; Supplemental Fig. 9e; Supplemental Fig. 16a-b, d; Supplemental Fig. 17a). Furthermore, the postnatal timing of their development is after the murine volar sweat gland formation has completed (Fig. 5a; Supplemental Fig. 5a-b).

Additionally, we have performed new bioinformatics analyses to integratively compare fetal porcine skin to postnatal skin, observing both a lack of basal bud morphology and transcriptional signature during rete ridge formation in porcine trunk skin and murine fingerpads (Fig. 3b-c; Supplemental Fig. 5b). We have validated these bioinformatics conclusions in-vivo using immunostaining, demonstrating that the basal bud markers LEF1 and SOX9 are not expressed in developing rete ridges in porcine trunk and murine fingerpad skin (Fig. 3a; Supplemental Fig. 5a-b; Fig. 5a).

Minor

The anti-correlation between hair density and epidermal thickness was not correctly stated (line 183).

Thank you for catching this typo. We have corrected the description of this relationship in the revised text: "Direct comparison of hair density and epidermal thickness revealed that there is an inverse relationship between epidermal thickness and hair density (Fig. 2d; Supplemental Fig. 2d)."

Multiple panels of Fig. 4 were not correctly cited (line 242-250).

Fig. 4 has been completely redesigned based on reviewer comments and new experiments. It seems the word “also” was mistakenly added in line 274.

We thank the reviewer for their attention to detail and have corrected this typo and checked the revised manuscript for other typos like this.

Fig. 7h is mislabeled and not cited anywhere.

In the course of the revisions, we have substantially modified almost all of the main and supplemental figures. We have double checked the new figure panel labeling and references in the text.

Referee #3 (Remarks to the Author):

The importance and function of Rete ridges (RR) in human skin has puzzled the skin and stem cell fields since decades. It is assumed that the RR are important for maintaining a high regenerative potential of the skin, but more general functional roles of RR remain elusive. To date, it is entirely unknown when (during skin development) RR develop, if they are distinct from fingerprint ridges, and which cellular and molecular mechanisms drive their establishment. This study, Thompson et al, investigate these intriguing questions and provide very exciting answers, such as RR development is molecularly entirely different from other skin appendages (hair follicles, sweat glands, and fingerprint ridges), RR develop perinatal (just after birth), and the presence of RR across the mammalian kingdom suggest that RR have a protective function in skin with reduced hair density (mouse vs grizzly vs pig and human skin). The latter notion also may help to answer the ever-recurring questions (not the why but the how): why did humans become (largely) hair less and how did they compensate for it?

The authors use an impressive array of mammalian species, analysis methods, and computations for their study. In addition to the novelty in skin/tissue biology, the study also provides a comprehensive single-cell transcriptomics resource of pig skin, which is highly valuable. There is no doubt that this study is novel and of high interest to a wide general readership (such as Nature), and I am certainly in support of this elegant and innovative study. However, the authors will see by the comments below, there is still much work needed to cement their claims and to dive more into the cellular and molecular mechanisms underlying RR formation.

Major comments and questions:

(1) Fig 1c has very low sample numbers for some timepoints and ideally all should have more than $n=1$. Yet, as it may be prohibitively difficult to increase the fetal sample number, and they show consistent results (notably, GW16-17 and GW18-20 have $n=7$ and $n=5$, respectively), those may be fine. However, it is important to include more samples in the 21-39yo group as it forms the first timepoint of the data with RR existence. A higher sample number will also clarify if the decrease of RR numbers between young adult to 60/80yo is significant, which would be expected based on literature showing that epidermis flattens with age (eg:

doi:10.1016/j.mad.2016.03.006; doi:10.3390/biom10121607; doi:10.1039/C5IB00238A).

Interestingly however, Fig1c seems to show no clear reduction in epidermal thickness. Could the authors comment on it, do we see this discrepancy because of the chosen body site(s) or the number of samples measured?

We attempted to collect more samples from <40yo humans, but due to the opportunistic nature of our human sample collection as a result of Mohs surgery discard tissue from skin cancer removal, we have not been able to collect anymore samples in that age group. We have collected skin from multiple body regions in the trunk and face regions, however the epidermal thickness measurements are generally consistent between body regions and show only minor differences with age (Rebuttal Fig. 4a-c). Critically, compared to previous literature as this reviewer aptly mentions, we did not observe a dramatic reduction in rete ridge density that was apparent to the naked eye or statistics. We attempted to transparently present this persistence of rete ridges throughout the epidermis in aged skin of the trunk and face using the zoom-out views of histology in Fig. 1b coupled with the zoom-ins. The persistence of rete ridges in aged face and trunk skin was consistent across biological replicates and body regions (Rebuttal Fig. 4a-c).

(2) Fig2: the authors claim that RR are associated with general epidermal thickness and reduced hair density. Does this also apply to human skin across body sites? For example, scalp skin has a higher hair density compared to forearm or abdominal skin; is the IR and RR thickness altered across those sites? If not, it would be valuable to provide a statement in the discussion. Related, Fig2b,d: It would be interesting to see the data for the Yucatan Hairless and Mangalitsa pig in all analyses.

We do not have sufficient sample sizes of human skin where both epidermal thickness and hair density we quantifiable to draw sound conclusions between body sites due to variance in the size and shape of surgical discard tissue samples obtained following Mohs surgery. Our analysis and quantification of human skin is almost exclusively focused on trunk skin and does not examine scalp regions. Some of the variance in hair density may also be a product of aging.

We thank the reviewer for their interest in the Mangalitsa and Yucatan hairless pigs, and have incorporated more of their data into the main figure instead of the supplemental figures (Fig. 2a; Supplemental Fig. 2a). Additionally, the Mangalitsa and Yucatan hairless are now directly compared to the other species instead of just compared across pig breeds as in the original submission. As these pig species have significantly thicker epidermis than rete ridge-less species and no difference in thickness compared to each other (Fig. 2c), their direct inclusion has strengthened our original statements related to the conserved function of rete ridges to support a thicker epidermis (Fig. 2a; Fig. 2c). Resultingly the direct comparison of the Mangalitsa and Yucatan with the rest of the histological “zoo” has greatly enhanced the manuscript. We thank the reviewer for this helpful suggestion.

(3) Line 148/149: please add a muscle staining. How do the authors know it's the PCM and not any other deeper back muscle like in mouse (see doi:10.1016/j.devcel.2023.07.015)? Related minor note: the statement of PCM not being present in human skin does not apply to all body sites in human skin but would apply to eg. back skin.

We have removed discussion of the panniculus carnosus from this manuscript. We thank the reviewer for their comments and interest.

(4) L202/3: Please specify the genes that were used for the characterization and show key genes for the classification as eg. dotplots. K14 in mouse marks exclusively basal layer cells, whereas in human skin it is expressed in basal and several suprabasal layer(s). In human skin, K15 seems a better marker for basal cells (doi: 10.15252/embj.2021110488), and considering the similarity of pig and human skin, you may consider checking K15 expression as an alternative. After this check, if you still think K14 is a valid marker for basal cells, please provide a K10/K14 stainings in pig skin.

We have added a new, detailed supplemental figure that presents marker genes used for cell type classification in DotPlots for every scRNA-seq dataset analyzed in this study (Supplemental Fig. 4a-b; 3-j). We have also provided a more detailed description of these markers and subsetting approaches in the revised Methods section of the manuscript. We have also added extensive immunostaining validation of basal markers, like KRT15 and KRT14, in human, pig, murine volar fingerpads, marmosets, and mice (Fig. 4a; Supplemental Fig. 9a, c, f). We also now present a comparison of epidermal stratification between pigs, mouse trunk and volar fingerpads, marmosets, and naked mole rats (Supplemental Fig. 9a, c, f; Supplemental Fig. 17b-c). We have also provided increased validation for the spatial consistency of dermal cell lineages in the dermal pocket during rete ridge formation in pigs and murine volar fingerpads as well as compared between human, cetacean, murine, and marmoset skin (Supplemental Fig. 16a-d; Supplemental Fig. 17a).

(5) Fig 3i: At least in the mouse, only certain HF types depend on EDAR signalling (eg: guard or tail HFs). The shown pictures of the EDAR-KO pigs do have (fewer?) hair follicles, which means that there is still some basal budding ongoing. Although I am convinced (eg from scRNA-seq analysis and stainings) that RR form via a different mechanism than HFs or fingerprint ridges, showing the EDAR-KO data for the purpose of making the case that RR do not form via basal budding is not convincing; it shows that EDAR signalling is not needed which is different to not forming via basal budding.

We thank the reviewer for their interest in the EDA-KO pig data. We have performed more detailed characterization of the EDA-KO pig versus WT pig skin during neonatal rete ridge development (Fig. 3e). Since LEF1/WNT-signaling are also known to be important for hair follicle development and LEF1 can directly regulate EDA^{12,13,16}, we have also performed new experiments using K14Cre Lef1 fl/fl mice to demonstrate that volar fingerpad rete ridge development does not require LEF1/WNT (Fig. 3d; Supplemental Fig. 6c). We have also greatly expanded our characterization of rete ridge development at the cellular and molecular levels in pig skin, which clearly shows a lack of basal bud-like morphology, molecular signature, and proliferative patterning compared to other epithelial appendages (Fig. 3a-c; Supplemental Fig. 6b-c; Supplemental Fig. 8a-d; Supplemental Fig. 9a; Supplemental Fig. 17a-b)^{2,5}. Critically, we have also identified that broad epidermal BMP-signaling activates during rete ridge formation in porcine trunk skin and murine volar fingerpad skin (Fig. 4c-e; Supplemental Fig. 10c-d; Supplemental Fig. 11a-b; Fig. 5b). Using two different lines of transgenic mice to selectively inhibit epidermal BMP-signaling in murine volar fingerpads, we have also demonstrated that epidermal BMP-signaling is required for rete ridge formation. These observations and functional validation have been incorporated into a summary figure of our proposed model for rete ridge formation (Fig. 5e).

(6) L233-235: The authors state: "While some fetal and neonatal mammals, such as mice, have the potential to regenerate hair follicles during wounding, it is unknown whether neonatal pigs

share this intrinsic regenerative capacity. Since we determined that the neonatal developmental window is when rete ridges form and that this window appears unresponsive of hair follicle or sweat gland formation, we hypothesized that neonatal pig wounds would be capable of regenerating rete ridges but not hair follicles and sweat glands since this is the same timeframe as when rete ridges are forming.” I.e. The authors say, and it's well known, that wounding allows de novo hair follicle formation, but then continue to say that they would not expect HFs to form in wounds of pigs but perhaps RR – which as stated does not make sense. I understand that the authors wanted to emphasize that in healthy settings, RR form neonatally and HFs/SGs form during embryogenesis. But assuming that only RRs would form in postnatal healed wounds would only make sense to me if the authors first refer to previous work or own data showing that de novo HF/SG formation has not been observed in neonatal wounds of pigs, while the formation of RRs has not yet been looked at.

We thank the reviewer for their critical feedback of our introduction of the porcine wound healing experiment. We have completely re-written this section on porcine neonatal wound healing to focus on the experiment itself and less on a discussion of the literature comparisons between neonatal porcine and neonatal murine wound healing. Thus, this section is now more concise and grounded in the direct motivation and results of the experiment. The section of the text now reads: “We next tested the developmental potential of rete ridge formation from neonatal pig skin via wound healing. Single, large square dorsal wounds on neonatal piglets heal to regenerate rete ridges and reform vascularized dermal pockets, suggesting that the neonatal signaling environment has an intrinsic potential to form and re-form the epidermal rete ridge and dermal pocket niches (Supplemental Fig. 7a-f).”

(7!) The authors often state that RR form via a novel mechanism, distinct from HF/SG/fingerprint ridges. However, did the authors eventually identify unique RR marker genes, distinct of IR gene expression? Or is the proposed new mechanism rather that RR utilize the ‘normal IFE differentiation’ process? I.e. could the authors elaborate more on the development of RRs in healthy skin (Fig3). Are there any differences in cell proliferation and differentiation between the RR and IR areas? And similarly, how do these aspects look like during the initial stages of RR development (e.g., p0 – p5 pig epidermis), is there higher proliferation in the areas of future RRs? Connected to the development of RRs, can dense vasculature already be observed in the early (before P5) dermal pockets or even before the initial formation of RRs (also see following comments)? Or could there also be other factors, such as nerves, involved in the process?

We thank the reviewer for their engagement and criticism of our original proposed mechanisms. We have performed extensive new bioinformatics and functional, in-vivo experimentation to address this reviewer's points.

1). We have performed new experiments to investigate proliferative patterning during rete ridge formation in pigs (Supplemental Fig. 8a-d), which revealed that proliferation does not seem to pre-pattern where rete ridges will form, as it does for hair follicles, sweat glands, and fingerprint ridges^{1,2,5}. Instead, proliferation is dynamically activated in both domains, with increased basal and suprabasal proliferation in the rete ridge through most of rete ridge development (Supplemental Fig. 8b-d).

2). We also performed functional experiments to examine differentiation between the rete ridge and inter-ridge domains in developing rete ridges using in-vivo BrdU-labeling (Supplemental Fig. 8e-i). We observed that both developing rete ridge and inter-ridge domains differentiate

vertically and function as epidermal proliferation units (Supplemental Fig. 8f-i). As a result, proliferative patterning after the initiation of rete ridge formation seems to support epidermal thickening, which is then maintained by shared differentiation programs between rete ridge and inter-ridge regions, since the rete ridge is thicker and has more spinous cell states to progress through to reach the skin surface (Supplemental Fig. 8f-l; Supplemental Fig. 9a).

3). It is possible that nerves or other cell lineages in the epidermis or dermis contribute to rete ridge formation. However, the focus of our current manuscript is on the epidermal and dermal fibroblasts, pericytes, and vasculature, since our previous and new data strongly supports interactions between these epidermal and dermal cell lineages morphologically and mechanistically in rete ridge formation and maturation. From this focus, we have identified novel epidermal signals to recruit dermal-vascular lineages, such as PDGFC (Supplemental Fig. 9e; Supplemental Fig. 10c-d; Supplemental Fig. 14a-d), dermal-epidermal interactions, such as FGF (Supplemental Fig. 10c-d; Supplemental Fig. 12a-b), a novel pro-vascular fibroblast subpopulation that arises in the dermal pocket (Supplemental Fig. 15a-d), and an epidermal requirement for BMP-signaling for rete ridge formation (Fig. 4c-e; Supplemental Fig. 10c; Supplemental Fig. 11a-b; Fig. 5b-e; Supplemental Fig. 17h-i). At this time, investigation of other cell lineages such as nerves is out of the scope of the current study. We have also changed the title of the paper to more accurately encompass the scope of the revised manuscript: “ Rete ridges form through evolutionarily distinct mechanisms from other appendages during mammalian skin development”. We thank the reviewer for their interest in the interactions between the rete ridge and dermal pocket.

4). Our new epidermal wholemount immunostaining, integrative scRNA-seq analyses, and Stereo-seq spatial transcriptomics analyses all support a developmental model wherein the epidermis is broadly compatible with rete ridge formation, instead of rete ridge and inter-ridge domains being pre-patterned by distinct transcriptional markers (Fig. 4a-e; Supplemental Fig. 9a-e). We observed a similar lack of spatial compartmentalization for these core basal state markers in the rete ridge and inter-ridge domains of human trunk skin and murine fingerpads (Supplemental Fig. 9f; Supplemental Fig. 17b).

(8) Connected to above: The authors have compiled an impressive scRNA-seq dataset from existing sources, as well as including their own samples. Can the authors separate the epidermal RR and IR areas based on the transcriptomic data? If possible to identify these areas, do they show differences in their proliferative and differentiation capacities? Which epidermal clusters are most involved in angiogenesis, are the IR keratinocytes secreting more of these factors than RR keratinocytes? Light-seq (<https://doi.org/10.1038/s41592-022-01604-1>) or new generation high-resolution spatial sequencing could be viable strategies to separate IR and RR and compare them.

To summarize and elaborate from our responses to (7):

1). From our new integrative scRNA-seq and Stereo-seq approaches, we have now demonstrated that the basal layer of postnatal epidermis is largely indistinguishable between the rete ridge and inter-ridge domains (Fig. 4b-c; Supplemental Fig. 9b-f). We have also validated this lack of transcriptional distinction using immunostaining and observed a lack of periodic patterning for these key basal markers, like KRT15, and signaling ligands, like BMPs and PDGFC (Fig. 4a-e; Fig. 5b; Supplemental Fig. 9a-f; Supplemental Fig. 17b). One of the only differences between the rete ridge and inter-ridge domains is the broader activation of BMP-

signaling into the suprabasal compartment of the rete ridge compared to the inter-ridge, as BMP-signaling is otherwise pan-basal in porcine trunk and volar fingerpad skin (Fig. 4e; Fig. 5b, e).

2). Proliferative patterning is dynamic during rete ridge formation. At P3, the inter-ridge basal cells are more proliferative than rete ridge basal cells, but immediately afterward the rete ridge basal and suprabasal cells become more proliferative than the inter-ridge (Supplemental Fig. 8b-d). Using in-vivo BrdU-labeling to track differentiation, we observed that rete ridges and inter-ridges were labeled comparably and both compartments exited the basal state to differentiate as an epidermal proliferation unit in similar proportions (Supplemental Fig. 8e-i).

3). Expression of pro-dermal and pro-vascular signaling pathways appeared broadly pan-basal in the porcine epidermis during rete ridge formation and maturation, at both the levels of spatial transcriptomics and when validated using immunostaining (Supplemental Fig. 9e-f; Supplemental Fig. 14a-d). We also validated that ITGA6, KRT14, KRT15, and PDGFC are pan-basal in the volar fingerpads during rete ridge formation and maturation (Supplemental Fig. 17b).

4). We thank the reviewer for their suggestion of Light-seq and spatial transcriptomics to more clearly define regional signaling gradients during rete ridge formation. We employed Stereo-seq to catalog spatial signaling during rete ridge formation and maturation in P3, P10, and 6mo pig skin (Supplemental Fig. 4d). This suggestion was pivotal in motivating us to pursue spatial transcriptomics, which completely transformed our ability to define the spatial compartmentalization of the key cell types and predicted signaling interactions during rete ridge formation and maturation.

(9) L302: “We hypothesized that epidermal-dermal interactions support dermal vascular lineage recruitment into the inter-ridge space, and that the proximity of these dermal cell lineages, including fibroblasts, pericytes, and blood vessels to the basal epidermis provides a parallel mechanism that jointly supports increased epidermal thickness.” This statement implies that the authors think RR formation precedes blood vessel dermal pocket establishment. Could it be the other way around? Vessels and/or nerves could be there first, and thus provide a distinct niche environment (signalling gradients/pattern in dermis) that induces RR formation (eg. Fig6 shows at E90 vessels touching the epidermis). It is noted from results in Fig6 that presence of RR does increase signals supporting vascularization, but this does not answer whether RR formation itself precedes or follows primary vascularization.

The rete ridge and dermal pocket are consistently associated with one another during rete ridge formation and especially in mature epidermis across mammalian species which have rete ridges (Fig. 2a; Supplemental Fig. 16a-e). Our new epidermal whole mount and cross-sectional immunostaining during rete ridge development in pigs also suggest that epidermal thickening and proliferative patterning are indicative of rete ridge formation prior to the establishment of the complete vascularized dermal pocket (Fig. 4a; Supplemental Fig. 8a-d). Critically, our new integrative scRNA-seq analyses coupled with CellChat identified a novel pro-vascular papillary fibroblast population which arises after rete ridge formation initiates, present in P10 and 6mo dermal pockets (Supplemental Fig. 15a-d; Supplemental Fig. 16a). These results suggest that the dermal pocket niche expands and matures following initial vascular recruitment.

We have demonstrated using both single-cell and spatial transcriptomics that dermal FGF-signaling activates during rete ridge formation. Compared to the lack of epidermal patterning for

rete ridge-associated signals like BMP, dermal FGF-signaling appeared concentrated around the dermal vasculature. Our new scRNA-seq and Stereo-seq analyses implicated epidermal BMP-signaling with rete ridge formation, with epidermal-derived BMP-signaling predicted to be received by the epidermis and also dermal cell lineages, including vasculature (Fig. 4b-e; Supplemental Fig. 10c-d; Supplemental Fig. 11a-b). Our functional studies of epidermal BMP-signaling in murine fingerpad rete ridge formation revealed that dermal pocket formation does not occur without corresponding rete ridges, suggesting that the dermal pocket is dependent upon the epithelial component of rete ridges (Fig. 5c-e; Supplemental Fig. 17h-i) As such, we conclude that epidermal and dermal components interact during rete ridge formation, but that the epidermal signaling environment is the instigator and required for appendage formation (Fig. 5e).

(10) In Fig4, the authors describe the formation of RRs in healed wounds. At p5+28dpw one can see that the RRs are already formed, but how does an earlier wound timepoint look like? Do RR start forming as soon as parts of the wounds are re-epithelialized or do they start forming only when the wounds are completely closed with a flat epithelium? Are there differences in RR maturation (size, density) between wound edges and wound fronts? As the vasculature seems important for the RR development, re-vascularization in association with wound healing could provide a great model system where you may be able to show if the RRs start forming only after vessels are established and have reached the epidermis.

Wound sections at the outermost edge of the wound site somewhat resemble psoriatic skin, with larger rete ridges and thicker epithelium than sections taken from deeper into the wound bed. Rete ridges appear to regenerate inwardly coinciding with re-epithelialization, with more “developmental-like” morphology most discernable at the edges of the wound (Supplemental Fig. 8).

(11) Fig6: Please provide an RNA staining (eg. RNAscope) confirming PDGFC expression in RR in pig, and its absence in mouse skin. PDGFC seems one of the few unique RR markers.

We have performed an additional P3 pig scRNA-seq to better understand early rete ridge formation and then updated our scRNA-seq CellChat analyses. PDGFC is still identified as a consistent epidermal-dermal signal during rete ridge formation and maturation, which we have also validated in-vivo using immunostaining in porcine trunk skin and murine volar fingerpad skin, which also has rete ridges (Supplemental Fig. 9e; Supplemental Fig. 14a, d; Supplemental Fig. 17b). We also validated our previous claim from murine scRNA-seq re-analysis⁴³ and confirmed that epidermal PDGFC is not expressed in murine trunk epidermis in-vivo (Supplemental Fig. 17c).

We have now performed Stereo-seq spatial transcriptomics on P3, P10, and 6mo pig skin and used Spatial CellChat to predict spatial signaling patterns between the epidermis and dermis during rete ridge formation and maturation. Spatial CellChat predicted PDGF- and PDGFC-signaling broadly from the epidermis to the dermal fibroblasts (Supplemental Fig. 14a; Supplemental Fig. 14d). We also validated the pan-basal expression pattern of PDGFC during rete ridge formation and maturation using immunostaining in pig skin and murine volar fingerpads to confirm that there is not periodic patterning of PDGFC between rete ridge and inter-ridge regions (Supplemental Fig. 9e; Supplemental Fig. 17b).

(12) Does the formation of RRs change the surrounding fibroblasts and the ECM components? As the authors find pro-fibroblast signals coming from the keratinocytes (Fig6), can they also

identify if these signals are mainly concentrated in the IR areas or beneath the RRs? Are the IR region fibroblasts distinct from other dermal fibroblasts?

Using Spatial CellChat across our P3, P10, and 6mo Stereo-seq datasets, we investigated global secreted signaling activity between the epidermis and the dermis (Supplemental Fig. 10a-b). We observed highest secreted signaling activity in the epidermis but also high signaling activity throughout the papillary dermis, which followed the general contour of the epidermal architecture rather than clustering into distinct dermal domains aligning with either the rete ridge or inter-ridge (Supplemental Fig. 10b). The papillary dermis has a distinctive ECM from the lower reticular dermis, and the ECM around the vasculature within the dermal pocket is also distinguishable from non-vascularized papillary dermis (Supplemental Fig. 1c; Supplemental Fig. 2a).

We identified from an integrative analysis of the dermal cell lineages throughout pig skin development that a pro-angiogenic, PECAM1+ papillary fibroblast subpopulation arises in the dermal pocket during rete ridge formation (Supplemental Fig. 15a-d). This scRNA-seq subpopulation expresses PDGFRA, PECAM1, and ITGA6, but not ACTA2, CDH5, or RGS5, indicating it is likely a novel fibroblast subpopulation with pro-vascular signaling activities (Supplemental Fig. 15c-d). We have spatially validated this proposed pro-vascular fibroblast subpopulation to the maturing dermal pockets using immunostaining based on PDGFRA+/ITGA6+ cells, which is also supported by Stereo-seq Spatial CellChat identifying PECAM1-signaling only within the vascularized dermal pockets (Fig. 4b; Supplemental Fig. 15a-d; Supplemental Fig. 16a).

(13) The 'reverse engineering' hypotheses establishment and experiments are very interesting, yet again beg for some more exploitation. (a) L390: "Histological assessment of TPA- and RA390 treated wounds revealed induced regions of rete ridge-like structures, complete with recruited vasculature into the inter-ridge space at both 1dpt and 4dpt (Fig. 7e-f; Extended Data 5f, h). The "recruited vasculature" would have to establish within 1 day, which seems fast. Again (related to point 9 above), isn't it more likely that RR appear between the already established inter-vascular dermal space? For example, the vascular niche pockets secreting proliferation/stratification inhibitory signals? (b) When discussing the role of RA treatment as inducing de-differentiation of keratinocytes, do the authors have any independent evidence for the occurrence of this de-differentiation? RA is known to induce hyperproliferation of keratinocytes (<https://doi.org/10.1038/jidsymp.1998.14>; <https://doi.org/10.1016/j.ydbio.2008.08.034>). As both TPA and RA treatment result in RR formation, induced hyperproliferation of keratinocytes would be sufficient to explain the results without necessity of de-differentiation.

TPA- and RA-treatments were performed daily for seven days, so 1 day post treatment was one day after the cessation of one week of topical treatment. We also observed induced rete ridges maintained by 4 days post treatment. However, we have removed the rete ridge induction experiments from this manuscript to focus instead on direct mechanistic validation of rete ridge formation using new pig and mouse models. We thank the reviewer for their critical engagement and constructive feedback on these experiments, and we will incorporate that feedback into a separate manuscript focusing on rete ridge induction in murine skin.

Minor and textual comments:

(a) Line 121: “apical ridges formed earlier than GW12 in human development and increased in length by adulthood”. Please mark the apical ridges the Fig 1a,b.

Apical ridges formed early in fetal development are shallow compared to postnatal skin and typically span the distance between hair follicles (Fig. 1a, f; Supplemental Fig. 1a-b). We have marked examples of apical ridges in Rebuttal Fig. 1a.

(b) Related to 121 (Methods): How is the apical ridge length measured? Do you mean “length” or “depth”? To me “length” is confusing. Please show either a cartoon or indicate on sample what you measure.

Apical ridge length is measured along the linear length of the apical surface, measuring from “trough to trough” between adjacent apical ridges. Though we have now removed apical ridge data and discussion from this manuscript since our developmental and mechanistic studies are focused on rete ridges, we have presented a graphical definition of how apical ridges were measured (Rebuttal Fig. 1a).

(c) Fig 1c. Please indicate which body sites were used for the measurements. Is it a combination of face and trunk skin or were also some other body sites included? Were the samples coming from donors of similar ethnic background? This can have an influence on epidermal thickness depending on body location (see doi:10.1111/jdv.18123).

We present histology from only face and trunk (defined as skin of the torso, back, or limbs) in Fig. 1. Skin from around the ears, eye, scalp, nose, volar palms or feet, or genital regions were not included in these definitions or this study. We present skin from face and trunk in Fig. 1b to reinforce the consistent persistence of rete ridges in the aged human samples that we observed in this study (Fig. 1b). Fetal skin presented in the manuscript was all collected from the trunk with an emphasis on the dorsum. Quantifications of human skin samples used in Fig. 1 were from trunk and face skin, respectively. We did not observe a difference in trunk and face epidermal thickness, but trunk skin had a slightly higher rete ridge density (Rebuttal Fig. 4a-c). The inclusion or exclusion of face skin samples did not change the pattern of the results in Fig. 1c-d for adult timepoints, namely that epidermal thickness and rete ridge did not significantly change during human skin aging (Fig. 1c-d; Rebuttal Fig. 4b-c). Quantifications of adult human samples used in the comparative analyses for Fig. 2 were performed on trunk skin. All human samples were deidentified prior to receipt and analysis, and ethnic background was not collected or reported, only age, sex, and body region. Due to a general lack of melanin visible in histology sections, most of the donors were likely white.

(d) L134: Fig 1e. Can you indicate the RRs? I cannot spot any.

We have now provided additional annotation of Figure 1's histology images to improve readability. Since rete ridge formation does not involve a basal bud step (Fig. 3a-e; Supplemental Fig. 8a-d; Fig. 4a), developing rete ridges are distinguishable by regional thickening and corresponding proximity to vertically-oriented underlying vasculature.

(e) Fig1i. header text in figure "RR vs apicR formation are independent processes". Measured is apical ridge length; to say something like RR vs apicR formation are independent processes, shouldn't the apicR measurement be somehow correlated to RR measurements? This graph alone is not an argument, is it? Or should it be apical ridge density?

We have removed comparison of rete ridges and apical ridges from this manuscript and will be presenting that data as part of a separate manuscript in preparation.

(f) L152/153: From your data at this point it's still an assumption that RR also form perinatally in human skin, right? As it is stated, you would have to show it in human samples, which of course is not possible. Please rephrase the sentence to conveying the message "as pig skin development is a great proxy for human skin development, it can be assumed that RR in human also form perinatal".

We thank the reviewer for this critical suggestion. We have directly incorporated this feedback into the text of the manuscript: "Previous studies have already demonstrated that rete ridges form in human skin within several months of birth³⁹. Here, we further determine that rete ridges begin forming perinatally in both humans and pigs, suggesting that porcine skin development is a close proxy for human skin development."

(g) Line 153: "Additionally, the developmental overlap between the onset of increased epidermal thickness and rete ridge formation suggest that these processes may be linked." Do you refer here also to the interridge thickening? If not (i.e. you only refer to RR in general) this is a circular argument. Please clarify what you exactly mean. Same for L167: please specify in the text to which parts of the epidermis (RR and/or IR) you refer to? Eg. simply add "...the epidermis (IR and RR) was thicker... "

We have now added greater clarity in both the presentation of quantification of RR and IR thickness during development (Fig. 1d, i) and when comparing mammalian species (Fig. 2c; Supplemental Fig. 2d). Rete ridge formation increases maximal epidermal thickness through the rete ridge, while the inter-ridge thickness does not change dramatically (Fig. 2i). We have also presented this data as a ratio of RR to IR thickness (Fig. 1d, i) to aid in visualization how the increase in maximal epidermal thickness is due to the formation and elongation of the rete ridge rather than increased thickening of the inter-ridge domain.

The updated description when comparing epidermal thickness between mammalian species is as follows:

"Directly comparing the thickness of the interfollicular epidermis across our histological zoo, species with rete ridges generally have a thicker epidermis than those without them, while the inter-ridge thickness (aside from the dolphin) is not as dramatically increased (Fig. 2c; Supplemental Fig. 2d). Comparison of the ratio of rete ridge to inter-ridge thickness amongst rete ridge-bearing species revealed a consistent ratio, wherein the rete ridge approximately doubles the thickness of the inter-ridge (Fig. 2c). Interestingly, as the inter-ridge thickness defines the ceiling of the dermal pocket, this ratio suggests an intimate link between the size of

the dermal pocket and the overall thickness of the epidermis, as exemplified by cetacean skin (Fig. 2a, c; Supplemental Fig. 2a, d)”.

Related to L153: Lines 167-169: The notion that epidermis is thicker where RRs are is not a surprise; this has been obvious since decades, thus this relationship is expected. Again, what is surprising is that skin with RR and lower hair density also has a thicker IR epidermis; so, every single part of the epidermis is thicker, but this does not come across from the text itself.

The updated description when comparing rete ridge and inter-ridge epidermal thickness between mammalian species is as follows:

“Directly comparing the thickness of the interfollicular epidermis across our histological zoo, species with rete ridges generally have a thicker epidermis than those without them, while the inter-ridge thickness (aside from the dolphin) is not as dramatically increased (Fig. 2c; Supplemental Fig. 2d). Comparison of the ratio of rete ridge to inter-ridge thickness amongst rete ridge-bearing species revealed a consistent ratio, wherein the rete ridge approximately doubles the thickness of the inter-ridge (Fig. 2c). Interestingly, as the inter-ridge thickness defines the ceiling of the dermal pocket, this ratio suggests an intimate link between the size of the dermal pocket and the overall thickness of the epidermis, as exemplified by cetacean skin (Fig. 2a, c; Supplemental Fig. 2a, d)”.

As far as the correlation between epidermal thickness and hair density, there is a correlation between increasing epidermal thickness and reduced hair density, both in the rete ridge and inter-ridge regions. However, the inter-ridge is only significantly thicker than rete ridge-less species in species with much thicker rete ridges, like pigs and humans compared to mice. Using the K14Cre Lef1 fl/fl mouse, we also demonstrated that genetic reduction in hair density alone does not cause epidermal thickness to increase, suggesting that there may not be a direct, causal relationship between reduced hair density and epidermal thickening (Supplemental Fig. 2d).

(h) Fig2. Are all skin samples across animals species taken and measured from the approximate same body location?

Yes, all skin samples in Fig. 2 are comparisons of dorsal trunk skin, except for human trunk skin, which also includes skin from the back, limb, chest, and abdomen trunk regions. Human face skin samples are not included in the Fig. 2 comparisons between species.

We have updated the methods section and figure legends to more clearly state the body regions used, where applicable.

(i) L156 and Fig 2b. Please add statistical testing, most importantly for second plot (epidermal thickness; IR), showing if the epidermal thickness differences with and without RR are significant.

We have now added statistical testing for all “histological zoo” quantifications. We have also re-written the text to add greater clarity to these statistical relationships:

“Directly comparing the thickness of the interfollicular epidermis across our histological zoo, species with rete ridges generally have a thicker epidermis than those without them, while the inter-ridge thickness (aside from the dolphin) is not as dramatically increased (Fig. 2c; Supplemental Fig. 2d). Comparison of the ratio of rete ridge to inter-ridge thickness amongst

rete ridge-bearing species revealed a consistent ratio, wherein the rete ridge approximately doubles the thickness of the inter-ridge (Fig. 2c). Interestingly, as the inter-ridge thickness defines the ceiling of the dermal pocket, this ratio suggests an intimate link between the size of the dermal pocket and the overall thickness of the epidermis, as exemplified by cetacean skin (Fig. 2a, c; Supplemental Fig. 2a, d)”.

(j) L183: should it be: hair density “decreases” instead of “increases”?

We have re-written the discussion of hair density and epidermal thickness to improve clarity:

“Additionally, the hair density across species is also variable, with rete ridge-less species generally having a higher hair density and thinner skin (Fig. 2c-d). Direct comparison of hair density and epidermal thickness revealed that there is an inverse relationship between epidermal thickness and hair density (Fig. 2d; Supplemental Fig. 2d).”

(k) L188: heading. Given your results and the recent study on fingerprint ridges and their development partly resembling hair follicle formation including high Lef1 expression (<https://doi.org/10.1016/j.cell.2023.01.015>), one could expand the heading to: “Rete ridges are a distinct class of epithelial appendage from hair follicles, sweat glands, and fingerprint ridges”.

We have renamed the section of the results to: “Rete ridges are a distinct type of epithelial appendage in skin”. We have also changed the title of the manuscript from “Evolutionary development, regeneration, and induction of rete ridges in mammalian skin” to “**Rete ridges form through evolutionarily distinct mechanisms from other appendages during mammalian skin development**”. This title change better reflects the revised narrative of the manuscript given the new experiments we have performed since the initial submission.

(l) L202: it would help the non-skin readership to state once more here when hair-bud formation stops; I assume it’s the same in pig skin (?) but in both mouse and human skin HF bud formation stops around birth or before. This information helps the reader to understand why the E90 and P10 is a great comparison.

We appreciate the suggestion from this reviewer and have done the following to improve clarity to a broad scientific readership of the transition embryonic to postnatal states and implications on basal budding in skin:

- 1) With the addition of a new P3 scRNA-seq timepoint and P3, P10, and 6-month old Stereo-seq timepoints from pig skin, we have expanded upon our initial comparison of basal budding vs rete ridge forming skin to include an earlier timepoint in rete ridge formation at P3. In addition, we have also defined the transition from basal bud formation to rete ridge formation in murine fingerpads. In both animals, the epidermis ceases to express LEF1 and SOX9 when sweat gland formation is complete and rete ridge formation begins (Fig. 3a-d; Fig. 5a; Supplemental Fig. 5a-f; Supplemental Fig. 6a-b).
- 2) We have also more clearly addressed basal bud versus rete ridge formation when introducing our new bioinformatics analyses focused on fetal vs postnatal basal epidermis in pigs:

“Based on our observation that rete ridge formation and epidermal thickening are independent of a reduction in hair density, we hypothesized that rete ridges form through a different molecular mechanism than basal budding. LEF1 is highly expressed in the basal buds of E90 (GW12-13) pig skin long before rete ridge formation, but LEF1 is not expressed in the

epidermis postnatally when rete ridges are forming or maturing (Fig. 3a). Additionally, we did not observe any points in rete ridge formation morphologically resembling a basal bud (Fig. 1g; Fig. 3a).”

- 3) Furthermore, we performed a new integration of E90, P3, P10, and 6mo basal and dividing keratinocytes from our porcine scRNA-seq data to demonstrate that fetal basal cells and basal buds cluster separately from postnatal basal cells (Fig. 3b-c).
- 4) We have added more discussion of the distinct developmental timing between basal buds and rete ridges to the Discussion section:

“The fetal origin of hair follicles and sweat glands has been clearly defined by previous studies in many species, including humans ^{1,30,40,75}. Additionally, fingerprint ridges and sweat glands, which in their early stages bear some superficial similarity to skin rete ridges, also form during fetal development in the hairless volar skin of humans and mice ^{2,4,25,26,76}. However, as we have demonstrated in multiple species, rete ridges form postnatally as the final type of epidermal appendage after basal buds stop forming.”

(m) L216/217: human foreskin (please also describe, as for volar skin, that it's devoid of HFs, etc..)

We thank the reviewer for pointing out the difference between the foreskin and trunk skin. We added acknowledgement of the difference from trunk skin when introducing this re-analysis in the re-written manuscript.

(n) L214-221: It is important to better describe/introduce fingerprint ridges development. The reader may get confused with rete ridges vs fingerprint ridges; i.e. its very important for this paper to make clear that fingerprint ridges form through a different (HF-like) molecular mechanism, which then makes it clear in L124-221 why you can use (human) fingerprint bud development as a proxy for (pig) HF bud development.

We have expanded upon the discussion of basal buds and their common markers for subsequent hair follicle, sweat gland, and fingerprint ridge formation. We have also highlighted some of the proposed fingerprint ridge-specific markers ¹ to show that these are not expressed during rete ridge formation (Fig. 3a-d). We have also added greater clarity to our presentation of re-analyses from the Glover et al. 2023 dataset, including circling the “basal buds” in the UMAP to aid in interpretability of FeaturePlots (Supplemental Fig. 5c; Supplemental Fig. 17d-e).

With our new investigation of fingerpad rete ridges in mice, we have attempted to be very clear and consistent in referring to these as “fingerpad rete ridges” and not “fingerpad ridges” in order to avoid perceived overlap with so-called fingerprint ridges. Additionally, we have now emphasized the distinct developmental timing between hair follicles, sweat glands, fingerprint ridges, to clearly distinguish these appendages from rete ridges. In particular, since fingerprint ridges express LEF1/EDAR ¹, our use of epidermal knockout models to demonstrate that LEF1/WNT is not required for fingerpad rete ridge formation emphasizes the distinction between these two types of epithelial appendages (Fig. 3d).

(o) Fig3: Please include another representation showing the expression of LEF1, EDAR and ITGA6 in more detail for the dataset, either as feature plots or expression levels in the different clusters.

We have now provided expanded supplementary materials presenting DotPlots of marker genes used to label the various epidermal states, including LEF1 and EDAR for basal buds, across all of our scRNA-seq epidermal subsets from human, pig, and mouse scRNA-seq (Supplemental Fig. 4e-f, i-j). Furthermore, expression of LEF1, EDAR, and SOX9 is presented as FeaturePlots for the integrated E90, P3, P10, and 6mo basal epidermis (Fig. 3b-c), where developing rete ridges clearly do not express these basal bud markers (Fig. 3c). We have also added FeaturePlots comparing expression of basal bud markers LEF1, EDAR, and SOX9 across our re-analyzed human fetal volar skin, neonatal foreskin, and adult trunk skin datasets (Supplemental Fig. 5c-d).

(p) Fig 3c, what are the clusters 3 and 6, are these the developing HFs?

In the former 3c, cluster 3 comprised dividing keratinocytes and cluster 6 a suprabasal state. Clearer classification of individual numbered clusters alongside DotPlots depicting expression of marker genes used to classify them has been added to each dataset's UMAP in our new scRNA-seq analyses (Supplemental Fig. 4b, e, g, i).

(q) Lines starting at 242 have references to Fig 1, but should be referring to Fig 4?

With the rewritten manuscript and substantially revised Fig. 2-5, this typo has been corrected.

(r) Fig4d-g: where are these zoom-ins taken from?

Zoom-ins were taken from central wound regions of the section.

(s) Figs 3 & 5: please include panels showing the gene expression signatures used for cluster annotations.

Clearer classification of individual numbered clusters alongside DotPlots depicting expression of marker genes used to classify them has been added to each dataset's UMAP in our new scRNA-seq analyses (Supplemental Fig. 4b, e, g, i). All datasets are also available and can be publicly queried using our webtool: [https://skinregeneration.org/papers/Thompson-et-al-2025/\(t\)](https://skinregeneration.org/papers/Thompson-et-al-2025/(t)) Fig 6c: Which timepoint(s) were used for CellChat analysis?

In our revised manuscript, we perform CellChat analysis on E90, P3, P10, 6mo porcine scRNA-seq datasets. We have added greater clarity for which computational methods were applied to which datasets to the revised Methods section. We have also updated our Github page to include all the scripts utilized for the data presented in the revised manuscript.

(u) The methods and GEO upload indicates that pig samples were acquired from *Sus scrofa* species, which is the wild boar. Is that really the case or do the authors mean *Sus Domesticus* (or *Sus scrofa domesticus*), which would be the domestic pig that's depicted in the figures?

Sus scrofa is a species name broadly encompassing both domestic and wild breeds of pigs, including the domestic and Mangalitsa pigs^{44,45}. In this study we utilized the *Sus scrofa* genome Sscrofa11.1⁴⁵ to align 10x Genomics scRNA-seq data using CellRanger, which is a reference genome built off of a Red Duroc-background (domestic) pig (NCBI RefSeq GCF_000003025.6). Pigs used in our study were from local farmers and were of a mixed Yorkshire and Red Duroc background, which is described in our Methods section. The *Sus scrofa* Sscrofa11.1 reference genome is commonly used in studies involving domestic and wild pigs^{46,47}.

Referee #3 (Remarks on code availability):

I have looked at <https://github.com/DriskellLab/Thompson-et-al.-2024>, and at a glance all code is there and looks fine. However, we have not re-run the code.

We appreciate the comment from the reviewer. We have updated our Github to reflect the code used to generate data in this revised manuscript.

References

1. Glover, J. D. *et al.* The developmental basis of fingerprint pattern formation and variation. *Cell* **186**, 940-956.e20 (2023).
2. Magerl, M. *et al.* Patterns of Proliferation and Apoptosis during Murine Hair Follicle Morphogenesis. *J. Invest. Dermatol.* **116**, 947–955 (2001).
3. Leung, Y., Kandyba, E., Chen, Y.-B., Ruffins, S. & Kobiela, K. Label Retaining Cells (LRCs) with Myoepithelial Characteristic from the Proximal Acinar Region Define Stem Cells in the Sweat Gland. *PLOS ONE* **8**, e74174 (2013).
4. Cui, C.-Y. *et al.* Involvement of Wnt, Eda and Shh at defined stages of sweat gland development. *Development* **141**, 3752–3760 (2014).
5. Lu, C. P., Polak, L., Keyes, B. E. & Fuchs, E. Spatiotemporal antagonism in mesenchymal-epithelial signaling in sweat versus hair fate decision. *Science* **354**, aah6102 (2016).
6. Montagna, W. The evolution of human skin(?). *J. Hum. Evol.* **14**, 3–22 (1985).
7. Montagna, W. Comparative Anatomy and Physiology of the Skin. *Arch. Dermatol.* **96**, 357–363 (1967).
8. Schwartz, G. G. & Rosenblum, L. A. Allometry of primate hair density and the evolution of human hairlessness. *Am. J. Phys. Anthropol.* **55**, 9–12 (1981).
9. Kamberov, Y. G. *et al.* Comparative evidence for the independent evolution of hair and sweat gland traits in primates. *J. Hum. Evol.* **125**, 99–105 (2018).
10. Kamberov, Y. G. *et al.* A genetic basis of variation in eccrine sweat gland and hair follicle density. *Proc. Natl. Acad. Sci.* **112**, 9932–9937 (2015).
11. Phan, Q. M. *et al.* Lef1 expression in fibroblasts maintains developmental potential in adult skin to regenerate wounds. *eLife* **9**, e60066 (2020).

12. Genderen, C. van *et al.* Development of several organs that require inductive epithelial-mesenchymal interactions is impaired in LEF-1-deficient mice. *Genes Dev.* **8**, 2691–2703 (1994).
13. Zhou, P., Byrne, C., Jacobs, J. & Fuchs, E. Lymphoid enhancer factor 1 directs hair follicle patterning and epithelial cell fate. *Genes Dev.* **9**, 700–713 (1995).
14. Li, J. *et al.* Limb development genes underlie variation in human fingerprint patterns. *Cell* **185**, 95-112.e18 (2022).
15. Jamora, C., DasGupta, R., Kocieniewski, P. & Fuchs, E. Links between signal transduction, transcription and adhesion in epithelial bud development. *Nature* **422**, 317–322 (2003).
16. Durmowicz, M. C., Cui, C.-Y. & Schlessinger, D. The *EDA* gene is a target of, but does not regulate *Wnt* signaling. *Gene* **285**, 203–211 (2002).
17. Leveque, J. L. & Corcuff, P. The Surface of the Skin — The Microrelief. in *Noninvasive Methods for the Quantification of Skin Functions: An Update on Methodology and Clinical Applications* (eds. Frosch, P. J. & Kligman, A. M.) 3–24 (Springer, Berlin, Heidelberg, 1993). doi:10.1007/978-3-642-78157-5_1.
18. Okajima, M. Development of dermal ridges in the fetus. *J. Med. Genet.* **12**, 243–250 (1975).
19. Penrose, L. S. & Ohara, P. T. The development of the epidermal ridges. *J. Med. Genet.* **10**, 201–208 (1973).
20. Jung, H.-S. *et al.* Local Inhibitory Action of BMPs and Their Relationships with Activators in Feather Formation: Implications for Periodic Patterning. *Dev. Biol.* **196**, 11–23 (1998).
21. Plikus, M. *et al.* Morpho-Regulation of Ectodermal Organs: Integument Pathology and Phenotypic Variations in K14-Noggin Engineered Mice through Modulation of Bone Morphogenic Protein Pathway. *Am. J. Pathol.* **164**, 1099–1114 (2004).
22. Mishina, Y., Hanks, M. C., Miura, S., Tallquist, M. D. & Behringer, R. R. Generation of *Bmpr/Alk3* conditional knockout mice. *genesis* **32**, 69–72 (2002).

23. Vasioukhin, V., Degenstein, L., Wise, B. & Fuchs, E. The magical touch: Genome targeting in epidermal stem cells induced by tamoxifen application to mouse skin. *Proc. Natl. Acad. Sci.* **96**, 8551–8556 (1999).
24. Ghazizadeh, S. & Taichman, L. B. Organization of Stem Cells and Their Progeny in Human Epidermis. *J. Invest. Dermatol.* **124**, 367–372 (2005).
25. Sellheyer, K. *et al.* Inhibition of skin development by overexpression of transforming growth factor beta 1 in the epidermis of transgenic mice. *Proc. Natl. Acad. Sci.* **90**, 5237–5241 (1993).
26. Bhora, F. Y. *et al.* Effect of Growth Factors on Cell Proliferation and Epithelialization in Human Skin. *J. Surg. Res.* **59**, 236–244 (1995).
27. Benjamin, L. E., Hemo, I. & Keshet, E. A plasticity window for blood vessel remodelling is defined by pericyte coverage of the preformed endothelial network and is regulated by PDGF-B and VEGF. *Development* **125**, 1591–1598 (1998).
28. Li, X. & Eriksson, U. Novel PDGF family members: PDGF-C and PDGF-D. *Cytokine Growth Factor Rev.* **14**, 91–98 (2003).
29. Tallquist, M. & Kazlauskas, A. PDGF signaling in cells and mice. *Cytokine Growth Factor Rev.* **15**, 205–213 (2004).
30. McDermott, U. *et al.* Ligand-Dependent Platelet-Derived Growth Factor Receptor (PDGFR)- α Activation Sensitizes Rare Lung Cancer and Sarcoma Cells to PDGFR Kinase Inhibitors. *Cancer Res.* **69**, 3937–3946 (2009).
31. Oike, Y. *et al.* Angiopoietin-related growth factor (AGF) promotes epidermal proliferation, remodeling, and regeneration. *Proc. Natl. Acad. Sci.* **100**, 9494–9499 (2003).
32. Xia, Y.-P. *et al.* Transgenic delivery of VEGF to mouse skin leads to an inflammatory condition resembling human psoriasis. *Blood* **102**, 161–168 (2003).

33. Mine, S., Fortunel, N. O., Pigeon, H. & Asselineau, D. Aging Alters Functionally Human Dermal Papillary Fibroblasts but Not Reticular Fibroblasts: A New View of Skin Morphogenesis and Aging. *PLOS ONE* **3**, e4066 (2008).
34. Paquet-Fifield, S. *et al.* A role for pericytes as microenvironmental regulators of human skin tissue regeneration. *J. Clin. Invest.* **119**, 2795–2806 (2009).
35. Ganier, C. *et al.* Multiscale spatial mapping of cell populations across anatomical sites in healthy human skin and basal cell carcinoma. *Proc. Natl. Acad. Sci.* **121**, e2313326120 (2024).
36. Wang, S. *et al.* Single cell transcriptomics of human epidermis identifies basal stem cell transition states. *Nat. Commun.* **11**, 4239 (2020).
37. Cheng, J. B. *et al.* Transcriptional Programming of Normal and Inflamed Human Epidermis at Single-Cell Resolution. *Cell Rep.* **25**, 871–883 (2018).
38. Anwer, A. W., Samad, L., Iftikhar, S. & Baig-Ansari, N. Reported Male Circumcision Practices in a Muslim-Majority Setting. *BioMed Res. Int.* **2017**, 4957348 (2017).
39. Miyauchi, Y. *et al.* Developmental Changes in Neonatal and Infant Skin Structures During the First 6 Months: In Vivo Observation. *Pediatr. Dermatol.* **33**, 289–295 (2016).
40. Erdem, E. *et al.* Histological and morphological development of the prepuce from birth to prepubertal age. *Investig. Clin. Urol.* **65**, 180–188 (2024).
41. Limandjaja, G. C. *et al.* Increased epidermal thickness and abnormal epidermal differentiation in keloid scars. *Br. J. Dermatol.* **176**, 116–126 (2017).
42. Chuong, C.-M. Morphogenesis of Epithelial Appendages: Variations on Top of a Common Theme and Implications in Regeneration. in *Madame Curie Bioscience Database [Internet]* (Landes Bioscience, 2013).
43. Liu, Y. *et al.* Hedgehog signaling reprograms hair follicle niche fibroblasts to a hyper-activated state. *Dev. Cell* **57**, 1758-1775.e7 (2022).

44. Broadway, M. S. *et al.* Interspecific effects of invasive wild pigs (*Sus scrofa*) on native nine-banded armadillos (*Dasypus novemcinctus*). *J. Mammal.* gyaf023 (2025)
doi:10.1093/jmammal/gyaf023.
45. Warr, A. *et al.* An improved pig reference genome sequence to enable pig genetics and genomics research. *GigaScience* **9**, giaa051 (2020).
46. Yang, H. *et al.* ABO genotype alters the gut microbiota by regulating GalNAc levels in pigs. *Nature* **606**, 358–367 (2022).
47. Jin, L. *et al.* A pig BodyMap transcriptome reveals diverse tissue physiologies and evolutionary dynamics of transcription. *Nat. Commun.* **12**, 3715 (2021).

Referees' comments:

Responses to reviewers and quotes from the revised manuscript have been color coded for visual clarity:

Original submission in "black"

First revision in "blue"

Second revision in "green"

Referee #1 (Remarks to the Author):

The authors have extensively revised the manuscript, including changing the title from "Evolutionary development, regeneration, and induction of rete ridges in mammalian skin" To "Rete ridges form through evolutionarily distinct mechanisms from other appendages during mammalian skin development". In this revision, the domestic pig is used as the main model to characterize when rete ridges develop. Bioinformatic analyses of pig and human skin are applied to infer mechanisms of ridge formation. The authors attempt to place their proposed mechanism in an evolutionary context by including both ridge-bearing and non-ridge-bearing mammals. Finally, they perform pharmacological treatments (TPA and RA) in newborn mouse skin to induce ridge-like structures, suggesting possible applications.

The revision represents significant effort and provides valuable data on ridge formation in pig, human, and mouse, and with potential translational implications. Ridge formation is a significant topic. Methodology are in general of good quality. The manuscript has improved a lot. However, several points require further clarification and revision. This is a strong body of work on pig, mouse and human, but the conclusion on evolutionary aspect is not sufficiently supported and should be tuned down. It is not necessary to extend the conclusions to encompass all mammalian skin appendages and evolution without a systematic comparative study, though such discussion is interesting and can be raised in a speculative manner.

R1-1A) Evolutionary aspect of rete ridges

- The authors emphasize the relationship between the order of hair, sweat gland, and ridge formation. Their detailed work in pigs, humans, and mice is strong.
- While data from other mammals (primates, grizzly bear, dolphin) add interest, this does not constitute a systematic survey of mammalian skin development. I do not suggest additional mammalian sampling, but without broader evidence, the evolutionary conclusions should be tempered and potential alternative mechanisms acknowledged.

We thank the reviewer for their interest in the breadth of species investigated in this work. We have added acknowledgement of potential ambiguity between the evolutionary and developmental mechanisms active in these other non-model species to the Discussion section of the manuscript.

R1-1B) The manuscript would benefit from discussing the following reference:

o Alibardi, L. (2004). Dermo-epidermal interactions in reptilian scales: speculations on the evolution of scales, feathers, and hairs. *J. Exp. Zool. Part B: Mol. Dev. Evol.* 302B(4): 365–383.

We thank the reviewer for this critical suggestion. We have incorporated discussion of epidermal-dermal interactions in broader appendage specification into the Introduction and Discussion sections:

- “Distinct epidermal and dermal signaling programs are key evolutionary mechanisms driving specification of diverse epidermal appendages within the skin of vertebrates, such as scales in reptiles, feathers in birds, and hair follicles in mammals ^{1,2,4,32–38}.” (lines77-79).

We have also acknowledged potential nuances or differences in the sequence or discrete timing of specification of different epithelial appendages during skin development, as it is possible that other species (or different breeds of pigs that form both primary and secondary hair follicles, as discussed by this reviewer in R1-4) may exhibit concurrent rather than sequential appendage specification:

- “Earlier waves of appendage morphogenesis influence the patterning of subsequent appendages, so future studies will need to understand how rete ridge formation is influenced by the balancing between the Turing reaction-diffusion patterning and expansion-induction patterning mechanisms operating in tissue spaces between previously established or potentially concurrently developing hair follicles and sweat glands” (lines493-498)

R1-2) Mechanisms of ridge formation

- The multi-species scRNA analyses are comprehensive and are a strength of the paper.
- Based on these, the authors highlight the role of dermal vasculature, suggesting epidermal–endothelial interactions are important in pig ridge formation, analogous to epidermal–dermal interactions in hair follicle formation.
- However, epidermal ridges represent a general morphological transformation and may form through diverse cellular mechanisms. The vascular model should be presented as one possible mechanism rather than the sole mechanism of ridge formation.

We thank the reviewer for their interest in the epidermal-dermal interactions contributing to rete ridge formation. Vascular integration and/or interaction is a consistent component of other epithelial appendages, including hair follicles ^{1,2} and feathers ³. However, the reviewer is correct that future studies will be needed to define how the relationship between rete ridges and dermal vasculature functionally compares to other epithelial appendages. We have added discussion to acknowledge these uncertainties as well as other potential epidermal-dermal interactions, including possible contributions from nerve and immune cell lineages:

- “Other epidermal- or dermal-resident cell types may also contribute to rete ridge formation and maturation, including nerve cell endings and immune cells ^{15,54,56,83,105}, which were beyond the scope of the current study. Future studies will be needed to understand how dermal and vascular contributions during rete ridge formation and

patterning differ from those involved in hair follicle, sweat gland, or fingerprint ridge formation^{1,2,4,26,38,42,45,102}." (lines509-514)

There is strong historical and clinical evidence implicating a functional and even mechanistic role for vasculature and rete ridge formation/maintenance. The dermal pockets between the towering rete ridges in cetacean skin are abundantly vascularized and innervated⁴⁻⁶. Clinically, in humans, infantile hemangioma is a usually benign tumor which exhibits dynamic dysfunction in angiogenic signaling and vascular expansion, with early stages of these tumors displaying sparse vasculature followed by upregulation in pro-vascular signaling, including VEGF and PDGF, and a rapid proliferation and expansion of vasculature in the tumor environment^{7,8}. Based on our scRNA-seq and Stereo-seq CellChat analyses, both VEGF and PDGF signaling are normally active during rete ridge formation (Supplemental Fig. 9e; Supplemental Fig. 10c-e; Supplemental Fig. 14a-b, d). Critically, hyper-vascularized infantile hemangiomas do not display rete ridges⁸, suggesting that the organized patterning of dermal vasculature may support rete ridge morphogenesis, as mentioned in the Discussion section:

- "In this work, we have highlighted the potential relationship between the epidermis and the underlying dermis in the establishment of rete ridges. The dermal pocket enables papillary fibroblasts, pericytes, blood vessel, and lymphatic vessel cells to assemble beneath the basal epithelium and establish a signaling source that potentially assists with epidermal thickening and rete ridge maintenance^{52,82-84}. Clinical support for the underlying dermal vasculature organization's potential contribution to rete ridge patterning can be observed in human infantile hemangiomas, benign skin tumors which display profound dermal hypervascularization and completely lack rete ridges^{103,104}." (lines502-509)

R1-3A) Regional specificity of rete ridges

- Fig. 2: The tissue collection sites for each species must be specified. Regional differences in rete ridge morphology could strongly affect conclusions (see Moyo 2018).
- Human rete ridge length and width vary regionally, as in cetaceans and most mammals. Comparable anatomical sites must therefore be compared across species.
- To avoid confusion, I recommend adding a supplemental figure showing the anatomical collection sites for each species.

We used consistent anatomical sites across species in almost all cases. We have added more written detail about these exact anatomical sites to every figure legend, the Methods section, and created a graphical summary in Supplemental Fig. 2f (and referenced this in the Methods). Specifics are also provided in response to individual reviewer questions about anatomical site below.

For brevity, we include this graphical summary here (Rebuttal Fig 1a) and summarize anatomical sites by species in bullet points (and referencing exact lines in the Methods):

- Naked mole rat, rhesus macaque, marmoset "back skin" (lines550-553, 562)
- Dolphins "trunk skin" (lines565-572)
- North American Grizzly Bear (*Ursus arctos horribilis*): rump skin of the lower back (lines572-578)
- Mice: back skin or forepaws/digits (lines578-594)

- Pigs: dorsal midline of the upper back (lines608-635)
- Humans: gestational samples from the back, adult samples mainly from the trunk with some from the face and a single sample from scalp and base of the nose for hair density quantifications (all clearly delineated now in the Methods and Figure Legends which regions were included in which figure panels) (lines594-605; lines733-735; lines739-744).

R1-3B) Authors may also discuss:

o Kowalczyk A., Chikina M., Clark N. (2022). Complementary evolution of coding and noncoding sequence underlies mammalian hairlessness.

We thank the reviewer for this critical suggestion and have discussed contributions from coding vs non-coding sequence variations in regulating hair density and hair formation across species to our discussion of our transgenic model inhibiting hair formation by ablating epidermal *Lef1*:

- "Critically, the dramatically reduced hair density in the absence of epidermal *Lef1* did not alter the thickness of the interfollicular epidermis at examined stages of postnatal development and maturation, suggesting that genetic reduction of hair density alone may not directly drive epidermal thickening or rete ridge formation (Supplemental Fig. 3g-i). Diverse combinations of coding and non-coding sequence variations are also associated with fine tuning hair density and hair shaft characteristics across *Mammalia*⁶¹. These studies alongside our murine model provide support for the combinatorial regulation of skin appendage density and specification, as the disparate hair density between naked mole rats and mice does not predict the presence of rete ridges, suggesting that rete ridge formation may be a separate process from modulations in hair density (Fig. 2a-c; Supplemental Fig. 2d)." (lines209-218)

This addition has reframed the presentation of the evolutionary independence of modulations in hair density and acquisition of rete ridge formation, as the "hairless" naked mole rat and densely haired mouse both lack rete ridges in their trunk skin regardless of their relative hair density (Fig. 2a):

- "Based on our histological zoo, we found that animals with rete ridges generally have a thicker epidermis and lower hair density than animals without rete ridges. However, modulations in hair density due to natural evolution or single-gene mutations support independence between hair density and acquisition of rete ridges^{57,92}, which we explored in this study using *Lef1* epidermal knockout mice. Selective breeding also

supports this, as Chinese crested dogs, hairless due to mutations in the *FOXI3* gene ⁷, have thin skin with sweat glands but lack rete ridges ⁹³. Similarly, naked mole rats have thin epidermis, sparse hairs, no sweat glands, and also lack rete ridges. Hypertrichosis in humans, such as in congenital generalized hypertrichosis, also does not appear to impact rete ridge formation ^{94–96}. In contrast, functionally hairless cetaceans, dolphins and whales, have extremely thick trunk skin with towering rete ridges, compared to other terrestrial mammals ^{53–56}." (lines461-471)

R1-4A) Ridges in pig skin

- Wild pigs possess both primary and secondary hair as well as rete ridges. In contrast, domestic breeds used here lack secondary hair and have reduced primary hair (Meyer 1986). This challenges the claim that “hairlessness coincides with rete ridge development.”
- Humans with scalp hair (and with congenital generalized hypertrichosis, “wolfman syndrome”) also display rete ridges, further weakening this association.

We thank the reviewer for making these insightful recommendations. Indeed, we agree that the reviewer’s point is valid. We have now tempered the language related to the reductions in hair density to the onset of rete ridge formation in the Abstract, Results, and Discussion sections.

In particular, we now emphasize that the “reduced hair density [in humans] coincides with the acquisition of epidermal rete ridges...” (lines 38-39) in our Abstract/Summary paragraph to avoid language implying a causative association between these two evolutionary phenomena.

In the Discussion section, we have clarified our original presentation of the evolutionary relationship between hair density and rete ridge formation, since a reduction in hair density does not intrinsically precipitate acquisition of rete ridges. We also incorporated discussion of congenital generalized hypertrichosis, as positive modulations in hair growth, such as increased hair coverage within species capable of forming rete ridges. Critically, the “fur-like” appearance of bears or humans with congenital generalized hypertrichosis is more a product of increased hair fiber growth than it is a product of increased hair follicle density (Fig. 2a-d) ^{9–11}, and does not itself seem to alter the rete ridges:

- "However, modulations in hair density due to natural evolution or single-gene mutations support independence between hair density and acquisition of rete ridges ^{57,92}, which we explored in this study using *Lef1* epidermal knockout mice. Selective breeding also supports this, as Chinese crested dogs, hairless due to mutations in the *FOXI3* gene ⁷, have thin skin with sweat glands but lack rete ridges ⁹³. Similarly, naked mole rats have thin epidermis, sparse hairs, no sweat glands, and also lack rete ridges. Hypertrichosis in humans, such as in congenital generalized hypertrichosis, also does not appear to impact rete ridge formation ^{94–96}. In contrast, functionally hairless cetaceans, dolphins and whales, have extremely thick trunk skin with towering rete ridges, compared to other terrestrial mammals ^{53–56}." (lines462-471).

Regarding wild pigs’ assortment of primary and secondary hairs, the presence of secondary hair follicles can also be observed in the heritage breed Mangalitsa pigs used in this study, which do have higher hair density than a mixed background “improved” breed pig in the United States (Fig. 2a-d). Both breeds have comparable rete ridges, so we have also rephrased the

Discussion section to further clarify claims related to hair density and rete ridges during evolution:

- “Due to their unique cellular and molecular characteristics, we propose that rete ridges be considered an appendage acquired *de novo* over the course of evolution *pari passu* with reductions in hair density and coverage, rather than merely as a derivative of a hair follicle, sweat gland, or fingerprint ridge.” (lines514-517).

Future comparative developmental studies will be needed to clarify whether the sequential formation of hair follicles, then sweat glands, and finally rete ridges occurs in orderly, non-overlapping waves, similar to what we observed in human/pig back skin and murine fingerpads, or whether there are instead overlapping gradients in appendage specification in non-model organisms, such as Grizzly bears or heritage breed pigs. Furthermore, as spatial patterning and paracrine signaling gradients can influence appendage specification and density in different anatomical sites or species¹²⁻¹⁹, future studies will be needed to understand how modulating the density of previous waves of epithelial appendages, like hair follicles or sweat glands, influences rete ridge patterning. We acknowledge this uncertainty in the Discussion section:

- “Earlier waves of appendage morphogenesis influence the patterning of subsequent appendages, so future studies will need to understand how rete ridge formation is influenced by the balancing between the Turing reaction-diffusion patterning and expansion-induction patterning mechanisms operating in tissue spaces between previously established or potentially concurrently developing hair follicles and sweat glands^{2,4,38,42,43,45}.” (lines493-498)

• **Specific clarifications needed:**

R1-5) Which pig breed was used for Fig. 1?

Pigs used in developmental studies were obtained from local farmers and were of a mixed Yorkshire/Red Duroc/Hampshire background and were generally unpigmented. Some of the piglets had fully pigmented skin or spots due to mixed background (e.g. Hampshire, which is known for having broad pigmented bands in the skin) and possible past interbreeding with regional Kunekune or Idaho pasture pig breeds. The 7mo-old pigs had pigmented skin due to this mixed background. Extra background details have been added to figure legends and the Methods section, and are quoted below:

- "Fetal pigs obtained from Biology Products were of unpigmented and of unknown background. Known embryonic day 90 pigs (used for histology, immunostaining, and scRNA-seq) were harvested from an unpigmented pregnant sow raised on a local farm of mixed Yorkshire and Red Duroc background. Postnatal pigs obtained from local farmers or butchers were of mixed backgrounds between predominantly Yorkshire, Red Duroc, and Hampshire breeds, with occasional regional interbreeding with Idaho Pasture pigs and Kunekunes, which have pigmented skin. Notably, the Hampshire is also known for exhibiting banded pigmentation in its skin. As such, most skin we studied was unpigmented, but some piglets or adults had pigmented skin or spots (as visible in Fig. 2b and Supplemental Fig. 7c). Every effort was made to collect skin samples from both

males and females across all timepoints. Adult male skin (6mo/7mo) collected from local farmers and butchers were presumably castrated prior to weaning in order to avoid boar taint in the meat." (lines618-629)

R1-6) How many breeds were studied across developmental stages?

We did not utilize pure breeds, aside from the Mangalitsa pig, Yucatan pig, and Hanford mini-pig for adult timepoints, in this manuscript. All pigs for postnatal developmental stages were obtained from local farmers on a mixed background (see detailed response to R1-5).

R1-7) Were all samples taken from the same body site? If not, please clarify.

Yes, in Fig. 1 all pig skin samples were obtained from the back skin along the dorsal midline approximately between the shoulder blades. Extra details about anatomical sites has been added to the Methods text for all species, and we have created a visual summary of anatomical site by species (Rebuttal Fig. 1a; Supplemental Fig. 2f).

R1-8) For fetal pig experiments, what breeds were used?

The fetal pigs obtained from Biology Products (Alexandria, MN) were of unknown background and had unpigmented skin. We harvested piglets of known embryonic day 90 from an unpigmented sow of mixed Yorkshire background obtained from a local farmer. Skin from these known E90 piglets were used for scRNA-seq and histology.

o Sinclair Biosciences lists Yucatan™ Miniature and Micro-Yucatan™ Miniature Swine, both described as "hairless." Which was used? Please correct the methods accordingly.

Yucatan Miniature. This clarification has been added to both the figure legends and the Methods.

R1-9) Hair and ridge formation in humans

- Individuals with congenital generalized hypertrichosis ("wolfman syndrome") develop both hair follicles and rete ridges. I recommend removing this abstract sentence: "This loss coincides with the acquisition of epidermal rete ridges..."

We have adjusted this statement to read "Reduced hair density..." instead of "loss". The hair density of species with rete ridges (e.g. humans) is dramatically lower than species without, even in bears which have dense bundles of hair follicles separated by expansive interfollicular epidermis containing rete ridges (Fig. 2b-d). As such, we would argue that external hair length more so than just hair density determines the external appearance and common classification of skin as "furry" (which bears are often considered). Human scalp hair is a good example of this principle, since prolonged anagen enables scalp hair elongation, especially compared to skin of the body, providing the perception and function (e.g. thermoregulation) of higher hair coverage, similar to the appearance of grizzly bear trunk skin despite both having dramatically lower hair density than other canonically furry species, like mice (Fig. 2b, d)^{9,20}. Relatedly, congenital

generalized hypertrichosis causes elongation of hair follicles in human trunk or face skin without necessarily altering the hair density itself or the rete ridges⁹⁻¹¹.

R1-10) Fig. 2: Human epidermis is established to be thinner than pig epidermis (Mayer 2009; Summerfield 2014, among others). The figure and data here suggest the opposite. Please reconcile this discrepancy.

- Revise Fig. 2a to show that pig epithelium is thicker than human.

We thank the reviewer for their interest in the comparison in epidermal thickness between species. In our analyses of both human and adult (6-7mo) domestic pig skin, we observed no significant differences in epidermal thickness, in either rete ridge or inter-ridge measurements (Fig. 1d,i; Fig. 2c; Supplemental Fig. 2d). However, we have re-arranged the ordering of humans and pigs on the epidermal thickness gradients in Fig. 2c and Supplemental Fig. 2d at this reviewer's request.

The Mayer 2009 study²¹ reported on the skin of wild, feral pigs of undefined adult age. Critically, this study does not appear to subdivide the thickness measurement of skin into the different sublayers. They note "the skin thickness averages about 1 to 2 mm. On adult boars, this thickness in the area of the shoulder may increase to 3.5 to 4mm"²¹. These measurements appear to more accurately describe the overall skin thickness. This interpretation meshes with our extensive experience collecting skin from the upper back region of a variety of ages and different breeds of pigs, and these measurements are consistent with the full-thickness of skin rather than just the epidermal thickness. Notably, only cetaceans appear to have epidermal thickness in the order of millimeters of thickness, and this thickness is many orders of magnitude thicker than other terrestrial mammalian epidermis other than hippos, whose skin exhibits much similarity with cetaceans^{4-6,22,23}.

Additionally, the Summerfield et al. 2015 study²² summarized previous studies comparing human and pig skin as follows, "Complete epidermis varies from 30 to 140um in thickness in pigs compared to 50-120um in humans", which aligns with the values we observed and reported from our human and domestic pig samples in this manuscript. It is worth noting that since most of our adult pig timepoints (6mo-7mo) were collected from local farmers and butchers, the males had likely been castrated early in postnatal life to prevent "boar taint" from ruining the marketability of the butchered meat. It is possible that this could influence the observed "maximum" epidermal thickness in our adult pig samples, so we have added more details about these animal backgrounds and possible effects on epidermal thickness to the Methods section.

- **R1-11)** Fig. 1 caption: please specify tissue sites ("Representative H&E stains of trunk and face skin of adult humans") in both figure and Methods.

Additional details on anatomical site from human samples have been added to the figure legend and explicitly described in the Methods for each figure. A visual representation is now provided in Supplemental Fig. 2f and above in Rebuttal Fig. 1.

R1-12) Reverse engineering in the mouse

- This is a strength of the manuscript, adding functional validation and translational potential.

- As authors acknowledge, the effect of TPA on cell proliferation / epidermal thickness is more comprehensible. The mechanism of RA action in this context should be more clearly explained.

We thank the reviewer for raising this point. In the original manuscript, we presented an inducible model of epidermal ridge-like structure formation via topical TPA and RA treatment. While this approach generated structures morphologically resembling rete ridges, we fully acknowledge the reviewer's concerns regarding mechanistic definitions and validation: the pleiotropic nature of these pharmacological compounds and poorly defined molecular mechanisms stimulated to influence epidermal differentiation, inflammation, or wound repair was a limitation of the approach in clearly elucidating molecular mechanisms underlying rete ridge formation. We took this concern seriously and revised our approach accordingly in the first revision:

In the revised manuscript, we substantially restructured our use of mouse models to reflect a more refined and biologically grounded strategy. We now define and emphasize rete ridge formation in murine volar fingerpads as a native, genetically tractable site of natural rete ridge formation (Fig. 3d; Fig. 5a-d; Supplemental Fig. 6a-c; Supplemental Fig. 17a-b). This model is not chemically induced, nor wound-dependent, and occurs naturally during postnatal development. As such, it provides a powerful opportunity to interrogate ridge morphogenesis in a physiologically relevant context.

Importantly, the volar fingerpad ridge structures are not just morphologically similar to those in human and porcine skin, they are molecularly and spatially conserved. These murine structures exhibit BMP and PDGFC signaling activity, thickened layers, and the expected undulating pattern of the basal epithelium (Figure 5a-d, Supplemental Fig 6a-b; Supplemental Fig 17a-b, h-i). We believe this directly strengthens the central claims of our manuscript.

To further demonstrate that these ridges represent a biologically distinct developmental program, we genetically ablated *Lef1* in the epidermis of mice using *K14Cre Lef1 fl/fl (Lef1-eKO)* mice (Figure 2e-f; Supplemental Fig 3a-i). These mice exhibit dramatically reduced hair follicle formation and still form rete ridges in the fingerpad (Figure 3d; Supplemental Fig. 3c-d). This key finding supports our central claim that rete ridge formation is governed by a distinct, non-follicular/glandular epithelial differentiation program.

We are grateful for the reviewer's critique, which helped us elevate this aspect of the study from descriptive to mechanistic. The characterization and implementation of the murine fingerpad as a model for rete ridges has strengthened the conceptual clarity of the work.

Minor Points:

R1-13) Fig. 1g: replace "dermal pockets" with dermal ridges to avoid confusion with hair dermal papillae.

Upon further consideration, we chose to keep the term "dermal pockets". In historical literature, what we term in this study as the dermal pocket is often referred to as a dermal papilla⁴⁻⁶, which we agree is absolutely confusing with the far more commonly described hair follicle-associated dermal papillae. Dermal ridges better define the underlying dermal architecture of skin microrelief, or the overall surface patterning and texture which creates hill-like patterns in

the macroscopic epidermal architecture independent of rete ridges, such as fingerprint ridges^{24–26}. This macroscopic epidermal and dermal ridging more closely resembles that of the dermal compartment of scaly skin than it does the rete ridge and vascularized dermal pocket^{27,28}. Furthermore, microrelief and epidermal/dermal architectural ridging intercalates hair and rete ridge patterning in skin, as visible in the variety of mammalian species included in this manuscript (see the H&E and Herovici stains in Supplemental Fig. 2a).

R1-14) Fig. 2b: clarify the term “hairiness.” If this refers to hair per unit area, please use hair density. Also, the gross image under “domestic pig” shows black skin—please specify the breed in text and figure.

We thank the reviewer for their attention to the details of this figure. The gradient is meant to be a relative representation of hair density between different species, so we have replaced the gradient label “Hairiness” with “Hair Density”.

We thank the reviewer for their attention to detail and porcine skin pigmentation. 6-month old and 7-month old pig skin analyzed in this manuscript were obtained from local butchers and pigs from different regional farms. The pigs in our region consist of generally mixed backgrounds from predominantly Yorkshire, Red Duroc, and Hampshire (which has pigmented banding in the skin) backgrounds. Farmers in the region also raise rarer breeds, Idaho Pasture pigs and Kunekunes, which sometimes have dark, pigmented skin, and frequently crossbreed to promote hardiness or other improved meat qualities in the progeny. As a result, pig skin utilized in this study displays a variety of pigmentation resulting from crossbreeding by different regional farmers, ranging from pink, spotted, to fully pigmented. Where possible, unpigmented skin from the dorsal midline (trunk skin of the upper back in line with the spine and between the shoulders) was collected to be consistent across samples and breed backgrounds. However, if pigmented spots were present in this region, then we would collect pigmented skin, as was the case with the adult pigs imaged for hair density quantification. We have provided additional details about the mixed backgrounds of our samples to the Figure Legends and Methods.

R1-15) Grizzly bear skin: published work (Tomiyasu 2017) indicates that back skin of grizzly bears lacks rete ridges. Bears and mice are considered loose-skinned mammals without rete ridges. Please state what anatomical site was sampled, as Fig. 2a does not convincingly show ridges. This discrepancy must be reconciled.

It is worth noting that the Japanese brown bears (*Ursus arctos lasiotus*) in these studies^{29,30} are a distinct subspecies of brown bear from the North American grizzly bears represented in our study (*Ursus arctos horribilis*)³¹. In addition, the histology presented in Tomiyasu et al. 2017³⁰ is from the upper back, and histology from this group frequently focuses on the upper back and related scent glands in male, which has vastly different morphology of hair follicles and sebaceous glands compared to the rump^{29,30}. Since the upper back is the site of “abundant sebum secretion” in male bears and related sebaceous glands change size dramatically in sync with breeding season^{29,30}, this anatomical site may not be generally indicative of the morphology of the rest of the trunk skin of the bear or that of other subspecies of brown bear. As this group consistently demonstrates, there are profound and dynamic changes to the sebaceous glands of the hair follicles and sweat glands in this body region seasonally (e.g.

breeding season) and dependent upon hormones. Furthermore, this group generally focuses on the glands of the grizzly bear, either sebaceous or apocrine in the secretory upper back secretory region, and their histology presents massive glands which we do not see similarly sized in our samples. This could be suggestive of a unique pro-glandular region of the upper back where these authors sampled from their bears, focusing on the bears' secretory activity, whereas our samples were collected from the dorsal rump (aka lower back) under the approved protocols from WSU IACUC. Furthermore, presented histology is generally very zoomed out to showcase the entire hair follicle and glands^{29,30}, and rete ridge recognition may thus be challenging without higher magnification images.

We consistently observed rete ridges in the dorsal rump skin from n=11 male and female grizzly bears ranging in age from 8 to 21 years of age, either born in wild or born in captivity, during the 2023 summer season (Fig. 2a; Supplemental Fig. 2a; Rebuttal Fig. 1a-b). Along the infundibulum-adjacent epithelium, fewer or smaller rete ridges are present (Rebuttal Fig. 1a), though there is minor variation in rete ridge size from bear to bear (Fig. 2c). Additional details regarding age and body region of sampling have been added to the methods and figure legends. All quantifications depicting bear histology quantifications are presented as box plots with biological replicate averages from multiple technical replicates transparently presented as individual data points (Fig. 2c; Supplemental Fig. 2d). We have replaced the image used in the main figures with a different replicate, since the reviewer is correct, it is not our best histology section.

We thank this reviewer for their critical engagement and interest regarding the skin anatomy of bear skin.

Rebuttal Fig. 2: North American Grizzly Bear Histology by Age and Sex. Representative H&E stains of rump skin from adult, 8-year old male and female North American grizzly bears born in captivity (top), 2 18-20-year old male and females born in captivity (middle), and 2 20-21-year old males and females born in the wild (bottom). Black dashed boxes indicate regions of zoom in. Orange dashed lines indicate the smooth infundibulum region. HF=hair follicle, Sw = sweat gland. Scale bars represent 100um. Note: the representative image of 20-year old female “born in wild” replicate in this figure was the biological replicate used in Fig.

2a/Supplemental Fig. 2a in the original submission. Fig. 2a/Supplemental Fig. 2a have been replaced in the second revision by a different biological replicate of 21yo male “born in wild” than was used in this rebuttal figure.

Referee #2 (Remarks to the Author):

The authors have addressed most of the reviewers' comments in the revised manuscript through large efforts and new experiments. These efforts have substantially improved the manuscript through the inclusion of new sequencing data, additional analyses, the establishment of a rete ridge model in mice, and functional testing of key signaling pathways. There are important findings in this manuscript, however in this resubmitted version some figure panels are missing, and several figure citations are incorrect. In addition, some language is misleading or confusing. Below suggestions to rectify the above observations to finalize this submission:

R2-1) The authors conclude that rete ridge development is driven by non-focal proliferation (line 281 and 302). However, the MKi67 staining data indicates that both basal and suprabasal proliferation are significantly higher in the rete ridge domain compared to the inter-ridge domain during the second postnatal week in pigs, when rete ridges are actively developing (Supplemental Fig. 8c). These localized differences in proliferation could reasonably be interpreted as focal, rather than non-focal. Furthermore, it is unclear how rete ridge and inter-ridge domains are defined when quantifying proliferation at P1-2 and P3 (Supplemental Fig. 8b). At these time points, the boundary between the domains seems uncertain, and the rete ridge domain looks broader than at later stages. This ambiguity could lead to inaccurate classification of proliferating cells as either inside or outside the rete ridge domain. In fact, both tissue sections (Supplemental Fig. 8b) and whole mount images (Fig. 4a and 4e) show that the rete ridge domain narrows over time, suggesting dynamic remodeling. As a result, cells initially counted in the inter-ridge domain may end up in the rete ridge domain at a later time point.

The epidermal domains were distinguished as the “inter-ridge” as the curving epithelial “roof” atop the dermal pocket. This reviewer accurately points out that the “width” of each domain can be dynamic over rete ridge formation and maturation. This ridging morphogenesis process underlying rete ridge formation makes the rete ridge and inter-ridge boundary more difficult to define than the discrete placode of individual hair follicles, sweat glands, and fingerprint ridges^{24,32-34}. We have added a statement to this section of the methods to acknowledge that inter-ridge and rete ridge domains may not be comprised of fixed populations of cells, and that the boundaries for these domains may be dynamic during rete ridge initiation:

- “It is possible, due to the dynamic architectural remodeling that occurs during rete ridge formation, that cells within the rete ridge versus inter-ridge boundaries do not represent fixed populations. However, we did not observe signs of lateral migration of BrdU-labeled cells between domains, consistent with previous studies⁶⁷.” (lines 760-764)

Based on our observations from BrdU-labeling during porcine rete ridge formation, we did not observe a difference in BrdU-labeling in the basal or suprabasal layers of the inter-ridge or rete ridge domains. If there was consistent migration from one domain to the other, we would expect to have observed differences in the proportion of short-term labeling between the rete ridge and

inter-ridge domains, particularly in the basal or supra-basal layers. We observed no significant differences between either domain within any sublayer of the epidermis, as described in the figure legend (Supplemental Fig. g-h). Furthermore, previous studies utilizing viral fluorescent tagging in xenografted human skin similarly observed little to no lateral migration between rete ridge and inter-ridge domain epithelial cells ³⁵.

R2-2) In the rebuttal, the authors argue that “during rete ridge formation and maturation, the suprabasal and spinous compartments expand as pigs develop rete ridges, while the epidermis is thinner and simpler in species that lack rete ridges.” They further claim that “epidermal thickening in rete ridge regions is primarily driven by suprabasal expansion, not just basal layer enlargement.” The primary data supporting these conclusions appear to be presented in Supplemental Fig. 9a and 9b. However, these data are neither described nor interpreted in the main text, and Supplemental Fig. 9a is not properly cited.

We thank the reviewer for their attention to this supplemental figure. In an accidental omission, we forgot to describe this panel in the manuscript’s text. We have now made this addition when introducing the spatiotemporal transcriptomics section in the results:

“Porcine interfollicular epidermis displays greater complexity in stratification over the course of rete ridge formation, with an expansion of the KRT10+ cell states within the suprabasal compartment of the rete ridges (Supplemental Fig. 9a). In contrast, interfollicular epidermis of non-rete ridge-bearing skin more closely resembles fetal pig skin, with a flatter layering between the between basal and KRT10+ differentiating layers (Supplemental Fig. 9a).” (lines317-321)

R2-3) The authors define the dermal space between rete ridges as the “dermal pocket—a prominently vascularized region of the papillary dermis”—and emphasize its interaction with the epidermis and its potential role in rete ridge development throughout the manuscript. They further discuss a co-dependency between rete ridges and dermal pockets in the rebuttal. However, this interpretation raises several concerns.

First, if the dermal pocket is defined as the space between rete ridges, then its formation is a structural consequence of rete ridge development. In that case, questioning whether dermal pocket formation depends on rete ridge formation becomes tautological and biologically meaningless. For example, “generate an interconnecting rete ridge network organized around vascularized dermal pockets” (line 51-52) implies that dermal pockets precede the formation of the rete ridge network—an interpretation that contradicts the definition of dermal pockets as secondary to rete ridge architecture.

Second, if the authors intend to define the dermal pocket based on its vascular characteristics, they have not provided sufficient evidence to support this claim. Due to the orientation of blood vessels in the skin, thin tissue sections may not reliably reveal regional differences in vascularization, A 3D whole mount analysis is more suitable. Moreover, the authors have not demonstrated any differences in angiogenic activity between rete ridge and inter-ridge regions during development. It is also possible that the dermal cells are passively enriched into the dermal pocket while expanding equally between different domains.

We thank the reviewer for their interest in and critical engagement with the dermal pocket component of rete ridges. To answer the reviewer's first comment, the sequence of rete ridge vs dermal pocket initiation is of great interest in defining this appendage’s epidermal and dermal niches ³⁶. It is no more meaningless than discussing the dependence of hair follicle formation on

the establishment of the placode, the epithelial component, or the establishment of the underlying dermal condensate, the dermal component. The rete ridge and vascularized dermal pocket appear to develop alongside one another (see also response to R2-5), and we have rephrased discussions of these two compartments to improve clarity and emphasize the requirement for epidermal BMP signaling in rete ridge formation, which we have functionally validated, and temper claims related to the dermal niche. Notably, in the absence of epidermal BMP signaling, there is no dermal niche resembling the vascularized dermal pocket (Fig. 5d; Supplemental 16e):

- From the Abstract/Summary Paragraph: “We propose that evolution of rete ridges in mammalian skin involved replacement of the molecular program for formation of discrete microscopic appendages, including hair follicles and sweat glands, with a distinct program for the interconnected appendage network. Broad epidermal activation of BMP is required for the development of rete ridge networks organized around underlying dermal pockets.” (lines48-52)
- From the Introduction: “Identified cellular and signaling interactions underlying rete ridge-specific development in the skin of humans and pigs support a new model for their formation and regeneration, which requires epidermal BMP signaling.” (lines106-108)
- From the Results: “Additionally, the space beneath the inter-ridge epidermis is occupied by dermal pockets – a prominently vascularized region of the papillary dermis (Fig. 1b; Supplemental Fig. 1a).” (lines123-125)

The presence of dermal vascularization between rete ridges has been described for decades in human as well as cetacean skin ^{4-6,20,28}. In historical literature, this dermal compartment was often referred to as the dermal papillae ⁴⁻⁶, which we have reframed to “the dermal pocket” in this study to avoid confusion with the far more commonly invoked hair follicle dermal papillae encapsulated within the base of hair follicles. Additionally, we have purged the text for references to relative or empiric differences in vascularity within the dermal pocket between species. However, the presence of vasculature is a hallmark feature of the dermal pocket and has long been recognized as such in a variety of species and using a variety of 2-dimensional and 3-dimensional experimental techniques for many decades ^{4-6,28,37}.

As such, we describe the dermal compartment between rete ridges as vascularized, since we observed this feature consistently in different species and anatomical sites that contain rete ridges using a variety of histological, immunohistochemical, and spatial transcriptomics approaches (Fig. 1b, g; Fig. 2a; Supplemental Fig. 2a; Fig. 3a; Fig. 4b; Supplemental Fig. 7c-d; Supplemental Fig. 9a; Supplemental Fig. 10a; Supplemental Fig. 16a-e; Supplemental Fig. 17a-b):

- “Since we observed that postnatal murine fingerpads have structures closely resembling rete ridges of humans and pigs, complete with vascularized dermal pockets (Supplemental Fig. 6b-c; Supplemental Fig. 16d), we investigated murine fingerpads to define the developmental timing of their formation.” (lines392-395)
- “Besides supporting epidermal thickening, the broad conservation of dermal pockets in different species and body regions containing rete ridges implies that there may be other key epidermal-dermal signaling interactions that are missing from the epidermis during trunk skin development in mice and other rete ridge-less species, such as BMP or PDGFC.” (lines473-477)

- "In this work, we have highlighted the potential relationship between the epidermis and the underlying dermis in the establishment of rete ridges. The dermal pocket enables papillary fibroblasts, pericytes, blood vessel, and lymphatic vessel cells to assemble beneath the basal epithelium and establish a signaling source that potentially assists with epidermal thickening and rete ridge maintenance^{52,82-84}. Clinical support for the underlying dermal vasculature organization's potential contribution to rete ridge patterning can be observed in human infantile hemangiomas, benign skin tumors which display profound dermal hypervascularization and completely lack rete ridges^{103,104}." (lines502-509)

We also used immunostaining for various markers of blood vessels (antibody used varies by species due to non-model species often not reacting to conventional mouse or human targeted antibodies). We commonly utilize ITGA6, which is broadly expressed in blood vessels (as well as the basal epithelium), overlapping with PECAM1 and CDH5 (see our scRNA-seq marker analysis: Supplementary Fig. 4b-c, g-h). Human-ITGA6 (antibody from BD Biosciences) is also reactive in many of the species examined in this manuscript, allowing for more consistent immunostaining comparisons between species (Fig. 3a, d; Supplemental Fig. 6a-c; Fig. 4a; Supplemental Fig. 7d-g; Supplemental Fig. 9a; Supplemental Fig. 16a, e; Fig. 5a; Supplemental Fig. 17a-b). The dermal pockets of human and cetacean trunk skin as well as murine fingerpads clearly contain vasculature, and immunostaining using aSMA demonstrates the broad coverage of these vessels with pericytes (Supplemental Fig. 16b-d). Additionally, we also demonstrate blood vessels in the dermal pocket of murine fingerpad and oral rete ridges using a mouse-PECAM1 antibody (Supplemental Fig. 16d) and have been sure to limit our description of these results to qualitative observations:

- "The vascularized dermal pocket's cellular composition is similar in other species and body regions that contain rete ridges, such as human and dolphin trunk skin, and in fingerpads of mice (Supplemental Fig. 16a-d). In contrast, rete ridge-less trunk skin of marmosets and mice does not morphologically resemble the dermal pocket (Supplemental Fig. 16e)." (lines376-380)

We are not the first to observe or describe the vascularization of the dermal pocket nor the first to describe its conservation across species. Historical literature has long recognized and referred to this highly vascularized dermal compartment as the "dermal papillae" which is confusing in modern nomenclature due to the far more commonly invoked hair follicle dermal papillae^{4-6,28}. Recent advances in non-invasive imaging technology, such as optical coherence tomography, support observations of these vascular histological cross-sections as indicative of the high degree of vascularization within the papillary dermis of human skin in a variety of anatomical sites^{37,38}. Optical coherence tomography does offer greater dimensionality than classical cross-sectional approaches, though these results are consistent in observing vascular changes during aging or between different body regions which were first recognized using histological approaches^{28,37-39}. All that being said, we have further tempered comparisons of the vascularity between species (see response to R2-5 below).

Through a combination of single-cell and spatial transcriptomics, immunostaining, and transgenic perturbation of the fingerpad rete ridges in mice, we identified and validated epidermal BMP signaling as a key driver required for rete ridge formation. Since the epidermis broadly sends and receives BMP signals (Fig. 4c-e; Supplemental Fig. 10c-d; Supplemental Fig. 11a-b; Fig. 5b-e), and rete ridge formation is significantly inhibited in the absence of functional

epidermal BMP signaling, we conclude that the epidermis actively contributes to the initiation and patterning of rete ridge formation. Minor protein gradients in key signals, such as BMPs/SMADs or PDGFC, may provide spatial nuances that support undulating patterning of rete ridge and inter-ridge domains, which are otherwise similar at the transcriptional and functional (e.g. differentiation) levels (Fig. 4e; Supplemental Fig. 9a-e; Supplemental Fig. 8b-h). How epidermal BMP signaling interacts with underlying dermal cell lineages, and whether other dermal-derived signals are independently required for rete ridge formation or patterning are out of the scope of the present study, and we have tempered our conclusions in the Discussion section to acknowledge these areas of uncertainty:

- “In this work, we have highlighted the potential relationship between the epidermis and the underlying dermis in the establishment of rete ridges. The dermal pocket enables papillary fibroblasts, pericytes, blood vessel, and lymphatic vessel cells to assemble beneath the basal epithelium and establish a signaling source that potentially assists with epidermal thickening and rete ridge maintenance^{52,82–84}. Clinical support for the underlying dermal vasculature organization’s potential contribution to rete ridge patterning can be observed in human infantile hemangiomas, benign skin tumors which display profound dermal hypervascularization and completely lack rete ridges^{103,104}. Other epidermal- or dermal-resident cell types may also contribute to rete ridge formation and maturation, including nerve cell endings and immune cells^{15,54,56,83,105}, which were beyond the scope of the current study. Future studies will be needed to understand how dermal and vascular contributions during rete ridge formation and patterning differ from those involved in hair follicle, sweat gland, or fingerprint ridge formation^{1,2,4,26,38,42,45,102}.” (lines502-514)

R2-4) It is exciting that the authors have established a new model to study rete ridge formation in the mouse fingerpad. This represents a significant advance in the field, and the authors may consider highlighting this breakthrough more prominently when introducing the finding for the first time (line 260).

We thank this reviewer for their interest and excitement regarding fingerpad rete ridges. We have taken their advice and added additional emphasis on the importance of characterizing the developmental timing of rete ridge formation in murine fingerpads to the results and discussion sections. This emphasis has also helped to clarify the distinction in developmental timing and signals between rete ridge formation and other epithelial appendages, and we the reviewer for this suggestion:

- “In murine volar skin, sweat glands start forming from LEF1⁺ basal buds during fetal development and continue forming briefly after birth (Supplemental Fig. 6a)^{4,25,26}. Transverse ridges form along the proximal volar surface of mouse digits during fetal development, resembling human fingerprint ridges^{2,65}, but do not form in the fingerpad skin at the digit tip, **where we instead observed rete ridge formation postnatally** (Supplemental Fig. 6a-c). As in porcine trunk skin, murine fingerpad rete ridges form after the completion of sweat gland morphogenesis, when volar epidermis lacks LEF1 expression (Supplemental Fig. 6b). We confirmed the independence of fingerpad rete ridge formation from *LEF1/WNT* signaling using *Lef1-eKO* mice, which exhibit impaired placode development (Fig. 2f; Supplemental Fig. 3b-d; Supplemental Figure 6c). Consistent with the hypothesis that rete ridge formation does not require *LEF1/WNT*

signaling to form, rete ridges in the fingerpads of juvenile mice were not affected by epidermal ablation of *Lef1* (Fig. 3d; Supplemental Fig. 6c).” (lines264-276)

- “Since we previously observed postnatal murine fingerpads have structures closely resembling rete ridges of humans and pigs, complete with vascularized dermal pockets (Supplemental Fig. 6b-c; Supplemental Fig. 16d), we investigated murine fingerpads to define the developmental timing of their formation.” (lines395-398)

R2-5) The authors present convincing data showing the critical role of epidermal BMP signaling in enabling rete ridge formation. However, they appear to overstate the role of signaling between epidermis and dermis in driving rete ridge formation, as the role of dermis in this process remains insufficiently defined. For example, the statement “Postnatal activation of epidermal-dermal signaling rete ridge and dermal pocket formation” (line 306) implies a causal relationship, yet no functional evidence is provided to support a driving role for epidermal-dermal signaling. Similarly, the claim that “these transcriptomics observations highlight the existence of divergent cellular and molecular interactions that drive species-specific differences in skin morphology” (lines 372–374) overextends the interpretation of correlative transcriptomic data without accompanying mechanistic validation. On the other hand, the statement “a new model for their formation and regeneration, which is led by epidermal cell lineages” implies epidermis is the driver, excluding a potential instructive role for the dermis.

We have further tempered the abstract and results section to emphasize epidermal BMP signaling, which we have validated *in vivo* and avoid implication of causal relationships between other predicted epidermal and dermal signaling activities during rete ridge formation:

- We have changed the header of this section to “Postnatal activation of epidermal-dermal signaling accompanies rete ridge formation” (line316) to remove implied causality between epidermal-dermal signaling and rete ridge formation.
- “Overall, these transcriptomics observations highlight the existence of distinct postnatal epidermal and dermal signaling activities during rete ridge formation. Concurrently, postnatal skin inactivates the fetal signaling programs associated with the generation of placodes for other discrete appendages, such as hair follicles and sweat glands (Fig. 3b-c; Fig. 4d-e; Supplemental Fig. 10c-e; Supplemental Fig. 11a-d).” (lines388-391)
- We have adjusted the statement in the introduction “...which are led by epidermal cell lineages” to be mechanistically focused and functionally validated, now reading “which requires epidermal BMP signaling” (line108).

We have moved further discussion of the potential role of vasculature and other dermal cell lineages in supporting rete ridge formation to the discussion section:

- “In this work, we have highlighted the potential relationship between the epidermis and the underlying dermis in the establishment of rete ridges. The dermal pocket enables papillary fibroblasts, pericytes, blood vessel, and lymphatic vessel cells to assemble beneath the basal epithelium and establish a signaling source that potentially assists with epidermal thickening and rete ridge maintenance^{52,82–84}. Clinical support for the underlying dermal vasculature organization’s potential contribution to rete ridge patterning can be observed in human infantile hemangiomas, benign tumors which display profound dermal hypervascularization and lack rete ridges^{103,104}. Other epidermal- or dermal-resident cell types may also contribute to rete ridge formation and maturation, including nerve cell endings and immune cells^{15,54,56,83,105}, which were beyond the scope of the current study. Future studies will be needed to understand how

dermal and vascular contributions during rete ridge formation and patterning differ from those involved in hair follicle, sweat gland, or fingerprint ridge formation^{1,2,4,26,38,42,45,102}.” (lines502-514)

R2-6) The authors use the term “basal budding” to specifically refer to the WNT-EDA/R mediated appendage formation process. However, this term can confuse the reader, as it can also be interpreted more generally to refer to any budding-like morphogenesis occurring at the basal layer—a process that is not necessarily excluded in rete ridge formation. Therefore, the statement “rete ridges must instead form through a distinct mechanism from basal budding” (line 277) is inappropriate. The authors may explicitly state that they are referring to the WNT–EDA/EDAR-mediated process.

We thank the reviewer for their critical engagement with the molecular and morphological process by which rete ridges develop compared to other cutaneous appendages. We had utilized the terminology basal budding to describe the evolutionarily-conserved morphological and molecular characteristics conserved in the initiation of many invaginating epidermal appendages, including hair follicles, sweat glands, and fingerprint ridges. A critical morphological feature of the epidermal placodes of these appendages is the initial rounded morphology and focal proliferation, which is quickly succeeded by downgrowth and cylindrical elongation, forming a single, discrete appendage per initiation site^{24,32–34,40,41}.

We have now added figure citation Fig. 4a to our description in formerly line277 (now line285), and alongside the cross-sectional results (line229) since this wholemount view demonstrates clearly the topographic, interconnecting ridge morphogenesis in stark contrast to the self-limited budding processes from which an epithelial placode elongates into a discrete epithelial appendage, as in the case of hair follicles and sweat glands. The developing rete ridges mature by deepening the surface undulations into mountainous ridges and valley-like inter-ridge domains, which are interconnected and 3-dimensional along the basal interfollicular epidermis, in stark contrast to the individualized hair follicle or sweat gland appendages (Fig. 4a; Fig. 5e)^{28,42}. Hence, rete ridge formation propagates via ridging morphogenesis rather than discrete, self-limited appendage initiation and downgrowth as in hair follicles or sweat glands.

We have clarified this *LEF1/WNT* and *EDA/R* independence and ridging morphogenesis process in the Results and Discussion. Furthermore, we have replaced most references to “basal buds” with references to the epidermal placode, which are summarized below:

- “To test the genetic and developmental determinants of hair density and epidermal thickness, we targeted an evolutionarily conserved step in hair follicle formation, LEF1/WNT and EDA/R-mediated epidermal placode formation. The WNT signaling transcription factor LEF1 is highly expressed in the fetal epidermis of humans, non-human primates, and rodents when hair follicles, sweat glands, and fingerprint ridges develop via epithelial placodes (Fig. 2e; Supplemental Fig. 3a, c)^{2,21,26,27,41}.” (lines192-197)
- “Rete ridges are a distinct type of cutaneous appendage”
- “Since a reduction in hair density does not spontaneously enable rete ridge formation or epidermal thickening, we hypothesized that rete ridges form through a different

molecular mechanism than the *LEF1/WNT* and *EDA/R*-mediated processes of hair follicles, sweat glands, and fingerprint ridges.” (lines222-226)

- Figure 3 figure title: " **Fig 3: Rete ridges form differently from other epithelial appendages.**"
- “We next examined the expression of shared markers for epidermal placode formation in hair follicles, sweat glands, or volar fingerprint ridges. *LEF1* and *EDAR*, shared markers of placodes involved in the formation of all three appendages^{2–5,8,10,21,26,27,39,41}, are highly expressed in E90 basal bud cells forming sweat glands but are only sparsely expressed in postnatal basal cells during rete ridge formation and maturation (Fig. 3c; Supplemental Fig. 4e-f; Supplemental Fig. 5b)^{49,62 2,27,28,49}. Postnatal basal cells do not express markers of the other appendages during rete ridge formation and maturation, except for BMP ligands (Fig. 3c; Supplemental Fig. 4e-f; Supplemental Fig. 5a-b). Postnatal basal cells involved in rete ridge formation highly expressed BMP ligands, such as *BMP7* and *BMP2*, which are not expressed outside of the budding fetal sweat glands in fetal skin (Fig. 3c).” (lines246-255).
- “We next set out to validate *in vivo* that rete ridge formation does not require placode-associated signals, such as *LEF1/WNT* and *EDA/R*, utilizing both mouse and pig models. In murine volar skin, sweat glands start forming from *LEF1*⁺ basal buds during fetal development and continue forming briefly after birth (Supplemental Fig. 6a)^{4,25,26}.” (lines263-266)
- “We confirmed the independence of fingerpad rete ridge formation from *LEF1/WNT* signaling using *Lef1-eKO* mice, which exhibit impaired placode development (Fig. 2f; Supplemental Fig. 3b-d; Supplemental Figure 6c). Consistent with the hypothesis that rete ridge formation does not require *LEF1/WNT* signaling to form, rete ridges in the fingerpads of juvenile mice were not affected by epidermal ablation of *Lef1* (Fig. 3d; Supplemental Fig. 6c).” (lines271-276)
- “Collectively, these results implicate BMP activation alongside the inactivation of *LEF1/WNT* and *EDA/R*-mediated processes as a crucial, evolutionarily conserved developmental milestone that controls appendage type specification and enables non-furry skin to develop rete ridges (Fig. 5e).” (lines441-444)
- “However, as we have demonstrated in multiple species, rete ridges form postnatally as the final type of epidermal appendage after other placode-based appendages stop forming. We propose that rete ridges function as a large-scale, interconnected appendage^{32,51}, which bestows “structural and functional complexities to the otherwise flat epithelia”⁹¹ in addition to possessing their own epidermal and dermal niches in postnatal human and porcine trunk skin as well as murine volar fingerpads.” (lines450-455)
- “In sum, we demonstrate that rete ridges form perinatally in pigs and postnatally in mouse fingerpads, after hair follicle and sweat gland formation is complete. We also demonstrate that rete ridges require distinct cellular and molecular mechanisms independent of *LEF1/WNT* and *EDA/R* signaling and critically require epidermal BMP signaling.” (lines541-544)

R2-7) The authors show that rete ridge and inter-ridge epidermal cells are molecularly and behaviorally equivalent, suggesting that rete ridges represent structural domains within the epidermis rather than functionally distinct units. Therefore, referring to rete ridges as skin appendages may be inappropriate, as most skin appendages are both structurally and functionally distinct from the interfollicular epidermis. Moreover, rete ridges have not traditionally been classified as skin appendages in the field. That said, the classification of fingerprint ridges as skin appendages introduces some ambiguity, and under that broader interpretation, rete ridges could potentially be considered a type of skin appendage.

We have revised our argument in the discussion section to better clarify why rete ridges should be considered a cutaneous appendage, based on long-standing definitions in the field ^{13,36,43}. Namely, that the formation of rete ridges modifies the epidermal architecture and has recognizable epidermal and dermal niches:

- "We propose that rete ridges function as a large-scale, interconnected appendage ^{32,51}, which bestows "structural and functional complexities to the otherwise flat epithelia"⁹¹ in addition to possessing their own epidermal and dermal niches in postnatal human and porcine trunk skin as well as murine volar fingerpads." (line450-453)

Whole mount immunostaining shown by us and others, as well as historical scanning electron microscopy of the basal epidermis clearly demonstrates the role of the rete ridge network in creating a unique interfollicular epidermal architecture that is not observed in mammals that lack rete ridges ^{28,44}. There are clearly defined structural and cellular domains within the hair follicle which are generally distinct from interfollicular epidermis ⁴⁵⁻⁴⁹. However, interfollicular epidermis has functional subdomains independent of the presence of rete ridges, such as the nerve-associated touch dome, which is distinct from both the hair follicle and the general interfollicular epidermis in mouse trunk skin ⁵⁰.

We argue that the rete ridges serve as a "large-scale, interconnected appendage" with both vertical (e.g. downward extension from the basal epidermis) as well as lateral (e.g. rete ridge versus inter-ridge domains) components. This is a critical distinction from the morphology of the hair follicle or sweat gland, which are tubular in shape and project deep into the dermis, culminating in a unique functional base (e.g. the matrix/bulb in the hair follicle which is the "factory" generating the external hair shaft, or the secretory coils in sweat glands which extrude the functional product of the sweat gland). Recent advances in the molecular and developmental characterization of human fingerprint ridges and the murine analog, transverse ridges, similarly represent a demonstrably unique epidermal appendage that has a large structural function in creating large-scale, interconnected epidermal and dermal ridging in skin, especially in human fingerpads ^{24,25}.

Critically, fingerprint ridges in fetal human and mouse volar skin were demonstrated to form through a basal bud which is, in the initial stages, morphologically and is molecularly similar to initial basal buds of hair follicles and sweat glands, both expressing and being patterned by *LEF1/WNT* and *EDA/R* ^{17,24,41,51-53}. However, a distinct WNT-negative dermal environment and lack of dermal condensate formation during volar fingerprint ridge formation contributes to their identity as a spatially distinct appendage from the hair follicles developing in dorsal skin ²⁴. Other work has identified distinct dermal niches involved in volar sweat gland development, which is also divergent from the hair follicle dermal niche ^{32,54}.

We agree with the reviewer that “skin appendages are both structurally and functionally distinct from the interfollicular epidermis.” As we have demonstrated in this manuscript, there are spatiotemporal distinctions between murine skin that naturally forms rete ridges (e.g. the fingerpad) and skin that does not (e.g. the back skin) at morphological, transcriptional, and protein-levels (Fig. 3a; Fig. 4a; Supplemental Fig. 8a-b; Supplemental Fig. 9a, e; Supplemental Fig. 16a-e; Fig. 5e; Supplemental Fig. 17b-c). Therefore, we observe these distinctions as evidence of the unique cellular and functional processes supported by rete ridges as an acquired epithelial appendage in the expanded interfollicular epidermis of the, generally, lower hair density mammals which have thickened epidermis (Fig. 2a-d). As rete ridges are interconnected and topographic in nature rather than individualized appendages, they uniquely modify the interfollicular epidermis compared to non-rete ridge-bearing skin. Future studies will be needed to validate the anatomical site distribution of these hallmarks in other non-model species of interest to rete ridge biology, including cetaceans and grizzly bears.

Furthermore, different anatomical sites in mice already possess unique assortments of accepted epithelial appendages, with hair follicles virtually omnipresent in the trunk and dorsal skin while almost entirely absent from the volar skin of the hands, with prevalent sweat glands and fingerprint ridges instead, especially in humans (Fig. 5e)^{18–20,24,32,54}. Additionally, distinct molecular and signaling environments, such as epidermal BMP and PDGF signaling, appear to be specific to the rete ridge-bearing skin in humans, pigs, and in the murine fingerpad but not in the murine back skin (Supplemental Fig. 17b-c). As a result, these compartmentalized, spatiotemporally regulated signaling environments support the interpretation that rete ridges are indeed a distinct type of epithelial appendage, which forms in an orderly fashion after the completion of hair follicle and sweat gland morphogenesis in multiple mammalian species. We expect future studies to elucidate more functional roles for the rete ridge in human and porcine skin as well as the murine fingerpad, which will further expand the interpretation and acceptance of rete ridges as a distinct, functional cutaneous appendage, as was recently done for volar fingerprint ridges^{24,25}.

R2-8) The use of the term “dermal maturation” (line 135) to describe the collagen maturation observed via Herovici staining is inappropriate, as “dermal maturation” typically encompasses a much broader range of structural and cellular changes within the dermis beyond collagen remodeling alone.

We thank the reviewer for their attention and interest in temporal Herovici staining, and we have made the recommended rephrasing to describe observation of “dermal connective tissue maturation” rather than the broader claim of “dermal maturation”.

R2-9) It is expected that rete ridges contribute to increased epidermal thickness in ridge-bearing species. However, it remains unclear whether they also influence the thickness of the inter-ridge regions. The authors note that “the inter-ridge thickness (aside from the cetaceans) is not dramatically increased” (lines 172–173), but no statistical analysis was performed. A quantitative comparison of inter-ridge thickness between ridge-bearing and ridge-less species would be valuable to determine whether inter-ridge regions also contribute to the overall epidermal thickening observed in ridge-bearing animals.

We thank the reviewer for their interest in the relationship between rete ridge formation and epidermal thickening. Statistical analyses were indeed performed in the first revision and were clearly labeled in both the figure panel and legend of Supplemental Fig. 2d. While the multiple comparisons from ANOVA are more difficult to interpret in the “inter-ridge thickness only” comparison versus the rete ridge thickness comparison, the cetaceans were the only species with universally thicker inter-ridge epidermis compared to all other mammalian species surveyed (Supplemental Fig. 2b). While there are relative differences in inter-ridge thickness (e.g. 2-fold increase in the pigs and humans compared to the mice), these relative differences were not always significantly different based on our statistical analyses which were equally applied across all sampled species. While there are some differences, such as human and domestic pig inter-ridge thickness being significantly thicker than that of mice, marmosets, and naked mole rat, Mangalitsa pig and grizzly bears were not. Thus, the presence of rete ridges does not appear to always result in inter-ridge thickening, since there were more nuanced gradients in increases in inter-ridge thickness across even rete ridge-bearing species. Critically, as we noted in Fig. 2c, the ratio of rete ridge to inter-ridge thickness is consistent across all rete ridge-bearing species we surveyed, so the granular finding that human rete ridge thickness is significantly greater than the rete ridge thickness of the grizzly bear is logical since the human inter-ridge thickness is also significantly greater than the grizzly bear inter-ridge thickness, maintaining a consistent ratio (Fig. 2c; Supplemental Fig. 2d). Cetaceans are great examples of this principle, since the ratio held true even in the bottlenose dolphin, suggesting that as the rete ridge gets dramatically thicker, so too does the inter-ridge. These observations mesh with our BrdU-labeling analysis, wherein we observed similar BrdU-label depletion from the basal layer and progression upward during differentiation, suggesting that altered differentiation between rete ridge and inter-ridge domains may not be the driver for the distinction in thickness between domains. Due to character limits on the length of the results section, we simplified the discussion of these 10-species comparisons of rete ridge and inter-ridge thickness to the overall patterns while the individual statistics are clearly presented for individual species comparisons. In summary, the overall epithelial thickening in rete ridge-bearing epidermis is overwhelmingly driven by the thickening of the rete ridge itself, with some proportionate increase in inter-ridge thickness (Fig. 2c; Supplemental Fig. 2d). This pattern in epidermal thickening is also clearly displayed by our temporal quantifications of both rete ridge and inter-ridge thickness throughout skin development in pigs and humans (Fig. 1c-d, h-i).

Additionally, we have now updated both the comparative epidermal thickness boxplots to have a y-axis with increased number of minor breaks to improve the readability of both of these graphs, since the y-axis is log₂-adjusted (Fig. 2c; Supplemental Fig. 2d; reproduced below in Rebuttal Fig. 3a for ease of access):

R2-10) To avoid confusion, the authors need to mention that the few hair follicles in the *Lef1-eKO* mice (line 200 and Supplemental Fig. 3c) are likely escapers, since some epithelial cells still express LEF1.

It is possible that some of the hairs represent escapers, and a global knockout of *LEF1* in mice has previously demonstrated that some hair follicles still form even in the complete absence of *LEF1*⁵¹. We have added acknowledgement of this potential phenomenon to the text:

- “Hair follicle density was dramatically reduced in *Lef1-eKO* mice and the few resulting follicles, potentially representing “escapers”⁸, failed to maintain an external hair fiber during the first hair cycle, establishing a cyclical pattern of hair growth and loss during subsequent hair cycles (Supplemental Fig. 3d-e).” (lines203-207)

R2-11) The authors claim that the rete ridge and inter-ridge epidermal cells are molecularly similar, and PDGFC and SMAD1 are broadly expressed across these domains. However, the whole mount staining images do not clearly show the broad expression of PDGFC (Supplemental Fig. 9e) or SMAD1 (Fig. 4e). Instead, these images appear to show differential expression of PDGFC and SMAD1 between the rete ridge and inter-ridge regions.

While these gradients are not clearly observed in our single-cell or spatial transcriptomics analyses, we have adjusted our description of the wholemount immunostaining to acknowledge potential gradients in the activities of PDGF and BMP signaling at the protein level, which may contribute to nuanced spatial patterning of rete ridge formation:

- “The rete ridge and inter-ridge regions both express SMAD1 *in vivo*, though the rete ridge regions exhibit higher BMP signaling than in the inter ridge regions (Fig. 4e).” (lines352-354)
- “Thus, the neonatal basal epidermis appears to be broadly supportive of rete ridge formation and dermal recruitment, while rete ridge initiation patterning may be queued by nuanced gradients at the protein level or local proximity to differential distribution of underlying dermal fibroblasts and vasculature (Fig. 4b; Supplemental Fig. 10b; Supplemental Fig. 12a-b; Supplemental Fig. 13a-c; Supplemental Fig. 14a-d).” (lines386-389).

R2-12) For Fig. 4d, Supplemental Fig. 10b, 11, 12, 13, 14, 15b, it is hard to judge the signal between different cell types, as cell types are not annotated in those graphs the presence of the legends.

We thank the reviewer for their attention to these details and interest in the details of our spatial analyses. We have added a dashed outline depicting the epidermal-dermal junction to improve readability and interpretability of these plots (Fig. 4d; Supplemental Fig. 10b; Supplemental Fig. 11a-d; Supplemental Fig. 12a-b; Supplemental Fig. 13a-c; Supplemental Fig. 14a-d; Supplemental Fig. 15d). Furthermore, we have provided more details in the Methods and figure legends regarding the cell type and Leiden cluster subsets utilized for Spatial CellChat analyses presented in the figures in the Methods. In brief, Spatial CellChat analyses was performed on a subset of bin20 spots based on Leiden clustering, for the similar epidermal and dermal cell lineages (e.g. basal keratinocytes, papillary fibroblasts, and vasculature/pericytes) which were subset from scRNA-seq datasets for CellChat analyses.

R2-13) The authors state that “rete ridge-less trunk skin of marmosets and mice has vasculature and pericytes which are generally distal from the interfollicular epidermis throughout development and maturation” (lines 363–365). However, this claim is not supported by quantitative analysis or comparison using vasculature-specific markers. Given that only segments of blood vessels are visible in thin skin sections, a 3D whole-mount analysis would be more appropriate to accurately assess the spatial relationship between the vasculature and the interfollicular epidermis.

Our immunostains used to describe fibroblasts (PDGFRA), blood vessels (ITGA6 or PECAM1), and pericytes (αSMA) in mouse and marmoset trunk skin are of thick, 60um cryo sections⁵⁵. However, we have tempered the language used to describe the spatial organization of mouse and marmoset dermis as described above in response to R2-3:

- "The vascularized dermal pocket's cellular composition is similar in other species and body regions that contain rete ridges, such as human and dolphin trunk skin, and in fingerpads of mice (Supplemental Fig. 16a-d). In contrast, rete ridge-less trunk skin of marmosets and mice does not morphologically resemble the dermal pocket (Supplemental Fig. 16e)." (lines376-380)

R2-14) Figure citation mistakes:

- Line 120: Supplemental Fig. 1a does not show dermal pockets. Instead, it shows fetal human skin when rete ridges are not formed.

We have corrected the citation to Fig. 1b.

- Line 243: Supplemental Fig. 5a does not show LEF1 and EDAR expression. Supplemental Fig. 5b should be cited instead. Do white signals indicate high level of expression or undetected expression? They can confuse readers.

We have corrected the citation to Supplemental Fig. 5b). We have also added in the figure legend to indicate that white spots represent spots which did not pass QC and were filtered out, which is discussed in the Methods and fully detailed on our Github. We thank the reviewer for drawing attention to this, so that we can provide greater clarity in the figure legend.

- Supplemental Fig. 5e is not cited anywhere.

Supplemental Fig. 5e was moved to a later figure (Supplemental Fig. 17e) during the writing/editing process. It has now been properly deleted from Supplemental Fig. 5e.

- Supplemental Fig. 7g is not cited anywhere.

Supplemental Fig. 7g was intended to be cited in conjunction with the rest of the neonatal pig wound healing panels but was forgotten by accident. We have now corrected that in the text.

- Line 287: Fig. 4a does not show basal bud formation. Instead, it shows interesting narrowing of the rete ridge area and expanding of the inter-ridge area over time.

This citation was intended to refer to the MKI67-associated supplemental figure demonstrating proliferative patterning in the budding and downgrowth of human hair follicles and sweat glands (Supplemental Fig. 8a). The reviewer accurately notes that Fig. 4a shows the formation of rete ridges, which do not morphologically resemble hair follicle and sweat gland basal budding. We have corrected this citation in the text.

- Figure legends for Supplemental Fig. 9d, 9e, 9f are missing.

We thank this reviewer for catching this mistake. We have updated this figure legend to describe panels d-f.

- Line 333: The correct citation should be Supplemental Fig. 10b–c, not Supplemental Fig. 8b–c.

We thank this reviewer for catching this mistake, and we have updated the figure citation.

- Figure legend for Fig. 4e is missing.

We thank the reviewer for catching this mistake. Fig. 4e was described in the legend but the preceding (e) identifier was missing. We have now corrected this.

- Fig. 4f-g are cited in the text but are missing in the Figure.

This is an oversight from a previous version of the figure. We have corrected these citations.

- In Supplemental Fig. 11b, BMP7 is shown in the graph, whereas BMP6 is labeled in the CirclePlot.

We thank the reviewer for catching this confusing labeling and are happy to be able to update multiple figure legends for improved clarity: Supplemental Fig. 11b emphasizes BMP ligands and zoomed out plots which are not included in the main figure 4d, which is focused on BMP signaling during rete ridge formation from P3 to P10 and specifically BMP7 signaling. The dashed boxes in the Spatial CellChat plots in Supplemental Fig. 11a-b indicate the zoomed-in regions presented in Fig. 4d. CirclePlots for P3/P10 BMP7 ligand-receptor interactions are already presented in Fig. 4d, so we placed the E90/6mo BMP pathway and BMP7 ligand-receptor CirclePlots in Supplemental Fig. 11a.

We have expanded the figure legend descriptions for these panels to more clearly describe which pathway or ligand-receptor plots are in which panel and to refer the reader to which plots are already presented in the main figure and thus apparently “missing” from the supplemental figure.

- Line 396: Supplemental Fig. 17c should be cited to show “murine trunk skin does not express Pdgfc”.

We thank the reviewer catching this. We now appropriately reference Supplemental Fig. 17a-c.

- Line 398: Supplemental Fig. 5c does not show EN1 expression.

We thank the reviewer for catching this erroneous citation. We have removed it.

- Line 399: The correct citation should be Supplemental Fig. 17d, not 17c.

We thank the reviewer for catching this mistake. We have corrected the citation.

- The title of Supplemental Fig. 17e is not appropriate, as JAG1 is active not only in neonatal skin but also in fetal skin.

The header specifically refers to the expression in the basal epithelium. JAG1 is highly expressed in the differentiating epidermal cells (rightmost clusters) and expressed at very low levels in the basal cells (leftmost clusters) (Supplemental Fig. 4i-j; Supplemental Fig. 17e). We have now changed the header to read “... NOTCH signaling ligands are more active in neonatal basal cells”.

- What do red dashed circles in Supplemental Fig. 17g indicate?

The red dashed circles draws attention to the subcluster containing LHX2+ hair follicle epithelial cells ⁵⁶, since this is the epidermal subcluster expressing Noggin in our re-analysis of the Liu et al. 2022 adult mouse trunk skin scRNA-seq dataset ⁵⁷. We have added more detail to this figure legend to clarify this.

In summary, we are grateful to the reviewer for their detail in catching and listing these figure panel citation mistakes. We apologize for these oversights and have made all the corrections.

Referee #3 (Remarks to the Author):

The authors have performed a rigorous revision, including a wealth of new experiments and analyses, thoroughly addressing my major and minor questions.

It is a true biological highlight that although rete ridges (RR) appear to form via initial bud emergence, like other epidermal appendages, they develop through distinct molecular and cellular mechanisms, which has been convincingly demonstrated in the revised manuscript. Further, it was particularly impressive that the authors undertook technically demanding experiments, including BrdU cell tracking in pigs, Stereo-seq, and spatial CellChat analysis. Beyond the novel biology, the manuscript also comes with a rich dataset, including a large collection of porcine scRNA-seq and spatial transcriptomics data, which is openly accessible via an online portal and tool.

In summary, the authors have conducted a rigorous cross-species analysis of rete ridge formation, a process debated for decades but previously unresolved. This is an exciting manuscript and represents a significant milestone in the field of tissue developmental biology.

Minor comments:

R3-1) The authors write in the abstract: “Therefore, we propose that the evolution of rete ridges in mammalian skin involved replacement of the molecular program for periodic patterning and subsequent downgrowth of discrete appendages, such as hair follicles and sweat glands, coupled with broad activation of BMP to generate an interconnecting rete ridge network organized around vascularized dermal pockets.” This sentence may be difficult for a general readership to follow and would benefit from rephrasing for clarity.

We thank the reviewer for drawing our attention to this long sentence in the abstract. We have now rephrased this for improved clarity and flow:

- “We propose that evolution of rete ridges in mammalian skin involved replacement of the molecular program for formation of discrete microscopic appendages, including hair follicles and sweat glands, with a distinct program for the interconnected appendage

network. Broad epidermal activation of BMP is required for the development of rete ridge networks organized around underlying dermal pockets.” (lines48-52)

R3-2) PDGFC, which is frequently highlighted in the manuscript and across species, would be valuable to show in pig skin as well. Could you display its expression in any of the pig scRNA-seq datasets, if not in the spatial data?

We thank the reviewer for their interest in PDGFC expression. Depiction of PDGFC expression in porcine and human skin was displayed in multiple supplemental figures, which are described in the following: PDGFC expression in integrated porcine scRNA-seq data was displayed in Supplemental Fig. 9e; PDGFC expression in our re-analyzed human scRNA-seq data was displayed in Supplemental Fig. 9f); expression of PDGFC in heatmap representations from E90, P3, P10, and 6mo pig skin scRNA-seq plus our re-analyses of GW14 volar skin, neonatal foreskin, and adult trunk skin from human scRNA-seq are in Supplemental Fig. 10d-e); spatial signaling analysis of PDGFC from Stereo-seq using Spatial CellChat and from scRNA-seq using CellChat is presented in Supplemental Fig. 14d.

scRNA-seq FeaturePlots can also be queried and visualized using our webtool for individual and integrated porcine scRNA-seq datasets: <https://skinregeneration.org/papers/Thompson-et-al-2025/>

R3-3) Please provide more details on the spatial analysis, such as the Stereo-seq dot sizes, the number of reads and genes per dot, whether dot-binning was used, whether cell segmentation was applied to identify underlying cells, etc.

We used bin size 20, which represents an overall binned dot size of 10um (0.5um “bin size 1” * 20 = 10um for “bin size 20”).

We have added more details about binning and processing to the Methods section (the complete details and source code is also publicly available on our Github), and we have added QC plots of reads and genes per dot for each dataset to the bottom row of Supplemental Fig.4d and reproduce it below for ease of access (Rebuttal Fig. 4):

We did not utilize cell segmentation. All analyses, including Spatial CellChat were employed on the bin20 dots. Especially for spatial signaling analyses, it can be difficult for cell segmentation to accurately classify dispersing paracrine factors back to their true cell of origin versus destination. We utilized Stereo-seq primarily for spatial context and analyzed this data in concert with our scRNA-seq data for a combined high resolution, single-cell type analyses with spatial context in predicting cell-cell communication.

Works Cited

1. Li, K. N. *et al.* Skin vasculature and hair follicle cross-talking associated with stem cell activation and tissue homeostasis. *eLife* **8**, e45977 (2019).
2. Yano, K., Brown, L. F. & Detmar, M. Control of hair growth and follicle size by VEGF-mediated angiogenesis. *J. Clin. Invest.* **107**, 409–417 (2001).
3. Ou, K.-L. *et al.* Adaptive patterning of vascular network during avian skin development: Mesenchymal plasticity and dermal vasculogenesis. *Cells Dev.* **179**, 203922 (2024).
4. Giacometti, L. The skin of the whale (*Balaenoptera physalus*). *Anat. Rec.* **159**, 69–75 (1967).
5. Haldiman, J. T. *et al.* Epidermal and papillary dermal characteristics of the bowhead whale (*balaena mysticetus*). *Anat. Rec.* **211**, 391–402 (1985).
6. Reeb, D., Best, P. B. & Kidson, S. H. Structure of the integument of southern right whales, *Eubalaena australis*. *Anat. Rec.* **290**, 596–613 (2007).
7. Ji, Y. *et al.* Signaling pathways in the development of infantile hemangioma. *J. Hematol. Oncol. J Hematol Oncol* **7**, 13 (2014).
8. Dadras, S. S., North, P. E., Bertoncini, J., Mihm, M. C. & Detmar, M. Infantile hemangiomas are arrested in an early developmental vascular differentiation state. *Mod. Pathol.* **17**, 1068–1079 (2004).
9. Chen, D. *et al.* Fibroblast bioelectric signaling drives hair growth. *Cell* **0**, (2025).
10. Figuera, L. E., Pandolfo, M., Dunne, P. W., Cantú, J. M. & Patel, P. I. Mapping of the congenital generalized hypertrichosis locus to chromosome Xq24–q27.1. *Nat. Genet.* **10**, 202–207 (1995).
11. DeStefano, G. M. *et al.* Position effect on FGF13 associated with X-linked congenital generalized hypertrichosis. *Proc. Natl. Acad. Sci.* **110**, 7790–7795 (2013).
12. Jung, H.-S. *et al.* Local Inhibitory Action of BMPs and Their Relationships with Activators in Feather Formation: Implications for Periodic Patterning. *Dev. Biol.* **196**, 11–23 (1998).
13. Ramos, R. *et al.* Parsing patterns: Emerging roles of tissue self-organization in health and disease. *Cell* **187**, 3165–3186 (2024).
14. Cooper, R. L. *et al.* An ancient Turing-like patterning mechanism regulates skin denticle development in sharks. *Sci. Adv.* **4**, eaau5484 (2018).
15. Best, A. & Kamilar, J. M. The evolution of eccrine sweat glands in human and nonhuman primates. *J. Hum. Evol.* **117**, 33–43 (2018).
16. Cheng, C. W. *et al.* Predicting the spatiotemporal dynamics of hair follicle patterns in the developing mouse. *Proc. Natl. Acad. Sci.* **111**, 2596–2601 (2014).

17. Sick, S., Reinker, S., Timmer, J. & Schlake, T. WNT and DKK Determine Hair Follicle Spacing Through a Reaction-Diffusion Mechanism. *Science* **314**, 1447–1450 (2006).
18. Kamberov, Y. G. *et al.* Comparative evidence for the independent evolution of hair and sweat gland traits in primates. *J. Hum. Evol.* **125**, 99–105 (2018).
19. Kamberov, Y. G. *et al.* A genetic basis of variation in eccrine sweat gland and hair follicle density. *Proc. Natl. Acad. Sci.* **112**, 9932–9937 (2015).
20. Montagna, W. The evolution of human skin(?). *J. Hum. Evol.* **14**, 3–22 (1985).
21. Mayer, J. J. Wild pig physical characteristics. *Wild Pigs Biol. Damage Control Tech. Manag.* 25–50 (2009).
22. Summerfield, A., Meurens, F. & Ricklin, M. E. The immunology of the porcine skin and its value as a model for human skin. *Mol. Immunol.* **66**, 14–21 (2015).
23. Springer, M. S. *et al.* Genomic and anatomical comparisons of skin support independent adaptation to life in water by cetaceans and hippos. *Curr. Biol.* **31**, 2124–2139.e3 (2021).
24. Glover, J. D. *et al.* The developmental basis of fingerprint pattern formation and variation. *Cell* **186**, 940–956.e20 (2023).
25. Li, J. *et al.* Limb development genes underlie variation in human fingerprint patterns. *Cell* **185**, 95–112.e18 (2022).
26. Okajima, M. Development of dermal ridges in the fetus. *J. Med. Genet.* **12**, 243–250 (1975).
27. Alibardi, L. Dermo-epidermal interactions in reptilian scales: Speculations on the evolution of scales, feathers, and hairs. *J. Exp. Zool. B Mol. Dev. Evol.* **302B**, 365–383 (2004).
28. Montagna, W. & Carlisle, K. Structural Changes in Aging Human Skin. *J. Invest. Dermatol.* **73**, 47–53 (1979).
29. Tomiyasu, J. *et al.* Association Between Back Scent Gland Development and Reproductive Status in Male Brown Bears (*Ursus arctos*). *J. Exp. Zool. Part Ecol. Integr. Physiol.* **343**, 629–635 (2025).
30. Tomiyasu, J. *et al.* Testosterone-related and seasonal changes in sebaceous glands in the back skin of adult male brown bears (*Ursus arctos*). *Can. J. Zool.* **96**, 205–211 (2018).
31. Miller, C. R., Waits, L. P. & Joyce, P. Phylogeography and mitochondrial diversity of extirpated brown bear (*Ursus arctos*) populations in the contiguous United States and Mexico. *Mol. Ecol.* **15**, 4477–4485 (2006).
32. Lu, C. P., Polak, L., Keyes, B. E. & Fuchs, E. Spatiotemporal antagonism in mesenchymal-epithelial signaling in sweat versus hair fate decision. *Science* **354**, aah6102 (2016).
33. Leung, Y., Kandyba, E., Chen, Y.-B., Ruffins, S. & Kobiela, K. Label Retaining Cells (LRCs) with Myoepithelial Characteristic from the Proximal Acinar Region Define Stem Cells in the Sweat Gland. *PLOS ONE* **8**, e74174 (2013).

34. Magerl, M. *et al.* Patterns of Proliferation and Apoptosis during Murine Hair Follicle Morphogenesis. *J. Invest. Dermatol.* **116**, 947–955 (2001).
35. Ghazizadeh, S. & Taichman, L. B. Organization of Stem Cells and Their Progeny in Human Epidermis. *J. Invest. Dermatol.* **124**, 367–372 (2005).
36. Chuong, C.-M. Morphogenesis of Epithelial Appendages: Variations on Top of a Common Theme and Implications in Regeneration. in *Madame Curie Bioscience Database [Internet]* (Landes Bioscience, 2013).
37. Ganier, C. *et al.* Multiscale spatial mapping of cell populations across anatomical sites in healthy human skin and basal cell carcinoma. *Proc. Natl. Acad. Sci.* **121**, e2313326120 (2024).
38. Hara, Y. *et al.* Visualization of age-related vascular alterations in facial skin using optical coherence tomography-based angiography. *J. Dermatol. Sci.* **90**, 96–98 (2018).
39. Paquet-Fifield, S. *et al.* A role for pericytes as microenvironmental regulators of human skin tissue regeneration. *J. Clin. Invest.* **119**, 2795–2806 (2009).
40. Jamora, C., DasGupta, R., Kocieniewski, P. & Fuchs, E. Links between signal transduction, transcription and adhesion in epithelial bud development. *Nature* **422**, 317–322 (2003).
41. Cui, C.-Y. *et al.* Involvement of Wnt, Eda and Shh at defined stages of sweat gland development. *Development* **141**, 3752–3760 (2014).
42. Pinkus, H. CHANGE 'RETE PEGS' TO RETE RIDGES. *Arch. Dermatol.* **88**, 225 (1963).
43. Chuong, C.-M., Chodankar, R., Widelitz, R. B. & Jiang, T.-X. Evo-Devo of feathers and scales: building complex epithelial appendages: Commentary. *Curr. Opin. Genet. Dev.* **10**, 449–456 (2000).
44. Jensen, U. B., Lowell, S. & Watt, F. M. The spatial relationship between stem cells and their progeny in the basal layer of human epidermis: a new view based on whole-mount labelling and lineage analysis. *Development* **126**, 2409–2418 (1999).
45. Cotsarelis, G., Sun, T.-T. & Lavker, R. M. Label-retaining cells reside in the bulge area of pilosebaceous unit: Implications for follicular stem cells, hair cycle, and skin carcinogenesis. *Cell* **61**, 1329–1337 (1990).
46. Vidal, V. P. I. *et al.* Sox9 Is Essential for Outer Root Sheath Differentiation and the Formation of the Hair Stem Cell Compartment. *Curr. Biol.* **15**, 1340–1351 (2005).
47. Nowak, J. A., Polak, L., Pasolli, H. A. & Fuchs, E. Hair Follicle Stem Cells Are Specified and Function in Early Skin Morphogenesis. *Cell Stem Cell* **3**, 33–43 (2008).
48. Jensen, K. B. *et al.* Lrig1 Expression Defines a Distinct Multipotent Stem Cell Population in Mammalian Epidermis. *Cell Stem Cell* **4**, 427–439 (2009).
49. Ito, M. *et al.* Stem cells in the hair follicle bulge contribute to wound repair but not to homeostasis of the epidermis. *Nat. Med.* **11**, 1351–1354 (2005).
50. Nguyen, M. B. *et al.* Tenascin-C expressing touch dome keratinocytes exhibit characteristics of all epidermal lineages. *Sci. Adv.* **10**, eadi5791 (2024).

51. Genderen, C. van *et al.* Development of several organs that require inductive epithelial-mesenchymal interactions is impaired in LEF-1-deficient mice. *Genes Dev.* **8**, 2691–2703 (1994).
52. Zhou, P., Byrne, C., Jacobs, J. & Fuchs, E. Lymphoid enhancer factor 1 directs hair follicle patterning and epithelial cell fate. *Genes Dev.* **9**, 700–713 (1995).
53. Botchkarev, V. A. & Fessing, M. Y. EGF Signaling in the Control of Hair Follicle Development. *J. Investig. Dermatol. Symp. Proc.* **10**, 247–251 (2005).
54. Dingwall, H. L. *et al.* Sweat gland development requires an eccrine dermal niche and couples two epidermal programs. *Dev. Cell* **59**, 20-32.e6 (2024).
55. Salz, L. & Driskell, R. R. Horizontal Whole Mount: A Novel Processing and Imaging Protocol for Thick, Three-dimensional Tissue Cross-sections of Skin. *J. Vis. Exp. JoVE* (2017) doi:10.3791/56106.
56. Folgueras, A. R. *et al.* Architectural Niche Organization by LHX2 Is Linked to Hair Follicle Stem Cell Function. *Cell Stem Cell* **13**, 314–327 (2013).
57. Liu, Y. *et al.* Hedgehog signaling reprograms hair follicle niche fibroblasts to a hyper-activated state. *Dev. Cell* **57**, 1758-1775.e7 (2022).

Responses to reviewers and quotes from the revised manuscript have been color coded for visual clarity:

Original submission in "black"

First revision in "blue"

Second revision in "green"

Third revision in "purple"

Referees' comments:

Referee #1 (Remarks to the Author):

Thank you for working hard to address my concerns and revise your initial conclusions.

My comments are addressed satisfactorily now.

We wish to sincerely thank this reviewer for their rigorous engagement with the concepts in our manuscript. Their critical feedback helped guide the manuscript towards greater functional validation and refined presentation of our experimental conclusions.

Referee #2 (Remarks to the Author):

The authors have satisfactorily addressed most of the reviewers' concerns. Only a few minor editing errors remain that should be corrected see below. Congratulations to the authors for their tremendous work.

We wish to sincerely thank this reviewer for their conceptual critiques and close reading of our manuscript for typos, especially related to figure panel citations. Their engagement with the concepts in our manuscript helped shape the narrative over the course of the review process to emphasize concepts which were functionally validated and refined from our original submission.

- Line 124-125: Supplemental Fig. 1a does not show dermal pockets. Instead, it shows fetal human skin when rete ridges are not formed. Although the authors indicated that this citation was corrected, it appears not to be.

We apologize for the oversight in the previous revision and have now corrected this: "Additionally, the space beneath the inter-ridge epidermis is occupied by dermal pockets – a prominently vascularized region of the papillary dermis (Fig 1b; Extended Data Fig 1b)" (lines 114-116).

- The panel identifiers in the figure legends for Supplemental Fig. 9d, 9e, 9f are incorrect.

We thank the reviewer for catching these typos and have now corrected them. After incorporating some Supplemental Figures as Extended Data Figures, these references are now Supplemental Fig 9d-f.

• Line 412: The correct citation for showing EN1 expression should be Supplemental Fig. 17d, not 17c. Although the authors indicated that this citation was corrected, it appears not to be.

We apologize for the oversight in the previous revision and have now corrected this in the reformatted Extended Data Figure 5:

“... since human fetal volar epidermis expresses *EN1*, but postnatal human foreskin and porcine trunk epidermis does not (Extended Data 5d).”

• Line 306 and 312: The word “patterned” was misspelled.

We thank the reviewer for catching this typo and have corrected it:

“Moreover, by P5, suprabasal proliferation was significantly elevated within the rete ridge domain, suggesting patterned regulation of epidermal proliferation and differentiation within the developing rete ridge (Supplemental Fig 6b-d)” (lines 297-299)

“Therefore, we conclude that spatially patterned proliferation and differentiation support the development and subsequent maintenance of the patterned rete ridge epidermal architecture (Supplemental Fig 6i).” (lines 304-306)

Referee #3 (Remarks to the Author):

This is an amazing manuscript! After re-reading it with all the implemented changes based on the thoughtful suggestions from all reviewers, I'd like to congratulate the authors on this work. All my previous questions have been fully addressed.

We wish to sincerely thank this reviewer for their engagement and suggestions for improving the manuscript's functional validation and conceptual clarity. Their critical feedback helped guide the manuscript towards the final state, with improved functional validation and refined presentation of our experimental conclusions.

* Nature Portfolio's authors website contains information about and links to policies and resources.

This email has been sent through the Springer Nature Manuscript Tracking System NY-610A-SN&MTS

Confidentiality Statement:

This e-mail is confidential and subject to copyright. Any unauthorised use or disclosure of its contents is prohibited. If you have received this email in error please notify our Manuscript Tracking System Helpdesk team at <http://platformsupport.nature.com> . Details of the confidentiality and pre-publicity policy may be found here <http://www.nature.com/authors/policies/confidentiality.html>
Privacy Policy | Update Profile